

# State of Wildfires 2024-25

Douglas I. Kelley[1],* Chantelle Burton[2],*, Francesca Di Giuseppe[3],*, Matthew W. Jones[4],*, Maria L. F. Barbosa[1,5], Esther Brambleby[4], Joe R. McNorton[3], Zhongwei Liu[6], Anna S. I. Bradley[2], Katie Blackford[7], Eleanor Burke[2], Andrew Ciavarella[2], Enza Di Tomaso[8], Jonathan Eden[9], Igor José M. Ferreira[10], Lukas Fiedler[11,12], Andrew J Hartley[2], Theodore R. Keeping[13], Seppe Lampe[14], Anna Lombardi[8], Guilherme Mataveli[4,10], Yuquan Qu[15], Patrícia S. Silva[16], Fiona R. Spuler[17,18], Carmen B. Steinmann[19,20], Miguel Ángel Torres-Vázquez[21], Renata Veiga[22], Dave van Wees[23], Jakob B. Wessel[24], Emily Wright[2], Bibiana Bilbao[25,26], Mathieu Bourbonnais[27], Gao Cong[28], Carlos M. Di Bella[29,30], Kebonye Dintwe[31], Victoria M. Donovan[32], Sarah Harris[33], Elena A. Kukvaskaya[34], Brigitte N'Dri[35], Cristina Santín[36,37], Galia Selaya[38,39], Johan Sjöström[40], John Abatzoglou[41], Niels Andela[23], Rachel Carmenta[42], Emilio Chuvieco[43], Louis Giglio[44], Douglas S. Hamilton[45], Stijn Hantson[46], Sarah Meier[47], Mark Parrington[8], Mojtaba Sadegh[48], Jesus San-Miguel-Ayanz[49], Fernando Sedano[49], Marco Turco[50], Guido R. van der Werf[51], Sander Veraverbeke[4,15], Liana O. Anderson[52], Hamish Clarke[53], Paulo M. Fernandes[54], Crystal A. Kolden[55]

**Affiliations:**
[1] Water and Climate Science, UK Centre for Ecology & Hydrology, Wallingford OX10 8BB, U.K.
[2] Hadley Centre, Met Office, Fitzroy Road, Exeter, UK, EX1 3PB
[3] European Centre for Medium-range Weather Forecast, Shinfield Park, Reading RG29AX, United Kingdom
[4] Tyndall Centre for Climate Change Research, School of Environmental Sciences, University of East Anglia, Norwich Research Park, Norwich, UK, NR4 7TJ
[5] Federal University of São Carlos, Rodovia Lauri Simões de Barros, km 12 - SP-189 - Aracaçu, Buri - São Paulo, 18290-000, Brazil
[6] National Centre for Earth Observation, University of Leicester, Space Park Leicester, 92 Corporation Road, Space City, Leicester, LE4 5SP, UK
[7] Department of Life Sciences, Imperial College London, Chemistry Building CHEM062, South Kensington, London, SW7 2AZ, UK
[8] European Centre for Medium-Range Weather Forecasts, Robert-Schuman-Platz 3, 53175 Bonn, Germany
[9] Centre for Agroecology, Water and Resilience, Coventry University, Wolston Ln, Ryton-on-Dunsmore, Coventry CV8 3LG, UK
[10] Earth Observation and Geoinformatics, National Institute for Space Research (INPE), Astronautas Avenue 1758, São José dos Campos, Brazil, 12227-010
[11] Institute of Oceanography, Center for Earth System Research and Sustainability, University of Hamburg, Bundesstraße 53, 20146 Hamburg, Germany
[12] International Max Planck Research School on Earth System Modelling, Max Planck Institute for Meteorology, Bundesstraße 53, 20146 Hamburg, Germany
[13] World Weather Attribution, Centre for Environmental Policy, Imperial College London, Weeks Building, 16-18 Prince's Gardens, London SW7 1NE
[14] Water & Climate, Vrije Universiteit Brussel, Pleinlaan 2, 1050 Brussel, Belgium
[15] Faculty of Science, Vrije Universiteit Amsterdam, De Boelelaan 1100, Amsterdam, Netherlands, 1081 HV
[16] School of the Environment, Yale University, 195 Prospect St, New Haven, CT 06511, USA
[17] Department of Meteorology, University of Reading, Brian Hoskins Building, Whiteknights Road, Earley Gate, Reading, RG6 6ET, UK
[18] The Alan Turing Institute, British Library, 96 Euston Rd., London NW1 2DB
[19] Institute for Environmental Decisions, Department of Environmental Systems Science, ETH Zurich
[20] Federal Office of Meteorology and Climatology MeteoSwiss, Operation Center 1, P.O. Box 257, 8058 Zurich-Airport, Switzerland
[21] Environmental Remote Sensing Research Group, Department of Geology, Geography and Environment, Universidad de Alcalá, Calle Colegios 2, Alcalá de Henares 28801, Spain
[22] Laboratory for Environmental Satellite Applications (LASA), Department of Meteorology, Federal University of Rio de Janeiro
[23] BeZero Carbon Ltd, 25 Christopher Street, London, EC2A 2BS, UK
[24] Department of Mathematics and Statistics, University of Exeter, Harrison Building, University of Exeter, North Park Road, EX4 4QF Exeter, United Kingdom
[25] Biology, Departamento de Estudios Ambientales, Universidad Simón Bolívar, Valle de Sartenejas, Aartado 89000, Caracas, Venezuela
[26] UMR Art-Dev 5281, Université Paul Valéry Montpellier, Site Saint-Charles, France , UMR 5281, Site Saint-Charles 1, Rue du Professeur Henri Serre, F-34090 Montpellier, France



[27] Earth, Environmental and Geographic Sciences, University of British Columbia - Okanagan, 1177 University
Way, Kelowna, BC, Canada V1V 1V7
[28] Department of Geography, University of Hong Kong, 10F, The Jockey Club Tower, Centennial Campus,
Pokfulam Road, Hong Kong
[29] Departamento de Métodos Cuantitativos y Sistemas de Información, University of Buenos Aires, Av. San
Martin 4453 (1417), CABA, Argentina
[30] IFEVA (Agricultural Physiology and Ecology Research Institute), Av. San Martin 4453 (1417), CABA, Argentina
[31] Department of Environmental Science, University of Botswana, Plot 4775 Notwane Rd, Gaborone Botswana
[32] West Florida Research and Education Center, School of Forest, Fisheries, and Geomatics Sciences, University
of Florida, 5988 Highway 90, Milton, FL 32583
[33] Fire Risk, Research and Community Preparedness, Country Fire Authority, Burwood East, Victoria, Australia
[34] Laboratory of Experimental and Applied Ecology, V.N. Sukachev Institute of Forest Siberian Branch of the
Russian Academy of Sciences - separate subdivision of the FRC KSC SB RAS, 50/28 Akademgorodok,
Krasnoyarsk, Russian Federation, 660036
[35] Natural Sciences, Nangui Abrogoua University, 02 BP 801 Abidjan 02
[36] Research Institute of Biodiversity (IMIB), CSIC-University of Oviedo-Principality of Asturias, IMIB, Research
Building, Mieres Campus, Mieres, Spain, 33600
[37] Biosciences, Swansea University, Wallace Building, Singleton Campus, Swansea, UK, SA2 8PP
[38] Research and action, ECOSCONSULT, Calle Tte. H. Balcazar No. 24, Santa Cruz de la Sierra
[39] Research, Fundacion Innova, Calle Tte. H. Balcazar No. 24, Santa Cruz de la Sierra
[40] Fire and Safety, RISE Research institutes of Sweden, Box 857, 51515 Borås, Sweden
[41] School of Engineering, University of California, Merced, 5200 N Lake Rd, Merced, CA, 95343, USA
[42] Tyndall Centre for Climate Change Research, School of Global Development, University of East Anglia,
Norwich Research Park, Norwich, UK, NR4 7TJ
[43] Environmental Remote Sensing Research Group, Department of Geology, Geography and Environment,
Universidad de Alcalá, Calle Colegios 2, Alcalá de Henares 28801, Spain
[44] Department of Geographical Sciences, University of Maryland, College Park, MD 20742
[45] Marine, Earth, and Atmospheric Science, North Carolina State University, 2800 Faucette Drive, Raleigh, North
Carolina, USA, 27603
[46] Program in Earth System Sciences, Faculty of Natural Sciences, Universidad del Rosario, Bogotá, Colombia.
[47] Land, Environment, Economics and Policy Institute (LEEP), Department of Economics, University of Exeter,
Rennes Drive EX4 4ST Exeter, United Kingdom
[48] Department of Civil Engineering, Boise State University, Boise, ID, USA
[49] Disaster Risk Management Unit (E.1), Directorate E (Space, Security, and Migration), European Commission
Joint Research Centre, European Commission, Rue du Champ de Mars 21, 1050 Brussels, Belgium
[50] Regional Atmospheric Modelling (MAR) Group, Regional Campus of International Excellence Campus Mare
Nostrum (CEIR), Department of Geography, University of Hong Kong
[51] Meteorology and Air Quality, Wageningen University and Research, Droevendaalsesteeg 3-3 A, 6708 PB,
Wageningen, the Netherlands
[52] Cemaden/MCTI, 500 – Distrito de Eugênio de Melo, São José dos Campos – São Paulo, Brazil
[53] FLARE Wildfire Research, School of Agriculture, Food and Ecosystem Sciences, University of Melbourne,
Grattan St, Parkville, 3010, Australia
[54] ForestWISE—Collaborative Laboratory for Integrated Forest & Fire Management, Centre for the Research and
Technology of Agro-Environmental and Biological Sciences, Universidade de Trás-os-Montes e Alto Douro,
Quinta de Prados, Vila Real, Portugal, 5000-801
[55] Wildfire Resilience Center, School of Engineering, University of California, Merced, 5200 N Lake Rd, Merced,
CA 95343, USA
*These authors contributed equally to this work.
Correspondence to:
doukel@ceh.ac.uk
chantelle.burton@metoffice.gov.uk
Francesca.DiGiuseppe@ecmwf.int
matthew.w.jones@uea.ac.uk
**Key words:** Wildfire, Extreme Fire, Attribution, Climate Change



## Abstract

Climate change is increasing the frequency and intensity of extreme wildfires globally, yet our understanding of these high-impact events remains uneven and shaped by media attention and regional research biases. The State of Wildfire Project systematically tracks and analyses global fire activity and this, its second annual report, covers the March 2024 to February 2025 fire season. During the 2024-25 fire season, fire-related carbon (C) emissions were totalled 2.2 Pg C, 9% above average and the 6th highest on record since 2003, despite below-average global burned area (BA; 3.7 million km2). Extreme fire seasons in South America's rainforests, dry forests and wetlands, and in Canada's boreal forests pushed up the global C emissions total. Fire C emissions were over four times above average in Bolivia, three times above average in Canada, and ~50% above average in Brazil and Venezuela. Wildfires in 2024-25 caused 100 fatalities in Nepal, 34 in South Africa, and 30 in Los Angeles, with additional fatalities reported in Canada, Côte d'Ivoire, Portugal, and Turkey. The Eaton and Palisades fires in Southern California caused 150,000 evacuations and US$140 billion in damages. Communities in Brazil, Bolivia, Southern California, and Northern India were exposed to fine particulate matter at concentrations 13-60 times WHO's daily air quality standards. We evaluated the causes and predictability of four extreme wildfire episodes from the 2024-25 fire season, including in Northeast Amazonia (January-March 2024), the Pantanal-Chiquitano border regions of Brazil and Bolivia (August-September 2024), Southern California (January 2025), and the Congo Basin (July-August 2024). Anomalous weather created conditions for these regional extremes, while fuel availability and human ignitions shaped spatial patterns and temporal fire dynamics. In the three tropical regions, prolonged drought was the dominant fire enabler, whereas in California, extreme heat, wind, and antecedent fuel build-up were the dominant enablers. Our attribution analyses show that climate change made extreme fire weather in Northeast Amazonia 30–70 times more likely, increasing burned area roughly fourfold compared to a scenario without climate change. In the Pantanal–Chiquitano, fire weather was 4–5 times more likely, with up to 35-fold increases in burned area. In Southern California, climate change made larger burned area 89% more likely, with burned area up to 25 times higher. The Congo Basin's fire weather was 3–8 times more likely with climate change, with a 2.7-fold increase in burned area. Socioeconomic changes since the pre-industrial period, including land-use change, also likely increased burned area in Northeast Amazonia. Our models project that events on the scale of 2024-25 will become up to 57%, 34%, and 50% more frequent than in the modern era in Northeast Amazonia, the Pantanal-Chiquitano, and the Congo Basin, respectively, under a middle-of-the-road scenario (SSP370). Climate action can limit the added risk, with frequency increases kept below 15% in all three regions under a strong mitigation scenario (SSP126). In Southern California, the future trajectory of extreme fire likelihood remains highly uncertain due to poorly constrained climate-vegetation-fire interactions influencing fuel moisture, though our models suggest that risk may decline in future. This annual report from the State of Wildfires Project integrates and advances cutting-edge fire observations and modelling with regional expertise to track changing global wildfire hazard, guiding policy and practice towards improved preparedness, mitigation, adaptation, and societal benefit. Thirteen new datasets and model codebase's presented in this work are available from the State of Wildfires Project's Zenodo community (https://zenodo.org/communities/stateofwildfiresproject, last access: 11 August 2025).

## Short Summary

The second State of Wildfires report examines extreme wildfire events from 2024 to early 2025. It analyses key regional events in Southern California, Northeast Amazonia, Pantanal-Chiquitano, and the Congo Basin, assessing their drivers, predictability, and attributing them to climate change and land use. Seasonal outlooks and decadal projections





168 are provided. Climate change greatly increased the likelihood of these fires, and without
169 strong mitigation, such events will become more frequent.

## 1. Introduction

### 1.1. Background

The potential for wildfires is growing under climate change, with increases in the frequency
and intensity of drought and periods of fire-favourable weather driving reductions in
vegetation (fuel) moisture and priming landscapes to burn more regularly, intensely, and
severely (Seneviratne et al., 2022; UNEP, 2022a; Jones et al., 2022; Abatzoglou et al., 2019;
Cunningham et al., 2024a). Additionally, human activities and land use change can
contribute to or exacerbate the risk of extremely large, fast-moving or intense fires,
especially in tropical forests where people are the primary cause of ignition and forest
degradation (Lapola et al., 2023). Recent years have been marked by a series of extreme
wildfire events spanning the globe (Abatzoglou et al., 2025), with record levels of burned
area (BA) occurring in the 2019-2020 Australian "Black Summer" bushfires (Abram et al.,
2021; Canadell et al., 2021) and a series of high-ranking wildfire seasons occurring in quick
succession in the western US (2020 and 2021; Higuera & Abatzoglou, 2020), Siberia (2020
and 2021; Zheng et al., 2023), varying parts of Europe (e.g. 2017, 2022, 2023; European
Commission Joint Research Centre, 2023, 2024, 2025), South America (2019, 2020, 2023,
2024; Kelley et al., 2021; Barbosa et al., 2022; Silveira et al., 2020; Mataveli et al., 2024,
2025), and Canada (2023, 2024; Jones et al., 2024b; Jain et al., 2024; Byrne et al., 2024;
Kolden et al., 2024). The 2024-25 fire season was marked by extreme fire extent and
emissions in Amazonia and the Pantanal-Chiquitano (Mataveli et al., 2025; Kolden et al.,
2025) and a second consecutive year of extreme fire extent and emissions in Canada
(Kolden et al., 2025; Parrington and Di Tomaso, 2025). The 2024-25 fire season also saw
extreme fire activity in equatorial Africa, which went broadly under-reported despite fires
triggering record rates of forest loss (stand-replacing fire extent) in the region (World
Resources Institute, 2025). Meanwhile, extremely destructive and costly individual fires
affected Southern California (Barnes et al., 2025; Woolcott, 2025) and Jasper National Park
in Alberta (Parks Canada, 2024; Insurance Bureau of Canada, 2025). Widespread regional
anomalies in high fire activity were also seen in northern India leading to severe haze events
(CAMS, 2024).
The prominence of recent extreme wildfires and wildfire seasons notably contrasts with
overall trends in the area burned by fires globally. A distinctive trend has emerged towards
enhanced fire activity and severity in forests and other fuel-rich environments, which is
occurring amidst increasingly frequent and intense droughts and heatwaves, particularly in
the extratropics (Jones et al., 2024a, Cunningham et al., 2024a). Due mostly to a reduction
in the global savannahs tied to landscape fragmentation and changing rainfall patterns,
global BA has fallen since the beginning of this century by around one-quarter (Andela et al.,
2017; Jones et al., 2022; Chen et al., 2024). Critically, this decline in fire extent masks major
shifts in the distribution of fires globally, with regions such as eastern Siberia and the
western US and Canada experiencing a more than 40% increase in BA since 2000 (Jones et
al., 2022; Zheng et al., 2021) and regions such as southeast Australia also showing
significant increases over longer periods despite high interannual variability (Canadell et al.,
2021). Likewise, there have been shifts in the global distribution of BA from non-forests to
forests globally and from the tropics to the extratropics, with the increased prevalence and
severity of forest fires emitting increasing quantities of forest carbon stocks each year and
driving increasing fire carbon (C) emissions globally (Kelley et al., 2019; Jones et al., 2024a).
Hence, focussing exclusively on global aggregated BA extent underplays the scale and
magnitude of the significant shifts in wildfire activity and impacts that are underway across
many world regions. An increase in forest and peatland burning is particularly concerning





due to the rich ecosystem services that these regions provide, including C storage and
biological and cultural diversity (UNEP, 2022b). The intensification of fire regimes in
environments that are less fire-adapted is of high importance because these ecosystems are
expected to be least resilient to such changes (Grau-Andrés et al., 2024) and because they
are often home to communities relying directly on the forest (Newton et al., 2022; Shepherd
et al., 2020; Schleicher et al., 2018).
The extreme wildfire events of recent years have significantly impacted societies and
ecosystems across the globe (Cunningham et al., 2024a, 2024b; 2025). Since 1990, wildfire
disasters have directly killed or injured at least ~18,000 people, a conservative measure
based on incomplete records and reporting biased to the global Northern countries (updated
from Jones et al., 2022; Centre for Research on the Epidemiology of Disasters, 2024). In
2023, 232,000 people were evacuated due to wildfires in Canada alone (Jain et al., 2024;
Kolden et al., 2024). Also since 1990, fires are estimated to have caused on the order of 1.5
million premature deaths globally per year through degraded air quality related to fine
particulate matter (PM$_{2.5}$; Johnston et al., 2021; Xu et al., 2024; Chen et al., 2021). Degraded
air quality related to fires is experienced most strongly in the tropics (Pai et al., 2022) and
often disproportionately affects the elderly, the young, the infirm, and traditional communities
with poor public services or means of protection (Carmenta et al., 2021; Johnston et al.,
240 2021).
As anthropogenic emissions of $CO_2$ remain persistently high, the world's natural C sinks in
forests, peatlands, and other ecosystems are increasingly pivotal to moderating increases in
atmospheric $CO_2$ concentration (Friedlingstein et al., 2025). Intact forests are often relied
upon for delivering national plans for reaching Net Zero (Smith et al., 2023) and offering sites
for nature based solutions (NBS). Yet, massive wildfire emissions from boreal forests and
soils in Siberia and Canada across the years 2020, 2021, and 2023 amount to over 1 billion
tonnes of C, a gross flux comparable in magnitude to annual $CO_2$ emissions from fossil fuel
combustion in India, the EU27 or the USA (Friedlingstein et al., 2025; Zheng et al., 2023). In
a natural fire regime, these gross emissions would likely be recuperated through post-fire
recovery. However, the greater vegetation mortality and loss of ecosystem function
associated with more widespread and severe fires contribute to shifts in local to regional
terrestrial carbon budgets from sinks to sources (Zheng et al., 2021; Gatti et al., 2021; Nolan
et al., 2021a; Phillips et al., 2022; Harrison et al., 2018; Jones et al., 2024a). Loss of
vegetation during extreme fire seasons can also have wider lasting effects on ecosystems,
for instance by reducing the habitat area available to endemic species (Ward et al., 2020;
Carmenta et al., 2025).
Extreme fires can moreover impact the livelihoods of various communities and landowners
who depend on intact natural landscapes. For example, the lands, territories and cultural
heritage of traditional communities and Indigenous Peoples can be degraded and
transformed by wildfires, raising climate justice issues that compound a legacy of
colonisation, dispossession and forced cessation of cultural practices (Garnett et al., 2018;
Barlow et al., 2018; Lapola et al., 2023; Pascoe et al., 2024). Further, conflating the
detrimental impacts of wildfire types has also stigmatised small-scale intergenerational fire
use and led to prohibitive fire governance that affects local communities (Carmenta et al.,
2021; Barlow et al., 2020; Pascoe et al., 2024).
Mitigating and adapting to increases in wildfire potential are growing priorities of
policymakers and require coordination with many other stakeholders. National and
international disaster management centres are seeking to enhance predictive capacity, while
fire management agencies are expanding or re-allocating their resources to rapidly suppress
fires to avoid them becoming too large, fast, or intense (e.g. Bowman et al., 2020). A number
of international organisations such as the UN Environment Programme (UNEP, 2022a), the
World Bank (2020, 2024a), the Organisation for Economic Co-operation and Development



(OECD, 2023), and a range of other inter- or non- governmental organisations are producing reports that consolidate evidence on the changing risk of extreme fires and identify best practices for mitigating their impacts, including through land management and urban/rural planning. Many land managers are developing and implementing approaches such as fuel reduction (Fernandes and Botelho, 2003; Stephens et al., 2012; Moreira et al., 2020; Chuvieco et al., 2023; Hsu et al., 2025). Wildfire response agencies are exploring innovative approaches to detecting and responding to fires, and there is rising interest in the prospect of integrated fire management around the world (Food and Agriculture Organization of the United Nations, 2024). Operators of C market projects and forest carbon-conservation initiatives, such as REDD+ are particularly wary of the risks that wildfires present to the permanence of C offsets, which often feature as a key tool in national policies and international initiatives for achieving Net Zero emissions (Barlow et al., 2012; Smith et al., 2023).

Amidst extreme wildfires and wildfire seasons, stakeholders increasingly turn to scientists for answers. How extreme was this fire event in a historical context? Is climate change amplifying fire occurrence? Can we disentangle the factors responsible in order to target those in policy and management? Will we see more wildfires like this in the future? Did land use or management factors exacerbate or ameliorate the problem? Could we have predicted these events and how can we improve early warning systems and preparedness in the future? What is the role of climate and socioeconomic factors, such as land use, in reducing risk of extreme wildfires in future?

While observational, statistical, and modelling tools for assessing extreme wildfire drivers and predicting wildfire occurrence are advancing rapidly, their application to studying extreme wildfire seasons or events on timescales relevant to public and political interest remains limited. The State of Wildfires report represents a new initiative to systematically catalogue extreme wildfire events at annual frequency and explain their occurrence, predictability and attribution to climate and land use changes. The report incorporates recent methodological advances in disentangling the drivers of four selected extreme wildfire events to fuel dryness, fuel load, and weather, and ignition and suppression factors. By applying these methodological advances in conjunction with models of global change, we quantify the change in likelihood of the past year's events under climate and land use changes. Observable fire metrics (e.g. BA) are the target variable of our causal inference and attribution work, which thereby advances on more common climate attribution studies that attribute change in fire-favourable meteorological conditions to climate change. Overall, this report capitalises on recent advances in the study of extreme fire events and seasons to provide timely information about shifting fire regimes and their causes. The findings of the report are relevant to organisations involved in prevention and combat efforts, policymakers, the media, and the wider public.

## 1.2. Objectives of this Report

The State of Wildfires report aims to deliver actionable information to policy and practice stakeholders and wider society. In rising to this challenge, we aim to spur scientific and technological innovation including stimulating development of better tools for understanding and predicting extreme fires. In this edition we:

1. Regionally identify extreme individual wildfires or extreme wildfire seasons of the period March 2004-February 2025, and place them in context of recent trends.
2. Quantify the impacts of extreme events in terms of the exposure of population, physical assets (built environment), and carbon projects to fire as well as degraded air quality.



3. Shortlist a selection of four extremes (extreme individual wildfires or extreme wildfire seasons) with notable impacts on society or the environment, which we term the 'focal events'.
4. Diagnose the contributions of both fuel dryness and load, ignitions, and suppression to the occurrence of each focal event.
5. Assess the capacity of operational predictive systems to predict each focal event.
6. Attribute each focal event to anthropogenic influences by testing the role of climate change and socioeconomic factors such as land use, land use change, and human ignitions.
7. Provide an outlook on the probability of extreme events in the coming fire season (commencing March 2025).
8. Project future changes in the probability of each focal event under future climate scenarios.

Key methodologies used to achieve the above objectives are summarised as follows. To address objectives 1 and 2, we build a comprehensive dataset of fire metrics including BA, fire counts, fire C emissions, and individual fire properties (size and rate of growth) for consistent world regions and quantitatively identify anomalies in these metrics during the past fire season (Giglio et al., 2018; van der Werf et al., 2017; Andela et al., 2019). To address objectives 3 and 4, we leverage weather forecasts from the European Centre for Medium-Range Weather Forecasts (ECMWF) at different time horizon from medium (1-15 days) to long range (up to 4 months ahead) and additionally employ two state-of-the-art fire models, *Controlar Fogo Local Analise pela Máxima Entropia* - English "Local Fire Control Analysis by Maximum Entropy" (ConFLAME; Kelley et al., 2019; Barbosa et al., 2025b) and *Probability of Fire* (PoF; McNorton et al., 2024) to pinpoint the causes of the extreme fire events of 2024-25. To address objective 5, we employ projections of fire weather from the Hadley Centre Large Ensemble (HadGEM3-A, Ciavarella et al., 2018) to attribute change in the Fire Weather Index (FWI) to climate change, and we drive ConFLAME (Kelley et al., 2019; Barbosa et al., 2025b) with outputs from both HadGEM3-A and separately with the Intersectoral Impacts Model Intercomparison Project 3a (ISIMIP3a) and Joint UK Land Environment Simulator Earth System model (JULES-ES; Mathison et al., 2023) to attribute extreme BA to climate and land use changes (Burton, Lampe et al., 2024). To address objective 6, we use seasonal outlook of FWI from the Copernicus Emergency Management Service (Di Giuseppe et al., 2024). To address objective 7, we again pair ConFLAME with JULES-ES (Mathison et al., 2023) to project future changes in BA under several future climate and land use scenarios and provide a comprehensive assessment of past and future extreme wildfire events.

The State of Wildfires report was launched in 2024 and is an annual report that can harness and adopt new methodologies brought forward by the scientific community in the interim between its yearly publication. Over the coming years and decades, we aim to enhance the tools presented in this report to predict extremes with increasing lead times, monitor emerging situations in near-real time, and explain their causes rapidly, thus enhancing our ability to deliver timely insights to decision-makers when they are most needed.

## 2. Extreme Wildfire Events of 2024-25

### 2.1. Methods

We catalogued the extreme regional wildfire events or annual fire seasons in the period March 2024-February 2025 based on a combination of anomalies in the distribution of several observable fire metrics from Earth observations (**Section 2.1.1** and **Section 2.1.2**). In this work, the global fire season is defined as occurring in March-February windows oriented around the annual minima of global fire activity in boreal spring (see further details





in **Section 2.1.1.2**). As a new development for this edition of the report, we added statistics describing anomalies in fire intensity during the 2024-25 fire season, complementing anomaly statistics provided in the prior edition related to BA, fire emissions, fire size, and rate of growth.

Due to the diversity of environmental settings in which fires occur and the range of ecological, economic, or societal impacts caused, defining an extreme fire or an extreme fire season remains inherently challenging. To date, extreme fires have commonly been defined by their BA extent, by their feedback on the global climate, and by their socio-economic and ecological impacts (Linley et al., 2022, 2025; Driscoll et al., 2024). We reviewed the range of approaches that can be taken to identify extreme wildfire events in our inaugural report (see **Appendix A** of Jones et al., 2024b) and so do not revisit this in the current article.

While an extreme fire event or extreme fire season may be visible as a significant anomaly against historical Earth observations, the scientific community seeks to apply a more comprehensive definition of extreme fire, including its impacts on society and the environment. To catalogue extreme events that were not necessarily visible in Earth observations, regional expert panels were constructed and given responsibility for identifying extreme events of the past fire season (**Section 2.1.3**). The expert panels were given flexibility to identify and catalogue wildfire characteristics or impacts that are considered regionally extreme but are not necessarily captured by Earth observations. Examples of extremes that can be captured by expert assessment (but not by Earth observations) include: suppression difficulty; fatalities and structure loss; impacts on human health and wellbeing; impacts on agricultural and other economic sectors; impacts on biodiversity, and; impacts on diverse ecosystem services such as recreation, tourism, or other cultural values. Hence, **Section 2.2** identifies a variety of impactful events displaying a broad range of characteristics and impacts that can occur across diverse fire regimes (e.g. Archibald et al., 2009; Cunningham et al., 2024a, 2024b; Keeley, 2009).

As a new development for this report, we added several new analyses providing context to the observed extremes in fire during the past fire season (**Section 2.1.4**). Specifically, we added an analysis of extreme fire weather days for the 2024-2025 fire season allowing the spatial and temporal context of extreme fires with extreme fire weather to be described.

## 2.1.1. Earth Observations of Fire

### 2.1.1.1. Input Datasets

We assembled observations of burned area (BA), synonymous with fire extent, for the period March 2002-February 2025 from the National Aeronautics and Space Administration (NASA) product MCD64A1 (collection 6.1). MCD64A1 provides daily BA observations at 500 m spatial resolution with global coverage and is based on retrievals from the Moderate Resolution Imaging Spectroradiometer (MODIS) sensors mounted to the Terra and Aqua satellites (Giglio et al., 2018, 2021).

We also produced a global record of individual fires for the period March 2002-February 2025 by updating the Global Fire Atlas (Andela et al., 2019) through February 2025, driven by the 500m MODIS BA data. The Global Fire Atlas algorithm clusters burned cells into individual fires, tracks their daily progression, and logs attributes such as fire size and mean daily rate of growth. Our updates are provided at Andela and Jones (2025). The Global Fire Atlas is one of several products tracking daily fire progression and identifying individual fires at global scale based on moderate resolution satellite data (Andela et al., 2019; Laurent et al., 2018; Artés et al., 2019). The product uses the MODIS BA product. The smallest unit of disaggregation is 500m and the shortest timestep on which the expansion of a fire can be





observed is daily. Given its resolution, the Global Fire Atlas is expected to represent the
dynamics of large fires better than smaller fast-moving fires.
In addition, we gathered estimates of fire carbon (C) emissions for the period March
2024-February 2025 from two models driven by Earth observations of active fires or BA:
firstly, the Global Fire Assimilation System (GFAS) product, provided operationally by the
Copernicus Atmospheric Services (CAMS) at 0.1 degree spatial resolution and daily
temporal resolution (Kaiser et al., 2012; European Centre for Medium-Range Weather
Forecasts, 2024), and; secondly, the Global Fire Emissions Database (GFED; version 4.1s)
product at 0.25 degree spatial resolution and daily temporal resolution (van der Werf et al.,
2017). GFAS is driven by the fire radiative power (FRP) retrievals in the MODIS active fire
product MCD14A1 and biome-level relationships between FRP and biomass consumed
based on GFED3 (Kaiser et al., 2012). For the 1997-2016 period, GFED4s is driven by
MODIS BA data (MCD64A1 collection 5) supplemented with small fire BA based on MODIS
active fire data, and a model for biomass productivity and fuel consumption (van der Werf et
al., 2017). For the post-2016 period, emissions are based on active fire detections scaled to
emissions using pixel-based scaling factors derived from the 2003-2016 overlapping period.
As a new analysis developed for the 2024-25 report, we added summaries of the peak (95th
percentile) intensity of the fires detected in the Global Fire Atlas. The underlying data for this
analysis were daily observations of fire radiative power (FRP) from the NASA active fire
products MOD14A1 and MYD14A1 (Giglio et al., 2016). FRP measures the rate of radiant
energy emitted by a fire, which is directly related to the fire's intensity and fuel consumption.
MOD14A1 and MYD14A1 each provide FRP observations at two different times of the day,
with the MOD14A1 dataset produced based on retrievals from the MODIS sensor aboard
NASA's Terra satellite, which overpasses at around 10:30 AM and 10:30 PM local time, and
the MYD14A1 dataset produced based on retrievals from the MODIS sensor aboard NASA's
Aqua satellite, which overpasses at around 1:30 PM and 1:30 AM local time. In our case,
daytime and nighttime observations of FRP were combined into a single dataset of active
fire detections obtained from any satellite overpass and either MODIS sensor. To minimize
potential uncertainties, we excluded FRP measurements associated with large MODIS scan
angles (>50°), and normalized the FRP measurements by pixel size (Li et al., 2024).
The upcoming decommissioning of the Terra and Aqua satellites on which the MODIS
instruments are mounted pose potential challenges for evaluating long-term data records of
BA and estimated emissions from wildfires. The wider community requires continued
development of BA and active fire products from sensors such as VIIRS (e.g., Parrington et
al., 2025).
**2.1.1.2.    *Input Data Uncertainties***
We note that the MODIS BA product data used in our analyses of anomalies in BA and
individual fire properties (via the Global Fire Atlas) are known to be conservative due to the
limitations to detecting small fires (e.g. agricultural fires) based on surface spectral changes
at 500m resolution. Recent work has shown that including detections of small active fires
increases global BA estimates by 93% (Chen et al., 2023). However, variability and trends in
regional BA totals using datasets that include small fires do not differ significantly from the
variability and trends present in the MODIS BA product (Chen et al., 2023). Hence, inclusion
or exclusion of small fires tends to generate biases in central estimates of BA in one
direction or the other, in line with the sensitivity of different sensors to different fire types.
Uncertainty in the detection of small fires is larger than in the case of fires detected in the
MODIS BA product, due to limited validation (van der Werf et al., 2017). The MODIS BA
product with resolution of 500 m is deemed highly suitable for addressing the research
questions of this report, which focus on more impactful fires that tend to burn larger areas.



Uncertainties in the BA estimation can be approached by comparing different existing global
BA products. For instance the estimations of BA from the NASA MCD64A1 product, which is
the basis for the calculations of this paper are 40% lower than ESA FireCCIS311, based on
Sentinel-3 reflectance and VIIRS active fires, and 20% lower than the estimations provided
by the Copernicus Land service (March 2024-Feb 2025 period). Comparing these estimates
with the BA derived from higher resolution sensors, such as Sentinel-2 MSI would probably
double the estimations of MCD64A1, as it was observed in Africa (Chuvieco et al., 2022) and
the GFED5 BA product (Chen et al., 2024).
Uncertainties in fire carbon emissions estimates from GFED4.1s are on the order of
±20-25% at 1 standard deviation for global totals (van der Werf et al., 2017; van der Werf et
al., 2010). Uncertainties in GFED4.1s stem from uncertainties in BA, the amount of biomass
consumed per unit BA, and the carbon emitted per unit biomass burned. Revisions to BA
input data, discussed above, have tended to influence GFED central estimates of fire C
emissions to a greater degree than the uncertainties around central estimates (van der Werf
et al., 2017; Chen et al., 2023). Uncertainties in fire carbon emissions estimates from GFAS
are on the order of approximately ±25% at 1 standard deviation for global totals.
Uncertainties are introduced by missed active fire detections, either below the detection
threshold of the MODIS instruments, or not observed during the limited diurnal coverage of
Low Earth Orbiting satellites, assumptions made for biome classifications, coefficients used
to convert observed thermal anomalies to consumed dry matter, and emission factors used
to estimate emitted quantities of carbon and pyrogenic pollutants. Variation in C emissions
estimates on the order of approximately 20-60% has been observed in studies comparing
multiple emissions products (Wiedinmyer et al., 2023).
The fire radiative power (FRP) data provided by the MOD14A1 and MYD14A1 products are
subject to several well-documented uncertainties that affect both the detection of active fires
and the precision of retrieved energy estimates (Giglio et al., 2016; Wooster et al., 2021).
Omission errors typically arise when fires are obscured by clouds or, in some cases, dense
smoke incorrectly flagged as clouds during masking procedures (Atwood et al., 2016).
Additional omissions occur when the mid-infrared (MIR) radiance levels of small,
low-intensity fires fall below detection thresholds, which is most common in the case of
sub-canopy or peatland combustion (Schroeder et al., 2008; Roberts et al., 2018). Temporal
gaps in satellite coverage also contribute, as MODIS instruments observe any given location
only up to four times per day, often missing short-lived events or peak fire activity in the late
afternoon (Roberts and Wooster, 2014). Commission errors, by contrast, typically occur
when non-fire thermal anomalies are misclassified as active fires. False positives can be
caused by sunglint on water or clouds or by thermally anomalous surfaces such as bare
soils, urban infrastructure, gas flares, and volcanic eruptions, which produce elevated MIR
radiance that mimics fire signatures (Wooster et al., 2021). Contextual detection algorithms
help mitigate these errors by comparing candidate pixels to local background conditions.
These approaches have been particularly successful in reducing commission errors, which
are often below 10% (Giglio et al., 2016; Wooster et al., 2021). In contrast, uncertainties in
omission errors and FRP observations remain less well characterised (Wooster et al., 2021;
Li et al., 2024).

### 2.1.1.3.   Regional Burned Area, Carbon Emissions and Fire Count Totals

We calculated regional totals of BA and C emissions based on a variety of regional layers
defined in **Table 1**. The regional layers represent a range of biogeographical boundaries
(e.g. biomes), geopolitical boundaries (e.g. countries), and values used in scientific reports
(e.g. by the Intergovernmental Panel on Climate Change; IPCC). We calculated monthly
totals of BA and fire C emissions for each region by aggregating monthly BA and daily C
emissions data, summing the data from the input datasets both spatially and temporally as





required. In the case of fire C emissions, we also calculated the mean estimate of fire C
emissions from GFED4.1s and GFAS, regionally.
We adopt a March-February definition of the global fire season (e.g. the latest global fire
season spans March 2024-February 2025). Due to an annual lull in the global fire calendar
in the boreal spring months, fire season BA totals are least sensitive to the shifts in fire
season cutoffs of 1-2 months if the fire season centres on spring (Boschetti and Roy, 2008).
This makes the global fire season centred on spring a pragmatic option for the study of
interannual variability or trends in fire extent (Boschetti and Roy, 2008). The period
March-February is specifically oriented at the end of the austral fire season and before
widespread fires have begun in the boreal extratropics. The regions where this global
definition of the fire season is most problematic are: northern hemisphere South America,
Southeast Asia, and Central America (Giglio et al., 2013).
In addition, we calculated totals of regional fire counts for each global fire season based on
the number of individual fire ignition points present within each region, using ignition point
vectors from the Global Fire Atlas. The resolution of the MODIS data supplied to the Global
Fire Atlas algorithm is 500 m and hence fires that are smaller in scale are omitted. Regional
or national systems may record greater fire counts due to the inclusion of smaller fires.





**Table 1:** Regional layers to which global Earth observations were disaggregated and used to define regions with extreme wildfire seasons or extreme individual wildfire attributes. Regional layers are available from Jones et al. (2025).

| Layer | Short Form | Source | Notes |
|---|---|---|---|
| Biomes | NA | Olson et al. (2001) | |
| Continents | NA | ArcGIS Hub (2024) | |
| Continental Biomes | NA | Olson et al. (2001), ArcGIS Hub (2024) | Spatial intersect of biomes and continents. |
| Ecoregions | NA | Olson et al. (2001) | Ecoregions are geographically inset within biomes. |
| Countries | NA | EU Eurostat (2020) | |
| UC Davis Global Administrative Areas (GADM) Level 1 | GADM-L1 | UC Davis (2022) | First sub-national administrative level, such as states of the US or provinces of China. Version 4.1. |
| Intergovernmental Panel on Climate Change *Sixth Assessment Report (AR6) Working Group I (WGI)* Reference Regions | IPCC AR6 WGI Regions | Iturbide et al. (2020) | |
| Global C Project *Regional C Cycle Assessment and Processes (RECCAP2)* Reference Regions | RECCAP2 Regions | Ciais et al. (2022) | |
| Global Fire Emissions Database (GFED) Basis Regions | GFED4.1s Regions | van der Werf et al. (2006) | |

### 2.1.1.4. Cross-Product Intercomparison of Regional Burned Area Totals

In this report, to characterise the dependence of our findings on BA product choices, we add a supplementary comparison between the regional BA totals detected by the MCD64A1 BA product and two other BA products. The first product was the ESA Climate Change Initiative FireCCIS311 product, derived from Sentinel-3 SYN reflectance and Visible Infrared Imaging Radiometer Suite (VIIRS) active fires (Lizundia-Loiola et al., 2022; see **Figure S1**). FireCCIS311 is provided at a spatial resolution of 300 m and is based on a contextual algorithm based on Sentinel-3 SYN surface reflectance (SYN combines OLCI and SLSTR reflectance), guided by active fire detections from VIIRS. The second product is NASA's VIIRS BA product (VNP64A1 v002; Zubkova et al., 2024; Giglio et al., 2024; see **Figure S1**), generated using an adaptation of the MODIS MCD64A1 Collection 6.1 algorithm, applied to 750 m VIIRS imagery and active fire detections. The hybrid algorithm uses dynamic thresholds on composite imagery derived from a burn-sensitive vegetation index and temporal texture measures, enabling it to distinguish fire-induced changes from other land surface changes. It identifies the burn date at 500 m resolution for each grid cell, with prior probabilities of burned/unburned areas informed by cumulative VIIRS active fire observations.





The FireCCIS311 product has been computed since 2019, and hence our cross-product comparisons focus on the fire seasons March 2019-February 2025. We followed identical approaches as described in prior sections to calculate regional BA totals and to quantify anomalies of the past fire season. With very few exceptions, we find a high level of consistency between the MCD64A1, FireCCIS311, and VIIRS VNP64A1 BA products with regards to both the regional BA totals and the geographical distribution of anomalies and rankings of BA in the 2024-25 fire season versus previous fire seasons since 2019 (**Figure S1;** Jones et al., 2025). This analysis adds confidence that regional anomalies identified in the MCD64A1 BA product are generally replicated across products from different space agencies using different algorithms applied to different combinations of Earth-observing sensors. The MCD64A1 BA product will soon discontinue due to the decommissioning of MODIS sensors aboard NASA's Terra and Aqua satellites. Consistency across products is an encouraging finding for the continuity of our annual reporting.

## 2.1.2. Identifying Extreme Fire Seasons and Events from Earth Observations

### 2.1.2.1. Regions with Extreme Wildfire Seasons

Anomalies in BA, fire C emissions, and fire counts in the latest global fire season (March 2024-February 2025) were calculated in several ways:

(i) as relative anomalies (expressed in %) from the annual mean during all previous March-February periods since 2002 (2003 for fire C emissions);
(ii) as standardised anomalies (standard deviations) from the annual mean during all previous March-February periods since 2002 (2003 for C emissions);
(iii) as a rank amongst all March-February periods since 2002 (2003 for fire C emissions), March 2024-February 2025 inclusive.

In this report, anomalies in fire C emissions are reported based on the two-model mean estimate from GFED4.1s and GFAS, however anomalies based on the GFED4.1s or GFAS estimates individually are also available via Jones et al. (2025).

We identified regions in which the latest fire season was potentially classifiable as 'extreme' based on the rank of BA, C emissions, and fire count amongst all fire seasons. For visualisation purposes, we identified regions in which the latest fire season ranked in the top 5 of all annual fire seasons on record (see **Section 2.2.1**). The BA data for the period March 2002-February 2025 includes 23 fire seasons, while the C emissions data for the period March 2003-February 2025 includes 21 fire seasons. Hence, a top-5 ranking translates approximately to a fire season in the upper quartile of those on record.

We further characterised the onset, peak, and cessation of anomalous monthly BA in March 2024-February 2025. First, we identified the month of the event's peak as the maximum difference between monthly BA values in March 2024-February 2025 and the climatological mean monthly values from the prior March-February periods. Thereafter, the event's onset and cessation were defined as the bounds of consecutive months with above-average BA prior to and following the peak but limited to the March 2024-February 2025 period.

The annual data and anomalies produced using these methods are available from Jones et al. (2025).



### 2.1.2.2. Regions with Extreme Individual Wildfire Attributes

We identified regions in which large or fast-moving fires occurred in the latest fire season based on records of individual fires from the Global Fire Atlas (Andela et al., 2019). For each region (**Table 1)** and year, we estimated the size of the largest fire, the daily rate of growth of the fire that spread most rapidly, the size of the 95th percentile fire, and the daily rate of growth of the 95th percentile fire. In the Global Fire Atlas, the daily rate of growth for any given fire is determined by calculating the average daily rate of growth at which the fire advanced across all its constituent cells. This method includes cells burned by the head, flank, and backfire and produces lower spread rates than if the calculation were based solely on the cells burned by the head fire.

As a new analysis developed for the 2024-25 report, we also identified regions in which intense fires occurred in the latest fire season based on the Global Fire Atlas and FRP observations from the MODIS active fire datasets (MOD14A1 and MYD14A1). Regional values were calculated per fire season across two steps as follows. First, each fire present in the Global Fire Atlas was assigned a peak intensity value equivalent to the 95th percentile of all FRP measurements (daytime and nighttime) occurring within the perimeter and date range of the fire. Second, the regional summary values were taken to be the mean of all peak (95th percentile) intensity values from the cohort of fires occurring in a region and fire season. This approach effectively masks FRP measurements to fires that occur in the Global Fire Atlas prior to averaging, meaning that the fire intensity anomalies presented here relate to the same set of fires as the fire size and fire rate of growth statistics.

Anomalies in each fire attribute were calculated relative to other fire seasons since 2003 using the same metrics as for BA (see ***i-iii*** above), and we identified regions in which the latest fire season featured fires with potentially extreme attributes based on the ranking of the individual fire metrics amongst all fire seasons.

The annual data and anomalies produced using these methods are available from Jones et al. (2025).

## 2.1.3. Identifying Extreme Fire Seasons and Events from Expert Consultation

### 2.1.3.1. Role of Expert Consultation

We assembled a panel of regional experts from each continent (**Table A1**) to contribute to the identification, description, and characterisation of extreme wildfire seasons or impactful events in the latest fire season. A key role of the expert panel was to catalogue regional events that significantly impacted society or the environment but which may not have been detected by Earth-observing satellites due to issues such as scale, short duration, timing of overpass, and cloud or canopy cover. This includes (but is not limited to) wildfires that impacted society by causing fatalities, evacuations, displacement (e.g. homelessness), direct structure or infrastructure loss or damage, degradation of air or water quality, loss of livelihood, cultural practice or other ways of life, and loss of economic productivity. This definition also includes (but is not limited to) wildfires that impact the environment via disturbance to vulnerable ecosystems, biodiverse areas, or ecosystem services such as C storage. This approach recognises that Earth observations do not provide a complete record of all impactful fires. We do not define ubiquitous quantitative thresholds of impact by any of the measures outlined above, but rather invite in-region experts to identify events that triggered impacts that were sufficient in magnitude to infiltrate public and political discourse. The sources of information available for cataloguing regional events include national/regional fire records, land and fire management agencies reports, disaster management reports,





news reports, and social media. A second key role of our expert panel was to describe and contextualise the impacts of the fire seasons highlighted as extreme by Earth observations or regional assessment (see **Section 2.2.3**).

The year in review by continent, produced by the expert panel, is presented in **Appendix A**.

### 2.1.3.2. *Shortlisting of Focal Events*

In later sections of this report, we conducted various analyses to understand the causes and predictability of a selection of extreme wildfire seasons or events during March 2024-February 2025 (see **Sections 4-6**). We limited the number of analyses to three globally prominent focal events of the 2024-25 global fire season because the approaches used are not operational and time is required to train and optimise our models regionally.

In discussion with our expert panel, we prioritised the three events studied in this report by weighing up the anomalies in Earth observations during the latest fire season as well as a suite of impacts that these extremes had on people and the environment. The focal events are notable for their international significance even where they have not attracted international media attention and where they have been highly relevant and recognized within and beyond their region.

## 2.1.4. Contextualising Analyses

### 2.1.4.1. *Contemporaneous Extremes in Fire Weather*

In the supplementary material edition of this report, we introduce routine summaries of the extreme (95th percentile) fire weather days during the March 2024-February 2025 global fire season based on the Fire Weather Index (FWI), a common metric of fire danger developed by the Canadian Forest Service as part of the Canadian Forest Fire Danger Rating System (CFFDRS; van Wagner, 1987). The FWI comprises various components that consider the influence of weather on fire danger, with 2m temperature, 10m wind speed, precipitation, and 2m relative humidity as prerequisite variables.Higher FWI values are generally seen during droughts, heatwaves and strong winds as these conditions are conducive to wildfires in environments with sufficient fuel load (Jolly et al., 2015; Di Giuseppe, 2016; Jones et al., 2022). We base our analysis of extreme (95th percentile) fire weather on the FWI dataset derived from the Copernicus Climate Change Service ERA5 reanalysis (Hersbach et al., 2020; Vitolo et al., 2020) and maintained by the Copernicus Emergency Management Service (CEMS, version 4.1, 2019). The same statistics are reported for the 2024-25 fire season as in the case of fire observational datasets, including (i) ranks, (ii) proportional anomalies, and (iii) standardised anomalies amongst all fire seasons since 2002 (**Figure S2**). Full discussion of the methodology and results are provided in **Supplementary Text S2**. The data produced using these methods are available from Turco et al. (2025).

### 2.1.4.2. *21st Century Trends in Burned Area*

To place recent extremes in the context of fire trends of the past two decades, we update our regional analyses of trends in annual BA from Jones et al. (2022). In addition to reporting trends in *total* BA, we also present trends in *forest* BA as these regularly diverge from total BA trends (**Figure S3**), following Jones et al. (2024a). Full discussion of the methodology and results are provided in **Supplementary Material S2**.





## 2.2. Results

### 2.2.1. Extreme Fire Seasons and Events of 2024-25 from Global Earth Observations

#### 2.2.1.1. Global Summary

According to the MODIS BA product, 3.7 million $km^2$ burned globally during the 2024-25 global fire season (March 2024-February 2025), 9% below the average of previous fire seasons (4.0 million $km^2$) since 2002 and overall ranking 16th (i.e., 8th lowest) of all fire seasons since 2002 (Jones et al., 2025). Despite this, fire C emissions were 9% above average at 2.2 Pg C during the 2024-25 global fire season, which ranks 6th amongst all fire seasons since 2003 (based on annual averages of GFED4.1s and GFAS estimates; see **Section 2.1.2**; Jones et al., 2025). The 2024-25 fire season therefore followed a similar pattern as in the 2023-24 fire season, with above-average emissions occurring despite below-average BA at the global level. These anomalies, signifying lesser fire extent but more severe fires than average, are emblematic of a reported trend towards increased fire extent and intensity in carbon-rich environments such as forests (Jones et al., 2024a). It is important to note that the MODIS BA product is uncorrected for missed small fire detections as in the case other estimates (e.g. Chen et al., 2023; Lizundia-Loiola et al., 2022), meaning that the estimated BA extents from MODIS are conservative (i.e., *at least* 3.7 million $km^2$ burned globally during 2024-25).

Stark regional contrasts in the anomalies in BA, fire C emissions and individual fire properties are visible in the Earth observations at various regional scales (**Figure 1**, **Figure 2, Figure 3**). The three countries with greatest positive anomalies in BA and C emissions during 2024-25 were Bolivia, Brazil, and Canada (**Table 2**, **Table 3**), marking a second consecutive year in which the Americas experienced an anomalous fire season.

On the scale of continental biomes (**Figure 1, Figure 2, Figure 3**), the greatest BA and fire C emissions anomalies of 2024-25 were seen in the North American boreal forests (mostly in Canada), the South American moist tropical forests (mostly in Amazonia), the South American dry tropical forests (mostly in the Chiquitano dry forests of Bolivia), and the South American grassland and savannah biome (mostly in the Cerrado region). On the other hand, it was a second consecutive year the African savannahs experienced a low fire season. In the world's tropical savannah regions, which contribute around 70% towards global BA, the total BA in the 2024-25 fire season was 290 thousand $km^2$ (12%) below average in Africa, slightly above average in South America, and slightly above average in Australia (**Figure 2**). Total BA across the global (sub)tropical grassland, savannah, and shrubland biome was 290 thousand $km^2$ (10%) below average, and the 6th lowest on record, but still contributed 70% towards total global BA during 2024-25. Correspondingly, the C emitted by fires in global savannahs was 102 Tg C (10%) below average in 2024-25.



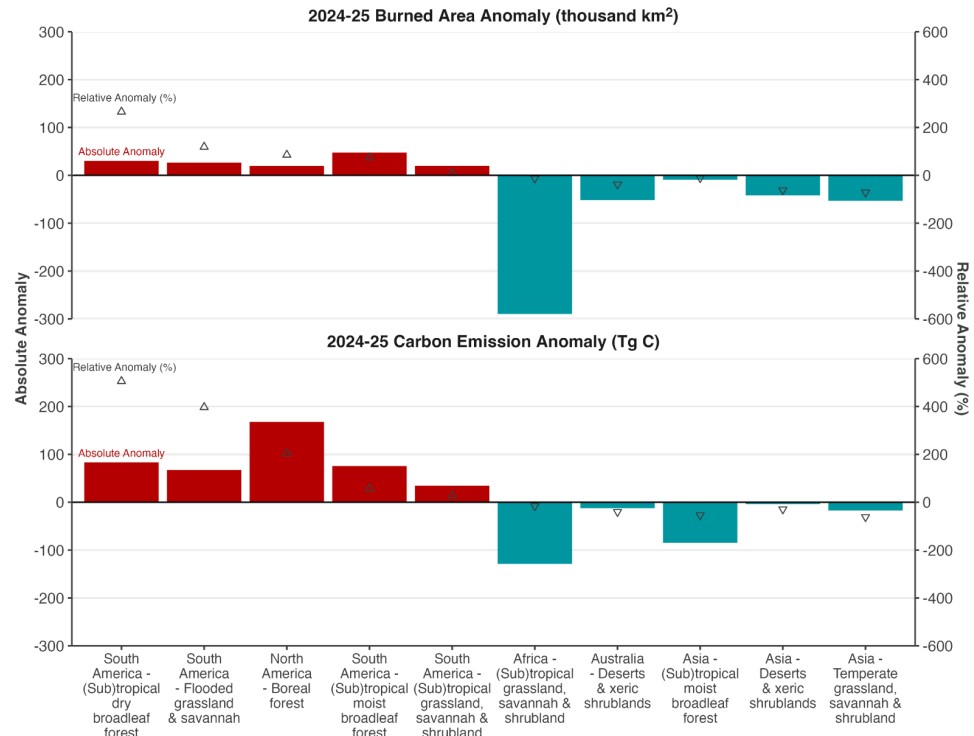

787
**Figure 1:** Anomalies in burned area (BA) and carbon (C) emissions for selected continental
biomes in the 2024-25 global fire season (March 2024-February 2025), versus the average
of prior fire seasons since 2002. The selected regions all experienced BA anomalies of over
±20 thousand km$^2$ or C emissions anomalies over ±30 Tg C during the 2024-25 global fire
season. Relative changes (%) are also marked by triangular symbols and can be read on the
secondary axis.

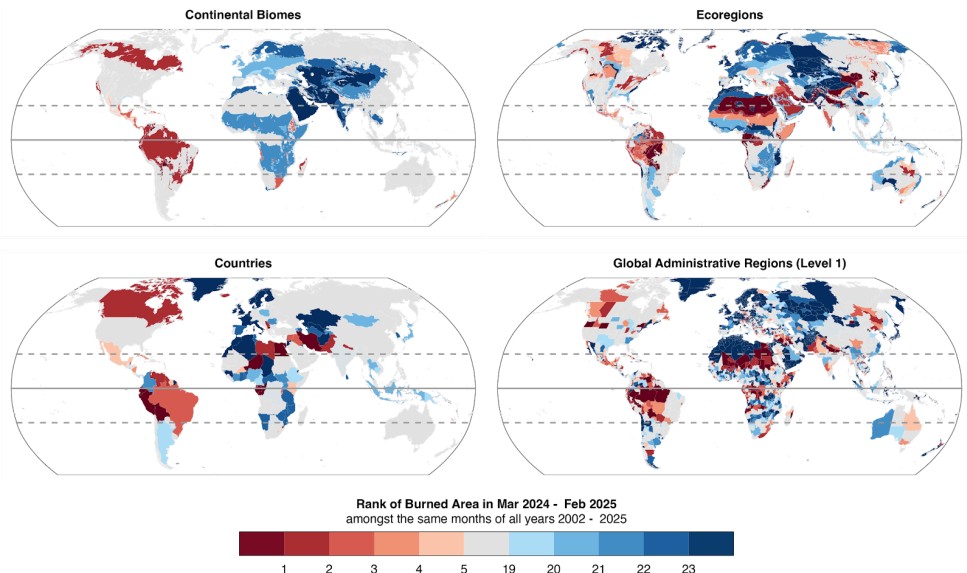

**794**

**795 Figure 2:** Ranks of BA during March 2024-February 2025 versus previous March-February
**796** periods (n = 23 global fire seasons), at the scales of **(top left)** continental biomes, **(top
right)** ecoregions, **(bottom left)** countries and **(bottom right)** level 1 administrative regions.
**798** Results for regions with high-ranking (top 5 years) or low-ranking (bottom 5 years) events
**799** are highlighted. The timing of BA anomalies is shown in **Figure S4**.

**800**

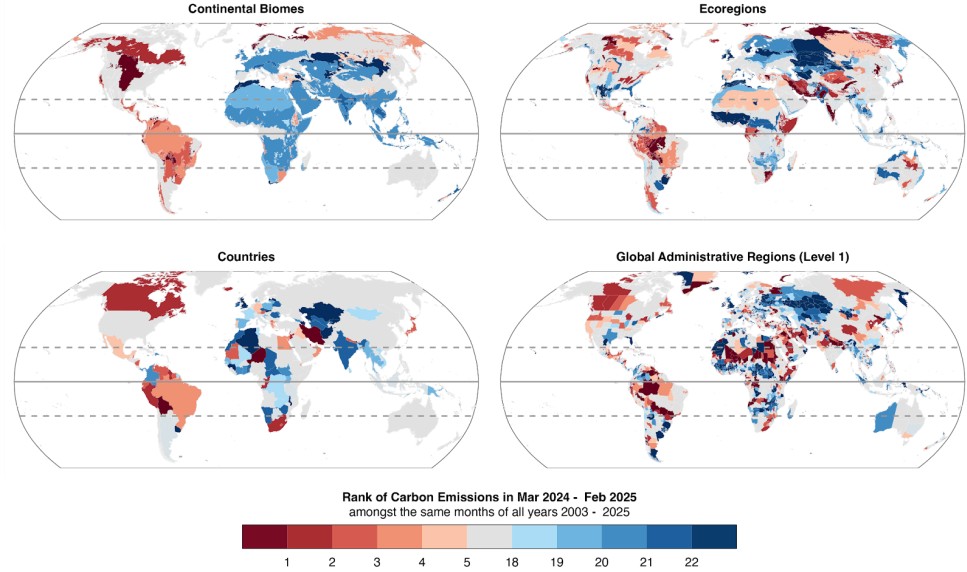

**801**

**802 Figure 3**: Rank of fire C emissions during March 2024-February 2025 versus all
**803** March-January periods since 2003 (n = 22 global fire seasons), at the scales of **(top left)**
**804** continental biomes, **(top right)** ecoregions, **(bottom left)** countries and **(bottom right)** level
**805** 1 administrative regions. We consider C emissions estimates from two products (GFAS and



GFED), first calculating the mean emissions value from the two products, then ranking the
values.
**Table 2:** Summary of the largest positive anomalies in burned area (BA) during the 2024-25
fire season on national and sub-national scales. Anomalies are expressed relative to all
previous fire seasons 2002-2024 (n = 23). The table includes the top ten countries ranked by
the magnitude of their absolute BA anomalies and the top 30 level 1 administrative regions
(e.g. states or provinces) grouped into countries where applicable. Extended data for all
countries and region layers are available from Jones et al. (2025).

| Region Name | BA during the 2024-25 fire season (thousand km²) | Absolute BA anomaly (thousand km²) | Relative BA Anomaly (%) | Ranking of the 2024-25 fire season |
|---|---|---|---|---|
| *Bolivia* | *107* | *+67* | *+169* | *1* |
| Santa Cruz (Department of Bolivia) | 65 | +49 | +311 | 1 |
| Beni (Department of Bolivia) | 36 | +15 | +74 | 4 |
| *Brazil* | *243* | *+59* | *+32* | *3* |
| Mato Grosso (State of Brazil) | 68 | +22 | +49 | 4 |
| Pará (State of Brazil) | 36 | +20 | +119 | 1 |
| Mato Grosso do Sul (State of Brazil) | 23 | +11 | +90 | 2 |
| Amazonas (State of Brazil) | 9 | +6 | +254 | 1 |
| São Paulo (State of Brazil) | 10 | +4 | +67 | 4 |
| *Canada* | *46* | *+21* | *+86* | *2* |
| Northwest Territories (Territory of Canada) | 16 | +12 | +281 | 3 |
| British Columbia (Province of Canada) | 8 | +5 | +154 | 4 |
| Alberta (Province of Canada) | 7 | +4 | +123 | 2 |
| *Venezuela* | *43* | *+15* | *+52* | *2* |
| Apure (State of Venezuela) | 16 | +5 | +41 | 2 |
| Bolívar (State of Venezuela) | 6 | +3 | +133 | 1 |
| *Niger* | *13* | *+10* | *+257* | *1* |
| Tahoua (Department of Niger) | 5 | +4 | +967 | 1 |
| *Burkina Faso* | *33* | *+9* | *+39* | *5* |
| Sahel (Region of Burkina Faso) | 6 | +6 | +1226 | 1 |
| *Angola* | *374* | *+9* | *+2* | *8* |
| Moxico (Province of Angola) | 61 | +8 | +15 | 3 |
| Huíla (Province of Angola) | 20 | +6 | +49 | 1 |
| Cunene (Province of Angola) | 18 | +5 | +35 | 5 |
| Bié (Province of Angola) | 20 | +4 | +25 | 1 |
| *Congo (Republic of the)* | *41* | *+8* | *+25* | *1* |
| *Sudan* | *82* | *+8* | *+11* | *8* |
| North Darfur (State of Sudan) | 15 | +9 | +168 | 1 |
| *Mali* | *77* | *+7* | *+10* | *6* |
| Gao (Region of Mali) | 13 | +12 | +1383 | 1 |
| *Other* | | | | |
| Queensland (State of Australia) | 100 | +19 | +24 | 5 |
| Heilongjiang (Province of China) | 23 | +14 | +164 | 2 |
| Zabaykal'ye (Territory of Russia) | 23 | +11 | +88 | 3 |
| North-Western (Province of Zambia) | 45 | +10 | +29 | 1 |
| Sakha (Republic of Russia) | 27 | +9 | +55 | 6 |
| Amur (Region of Russia) | 20 | +8 | +70 | 4 |
| Zamfara (State of Nigeria) | 9 | +5 | +95 | 4 |
| Oregon (State of United States) | 7 | +5 | +285 | 1 |
| Jilin (Province of China) | 7 | +4 | +186 | 4 |
| Sankuru (Province of Dem. Rep. Congo) | 11 | +4 | +58 | 1 |






**Table 3:** Summary of the largest positive anomalies in carbon (C) emissions during the 2024-25 fire season on national and sub-national scales. Anomalies are expressed relative to all previous fire seasons 2003-2024 (n = 22). The table includes the top ten countries ranked by the magnitude of their absolute C emissions anomalies and the top 30 level 1 administrative regions (e.g. states or provinces) grouped into countries where applicable. Extended data for all countries and region layers are available from Jones et al. (2025).

| Region Name | C emitted during the 2024-25 fire season (Tg C) | Absolute C emissions anomaly (Tg C) | Relative C emissions Anomaly (%) | Ranking of the 2024-25 fire season |
|---|---|---|---|---|
| *Canada* | *282* | *+189* | *+204* | *2* |
| Northwest Territories (Territory of Canada) | 104 | +85 | +441 | 2 |
| Alberta (Province of Canada) | 56 | +42 | +297 | 2 |
| British Columbia (Province of Canada) | 55 | +36 | +196 | 2 |
| Saskatchewan (Province of Canada) | 43 | +28 | +184 | 3 |
| Manitoba (Province of Canada) | 11 | +5 | +74 | 4 |
| *Bolivia* | *187* | *+148* | *+383* | *1* |
| Santa Cruz (Department of Bolivia) | 157 | +136 | +637 | 1 |
| Beni (Department of Bolivia) | 23 | +11 | +86 | 3 |
| La Paz (Department of Bolivia) | 4 | +2 | +79 | 4 |
| *Brazil* | *314* | *+111* | *+55* | *4* |
| Mato Grosso (State of Brazil) | 86 | +29 | +50 | 6 |
| Amazonas (State of Brazil) | 35 | +25 | +237 | 1 |
| Mato Grosso do Sul (State of Brazil) | 30 | +23 | +323 | 1 |
| Pará (State of Brazil) | 59 | +22 | +61 | 4 |
| Tocantins (State of Brazil) | 22 | +5 | +33 | 5 |
| São Paulo (State of Brazil) | 8 | +5 | +190 | 1 |
| Rondônia (State of Brazil) | 22 | +3 | +16 | 7 |
| Roraima (State of Brazil) | 5 | +2 | +81 | 5 |
| *Venezuela* | *26* | *+8* | *+47* | *3* |
| Bolívar (State of Venezuela) | 5 | +2 | +97 | 1 |
| *Mexico* | *29* | *+6* | *+26* | *5* |
| *South Africa* | *18* | *+3* | *+24* | *2* |
| *Angola* | *146* | *+3* | *+2* | *9* |
| Moxico (Province of Angola) | 28 | +5 | +21 | 3 |
| Bié (Province of Angola) | 9 | +2 | +35 | 1 |
| Huíla (Province of Angola) | 7 | +2 | +37 | 1 |
| *Peru* | *7* | *+2* | *+51* | *2* |
| *Russian Federation* | *179* | *+2* | *+1* | *9* |
| Sakha (Republic of Russia) | 75 | +32 | +74 | 3 |
| Zabaykal'ye (Territory of Russia) | 31 | +14 | +78 | 4 |
| Amur (Region of Russia) | 25 | +8 | +46 | 5 |
| Arkhangel'sk (Region of Russia) | 2 | +2 | +1776 | 1 |
| *Congo (Republic of the)* | *10* | *+2* | *+24* | *2* |
| *Other* | | | | |
| Queensland (State of Australia) | 31 | +4 | +14 | 7 |
| Oregon (State of United States) | 7 | +4 | +130 | 3 |
| Idaho (State of United States) | 5 | +3 | +139 | 3 |
| North-Western (Province of Zambia) | 22 | +2 | +12 | 1 |
| Alto Paraguay (Department of Paraguay) | 6 | +2 | +55 | 2 |
| Mai-Ndombe (Province of Dem. Rep. Congo) | 7 | +2 | +36 | 1 |





### 2.2.1.2. An Unprecedented Fire Season in South America

There were pronounced and widespread positive anomalies in BA in 2024-25 across South America during 2024-25 (**Figure 1, Figure 2**). Several South American biomes experienced extremely high or even record-setting BA in the 2024-25 fire season (**Figure 1**). The South American (sub)tropical dry broadleaf forests, principally comprising the Chiquitano and Chaco dry forests, experienced a record-breaking fire season, with the 42 thousand km$^2$ burned exceeding the average since 2002 by a factor of 3.6 and the 100 Tg C emitted exceeding the average since 2003 by a factor of 6. In the South American (sub)tropical moist broadleaf forests, principally comprising the Amazon rainforest, BA was 47 thousand km$^2$ (75%) above the average since 2002, which is the second-highest year on record, and C emissions were correspondingly 76 Tg C (58%) above average. Finally, in the South American Flooded grassland and savannah biome, which principally includes the seasonally inundated Pantanal region, BA was 26 thousand km$^2$ (119%) above the average since 2002, which is also the second-highest year on record, and C emissions were correspondingly 67 Tg C (397%) above average. Across South America as a whole, BA was 120 thousand km$^2$ (35%) above average and C emissions were 263 Tg C (84%) above average, producing the highest C emissions total on record for the continent.

The spatial breadth of the record-setting or high-ranking anomalies in fire extent, emissions, size, rate or spread and intensity (**Figure 2, Figure 3**, **Figure 4**), as well as their impact on society and the environment, made the last fire season unprecedented on the continent. **Appendix A** (**Section A6**) discusses the unprecedented South American fire season of 2024-25 in greater detail, including its impacts and regional context, relying also on information from regional fire monitoring systems and reporting.

Fifteen of South America's 115 ecoregions experienced new record levels of BA or C emissions during 2024-25 (**Figure 2, Figure 3**) and 72 of South America's ecoregions experienced BA or C emissions in the top three years on record (**Figure 2, Figure 3**). Regions with record levels of BA or C emissions included the Chiquitano dry forests and the Pantanal wetlands of Bolivia and central-west Brazil. In nearby southern and southwestern parts of Amazonia, five moist forest and seasonally flooded (várzea) ecoregions also showed record-breaking BA or C emissions. The widespread positive BA anomalies in southern and southwest Amazonia, the Chiquitano and the Pantanal were visible in the MODIS BA dataset from March and April 2024, peaking in August-November 2024 before subsiding around November (**Figure S4**). In the Guianan shield region, encompassing much of Northeast Amazonia (north of the Amazon river and the Rio Negro tributary) and the Guianan forests of Venezuela, Guyana, and Suriname, four moist forest and swamp forest ecoregions also experienced record-breaking levels of BA or C emissions (**Figure 2, Figure 3**). Here, BA anomalies peaked around March-April before subsiding in May in northern parts but persisted through to December in areas closer to the equator (**Figure S4**).

At the national level within South America, the most significant anomalies in BA during the 2024-25 fire season occurred in Bolivia, where BA was 67 thousand km$^2$ (169%) above average and fire C emissions were 148 Tg C (383%) above average, the greatest values on record in the country (**Figure 2**, **Figure 3; Table 2, Table 3**). In Brazil, BA was 59 thousand km$^2$ (32%) above average and emissions were 111 Tg C (55%) above average during 2024-25, making it the country's third highest fire season on record for BA after 2007-08 and 2010-11. Additionally, Venezuela recorded an anomaly of +15 thousand km$^2$ (+52%), its second-highest BA total after 2023-24. Anomalies in these three countries are highlighted due to global totals of BA and C emissions (**Table 2**, **Table 3**). On sub-national scales, the 2024-25 fire season saw record-breaking BA or C emissions in four states of Brazil (Pará, Amazonas, Mato Grosso do Sul, and São Paulo), one department of Bolivia (Santa Cruz), 3 States of Venezuela (Bolivar, Delta Amacuro, Monagas). Other record-breaking anomalies were seen at sub-national levels across South America (**Figure 2, Figure 3**), including in 6

regions of Guyana, 7 regions of Peru, 2 districts of Suriname, 8 provinces of Ecuador, as well as some parts of Chile and Colombia (**Figure 2**, **Figure 3**), clearly signalling the large geographical breadth of the extremes on the continent during the 2024-25 fire season.

For most regions of South America, the anomalies in BA and C emissions were explained by particularly large, fast moving and intense fires, rather than above-average fire counts (**Figure 4**). In Brazil, data on individual fire characteristics from the Global Fire Atlas showed new record fire sizes at the 95th percentile threshold for 6 states (Amapá, Mato Grosso, Mato Grosso do Sul, Paraná, Rondônia, and São Paulo). In Mato Grosso, Mato Grosso do Sul, and São Paulo, 95th percentile fire sizes were 105-266% above average, driving record breaking BA despite fire counts being 18-54% below average. Meanwhile, three states (Mato Grosso, Mato Grosso do Sul, and São Paulo) all saw the fastest rates of growth at the 95th percentile threshold, and 5 states (Mato Grosso do Sul, Paraná, Rio de Janeiro, Roraima, and São Paulo) experienced the most intense fires on record (measured per the average fire's 95th percentile intensity value; **Figure 4**). Unlike in other parts of Brazil, the fire count anomaly (+154%) was record-breaking in Amazonas during 2024-25, combining with the 95th percentile fire size anomaly (+60%) to produce the record-breaking BA. Similar patterns were observed across South America, with anomalies in fire size, rates of growth, and intensities generally being more widespread than anomalies in fire count (**Figure 4**). Some notable exceptions were 5 regions of Peru, 5 regions of Ecuador, 3 regions of Colombia, and 3 regions of Guyana, where record-setting fire counts were observed, as well as in parts of Venezuela where high-ranking fire counts occurred (**Figure 4**).

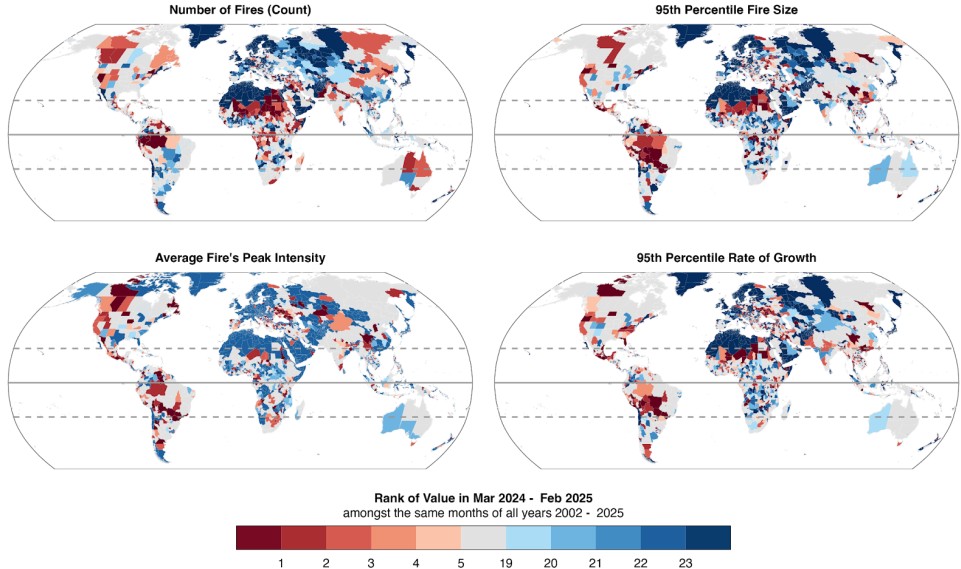

**Figure 4:** Ranks of selected individual fire properties during the March 2024-February 2025 fire season versus previous March-February periods (n = 23 global fire seasons), including **(top left)** fire count, **(top right)** 95th percentile fire size, **(bottom left)** the average value of the peak intensity (95th percentile FRP within fire perimeters) considering all regional fires, and **(bottom right)** 95th percentile daily rate of growth. Results are shown at the scale of states or provinces (GADM administrative level 1 regions).



### 2.2.1.3. A Second Consecutive Extreme Fire Year in North America

The 2024-25 fire season was the second-highest fire year on record for BA and C emissions in the North American boreal forests, with BA 86% above the average since 2002 (+20 thousand km$^2$) and C emissions 3 times the average since 2003 (+168 Tg C). These large anomalies follow the record-breaking 2023-24 fire season when BA was five times above average and C emissions were seven times above average, marking two consecutive years of extreme fire activity in the North American boreal forests. Elsewhere, BA extent was in the top three years on record in the North American (sub)tropical moist broadleaf forest (concentrated in Latin America), and in the North American mediterranean forests, woodlands and scrub (concentrated in Southern California). Across North America as a whole, BA was 31 thousand km$^2$ (35%) above average and C emissions were 194 Tg C (112%) above average, the second highest totals on record for both metrics.

Eleven of North America's 189 ecoregions experienced new record levels of BA or C emissions during 2024-25 (**Figure 2, Figure 3**), with these regions principally concentrated in northwest Canadian taiga and tundra, mountain forests of the northwest US and southwest Canada (principally in Oregon and Alberta), and moist tropical forest ecoregions of mesoamerica (principally in Mexico), but also including the Central Valley grasslands of California and the northeast coastal forests of the US. More broadly, but with a similar geographical distribution, 44 North American ecoregions experienced BA or C emissions in the top three years on record (**Figure 2, Figure 3**). The positive BA anomalies in extratropical North America were visible in the MODIS BA dataset from April 2024 in western regions (e.g. mountain forests of the northwest US and southwest Canada), July-August 2024 in the central regions (e.g. Canadian tundra and taiga), and late into the 2024 summer in eastern regions (e.g. northeast coastal forests; **Figure S4**). Thereafter, BA anomalies were consistently observed through summer (July-September 2024) and in some cases persisted through October 2024.

In Canada, BA was 21,000 km$^2$ (86%) above average and C emissions were 189 Tg C (204%) above average during 2024-25, marking the country's second highest fire season on record immediately following the record-breaking fire season of 2023-24 (**Figure 2**, **Figure 3; Table 2**, **Table 3**). Notably, the anomalies of 2024-25 were concentrated in the western Canadian states of British Columbia, Alberta and northwest Territories which all saw the second-highest BA or C emissions on record, with large anomalies in the range of 120-440%, second only to the 2023-24 fire season. More generally, record levels of BA or C emissions were less spatially extensive in North America than in South America, though the US states of Oregon, Wyoming, and New York saw record BA, as did several mesoamerican states of Mexico, Guatemala, and Costa Rica (**Figure 2**, **Figure 3**).

For western Canada, individual fire metrics from the Global Fire Atlas were also anomalous and highly-ranked amongst previous years, but generally fell short of the records set in the 2023-24 fire season (**Figure 4**). For example, fire counts were 170-190% above average in Alberta and British Columbia, ranked second (behind 2023-24), whereas anomalies in 95th percentile fire size and rate of growth were not particularly large. Meanwhile, the explanation for the anomalous BA in some states of the northwest US was not consistent, with some states experiencing above-average fire counts, some experiencing above-average fire sizes, but few experiencing both.

**Appendix A** (**Section A4**) provides a more complete summary of the fire season in North America based on the regional panel assessment.





### 2.2.1.4.    *A Mixed Picture in Africa*

For the second consecutive year, BA was around 290 thousand km$^2$ (12%) below the average of previous fire seasons in the African (sub)tropical grassland, savannah and shrubland biome, and the 3rd lowest on record (**Figure 2**), but still contributed 56% towards the global BA total and 86% towards total BA in Africa. BA anomalies in the African savannahs have a significant influence on the continental BA anomalies, and indeed BA across Africa as a whole was 313 thousand km$^2$ (12%) below average.

Despite the low fire activity in Africa during 2023, several exceptions emerged in both central and northern Africa. Record levels of BA were observed in several parts of the Congo Basin (**Figure 2, Figure 3**) due to an unusually high number of fires (**Figure 4**). BA in the Republic of Congo was 25% above average, the highest on record, and similarly fire C emissions were 25% above average (**Table 2**, **Table 3**). In the Democratic Republic of the Congo, the Mai-Ndombe and Sankuru provinces each experienced record levels of BA or fire C emissions with anomalies in the range of 36-58% (**Table 2**, **Table 3**). These anomalies were centred on several western ecoregions of the Congo Basin, including the Atlantic Equatorial coastal forests where BA was more than triple the annual mean, Western Congolian swamp forests where BA was twice the annual average and the Central Congolian lowland forests where BA was 77% above average, and the Northwestern Congolian lowland forests where BA was 55% above average.

Likewise, several northern regions of Angola experienced record BA (**Figure 2**, **Figure 3**, **Table 2**, **Table 3**). In northern Africa, Mali, Niger, Chad and Sudan all saw high BA in various states or regions that encompass the semi-arid Sahel region, though these anomalies notably occur against a low baseline in most cases due to the typically sparse vegetation fuel loadings in such regions. **Appendix A** (**Section A1**) provides a more complete summary of the fire season in Africa based on the regional panel assessment.

### 2.2.1.5.    *A Low Fire Year in Eurasia*

Asian and European biomes generally experienced a low fire year that contributed towards the below-average global BA total in 2024-25 (**Figure 1**, **Figure 2**). BA was around 50 thousand km$^2$ (71%) below average in the Asian temperate grassland, savannah and shrubland biome, 42 thousands km$^2$ (62%) below average in the Asian xeric shrublands, and 9 thousand km$^2$ (11%) below average in the Asian (sub)tropical broadleaf forests. The below-average fire extent in all of these regions translated into below-average C emissions, though not in direct proportion because the combustion of vegetation per unit BA also varied compared with previous years (**Figure 1**). For example, while BA was 11% below average in the Asian (sub)tropical broadleaf forests, C emissions were 54% (85 Tg C) below average signifying that areas that did burn tended to do so with anomalously low severity. Across Asia as a whole, the total BA was 99 thousand km$^2$ (26%) below average during 2024-25, the 4th lowest annual total on record, and C emissions were 119 Tg C (28%) below average, the 5th lowest on record.

While most regions of Asia experienced a low fire year in general, there were some notable exceptions. Many states of northeast India and Nepal experienced high-ranking or record-breaking levels of BA or C emissions (**Figure 2, Figure 3**), highlighting a coherent regional-scale anomaly during 2024-25. Similarly, in northeast Asia where 2 provinces of China (Heilongjiang and Jilin; **Table 2**), 2 provinces of South Korea, and 7 prefectures of Japan experienced record-breaking BA or C emissions and many neighbouring regions likewise experienced high-ranking fire years (**Figure 2**, **Figure 3**). **Appendix A** (**Section A2**) provides a more complete assessment of the fire season in Asia.



Though less impactful on the global BA and C emissions totals than in the vast Asian biomes, the 2024-25 fire season was notably another low fire year in Europe. For example, BA was 13 thousand km$^2$ (59%) below average in the European temperate broadleaf and mixed forests, 12 thousand km$^2$ (40%) below average in the European temperate grassland, savannah and shrubland biome. Across Europe as a whole, the total BA was 30 thousand km$^2$ (49%) below average during 2024-25, the 4th lowest annual total on record, and C emissions were 5 Tg C (22%) below average, the 7th lowest on record.

Despite the low fire activity in Europe, there were several exceptions in southeast Europe. In regions of Serbia, North Macedonia, and western Turkey experienced record high BA or C emissions in 2024-25. Further north, several eastern regions of Ukraine experienced record-breaking fire C emissions, with some suggesting a link between elevated ignitions and the ongoing conflict in the country (European Commission Joint Research Centre, 2025). **Appendix A** (**Section A3**) provides a more complete assessment of the fire season in Europe based on regional panel assessment.

### 2.2.2. Focal Events of this Report

In this year's report, we identify four focal events with global relevance for further study across **Sections 4-6**. The four events are Northeast Amazonia, the Pantanal-Chiquitano, Southern California, and the Congo Basin (**Figure 5**), and our reasons for selecting these particular events are detailed below. In **Sections 4-6**, our analyses explain the causes of each of the events (**Section 4**), evaluate the predictability of the events (**Section 4**), attribute the events to climate change and land use factors (**Section 5**), and predict the likelihood of similar events under future climate change scenarios (**Section 6**).

Earth System
Science
Data

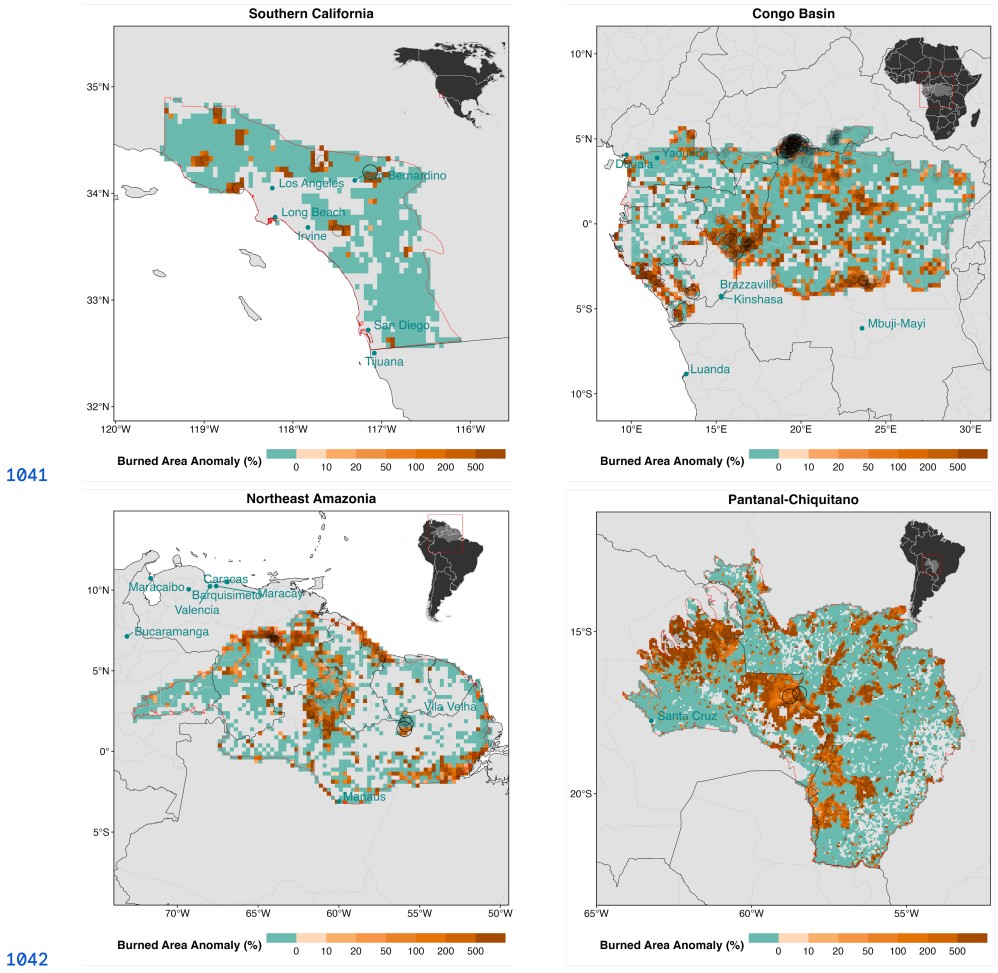

**Figure 5:** Spatial distribution of burned area (BA) anomalies during 2024-25 relative to the mean annual BA (%). BA is shown at 0.25° resolution (Northeast Amazonia and Congo Basin) or 0.05° resolution (Pantanal and southern California). Fire ignition points (open circles) from the Global Fire Atlas are also shown for the fires with sizes in the upper quartile regionally during 2002-2025, with the largest fires for each region displaying as the largest and most visible circles.

### 2.2.2.1. Northeast Amazonia (January-March 2024)

The Northeast Amazonia region here refers to the moist tropical forest ecoregions northeast of the Amazon river and the Rio Negro tributary, mostly including Amazonia but also including the Guianan Shield forests that extend into Venezuela, Guyana, Suriname, and French Guiana (**Figure 5**). We specifically target the period January-March 2024. In this region, as in other parts of the northern hemisphere tropics, our global March-February fire season definition can be misaligned with local fire seasonality, specifically where fire seasons span two calendar years. Although this event straddles the boundary between the 2023-24 and 2024-25 fire seasons, we include it here to ensure that significant fire activity is not excluded solely due to the constraints of our reporting framework. **Section 2.2.1**



discusses the regional anomalies that led this region to be identified (e.g. **Figure 2**, **Figure 3**), with further review of the fire season provided by our expert panel in **Appendix A** (**Section A6**). It emerges as a major event of global relevance for the following reasons:

- **Record-breaking burned area in forests:** The area of forest burned was more than four times (+332%) the average, and the highest on record, while total BA (including non-forests) was also 67% above average. In forests, 8 continuous months of the fire season (March-September 2024) had BA above the climatological mean, peaking in March 2024. The most pronounced anomalies occurred in the Northern Amazonian savannas around Roraima, the forest-savanna transition zones of northern Venezuela and southern Guyana, and the coastal ecosystems near the Guyana-Suriname border (**Figure S5**).
- **Carryover from the previous fire season:** A new record for total BA had been set in the previous fire season (2023-24) mostly due to an anomalously high count of large fires in non-forests. The transition of anomalously high fire activity into forest environments during the 2024-25 fire season was a distinguishing factor.
- **Anomalous fire counts:** The large BA anomalies were explained by an anomalously high number of fires, with 1,500 (52%) more fires than the average fire season.
- **Widespread forest loss:** Highest rates of forest loss (stand-replacing fire extent) since 2016 recorded in Amazonia with 60% attributed to wildfires.
- **Disproportionate impact on rural, traditional populations and Indigenous territories:** Fires degraded air quality and destroyed crops, homes, and native vegetation, intensifying food and water insecurity for those living in the region, including Indigenous peoples. The compounded effects of fire and drought deepened the humanitarian crisis in the Yanomami Territory and local organizations estimate at least 70,000 people across urban and rural areas without access to clean water.

### 2.2.2.2.    Pantanal and Chiquitano (August-September 2024)

The Pantanal-Chiquitano region here refers to the areas draining into the Pantanal (IBGE, 2021), the world's largest tropical wetland area, and the Chiquitano dry forest ecoregion in Bolivia (**Figure 5**). We specifically target the period January-March 2024, when the most substantial anomalies in BA were observed (**Figure S6**). **Section 2.2.1** discusses the regional anomalies in Brazil and Bolivia that led this region to be identified (e.g. **Figure 2**, **Figure 3**), with further review of the fire season provided by our expert panel in **Appendix A** (**Section A6**). It emerges as a major event of global relevance for the following reasons:

- **Record-Breaking burned area:** BA in the Pantanal-Chiquitano region was almost triple (+196%) the annual average, and the highest on record. This anomaly included a +466% BA anomaly in forests. There were 8 continuous months (March-October) with BA above the climatological mean, oriented around a peak in August 2024.
- **Record-Breaking carbon emissions:** Fire C emissions were 6 times (+502%) the annual mean, driven up by the large anomaly in forest fire C emissions in the period.
- **Record fire size and spread:** The 95th percentile fire size for the region was over three times (+226%) the average and the 95th percentile rate of growth was 88% above average, signifying that large, fast-spreading fires drove up the anomalous BA total in the region.
- **Severe air quality degradation:** Over 900 μg/m³ of fine particulate matter ($PM_{2.5}$) was recorded in September 2024, which is 60 times above WHO standard.
- **Economic losses:** Agribusiness losses due to wildfires reached R$ 1.2 billion (~US$222 million) in the Pantanal, the biome's main economic sector.
- **Challenges in response:** 78 days of firefighting effort which involved multiple actors was marked by significant access and logistical challenges in remote regions, making it difficult to reach and support isolated communities.



### 2.2.2.3. Southern California (January 2025)

Southern California here refers to the Mediterranean portions of seven counties in California (Los Angeles, Orange, Riverside, San Bernardino, San Diego, Santa Barbara, and Ventura; **Figure 5**). The Mediterranean portions are defined based on the ecoregional definition of the US Environmental Protection Agency (EPA, 2024). Although California as a whole did not experience a particularly strong fire season in 2024-25 from the vantage of BA or fire C emissions (e.g. **Figure 2**, **Figure 3**), the regional expert panel identified the numerous wildfires affecting LA and surrounding counties in January 2025 as a major event of the 2024-25 fire season (see **Appendix A Section A4**), with the Palisades and Eaton fires in particular leading to loss and damage in the suburbs of LA. We specifically target the period January 2025 when the most substantial anomalies were observed (**Figure S7**). Southern California emerges as a major event of global relevance for the following reasons:

- **High fatalities and structure loss.** Over 11,500 homes were destroyed across Los Angeles County, and at least 30 lives were lost (Los Angeles County Coroner, 2025; Wikipedia, 2025). The Palisades Fire damaged or destroyed nearly 8,000 structures, while the Eaton Fire impacted over 10,000 structures (CALFIRE, 2025; Wikipedia, 2025).
- **Mass evacuations.** At least 153,000 people evacuated, with up to 200,000 under evacuation warnings or orders during the peak of the crisis (USGS, 2025b; NPR, 2025; Wikipedia, 2025).
- **Air quality impacts**. Air and municipal water quality were heavily impacted by the fires, contributing to negative health outcomes for thousands. During the fires, peak $PM_{2.5}$ levels were recorded at 483µg/m³ (an order of magnitude greater than the 35 µg/m³ daily standard set by the US Environmental Protection Association), part of a prolonged period of Hazardous air quality (California Air Resources Board, 2025).
- **Water quality impacts.** Municipal water supplies were considered unsafe for several weeks following the fires for tens of thousands of residents in the affected areas (City of Pasadena, 2025). In response to the fires outside Los Angeles, over 8.3 million cubic meters of water from federal reservoirs in central California, a move which has been criticised because this water did not supply southern California, happened well after the fires were controlled, and because it would otherwise have been used for irrigation in the Central Valley (Levin et al., 2025).
- **Exceptional economic loss.** Total economic losses were estimated at US$140B including property destruction, health costs, business disruption, and infrastructure impacts, making this one of the most costly wildfire events in US history (LAEDC, 2025; UCLA Anderson School of Management, 2025).
- **Wider economic disruption.** The fires are projected to cause US$4.6-8.9 billion in lost economic output over five years, 25,000-50,000 job-years lost, and labour income reductions of US$1.9-3.7 billion (LAEDC, 2025). The Palisades and Eaton fires affected almost 2,000 businesses (LAEDC, 2025). As LA is also the largest port on the US Pacific coast, the fires impacted broader supply chains that run through the port of LA (Terrill, 2025).
- **High insured losses.** Industry estimates have placed insured losses in the range of to US$20-75 billion (Li and Yu, 2025; Morningstar DBRS, 2025; Insurance Insider, 2025), placing substantial additional stress on the already volatile home insurance market in California and on most global reinsurers.
- **Housing and affordability crisis.** Thousands of affordable housing units were destroyed, worsening Southern California's housing shortage, displacing large numbers of lower-income residents, and exacerbating the problem of homelessness in the region (Mattson-Teig, 2025; Li and Yu, 2025; Booth, 2025). This triggered a ripple mass displacement into both surrounding communities and beyond in the months following the fires (NYT, 2025).





● **Debris flows.** The geology of southern California is highly conducive to erosion and
debris flows after wildfires. Several debris flows following high-intensity rainfall events
in the weeks after the fire produced further damage and required hundreds of
additional evacuations in and near the affected areas (USGS, 2025a).

The fires in Southern California have already been subject to several detailed investigations,
which found that the fires were driven by exceptionally late onset of winter rains that
extended the fire season into January, unseasonably warm winter temperatures, fuel buildup
from very wet conditions in the prior year to two, and powerful Santa Ana winds exceeding
130 km/h, creating extreme fire weather conditions that propelled fires to progress downhill
from wildlands into the built environment and become an urban conflagration (Barnes et al.,
2025; Garrett, 2025). The potential for extreme wildfires to develop under dry downslope
winds was predicted several days in advance, including by the National Interagency Fire
Center (NIFC), the National Weather Service (NWS), and the Storm Prediction Center (SPC;
see summary by Wikipedia, 2025) as well as by specialist commentators (e.g. Swain, 2025).

### *2.2.2.4. Congo Basin (July-August 2024)*

The Congo Basin region here refers to the moist tropical forest ecoregions of equatorial
Africa (**Figure 5**). **Section 2.2.1** discusses the regional anomalies that led this region to be
identified (e.g. **Figure 2**, **Figure 3**), with further review of the fire season provided by our
expert panel in **Appendix A** (**Section A6**). It emerges as a major event of global relevance
for the following reasons:
● **Record-breaking burned area:** Highest-ranked BA on record at 28% above the
annual mean due to there being 4,000 (20%) more fires than in the average year.
There were 7 continuous months with BA above the climatological mean. The largest
fire anomalies were observed during July and August (**Figure S8**), especially in
southern Democratic Republic of the Congo, northern Angola, and parts of the
Republic of the Congo.
● **Unprecedented role of fire in primary forest loss:** Forest loss statistics from the
recent Global Forest Watch report (Goldman et al., 2025) showed that wildfires were
the dominant driver of a more than doubling (+150%) of rates of forest loss in the
Republic of the Congo and the Democratic Republic of the Congo during 2024
versus 2023, representing the highest rates of primary forest loss since 2015.
● **Sparse reporting and poor media coverage:** Reporting on the occurrence, drivers,
and consequences of fire is extremely sparse in this region, including by government
agencies and the international and national news media. This demonstrates that
extreme fire events in this region are often overlooked, making it an intriguing case
study to investigate in this report.

## 1207 3. Impact Assessments

In this edition of the report, we introduce new routine regional assessments of fire impacts
on society in terms of population exposure to fire, physical asset exposure to fire, the
exposure of carbon projects to fire, and the degradation of air quality through emissions of
fine particulate matter ($PM_{2.5}$). For our air quality analysis, estimates are generated for the
focal events only (**Section 2.2.2**). In all other cases, estimates are provided for each of the
regional layers detailed in **Table 1**, mirroring our approach to providing regional summaries
of BA, C emissions, and individual fire properties (**Section 2.1.2**).



### 3.1. Methods

### 3.1.1. Population Exposure Assessment

Population exposure estimates are produced using the global risk assessment platform CLIMADA (Aznar-Siguan and Bresch 2019). CLIMADA has previously been validated and applied to systematically quantify exposed population to a variety of natural hazards globally, such as river floods (Kam et al. 2021) and tropical cyclones (Stalhandske et al., 2024; Kam et al. 2024). The BA hazard set is set up using the MCD64A1 MODIS BA product (Giglio et al. 2018). The original BA data are aggregated monthly on a regular grid with a resolution of 150 arcsec and expressed as the fraction of total cell area burned. For the spatial distribution of exposed population, we use Gridded Population of the World (Doxsey-Whitfield et al., 2015), which is spatially reaggregated on the same grid as the hazard using the LitPop exposure layer (Eberenz et al. 2020). The population exposed to wildfires is estimated by multiplying the BA fraction (BA expressed as a fraction of burnable area) of each cell by the population present in each grid cell. As a complementary approximation to the main analysis, a single displacement share is derived by comparing population exposure estimates with reported displacement figures from the Internal Displacement Monitoring Center (IDMC, 2025), acknowledging that exposure only partially translates into impact. Event records are matched to BA observations following the methodology described in Riedel et al. (2025). We compute the ratio between recorded impacts and exposed values for each event and provide the median of these damage ratios across events.

The data produced using these methods are available from Steinmann et al. (2025).

### 3.1.2. Physical Asset Exposure Assessment

Physical asset exposure estimates are produced using the global risk assessment platform CLIMADA (Aznar-Siguan and Bresch 2019). CLIMADA has previously been validated and applied to systematically quantify economic impacts resulting from exposure of physical assets to a variety of natural hazards globally (Stalhandske et al. 2024), including fires (Lüthi et al. 2021). The exposure layer LitPop (Eberenz et al. 2020) was used to spatially distribute national-scale macroeconomic indicators as a function of night light intensity (Román et al. 2018) and population density (Doxsey-Whitfield et al., 2015) within national geographical domains. We disaggregate country-based produced capital estimates (World Bank, 2024c) for the year 2018 to approximate physical asset density in US dollars (US$). Physical asset exposure to wildfires is estimated by multiplying the BA fraction of each cell by the physical asset totals present in each grid cell (analogous to our analysis of population exposure, **Section 3.1.1**). In addition to this analysis, a single overall loss fraction is provided recognising that exposure tends to overstate actual asset damage. This fraction is derived by comparing modelled exposure estimates with asset damages from wildfire events, as reported in the Emergency Events Database (EM-DAT; Delforge et al. 2025) maintained by the Centre for Research on the Epidemiology of Disasters (CRED). Event records are matched to BA observations following the methodology described in Riedel et al. (2025). We compute the ratio between recorded impacts and exposed values for each event and provide the median of these damage ratios across events.

The data produced using these methods are available from Steinmann et al. (2025).

### 3.1.3. Carbon Projects Exposure

We estimated the exposure of carbon offset projects to fire by combining a large set (n=927) of project boundaries for forestry projects in Latin America (n=394), northern America (n=316), Eurasia (n=150), Africa (n=60), and Australasia (7) with information on fire and





climate. Project boundaries were sourced from BeZero Carbon Ltd., who have collated and digitised boundaries for all nature-based projects in the Voluntary Carbon Market (VCM). Information on annual BA was derived from the MCD64A1 collection 6.1 data (Giglio et al., 2018) and this was combined with information on land cover from MCD12Q1 collection 6.1 (Sulla-Menashe et al., 2019) to separate forest from non-forest fires. To evaluate drought conditions, we calculated the 12-month Standardized Precipitation Evapotranspiration Index (SPEI) using data from ERA5-Land (Muñoz-Sabater et al., 2021) calibrated over the 1980-2014 period.

We evaluated fire activity during the 2024 calendar year in the context of long-term trends in drought and fire risk. First, to assess how 2024 compared to previous years since 2001, we calculated the number of carbon projects affected by fire in each year and the average percentage of project area burned per year (%). Second, to place this in the context of climate change, we calculated the 2024 drought anomaly as the 2024 SPEI minus the long-term average SPEI (1980-2023).

### 3.1.4. Air Quality Impact Assessment

The human health risks associated with fire smoke pollution are well established. Smoke contains a toxic mix of gases, including ozone and carbon monoxide, as well as fine particulate matter ($PM_{2.5}$) that can carry heavy metals and environmentally persistent free radicals (Hamilton et al., 2021; Andreae, 2019; Fang et al., 2023). Even short-term exposure to these pollutants has been associated with increased risk of cardiovascular and respiratory illnesses, including asthma exacerbation, reduced lung function, and acute infections (Johnston et al., 2021; Xu et al., 2024; Chen et al., 2021; Xu et al., 2023; Aguilera et al., 2021; Zhang et al., 2025). Furthermore, wildfire smoke contributes to increased mortality, particularly among vulnerable populations. In addition to these physiological effects, heavy smoke can significantly reduce visibility, compounding health risks by increasing the likelihood of injuries during regular driving, evacuation, or emergency response (Gill and Britz-McKibbin, 2020), and generates lasting mental health effects amongst exposed or displaced communities (Humphreys et al., 2022).

To quantify the contribution of fires to degraded air quality we used the global model framework utilised by the Copernicus Atmosphere Monitoring Service (CAMS) to simulate concentrations of fine (<2.5 μm diameter) particulate matter ($PM_{2.5}$; Peuch et al., 2022). One of the key objectives of CAMS is to monitor and forecast global atmospheric composition including smoke from vegetation fires. Fires in CAMS are prescribed by the Global Fire Assimilation System (GFAS; Kaiser et al., 2012), which calculates hourly estimates of biomass burning emissions by assimilating fire radiative power (FRP) observations from satellite-based sensors and by means of land cover-dependent conversion (FRP to dry matter) and emission factors (dry matter to emitted gas or aerosol species per biome) describing the rate at which about 40 smoke constituents are released into the atmosphere. This study uses GFAS version 1.4, which is the version used currently for the NRT production input of CAMS global and regional forecast services, plus some improvements that include the use of VIIRS FRP retrievals. Spurious FRP observations of no vegetation fire origin are filtered out in GFAS with a static map. GFAS ingests active fire information together with a characterization of its uncertainty, including an uncertainty component related to the satellite sensor detection limit and a solution for partial observational cloud coverage.

Simulations are run with the Integrated Forecasting System extended with modules of atmospheric composition (IFS-COMPO), which describe source, sink, and transport processes of the main reactive trace gases (Flemming et al., 2015; Huijnen et al., 2016) and aerosol species (Morcrette et al., 2009; Remy et al., 2022, 2024) and which, together with satellite observations, is at the core of the CAMS system for the global domain. Mass fluxes of atmospheric constituents from the surface into the atmosphere are either prescribed from





CAMS pre-compiled emissions inventories, with some aspects of on-line simulated temporal variability, as in the case of pollutants from the burning of fossil fuels for transportation and electricity, or estimated online at every time step in the IFS when strongly dependent on meteorological conditions as in the case of desert dust and sea salt aerosol and of biogenic fluxes of $CO_2$. The resolution used is the current operational resolution of 40 km, with 137 vertical levels. GFAS biomass burning emissions are estimated at 0.1° resolution based on FRP observations from the MODIS sensor on both the Terra and Aqua satellites (Giglio et al., 2016) and from the VIIRS sensor on the Suomi NPP satellite (Csiszar et al., 2014). The vertical distribution of fire emissions within the simulation follows the GFAS IS4FIRES injection height estimation (Sofiev et al., 2012; Remy et al., 2017).

To isolate the contribution of extreme fire events to atmospheric $PM_{2.5}$ concentrations, two sets of forecast experiments are run for specific focal events using a similar assessment framework. In the first ("with local fires"), all emission sources of $PM_{2.5}$ were considered including those of anthropogenic, dust, biogenic and other natural origin. In the second ("no local fires"), biomass burning emissions from within the focal event are excluded. The difference in simulated $PM_{2.5}$ concentrations between the two runs then represents the fire contribution to $PM_{2.5}$ within the region.

After $PM_{2.5}$ concentrations had been simulated at a 3-hourly temporal, and 40 km spatial, resolution, we summarised the influence of fires in the region to a daily population-weighted mean $PM_{2.5}$ concentration at ground-level (in units of $\mu g/m^3$) for each focal region. Population data for the year 2020 from the Gridded Population of the World (GPW) dataset version 4 (Doxsey-Whitfield et al., 2015) were used to weight the values of $PM_{2.5}$ concentration in each grid cell of the focal regions, producing a weighted mean value for $PM_{2.5}$ concentration for each simulated date. This process was repeated for each simulation ("with local fires" and "no local fires"), and daily differences between the simulations were used to assess the additional number of days with poor air quality caused by fires in the focal regions.

To illustrate the scale and intensity of wildfire smoke health-relevant exposure within the 2024-2025 fire season, total population-weighted $PM_{2.5}$ and the isolated contribution of fires to population-weighted $PM_{2.5}$ in a focal region are compared against the World Health Organisation 24-hour mean (15 $\mu g/m^3$) standard for daily $PM_{2.5}$ exposure (WHO, 2021).

## 3.2. Results

### 3.2.1. Population Exposure

During the 2024-25 fire season, we estimate approximately 100 million people to have been exposed to wildfires worldwide. Exposure was most pronounced across South and Southeast Asia, as well as Central and East Africa. At the country level, India and the Democratic Republic of the Congo show the highest numbers, with around 15 million people affected in each **(Figure 6; Figure S9)**. Nigeria, China, Mozambique, and South Sudan also were also exposed substantially, each with more than 5 million people affected. At the subnational level, we estimate the highest population exposures in Uttar Pradesh State (India) with over 4.6 million people, Heilongjiang Province (China) with 3.7 million, and Punjab State (India) with 3.6 million exposed **(Figure 6; Figure S9)**. Several provinces in the Democratic Republic of the Congo also exceed 2 million, illustrating how national-level exposure is often driven by a few highly affected administrative regions.

Some of the countries with the most extreme anomalies in fire BA and C emissions, most notably Bolivia, Brazil, and Canada, accounted for only a small share of absolute global population exposure, and showed negative (Canada) to modest positive (e.g. Bolivia and Brazil) anomalies **(Figure 6; Figure S9)**. This decoupling highlights the relevance of the



spatial distributions of both BA and population to population exposure, which might be low when extensive fires occur in remote places.

Several of the countries with the highest absolute exposures, such as in India and the Democratic Republic of the Congo, showed negative anomalies on a national level, related to the fact that fire-related population exposure in these regions is more recurrent. Nonetheless, on a subnational level, some regions of these countries show considerable positive anomalies, such as in India's Uttar Pradesh State where 4.6 million people were exposed (146% above average) and about half a million being exposed in the provinces Kasaï-Central (+33%) and Kongo-Central (+27%) in the DRC.

Population exposure anomalies were also high in relative terms across parts of the Middle East and the Balkans (e.g, Jordan, Iran, Iraq, North Macedonia, Albania), the Andes region (e.g., Peru, Ecuador), the Northern coast of South America (Venezuela, Guyana, Suriname; broadly encompassing our focal region of Northeast Amazonia), and Central Sahel (e.g., Niger), as well as isolated cases such as Nepal and Iceland. For example, Jordan shows divergent anomalies in population exposure (+201%) resulting from large subnational regions of Balqa (+322%) and Irbid (+393%). These patterns of exposure mostly align with patterns in BA and carbon emissions **(Section 2)**, in the Middle East and the the Balkans, Andes, Northern coast of South America and Central Sahel. Although the absolute number of people affected in some of these countries remains low, the relative anomaly marks a sharp departure from historical patterns.

It is important to distinguish between the exposed and affected population. Based on 521 events in the years 2008-2025 recorded by IDMC (2025), we estimate the damage ratio of exposed to displaced population to amount to 3.0%. While nearly 100 million people were exposed to wildfire activity in the 2024-25 season, only a small fraction - 20,046 people (IDMC, 2025) - were formally displaced (0.02%). Note, however, that this figure likely understates the true scale of disruption, as displacement records are incomplete. Many affected individuals may not be forced to leave their homes but still experience substantial short- and long-term consequences, including health burdens (Gould et al., 2024) and financial distress such as short-term earning disruptions (Borgschulte et al., 2024), increased missed mortgage payments (Ho et al., 2023), declines in property values (Huang and Skidmore, 2024), and lasting reductions in income later in life (Meier et al., 2025). Moreover, recent cases have emphasised that the number of people impacted by wildfire smoke can be many times higher than the number of people directly exposed to fire (Jones et al., 2024b; Kolden et al., 2024, 2025; Johnston et al., 2021). As such, these records should be viewed as a conservative lower bound on the broader human impacts of wildfire exposure.

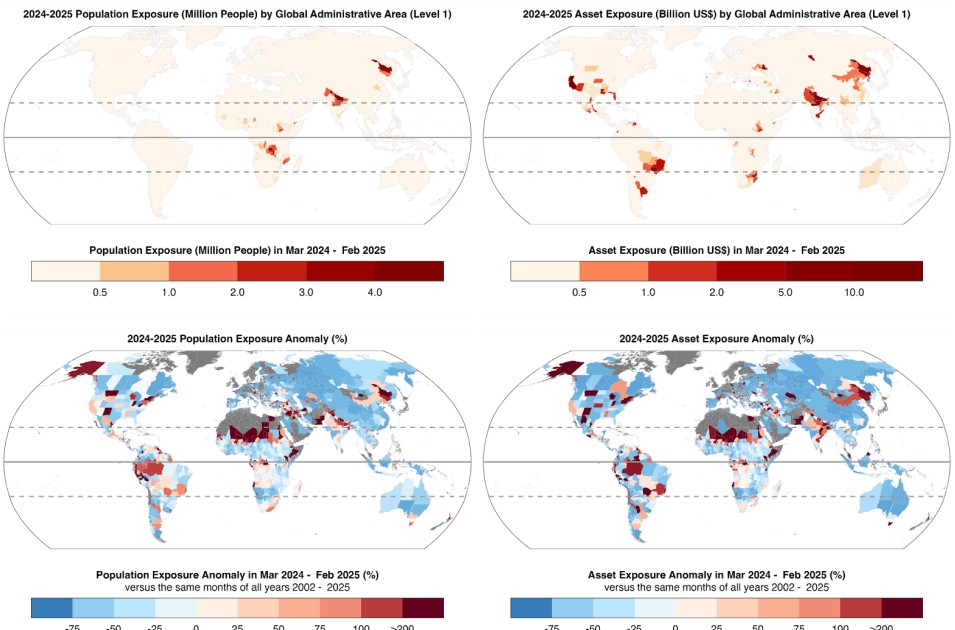

**Figure 6: (left panels)** Population and **(right panels)** physical assets exposed to burned
area (BA) during the 2024-25 global fire season. The figure shows **(top panels)** the number
of people or the asset value (billion US$) exposed to fire and **(bottom panels)** the relative
anomaly versus all years since 2002. Results are shown at the national scale in **Figure S9**.

### 3.2.2.    Physical Asset Exposure

We estimate that physical assets exposed to wildfires during the 2024-25 season amounted
to US$215 billion worldwide. The highest asset exposures were concentrated in a mix of
middle- and high-income countries, led by India (US$44 billion), the United States (US$26
billion) and China (US$17 billion), followed by Venezuela, South Africa, and Brazil **(Figure 6;
Figure S9)**. While India, and to a lesser extent Brazil and China, ranked highly in both
population and asset exposure, the asset exposure landscape broadens to include
developed countries such as the United States and South Africa (US$14 billion). This
divergence not only reveals different spatial patterns of wealth and infrastructure but also the
concentration of high-value assets in certain subnation regions **(Figure 6; Figure S9)**. For
instance, South Africa's Gauteng province, its economic hub, ranked among the most
exposed globally at US$8 billion, despite the country's moderate population exposure.
Similarly, in the United States, California alone accounted for over US$17 billion in exposed
assets, driven largely by the severe January 2025 wildfires (US$14 billion) discussed in
**Section 2.2.3**. These estimates are still low in comparison to damage records provided by
EM-DAT for the LA fires (US$52.5 billion). This difference is likely caused by an
underestimation of the affected exposure, which consisted of exceptionally high-value
structures not represented by LitPop. This also explains the underestimation of the asset
exposure anomaly in California, which is less pronounced (+60%) than in other states and
regions of the world (**Figure 6**).

In contrast, Central African countries such as the Democratic Republic of the Congo,
Nigeria, and Mozambique, which featured prominently in population exposure, did not rank
highly in terms of exposed assets **(Figure 6; Figure S9)**. The exception is South Sudan



(US$4 billion), where asset exposure remains substantial. The data also highlights high
absolute asset exposure in Mexico, Turkey, and the Russian Federation, each with national
totals around US$8 billion. At the subnational level, exposure was concentrated in
economically important regions, including Izmir in Turkey (US$3 billion), Mexico City (Distrito
Federal; US$3 billion), and Russia's Kemerovo and Rostov regions (approximately US$3
billion each). These patterns underscore how wildfire-related asset exposure is shaped by
the intersection of fire occurrence with concentrated infrastructure and economic activity.
Asset exposure anomalies for the 2024-25 fire season, expressed relative to the same
months of all previous fire seasons from 2002-2024 (n = 23), reveal several hotspots with
unusually high physical asset exposure. Notable positive national-level anomalies were
concentrated across the Middle East (e.g., Iraq, Syria), Southeast Europe and the Balkans
(e.g., Albania, Bosnia and Herzegovina, Greece), parts of the Sahel and Horn of Africa (e.g.,
Niger, Eritrea), and the northern tropical of South America (e.g., Ecuador, Colombia,
Guyana) **(Figure 6; Figure S9)**. At the subnational level **(Figure 6; Figure S9)**, pronounced
relative anomalies were observed in regions not necessarily among the highest in absolute
asset exposure. For example, many of the strongest asset exposure anomalies were highly
localised, including regions of Chad, Sudan, Brazil, and Pakistan, where this season's
values sharply deviate from past levels **(Figure 6; Figure S9)**. In contrast, while California
recorded the highest total asset exposure, its relative anomaly was modest, reflecting its
regular exposure to fire. These spatial contrasts underscore that extreme fire seasons can
affect both high-value regions and those with historically lower risk.
A comparison between asset exposure anomalies and BA anomalies **(Figure 6)** shows
areas of both alignment and divergence. Overlaps are evident in Venezuela, western Brazil,
Niger, and parts of India and Bolivia, where elevated fire activity coincided with high asset
exposure. In contrast, strong BA anomalies in parts of equatorial Africa and Russia were not
matched by anomalous asset exposure. This disconnect underscores that fire activity alone
is not a sufficient proxy for physical asset impact. Rather, extensive burns in remote or
forested areas may have limited consequences for built infrastructure, whereas smaller fires
near wildland-urban interfaces can generate disproportionately high asset exposure (Calkin
et al., 2023).
As with population exposure, asset exposure does not equate to realised impact. Comparing
modelled exposed assets with reported EM-DAT figures, economic losses from 105 historic
wildfire events in the time period 2002-2025 show a damage ratio of around 29% of exposed
asset value. While a modelled US$215 billion in physical assets were exposed to wildfires in
2024-25, a smaller sum of US$57 billion in realised damages was recorded by EM-DAT, or
around one-quarter of our exposure estimate. Note, that these figures reflect differences in
scope and data quality. EM-DAT's total economic damage records may include indirect
losses, such as business interruption and sectoral impacts. Its definition is broad,
source-dependent, and rarely disaggregated. Thus, reporting is uneven and regionally
biased due to variation in local capacity and data availability (Mazhin et al., 2021, Jones et
al., 2023). In contrast, our modelled asset exposure offers a consistent estimate of physical
assets at risk, representing the maximum potential asset loss. Yet, it does not represent
realised or total economic damage. While both measures have limitations, together they help
to characterise the scale of global wildfire-related economic impacts.

### 3.2.3.    Carbon Projects Exposure

Forestry projects can provide cost effective climate mitigation and co-benefits to society and
biodiversity, though their outcomes depend on complex interactions between project
activities and their local ecological and social context (Holl and Brancalion, 2020). Wildfires
present a growing threat to forest carbon offset projects, posing risks to the permanence of
stored carbon (Anderegg et al., 2020) and thus credit integrity (Badgley et al., 2022) and the



financial viability of project activities (Conte and Kotchen 2010, Michaelowa et al., 2021).
Forestry projects can focus on emissions avoidance (e.g. REDD+), emissions removal (e.g.
afforestation or forest restoration), or a combination (e.g. improved forest management).
Here we evaluate fire activity during the 2024 calendar year across an unprecedented
number of forestry projects in the Voluntary Carbon Market (VCM), and place results in the
context of long term trends in fire risk.
The 2024 fire season was characterized by anomalously high fire activity across the 927
projects evaluated. In total 169, or 18% of projects recorded BA in 2024, a record over the
observational period (2001-2024) (Figure S10 (a)). This coincided with record annual BA
with 1.6% of project areas affected on average (Figure S10 (b)). Regional drought extremes
were likely responsible for the observed uptick in fire activity during 2024, with drought
conditions in 72% of projects exceeding the long-term (1980-2023) average and, in 13% of
projects, exceeding extreme (SPEI < -2) drought conditions (Figure S10 (c)).
Interestingly, observed anomalies vary regionally and further depend on project activities.
Exceptional drought conditions in Latin America resulted in a record number of projects
being affected by fire but total BA was just short of previous peak years. In this region, many
projects focus on the avoidance of deforestation (38%), and in addition to climate, fire risk is
driven by changing land cover and land use over time (Alencar et al., 2015). In comparison,
in northern America a smaller number of projects are prone to fire annually and the majority
(93%) of projects focus on improved forest management. Here, a record average burned
area was observed, but the total number of projects affected was modest and aligned with
average drought conditions. Africa had the highest average BA but 2024 was a low fire year,
aligned with long term BA trends in African savannas and woodlands (Andela et al., 2014),
and a relatively large number of projects focused on afforestation or forest restoration (52%),
which may result in decreasing fire activity over time.
Notably, despite increasing fire risk, about 46% of projects did not experience any BA over
the observational period, and 67% of projects were at moderately low risk from fire (with less
than 0.5% burned annually in the forests within a 50-km buffer zone around the project).
Aligned with long-term changes in fire weather (Jolly et al., 2015, Abatzoglou et al., 2019),
we found that the majority of forest carbon projects faced anomalous drought conditions in
2024. The 2024 fire season affected a record number of forest carbon projects globally,
resulting in an unprecedented annual percentage of BA within project boundaries.
High-integrity forest carbon projects can help to mitigate global climate change, and we find
some evidence that these interventions are also reducing fire risk locally. Nonetheless, the
quality of carbon credits issued by nature-based projects depends on the permanence of the
carbon emissions avoided or removed, which we show to be increasingly at risk.
### 3.2.4.    Air Quality Impact
Here, we present estimates of the concentration of fine particulate matter ($PM_{2.5}$) that the
average person in the Pantanal-Chiquitano region experienced due to wildfire smoke
emissions (the population-weighted $PM_{2.5}$ concentration; **Figure 7**). In the
Pantanal-Chiquitano, the population-weighted $PM_{2.5}$ concentration exceeded the WHO daily
$PM_{2.5}$ daily standard of 15 µg/m³ on most days from August to November (**Figure 7**), with
only 30 days between July and October falling below the threshold, most of which were in
early July. Considering fire emissions alone, the average person experienced $PM_{2.5}$ above
15 µg/m³ on 16 additional days between July to October due to local fire emissions, which is
slightly lower (20%) than previous 30% estimate of the contribution of Brazilian deforestation
fires to $PM_{2.5}$ (Reddington et al. 2015). September marked the peak pollution month where
the average person experienced $PM_{2.5}$ concentrations of 61 µg/m³ and fires accounted for
approximately 59% of the pollution mass (~36 µg/m³). In comparison to **Figure 7**,



non-population weighted daily concentrations met or exceeded the US Environmental Protection Agency's 24-hour maximum standard of 150 µg/m³ on five days. Though no comparable single-day maxima standard exists under WHO or Brazilian air quality regulations, this highlights the potential of extreme pollution exposure in low population regions closer downwind of South American fire occurrence. Furthermore, even in the absence of fires, background pollution levels are already severely degraded; the presence of fire emissions, however, significantly worsens air quality conditions. Furthermore, this analysis has focused only on the impact of local fires, yet the overall seasonality of PM matches the fire season in South America. This suggests that while local fires are enhancing exposure to pollution there is likely to be a significant contribution from longer-range fire smoke transport to the region.

To help contextualize model findings, we also examined model results for the January 2025 Los Angeles (LA) wildfire (not shown). The modelled $PM_{2.5}$ results for the LA region were muted, with a maximum population-weighted daily concentration of 29 µg/m³ on January 17. However, observational reports of the LA fire document much more extreme pollution, including a 480 µg/m³ one hour peak and a 93 µg/m³ daily mean peak on January 8th (US EPA, 2025; Briscoe and Rainey, 2025). This discrepancy likely stems from insufficient spatial and temporal resolution in both the model and the analysis region, which cannot capture the rapid and highly localized plume behaviour typical of urban or wildland-urban interface fires. It illustrates why high-resolution modelling that captures community scale air quality analysis of short-lived extreme events are needed for comprehensive impact assessments of fires as they encroach into populated regions. Benchmarking model performance against documented local maxima could guide improvements and enhance reliability for future health risk evaluations in all burning environments.

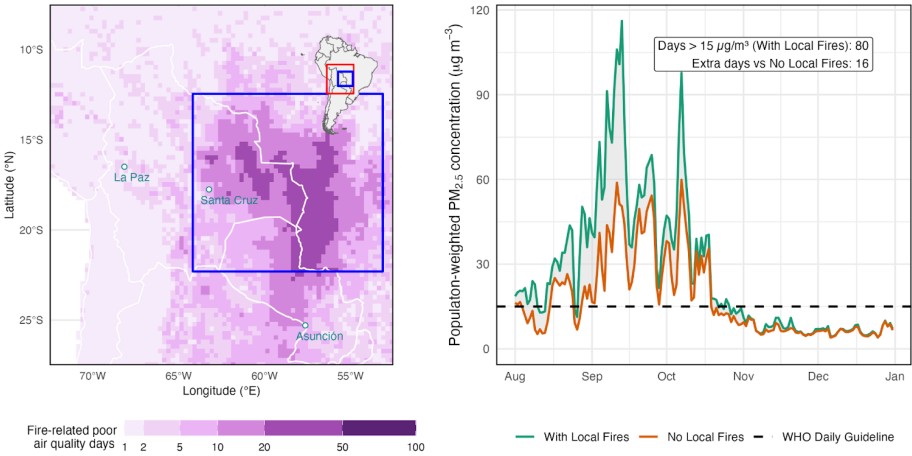

**Figure 7:** Poor air quality days caused by anomalous fire activity in the Pantanal-Chiquitano region during the 2024-25 fire season. **(Left panel)** shows the additional number of days exceeding the World Health Organisation (WHO) daily standard of 15 µg/m³ as a result of fire emissions occurring within the defined regions (red outlines), over and above the number of poor air quality days caused by all other sources of $PM_{2.5}$ (e.g. industrial, transport, and residential) and from fires occurring outside of the defined regions. **(Right panel)** shows a daily time series of population-weighted $PM_{2.5}$ concentrations (µg/m³) under scenarios that include or exclude local fires. The WHO daily standard of 15 µg/m³ is shown and days exceeding that threshold are counted as poor air quality days.





## 4.    Diagnosing Causes and Assessing Predictability

### 4.1.    Methods

#### 4.1.1.    Predictability of Focal Events of the 2024-25 Fire Season

##### *4.1.1.1.    Short to Medium Range Forecasts*

We evaluated the capacity of two distinct methods to predict fire occurrence over short to medium -range time periods (1 to 15 days): the Fire Weather Index (FWI; van Wagner, 1987; Vitolo et al., 2020) and the Probability of Fire (PoF; McNorton et al., 2024).

FWI is a well-established empirical indicator of fire danger mostly reflecting the influence of meteorological conditions on landscape flammability. It is based on 4 prerequisite weather variables and it describes the impact that atmospheric conditions have on fuel dryness (see **Section 2.1.4.1**). It was originally calibrated for the boreal forests of Canada and assumes constant fuel characteristics. Due to its ease of implementation it is now widely used around the world (Bedia et al., 2015; Di Giuseppe et al., 2016; Abatzoglou et al., 2018; Vitolo et al., 2020; Jones et al., 2022). Here, we used weather inputs from the ECMWF Integrated Forecasting system in its operational configurations at 9 km resolutions.

PoF is one of the outputs of the ECMWF Sparky fire model and aims to improve upon the fire forecasting skill of the FWI (McNorton et al., 2024; Di Giuseppe et al., 2025). The Sparky-PoF is a data-driven fire prediction system that advances on fire danger metrics by modelling not only the effect of meteorological variables on fire likelihood but also (i) the temporal evolution of fuel load and fuel moisture content and (ii) ignition events informed by lightning forecasts, human population density, and road networks. PoF is an example of a new generation of indicators based on machine learning methods that have recently been created to produce more informative operational predictions of wildfire (Shmuel et al., 2025; Di Giuseppe, 2023). One of the practical advantages of PoF is that it can directly output a prediction of the number of fire hotspots when averaged over vast areas which is directly comparable to active fire observations. While these approaches are relatively new, they hold great promise for improving fire forecasting, particularly in fuel-limited biomes where FWI is a weaker predictor of fire activity (Bedia et al., 2015; Jones et al. 2022). PoF leverages medium-range (up to 15 days horizon) weather forecasts and fuel variables that are available from an experimental configuration where IFS is coupled with the Sparky fire model to drive a data-driven classifier, trained on observed hotspots using a XGBoost methodology (Shmuel and Heifetz, 2025; Jain et al., 2020). Predictions of PoF from Sparky showed better skills than FWI in recent events and are available operationally with forecasts up to 10 days in advance (Di Giuseppe et al., 2025).

In general, FWI is effective at capturing the immediate emergence of fire-conducive weather conditions across much of the globe. However, it does not consider the fuel build-up and the state of vegetation in specific biomes other than boreal forests, which is often a critical factor in fire occurrence As a result, FWI-based systems may predict fire-prone conditions too far in advance of actual fire emergence, particularly in ecosystems where vegetation availability (i.e., fuel) governs ignition potential. In contrast, data-driven models like PoF, which incorporate information on both dead and live fuel moisture content are better able to reflect the delayed response of ecosystems to dry conditions. These models provide a more realistic representation of fire potential in fuel-limited landscapes or in regions where the hydroclimatic cascade delays fire onset. This is especially relevant for wetland biomes, which have been a key focus of analysis this year.



### 4.1.1.2. Subseasonal to Seasonal Forecast

#### 4.1.1.2.1. Fire weather

The prediction of fire weather over sub-seasonal to seasonal (up to 6 months ahead) is a relatively unexplored field of research (Roads et al., 2005). Until recently, only a few studies had specifically examined the prediction and predictability of fire weather-related quantities and their connection to actual fire activity globally (Di Giuseppe et al., 2024). Here, we evaluate the ability of cutting-edge seasonal prediction systems to predict anomalies in the FWI, using data available through the Copernicus Emergency Management Service which uses ECMWF's SEAS5 seasonal forecasts as forcing (Di Giuseppe et al., 2024). We probabilistically quantify the likelihood of FWI values exceeding the seasonal mean prediction time steps ranging from 1 to 3 months considering a climate that spans the period 1991-2016. These predictions are not designed to inform on the exact location of fire outbreaks, but rather to serve as an indicator of landscape preconditioning to burn. The predictions highlight regions where anomalous fire weather may emerge and thus merit closer monitoring, offering an early signal of where fires could become a concern.

On seasonal timescales, patterns of fire weather are significantly influenced by large-scale climate modes such as the El Niño-Southern Oscillation (ENSO) through variation in temperature and rainfall patterns across the tropics (Latif et al., 1998; Chen et al., 2017; Bedia et al., 2018). In some tropical countries, forecasts of ENSO have been used directly to predict risk of fire and to implement preemptive fire management actions including bans on fire (Pan et al., 2018). For example, major fire anomalies and regional haze events in southeast Asia are thought to have been avoided during the 2023-2024 El Niño, following the implementation of new predictive systems and policy interventions since earlier El Niño years (e.g. 2015) (World Resources Institute, 2016). The effect of other large-scale climate modes is also present in other world regions, such as in the case of the Indian Ocean Dipole (IOD) in the case of Australia (Harris and Lucas, 2019) and several Atlantic and Pacific oscillations in the case of Amazonia (Aragão et al., 2018). The ECMWF's SEAS5 forecasts have been shown to accurately predict the meteorological variability associated with ENSO and their effects on fire activity over timescales of 1 to 2 months ahead (Johnson et al., 2019; Di Giuseppe et al., 2024).

#### 4.1.1.2.2. Burned Area

While FWI forecasts can successfully identify regions with elevated fire danger aligning with observed BA anomalies, they tend to indicate broad areas at risk and lack the specificity needed to pinpoint where fires are most likely to occur. This reflects a key limitation: translating fire weather anomalies into accurate predictions of seasonal fire activity is not straightforward, as it requires incorporating additional drivers, namely fuel availability, ignition sources, and suppression capacity. Modeling the complex dynamics among fire and its bioclimatic and human drivers remains a challenge and is the focus of extensive research (e.g. Jones et al., 2022). Nevertheless, when considering forecasting ability in the long range and accuracy, climate remains the most reliable parameter among the drivers of fire activity. Accordingly, we examine the potential of machine learning techniques to forecast BA anomalies, which are being developed to provide targeted forecasts that guide the deployment and coordination of limited firefighting resources amidst increasingly synchronous wildfires (Torres-Vázquez et al., 2025a; Abatzoglou et al. 2021). We employ the model developed by Torres-Vázquez et al. (2025b), which is a hybrid approach combining dynamical seasonal drought forecasts with a statistical climate-fire model based on the Random Forest (RF) algorithm. This model leverages the Standardised Precipitation Index (SPI), aggregated over periods of 3, 6, or 12 months, to capture both antecedent and concurrent climatic conditions that influence fire activity. Calibrated with historical BA and SPI data, the RF model forecasts BA anomalies one month ahead of the fire season. The





system has shown promising predictive skill, successfully capturing BA anomalies across the
globe (Torres-Vázquez et al., 2025b).

### 4.1.1.3. *Uncertainties and forecast skills*

Uncertainty is a key factor in prediction and is likely to increase with forecast horizon. The
forecast uncertainty is provided as the spread across a set of ensemble simulations from
possible scenarios or by expressing the forecast as probability. Variability across the
ensemble of forecast realizations was previously estimated to be in the range of 10%-15%
for FWI (Vitolo et al., 2020), and in this study is reported as variance in the forecast values.
PoF is a measure that is probabilistic in nature and is reported as probability of occurrence.
For long-range predictions, uncertainty is also explicitly incorporated by expressing forecasts
in probabilistic terms, specifically as the probability of exceeding (or falling below) certain
thresholds, such as the upper and lower tercile.
The quality of fire forecasts is assessed by visually examining how well the forecasts capture
the likelihood of key focal fire events. This approach mirrors the way fire management
agencies typically interpret and use these indicators during the fire season. It is designed to
partially reflect the operational context in which such indices are applied. Similarly, the
seasonal predictions of FWI and the probability of above-median BA aim to demonstrate the
type of information currently available to support informed decision-making for resource
planning at extended lead times.

### 4.1.2. Identifying Causes of Focal Events

We assess the main or concurrent causes of the 2024-25 focal fire events using two
complementary modelling frameworks: the Probability of Fire as part of the Sparky modelling
complex (McNorton et al., 2024) and the ConFLAME attribution framework (Kelley et al.,
2021; Barbosa et al. 2025b). PoF is applied to satellite observations of active fires (Giglio et
al., 2018; regridded to 0.1°) and targets a prediction of absolute fire counts on daily
timescales. Meanwhile, ConFLAME is applied to satellite observations of BA from MODIS
(Giglio et al., 2018; regridded to 0.5°), enabling causality analysis of fire events to key
environmental and human-related causes. The ConFLAME analysis is performed on
absolute BA fraction and anomalies from the 2002-2025 climatological mean and includes
full regional summaries to provide broader context and to better support interpretation of
region-wide drivers and trends. Used together, as in this report, the two systems provide
complementary analyses of the causes of both active fire hotspots and BA anomalies.
Each model groups predictors into broader categories of causation: weather, fuel and
ignitions (**Table S1 in Supplementary Material S4**). Some predictors are shared or overlap
between categories due to their interconnected nature (e.g., fuel moisture and weather), but
the models are designed to avoid double-counting. To identify the main causes of the fire
event, PoF uses an ensemble-based gradient-boosted decision tree classifier (XGBoost),
with attribution provided through SHapley Additive exPlanations (SHAP) method taken from
the SHAP library (Lundberg and Lee, 2017) values to quantify the influence of each driver
group on predicted fire hotspots.
ConFLAME, in contrast, uses a probabilistic Bayesian approach to assess the contribution of
each driver group to observed BA, accounting for model uncertainty and fire stochasticity.
While PoF is trained globally, ConFLAME is trained separately for each region to capture
regional variation in the relationship between fire drivers and BA. Regional influence is
particularly relevant for explaining the final BA as it depends on the local variations of fuel
and ignition. Local ecology shapes how vegetation and biomass affect burning (Lehmann et
al. 2014) and human control can result in promoting fire (e.g. through deforestation or water
extraction) or suppressing it (Andela et al. 2017). In ConFLAME causes are combined



through logistic functions, with results expressed in terms of likelihoods for a detectable BA to be associated to a specific cause.

Both systems include uncertainty estimates. PoF reflects uncertainty via probabilistic ensemble outputs and a measure is provided by the error in the predicted number of hotspots. ConFLAME directly quantifies uncertainty from both drivers and model structure, providing confidence intervals for predictions. While neither system is free of limitations, this dual-model setup allows for a more robust assessment of fire causes across different spatial and temporal scales, with prediction of hotspots providing a fine-scale measure of fire activity and BA an integrated assessment of landscape impacts.

The PoF model does not assume that each factor always pushes fire activity in the same direction. For example, while increased fuel moisture generally reduces fire activity by dampening ignition and spread, in some regions, higher antecedent rainfall can lead to greater vegetation growth, increasing available fuel and potentially resulting in more intense fires later. In fuel-limited regions, where grasses and herbaceous plants dominate, high rainfall can boost fuel growth and lead to more burning. But in fuel-rich areas with lots of trees, that same rainfall mostly increases fuel moisture, potentially decreasing fire activity. In contrast, ConFLAME allows you to specify the expected direction of influence. When a factor can both increase or decrease fire activity depending on context, those effects are represented separately in the model. See **Supplementary Material for Section 4** for a detailed description.

## 4.2. Results

### 4.2.1. Predictability of Focal Events

#### 4.2.1.1. *Short to Medium Range Forecasts*

##### 4.2.1.1.1. *Northeast Amazonia*

Between January and March, satellites detected over 30,200 fire hotspots, marking the highest number recorded for that period since monitoring began in 1999 (Eschenbacher, 2024). These fires were intensified by persistent extreme drought conditions associated with the El Niño phenomenon, which led to higher temperatures and reduced rainfall (NASA Earth Observatory, 2024c; **Figure 8**). This part of the region lies in the Northern Hemisphere tropics, where the peak of the fire season aligns with boreal winter months. The region is lesser-studied than parts of southern hemisphere Amazonia (Brando et al., 2020; Alencar et al., 2015).

Both FWI and PoF systems identified two main fire seasons in 2024: February-April and August-November. However, around 80% of the total BA concentrated in the early months of the year and only 20% during the second dry season. The total probabilities of PoF values over the focal region translates into a number of predicted hotspots and this is directly comparable to the detections from MODIS. In March, when approximately 60% of the annual BA was recorded, the PoF reached its peak predicting nearly 700 fire hotspots in a single day, closely matching the ~600 observed hotspots. While the FWI also indicated anomalously high fire risk, its peak occurred later in September. The onset of the first fire wave aligned closely with the emergence of fire-prone weather conditions (**Figure 8**), highlighting the role of weather in enabling fire activity. In this region, where over 90% of ignitions are human-driven and fuel availability remains high, atmospheric conditions primarily act as the trigger for widespread burning.



Interestingly, both the PoF and FWI systems failed to capture a lull in fire activity during the second emergence in August-November of fire-conducive conditions showing the limitations of forecasting fire activity rather than fire danger. One possible explanation is that these conditions fell outside typical burning cycles, for example, in agricultural areas where fires are often timed around harvest. This raises an important possibility that the models failed to represent the quiet September period because they have only limited information on human ignition patterns, land ownership and use types, and lesser-studied factors such as fire suppression, policy interventions, and managed or cultural burning practices, underscoring the need for improved human activity data that could significantly improve fire prediction (Jones et al., 2022).

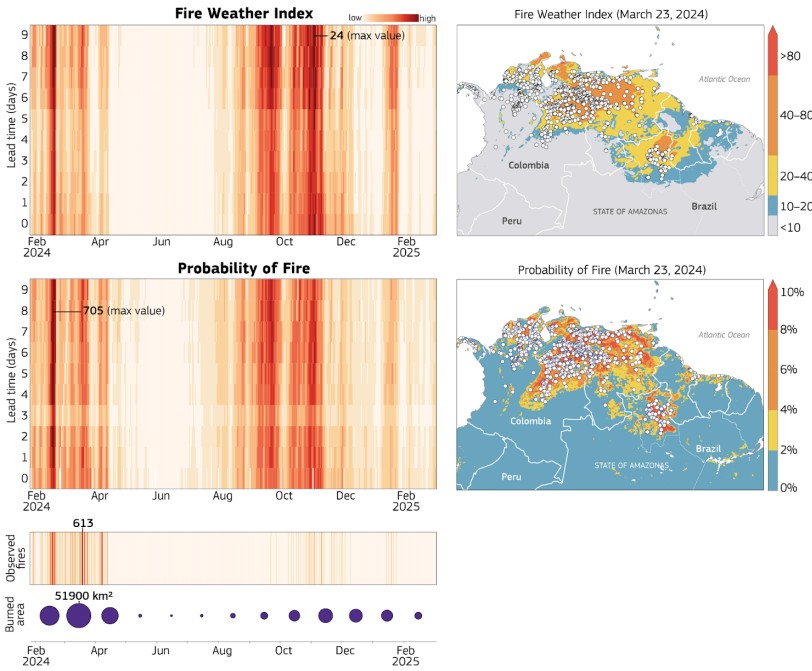

**Figure 8:** Northeast Amazonia forecasts of the FWI and PoF with lead times up to 10 days prior for the period February 2024-February 2025 as an average value over the focal area. The total percentage of PoF values over the focal region translates into a number of predicted hotspots and this is directly comparable to the detections observed by MODIS. The x-axis corresponds to specific dates throughout the year, while the y-axis denotes either observations or the time leading up to the date when a forecast was generated. The vertical colour coherence allows for quick identification of the time windows of predictability associated with the observed fire activity both provided in terms of number of detected active fires per day and total monthly BA (circles). The maps represent a snapshot in time at day 0 to allow the comparison of the spatial distribution of the forecasts and the recorded fire activity by MODIS.



### 4.2.1.1.2. *Pantanal and Chiquitano*

The Pantanal and Chiquitano have been enduring a prolonged dry period since 2019 leading to the 2024 worst water crisis ever recorded in the biome (World Wildlife Fund, 2024). Notably, the Pantanal did not experience its typical flood season in early 2024 and the average area covered by water during the first four months was smaller than that of the previous year's dry periods (Van Dijk et al., 2025). By the end of May 2024, almost the entire Pantanal and Chiquitano region was classified as experiencing extreme drought, the second-highest classification of drought intensity on the Integrated Drought Index (NASA Earth Observatory, 2025a, 2024c). As the Pantanal is a wetland ecosystem, the establishment of dry conditions is a prerequisite for the onset of fire activity. A full hydrological cascade must occur before widespread burning can take place: prolonged precipitation deficits must lead to the reduction of flooded areas, their replacement by grasslands, and the progressive desiccation of both live and dead vegetation. This sequence introduces a natural delay, which explains why fire activity in the region peaked in August and September, well after the onset of dry weather in June (**Figure S11** in **Supplementary Material S4**).

The total percentage of PoF values over the focal region translates into a number of predicted hotspots and this is directly comparable to the detections from MODIS. The most severe PoF forecast, predicting 971 hotspots, closely matches the 885 observed in late August. At large scales, the FWI offers a useful overview of fire-conducive weather conditions. However, it is the inclusion of fuel characteristics in the PoF that provides the finer spatial granularity (maps in **Figure S11** in **Supplementary Material S4**) needed for more accurate and actionable fire risk assessments.

### 4.2.1.1.3. *Southern California*

California is arguably one of the most extensively studied regions in terms of shifts in fire regimes (see, e.g., Billmire et al., 2014; Littell et al., 2016; Williams et al., 2019; Swain et al, 2025). In 2024, Southern California experienced severe burning in September, with a total of 1,200 km$^2$ burned. Although these fires fell within the typical fire season, the total BA was unremarkable for the region compared to previous years. However, the most significant fire event took place much later, in January 2025, well outside the typical fire period, when the Palisades and Eaton fires broke out in Los Angeles county. The events sparked widespread public debate about how prepared we are to anticipate off-season fires (Woolcott, 2025).

As shown in **Figure S12** (**Supplementary Material S4**), fire-prone weather conditions persist across much of the year, extending well into autumn, a reflection of the expanding fire season driven by climate warming. Yet, in regions like Southern California, fire prediction based solely on weather indicators is often inadequate. The primary causes of the severity of these events was an intensification of the hydrological cycle that exacerbated both wet and dry extremes. Southern California experienced an unusually wet antecedent period prior to intense drying in an unusually dry winter, which created an accumulation of dry fuel setting the ideal conditions for intense fire activity (Swain et al., 2025). Fuel accumulation is a persistent feature throughout the fire season, and therefore does not result in a large difference between the PoF and FWI forecasts when averaged over the Mediterranean areas of California. However, its inclusion in the prediction system allows for the identification of zones with higher susceptibility, which are clearly visible in the accompanying map. Neither the FWI nor PoF metrics could provide adequate warning regarding the magnitude of the winter fire event affecting the wildland-urban interface. These events were driven by atmospheric phenomena influenced by steep orography, which are not resolved by current weather forecasting models. The lack of the required resolution impacts equally on empirical and machine learning methods. This highlights the need for improved high resolution forecasting for fire danger in the wildland-urban interface





### 4.2.1.1.4. Congo Basin

The 2024 dry anomaly in Central Africa has been partly attributed to the co-occurrence of a positive El Niño phase and a warm Indian Ocean Dipole (McPhaden et al., 2024). These conditions tend to shift the West African monsoon northward, leading to suppressed precipitation over the Congo Basin during the core of the rainy season, a pattern observed globally in recent years (Toreti et al., 2024). This event also aligns with a broader trend of a lengthening and intensifying dry season in the Congo rainforest. Satellite analyses over the past few decades show that the dry season is starting earlier and ending later, increasing the region's vulnerability (Jiang, 2019). There are typically two main fire seasons in the Congo Basin: from December to March north of the equator, and from June to September south of the equator. In the equatorial zone, however, fires are not naturally occurring, as precipitation is distributed throughout the year. The expansion of dry seasons both north and south of the equator has led to a situation where fire seasons in the Congo Basin now span almost the entire year with peak activities between July and August and December and March. Compounding this, a decline in lightning activity over the region (Chakraborty and Menghal, 2025) suggests that fires are increasingly of human rather than natural origin. This combination of persistent drier-than-average conditions and human-driven ignition means that fire activity is now widespread and weakly correlated with weather patterns. As a result, predictions have a very short predictability window of only a few days (horizon at correlation of lines in **Figure S13**). The detachment of fire activity from natural conditions in the Congo Basin presents a significant challenge for forecasting (**Figure S13**). In these regions, the discriminatory power between fire-prone and non-prone conditions is greatly reduced, and both FWI and PoF tend to overpredict fire occurrence. In particular, FWI fails to capture the complex interactions among fuel availability, ignition sources, and human activity. This limitation is especially pronounced in areas where natural ignitions are infrequent, and fuel dynamics, rather than weather alone, drive fire occurrence and behaviour (**Figure S13**).

### 4.2.1.2. Seasonal Predictability from Fire Weather Forecasts

The year 2024 has been officially declared the warmest year on record, surpassing previous temperature benchmarks (WMO, 2025; NOAA, 2025a). This exceptional warmth has been driven not only by long-term global warming (IPCC, 2023), but also by a combination of short-term ocean-atmosphere anomalies. In particular, extensive and persistent oceanic heat waves have been observed across multiple ocean basins, contributing to elevated sea surface temperatures (Holbrook et al., 2019). These marine heatwaves have been further reinforced by an unusual reduction in low-level cloud cover over parts of the Atlantic Ocean, allowing for increased solar radiation absorption at the ocean surface and amplifying the warming (Ceppi and Nowack, 2021).

Given this overall picture, seasonal forecasts of FWI anomalies successfully captured the broad regional patterns of elevated fire danger, particularly in Northeast Amazonia and parts of Bolivia and Venezuela (**Figure 9**). These forecasts aligned with the widespread drought and above-average temperatures linked to the strongest El Niño since 2015, a concurrent positive Indian Ocean Dipole, and record-breaking ocean heatwaves. Together, these factors intensified drying across equatorial South America, expanding fire-prone conditions well beyond the regions that ultimately experienced the most extreme burning.

All forecasts issued one month before the fire season showed high confidence (between 60 and 90%) in the development of above-normal conditions in our focal regions, all exceeding the 66th percentile of climatological values.

**Figure 9** demonstrates both the strengths and limitations of FWI-based seasonal forecasts. While they provide valuable early warnings by detecting fire weather anomalies, their broad-scale nature can lead to overestimations of fire impact if not combined with





information on fire susceptibility. These findings reinforce the value of FWI in anticipating periods of increased landscape flammability, but also highlight the need to more appropriately model anomalies in fuel load and moisture and to integrate non-climatic factors, such as ignition sources, land use practices, suppression capacity, and landscape accessibility, into fire impact forecasting models to improve precision and operational relevance. Future seasonal-scale forecasts may seek to implement PoF as a predictive tool, which improves upon FWI by tracking fuel loads and moisture and thus the legacy effects of antecedent conditions on landscape flammability.

**Figure S14** presents an example of the burned-area anomaly forecasting system using our hybrid dynamical and Random Forest (RF) approach (**Section 4.1.1.2.2** and Torres-Vázquez et al., 2025b). The maps illustrate the predicted probability of a BA anomaly and whether these predictions could trigger alerts for BA anomalous  seasons within a potential early-warning system. Following Torres-Vázquez et al. (2025b), alerts are issued when predicted probabilities exceed thresholds optimized to balance correct detections and false alarms. For the 2024 season, anomalies in South America, notably in drought-affected regions influenced by El Niño conditions, were reasonably well anticipated. However, in other regions, particularly parts of Africa including the Congo basin, there were numerous false alarms, reflecting current limitations in fully capturing regional complexities and non-climatic fire drivers. This first implementation demonstrates operational potential, and future refinements (such as incorporating extended fire records and adjusting region-specific thresholds) could  enhance skills  by  reducing  false positives.


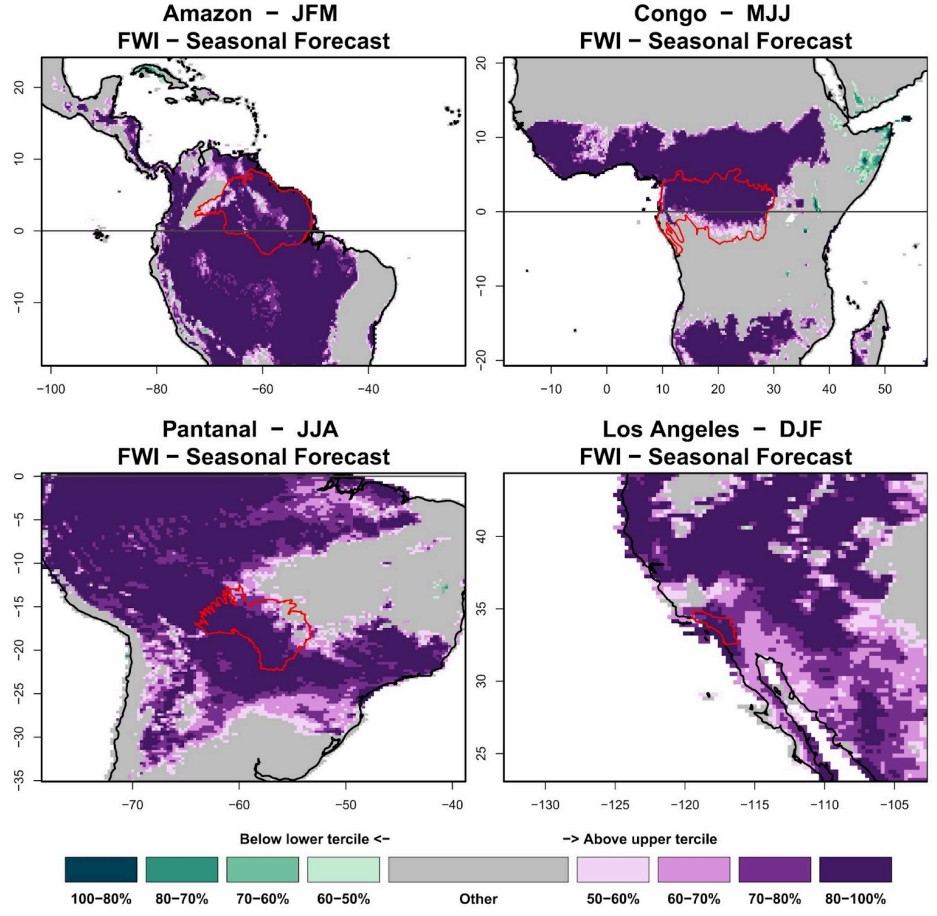

**Figure 9:** Seasonal prediction of Fire Weather Index (FWI) during the periods relevant to our focal events, presented in probabilistic terms that indicate the likelihood of an increased or decreased anomalous fire season.

### 4.2.2.    Identifying Causes of Focal Events

Weather, fuel, and ignitions are the three primary controls influencing the occurrence and intensity of fire events (Di Giuseppe el., 2025). These broad categories can be further examined to pinpoint individual factors, for example, precipitation and temperature within the weather category or fuel moisture from dead and live vegetation in the fuel category. Analysing the single factors can give an idea not only of the probability of the fire to occur but also on their intensity and behaviour. For example anomalies above the expected climate (here 2003-2023) in the moisture of dead fuel, due to its lower moisture content and higher combustibility, often plays a significant role in determining ignition potential. Low live fuel moisture increases vegetation flammability, thereby contributing significantly to greater fire severity and intensity.

Beyond this descriptive approach, the PoF and ConFLAME causality models enable a probabilistic attribution of fire occurrence to the three primary controls. These models





provide attribution even when no fire is recorded: a low probability across all controls reflects an accurate forecast of no fire, while a high probability without observed fire activity could point to successful suppression efforts, fire-prevention policies, or other unaccounted human factors not included in the model forecasts. The discrepancies between the model prediction and the observed fire activity (fire hotspots or BA anomalies) are included to provide a measure of the model uncertainties.

### 4.2.2.1.    Causes of Fire Hotspots during Focal Events

#### 4.2.2.1.1.    Northeast Amazonia

According to our Sparky-PoF analysis, the extreme fire activity during the 2024-25 fire season in Northeast Amazonia (described in **Section 2.2.2.1**), was predominantly driven by anomalous dry weather. Northeast Amazonia experienced an exceptionally severe fire season between January and April (**Figure 10**), driven by extreme drought which started in 2023, intensified by the combined effects of El Niño and the Atlantic Meridional Mode, which brought unusually high temperatures and suppressed rainfall. At the peak of the season, during the week of 20-26 March, nearly 2,000 fire hotspots were observed. Fires were fueled by prolonged and intense drying across the entire landscape, which made vegetation highly flammable and enabled rapid fire spread across large areas. On the most severe week of burns our causation analysis shows that weather conditions were the dominant factor, accounting for about 60% of fire activity, while fuel availability and ignition sources each contributed around 20%. During the first part of the year the exceptional dryness meant that soil humidity levels and moisture in both dead and live vegetation fell to among the driest 2% of historical conditions, while deep soil moisture dropped below 1%. The time series of lightning activity (**Figure 10**, bottom panel) further illustrates that ignitions in the region are predominantly human-driven. During the May-August period, lightning activity is high and is linked to storms and rainfall, which tend to suppress fire ignition and spread. As a result, even though lightning increases the relative contribution of ignition to predicted fire activity, doubling its weight to around 40%, this is not reflected in actual fire occurrence or BA. A second, less intense onset of fires occurred between September and January. This was driven by a more superficial drying of the landscape that did not extend into deeper soil layers. Unlike the earlier season, which was associated with hydrological drought, this later period was more reflective of meteorological drought (precipitation deficit).

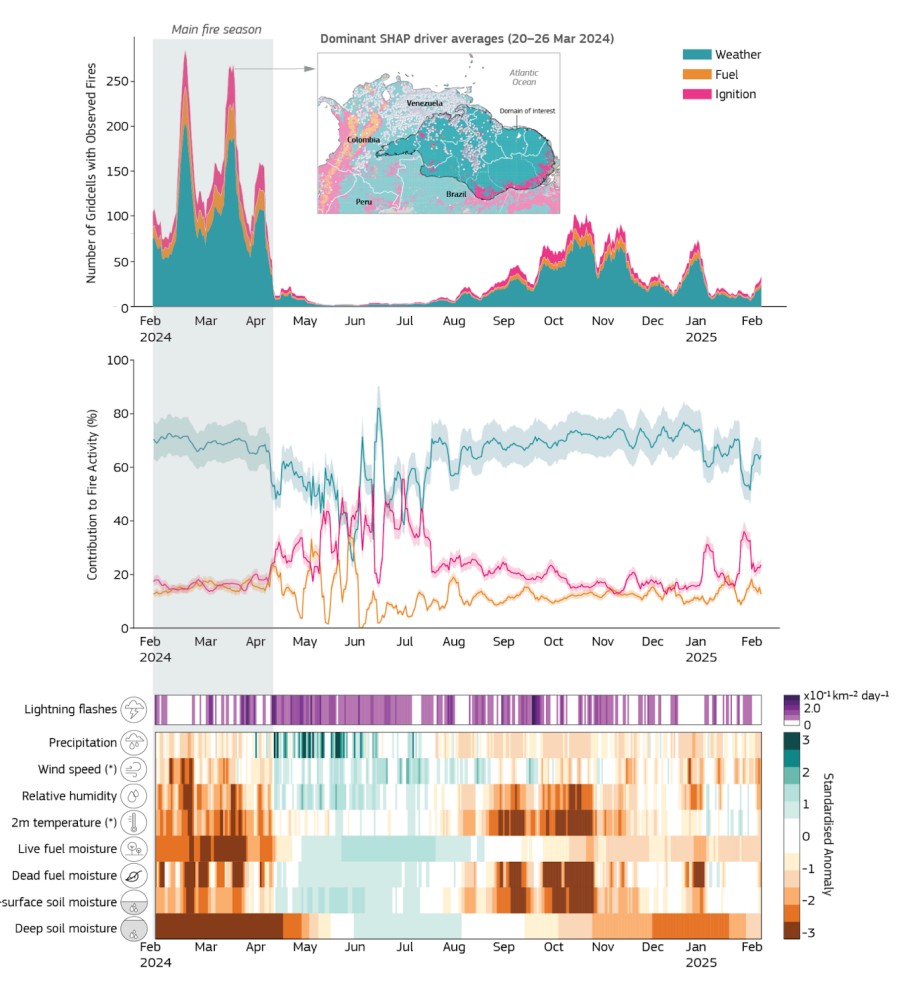

**Figure 10:** Drivers explaining fire hotspots prediction in Northeast Amazonia. Daily fire activity and contributing drivers from February 2024 to February 2025. (**Top panel**) Daily count of grid cells with detected fire hotspots, stacked by dominant driver category, fuel, weather, or ignition/suppression. A dominant driver is assigned only if its contribution exceeds 50% of the total attribution; otherwise, the grid cell is left unclassified (gray). An inset map shows the spatial distribution of dominant drivers during the peak fire week, highlighting regional heterogeneity in fire causation. (**Middle panel**) Relative contributions (%) of each driver category to predicted fire occurrence, with shaded bands indicating model-observation uncertainty. (**Bottom panel**) Standardized anomalies (in units of standard deviation) for each input driver variable, including lightning flash density. Asterisks (*) indicate reversed anomalies.





### *4.2.2.1.2. Pantanal and Chiquitano*

According to our Sparky-PoF analysis, the extreme fire activity during the 2024-25 fire season in the Pantanal-Chiquitano (described in **Section 2.2.2.2**), was mainly the result of extremely dry weather which had started since 2023. Drought conditions affecting the Pantanal and Chiquitano continued into the early months of the 2024-25 fire season following multiple years of below-average rainfall (**Figure 11**). Although the year began with relatively moist surface conditions, deep soil moisture remained in the driest 15% of observed records or 1-2 standard deviations below the mean (**Figure 11**). A wet phase in February-April allowed moisture transfer from the atmosphere to surface fuels, but it did not infiltrate deeply into the soil. As a result, when surface conditions dried out again at the beginning of June, vegetation quickly became highly flammable and primed to ignite.

While fire activity in this region was predominantly controlled by weather (71% mean contribution throughout the year), the role of fuel became increasingly important during the most intense burning phases (up to 40% during the most intense week between 5 and 14 August 2024). In fact, the contribution of fuel conditions doubles during these peak events, indicating that the persistence of fire-conducive weather over time, rather than the specific daily weather, plays a dominant role in driving the most severe fires.

In the Pantanal and Chiquitano, the lack of correlation between fire occurrence and natural ignition sources, such as lightning density (**Figure 11**, bottom panel), is even more evident than in other regions. When lightning does occur, it is typically accompanied by rainfall due to the convective nature of tropical storm systems, further reducing the likelihood of fire ignition. The only notable 'dry lightning' event, observed in mid-May, caused a spike in the modelled PoF which translated into a spike of fire activity that was observable though small in magnitude. Humans are the main source of ignitions in the region (Menezes et al., 2022) and, while weather remains the main driver of fire activity overall, fuel conditions are playing an increasingly important role in determining the severity and extent of extreme fire events (**Figure 11**).





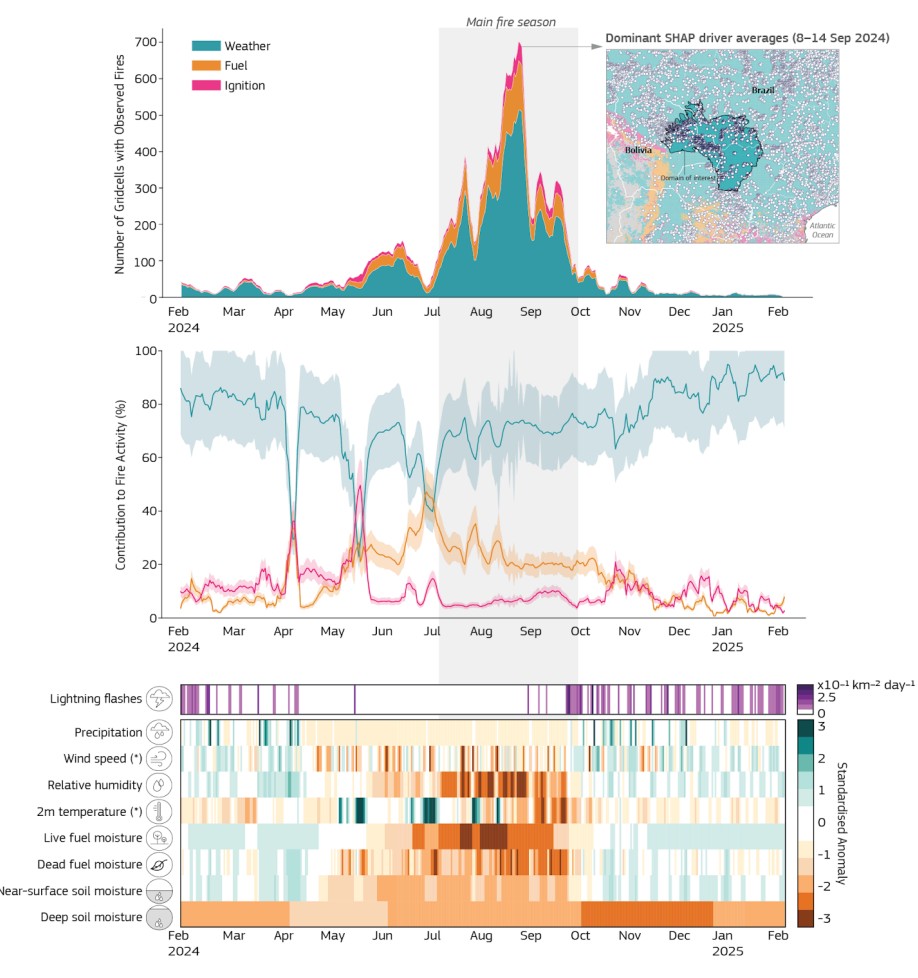

**Figure 11:** Drivers explaining fire hotspots in the Pantanal-Chiquitano (as for **Figure 10**).

### 4.2.2.1.3. Southern California

According to our Sparky PoF analysis of the extreme fire activity during the 2024-25 fire season in Southern California (described in Section 2.2.2.3), the results point to a combination of drivers, weather, fuel, and ignitions, each playing an almost equal role in creating the fire prone conditions observed during the two major events in January 2025 (Palisades and Eton fires).

Early in the 2024-25 fire season, Southern California was emerging from a two-year period of very wet conditions, with deep soil moisture levels at 2 to 3 standard deviations wetter than the climatological average (**Figure S15**). During the summer of 2024, lightning may have contributed to ignitions, although in these areas most fires are typically human-induced. Overall, fire activity remained relatively low and below the climatological average.



However, the Palisades and Eaton fires in January 2025 were well outside the typical fire season. These fires were clear outliers in terms of their seasonality, triggered by a rare alignment of short-lived but intense fire-prone conditions while fuel moistures remained low (**Figure S15**). Between 5 and 25 January, favourable weather, fuel availability, and ignition sources aligned leading to create ideal conditions for ignition and rapid fire spread.

In the week preceding the fires, fire weather conditions contributed around 40% to the predicted fire probability, fuel availability 30%, and ignition sources the remaining 20%. Despite the generally moist deep soil conditions, a brief but extreme episode of surface drying (reaching 3 standard deviations below normal) combined with unusually strong winds (also 3 standard deviations above average), was sufficient to create highly flammable conditions at the wildland-urban interface, enabling the fires to ignite and spread rapidly.

### 4.2.2.1.4. Congo Basin

According to our Sparky-PoF analysis, the extreme fire activity during the 2024-25 fire season in the Congo basin (described in **Section 2.2.2.4**), was the result of the extreme drought that has affected the regions in recent years.

In 2024-2025, fire activity in the Congo occurred year-round in a region marked by abundant and widespread vegetation cover. The spring wet season (March-May) did not materialise due to extreme and persistent drought conditions. As a result, the second wet season later in the year also brought limited relief, leaving deep soil layers significantly dry (up to 2 standard deviations below climatological norms). The region remains in a prolonged state of water deficit until now (**Figure S16**).

Throughout the year, weather conditions were the dominant and most stable factor influencing both the number, intensity and duration of fire events. A combination of low rainfall (67% below the climatological average) and elevated temperatures (90% above the climatological average) led to sustained drying of both vegetation and soil, placing them among the driest 2% and 1% of the climatological record, respectively. These conditions maintained highly flammable landscapes across the region (**Figure S16**)

Most fire ignitions in the Congo basin can be attributed to human activity. Although lightning occurs year-round (**Figure S16**), it is more frequent during the wet season due to the convective nature of tropical precipitation. However, during these wetter periods, high moisture levels typically prevent fire ignition and spread. In contrast, during prolonged dry spells, even a small number of human-caused ignitions can trigger widespread and persistent fire outbreaks, owing to the highly combustible state of the vegetation.

### 4.2.2.2. Causes of Burned Area Anomalies during Focal Events

### 4.2.2.2.1. Northeast Amazonia

According to our ConFLAME analysis of the extreme BA during the 2024-25 fire season in Northeast Amazonia (described in **Section 2.2.2.1**), weather conditions explained about 40-60% of the BA anomalies, though with fuel conditions acting as an important determinant cause during the periods with greatest fire extent (**Figure 12**). In the peak month of March 2024, BA exceeded the long-term average (2002-2024) by over 12,000 km². Nearly half of the March 2024 anomaly could be attributed to fuel conditions, while weather anomalies potentially accounted for between 50% and 150% of the BA anomaly (a high-end value of 150% would suggest that weather alone would have caused anomalies exceeding the observed values, but below-average ignitions moderated the BA response; **Figure 12**). During the secondary peak in BA anomalies during October-November, fuel and weather contributed similarly with fuel rising in importance due to the insufficient water recharge from



the wet season. Weather and fuel together accounted for between 1,000 km² and over 10,000 km² of BA anomalies.

Consistent with active fire analysis (**Section 4.2.1.1**), fuel played a key role in determining the geographical distribution of BA during the 2024-25 fire season (**Figure S19**). This is visible in northern regions such as the Forest-Savanna Transition Zone in northern Venezuela and southern Guyana, and the Northern Amazonia Savannas of Roraima and northern Pará, where savanna outcrops are surrounded by rainforest (see maps in **Figure S19**). In the forest landscapes, fuel anomalies and fire weather anomalies drove the predicted anomalies in BA. Interestingly, predicted BA anomalies were large in some parts of the region (e.g. Suriname) but went undetected by the MODIS BA product. The causality framework is very confident in its prediction, raising the question of whether detections were missed, possibly due to dense canopy and persistent cloud cover (Giglio et al., 2006).

Despite widespread BA in early 2024, many parts of the region remained largely unburned. Understanding why is as important as knowing what drove the fires. Our analysis shows that in areas with very low BA fraction (less than 0.5% of burnable area), no single factor (fuel, weather, or human activity) clearly limited fire spread (refer to **Figure S18**). Instead, a combination of factors, such as low ignition rates, patchy fuels, or short dry spells, likely prevented fires from taking hold. On the other hand, in the most severely burned areas (top 5% of BA), the relative importance of fuel and weather was reversed compared to broader patterns. Here, fuel moisture emerged as the primary driver of BA. Drier conditions could have increased BA by 30-40%. Weather still played a role contributing an additional 20% increase, but its influence was secondary to that of fuel.



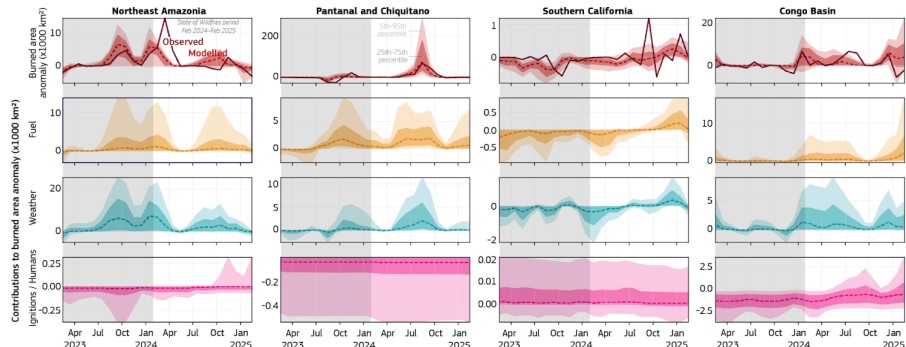

**Figure 12:** Anomalies in burned area (BA) and driver contributions for each focus region during 2024, relative to the 2002-2024 average. Columns represent regions; rows show different variables. **(Top row)** Modelled and observed BA anomalies, expressed thousand km². Model output shows median, interquartile range (shaded), and 5th-95th percentile range (lighter shading). **(Other rows)** Modelled contributions to BA anomalies from fuel conditions, fire weather, and human/ignition-related factors, also shown in thousand km². These panels highlight which drivers contributed most to regional fire deviations from the historical average in 2024.

### 4.2.2.2.2. Pantanal and Chiquitano

According to our ConFLAME analysis of the extreme BA during the 2024-25 fire season in the Pantanal-Chiquitano (described in **Section 2.2.2.2**), weather conditions explained half of the BA anomalies and fuel conditions explained almost 30% (**Figure 12**). June, July, and August accounted for the most extensive burning in the Pantanal, with 25-75% of the landscape experiencing some fire activity, even if large parts featured only small anomalies. The peak occurred in June, when the BA exceeded climatological values by more than 5,000 km² (almost triple than the annual mean). This anomaly was primarily driven by weather conditions (50-60%) with fuel (10-20%) and ignition (10-20%) contributing equally. Although fire weather remained favourable in September and October due to persistently high temperatures, overall fire activity was lower than during the earlier peak (**Figure 12 and S20**).

We found that ignition sources contributed only 10-20% to the anomalously high fire activity in 2024. However, we caution that our modelling framework only partially captures ignition dynamics, particularly those related to human activities such as farming. This limited representation is reflected in the wide uncertainty range assigned to ignition within the causality framework. Key factors like land clearing, water extraction, and the proximity of ignitions to protected areas are known contributors to extreme fires in the Pantanal (Barbosa et al., 2022) and they are not fully accounted for in our analysis.

Regional differences in fire drivers were evident (**Figure S20**). Fuel conditions played a key role in the fine-scale geographical distribution of BA anomalies. Exceptionally dry fuels affected the Chiquitano dry forests in the east, while weather was the dominant driver in upland regions along the edge of the Pantanal wetlands, such as the Serra do Amolar hills in western Brazil. The most extreme fires were observed where these two influences overlapped, where vegetation was both unusually flammable and atmospheric conditions were conducive to burning.



As for what prevented the fires from becoming even more severe (**Figure S18**), no single
factor alone limited fire spread, even in the areas that burned most intensely. However, small
shifts in conditions, such as drier weather, drier fuels, or fewer land-use barriers, could have
led to 2-12% more BA in the model cells experiencing the greatest fire anomalies regions
(top 5% of anomalies).

### 4.2.2.2.3. Southern California

According to our ConFLAME analysis of the extreme BA during the 2024-25 fire season in
Southern California (described in **Section 2.2.2.3**), the most important cause of the extent of
burned areas was fuel (30% to 60%) closely followed by weather (20-40%), while ignitions
(20%) was less pronounced that in previous years and acted as reducing factor (**Figure 12**).
During January 2025, unusually dry fuel conditions played a key role in promoting BA
anomalies, explaining up to 500 km² of the 800 km² of the anomalous BA in that month. Fire
weather conditions, starting as early as October 2024, were also anomalous versus previous
years. Focusing on the areas with the most extensive burning (top 5% of BA), we found that
anomalies could have been 30-60% larger under drier fuel conditions and more extreme fire
weather, with an additional 5% increase if fuel availability anomalies had also been higher
(**Figure S17 and Figure 4.11**). The substantial suppression efforts deployed is unaccounted
for in our modelling framework and could be one of the possible reasons the fires did not
escalate even further.

### 4.2.2.2.4. Congo Basin

According to our ConFLAME analysis of the extreme BA during the 2024-25 fire season in
the Congo basin (described in **Section 2.2.2.4**), weather conditions explained about 30-60%
of the BA anomalies, with fuel conditions acting as an important secondary control during the
periods with greatest fire extent (**Figure 12**). Fuel conditions in the Congo Basin remained
relatively stable throughout 2024, contributing between 10-35% to fire activity year-round. In
contrast, the influence of weather conditions varied more substantially, with virtually no
fire-conducive weather in October-November, outside the typical fire season, and moderate
levels (5-15%) during peak fire periods, particularly in January and July (see also **Figure
S17**). July stood out as the month with the largest deviation from typical fire patterns. During
this time, fuel conditions and fire weather contributed almost equally to the BA (**Figure S22**).
ConFLAME indicates widespread anomalous BA across the southern part of the Congo
basin. These model estimates of BA are larger than the BA detected by satellites (Figure
S22). Dense canopy in these remote regions may have led to missed detections of BA.
Particularly high fire-conducive conditions were predicted across much of southern
Democratic Republic of the Congo (DRC), as well as northern Angola and parts of the
Republic of the Congo. However, two notable pockets, in the far northeast of the basin,
around the border of northeast DRC and South Sudan, and a smaller zone just east of the
border between the DRC and Republic of the Congo in the north, did not emerge in our
analysis.
Despite these broad areas of fire-favourable conditions, fires did not become much larger in
many places. The key reasons for this were moisture and weather limitations. Looking at
areas at the top 5% of burning, up to 15% more fire could have occurred if fuel had been
even drier or if atmospheric conditions had been slightly more favourable.



# 5.     Attribution to Global Change Factors

Many of the direct drivers and controls on fire events, outlined in **Section 4** (e.g. weather, fuel, moisture, ignition and suppression), are influenced by global change factors such as climate and land-use change. Since the pre-industrial era, global mean temperature has increased by ~1.3˚C (Betts et al., 2023; Forster et al., 2025), with greater rates of warming at higher latitudes, adding potential for fuel drying. Climate change has also resulted in altered precipitation patterns, with total rainfall and dry season length increasing or decreasing variably across regions (Polade et al., 2014; Swain et al., 2018; IPCC, 2023a). Meanwhile, changes to fuel load and ignition rates are driven by emissions, climate change and land-use change, with varying effects regionally (Foley et al., 2005; Finney et al., 2018; Romps, 2019; Wang et al., 2024).

## 5.1.     Methods

### 5.1.1.     Overview of Attribution Approaches

Fire is a complex phenomenon that impacts societies and ecosystems in many ways, from the extent of BA to the severity of individual fire events. Different user groups seek information on different aspects of fire risk, whether policymakers, communities, fire managers, litigators, or those working to build a broader scientific evidence base. To provide results relevant for a wide range of stakeholders we apply various modelling approaches to fire attribution, drawing on different metrics and attribution techniques, to build a more comprehensive understanding of human influence on recent extreme fire activity. Our approach includes analyses of fire weather indices and BA, alongside a range of attribution metrics suited to these different contexts. Our BA attribution also provides the evidence, in the form of a calibrated probabilistic model, needed to perform future risk projections in **Section 6**.

While most attribution research has focused on the contribution of anthropogenic climate change, humans influence fire occurrence and risk in multiple other ways: the direct influence of people via activities such as land use change and landscape configuration; changes in ignition probability, fire suppression, among others. Considering human-driven climate change separately to changes in human activity, in addition to their combined effect, allows us to disentangle the contributions of local and global environmental change.

Understanding the influence of people or climate on fire and its drivers is inherently challenging, given the complexity of fire processes and the interactions between natural and human systems. Integrating these range of complementary methods - each with its own strengths and limitations, additionally helps build confidence in attribution results that no single method could provide alone. We can therefore identify where there is broad agreement across methods.

To quantify the different ways people affect fire, we apply four types of attribution in this report (**Table 4**), designed to meet diverse user needs and to align with the modelling frameworks currently available:

- i) Firstly, our attribution to *anthropogenic climate forcing* explicitly targets the changes driven by anthropogenic greenhouse gas emissions and land-use change, following the IPCC WGI definition (Hegerl et al., 2009; Mengel et al., 2021). We prescribe these emissions in a model to specifically isolate human forcing from natural variability **(Section 5.1.2 and 5.1.3)**.
- ii) Our attribution to *total climate forcing* considers changes driven by climate change since the pre-industrial period, including both anthropogenic climate forcing and





natural variability in line with the IPCC WGII and the Intersectoral Impacts Model Intercomparison Project 3a (ISIMIP3a) definition of climate change impact attribution (IPCC, 2023b; IPCC 2023c; Mengel et al., 2021). This involves comparing simulations driven with historical reanalysis, our factual, to a detrended counterfactual simulation, where the trend in each climate variable is removed (with both simulations including historical transient land-use change). Therefore only the impacts of climate change are attributed, not distinguishing between anthropogenic or natural causes (Mengel et al., 2021; Burton, Lampe et al., 2024). We perform this between 2003-2019, the overlap between available counterfactual simulations and satellite data used for training in Burton, Lampe et al., 2024.

- iii) Our attribution to *socio-economic factors* is applied via the same set of simulations as our attribution to *total climate forcing*. We isolate the role of socio-economic factors by comparing the early industrial period to the late industrial period (1901-1917 versus 2003-2019) using detrended ISIMIP3a data, in which only land-use and population density are allowed to change (Burton, Lampe et al., 2024).
- iv) Our attribution to *all forcings* compares the early industrial period in the counterfactual scenario to the last industrial period in the factual scenario, which gives the net effect of all forcings combined (*anthropogenic climate forcing + total climate forcing + socio-economic factors*).

The attribution methods described above enable us to assess the influence of climate and socio-economic forcings on fire in each focal region with respect to three different target variables:

- i) **Extremes in fire weather during 2024-25**. The FWI is a weather-based indicator of landscape flammability and can provide insight into how fire-prone conditions are likely to be affected by a changing climate.

  Using the HadGEM3-A large ensemble, we attribute changes in the probability of extreme fire weather conditions to *anthropogenic climate forcing*. This analysis specifically targets the months identified as extreme for each focal event as outlined in **Section 2.2.2** focusing on sub-regional extremes that occur in the model grid cells with the highest FWI values (top 5% of all regional grid cells). By focusing exclusively on these areas of most severe fire weather, this approach provides a proxy for understanding how each forcing influences the locations and times of highest fire risk within the region. We used this methodology as in last year's report. See **Section 5.1.2** for details.

- ii) **Region-wide extreme BA during 2024-25 focal events.** Event specific BA reflects how climate and human factors jointly influence the actual extent of burning during major fire events, offering a direct measure of fire impact on people and ecosystems.

  Using the ConFLAME model framework we attribute changes in the likelihood of the 2024-25 observed total BA across the entire focal region to *anthropogenic climate forcing*, *total climate forcing, socio-economic factors,* and *all forcings* combined. Like our FWI analysis focuses on the observed peak burning months and captures the overall influence of each forcing on the extent of fire activity at the regional scale. See **Section 5.1.3** for details.

- iii) **Background changes in BA this century** using median monthly over recent decades. Background BA shows how climate change is reshaping regional fire regimes over the long term, revealing gradual shifts in baseline fire activity that may go unnoticed in year-to-year variability.





Using fire-enabled dynamic global vegetation models (DGVMs) participating in the Fire Model Intercomparison Project (FireMIP), we attribute changes in median monthly BA averaged over recent decades (2003-2019) to *total climate forcing*, *socio-economic factors*, and *all forcings* combined. This approach provides context on longer-term background fire activity and applies the same methodology as last year's report, though this year focussing on specific focal regions. See **Section 5.1.4** for details.

In each approach we include an explicit estimate of uncertainty. We use bootstrapping to give uncertainty estimates for the FWI Risk Ratios (RR) defined as the ratio between the probability of seeing the observed FWI with the target forcing vs without anthropogenic climate forcing, reported here at 90% confidence intervals. ConFLAME is designed as an uncertainty quantification model (as per our driver assessment, **Section 4.2.4**), giving the likelihood of all possible BA outcomes for each region based on a probabilistic analysis of past burn patterns and environmental conditions. We combine the information from the FireMIP models in a weighted multi-model ensemble to give uncertainty ranges across the models. Each result therefore presents a 5-95th percentile probability estimate.

For consistency with last year's report we also report attribution estimates based on methods used in the State of Wildfires 2023-24 report (Jones et al., 2024b):

- iv) **Sub-regional extreme BA during 2024-25.** We attribute changes in the likelihood of extreme BA occurring within the model grid cells with the highest BA (top 5% of all regional grid cells), focusing on areas where fire activity was most spatially concentrated during peak burning months. This analysis uses the same ConFLAME simulations and forcing scenarios as the region-wide BA attribution and provides insight into how forcings affect the most severely impacted locations within the region. See **Supplementary Text S5.2.3** for discussion of results.

In the coming years, our project seeks to incorporate attribution results based on a broader set of Earth System Models (ESM) to better sample the structural uncertainty arising from differences in process representation across different models (i.e. beyond HadGEM3-A). In this report, we introduce results based on the one ESM as follows:

- v) **Background changes in fire weather this decade.** Using the Canadian Earth System Model (CanESM5; Swart et al., 2019), we attribute changes in the frequency of extreme fire weather to *total climate forcing* with the Canadian Fire Weather Index (FWI), identifying how the likelihood of extreme fire weather has changed by comparing the frequency of high Fire Weather Index (FWI) values in pre-industrial and present-day climates. Our analysis covers the years 2016 to 2025, focusing on the climatological months of peak burning during the 2024-2025 fire season. See **Supplementary Text S5.1.2** for methodology and **Supplementary Text S5.2.2** for discussion of results.





**Table 4**: Summary of the attribution approaches used in this report. See **Table S2** for a
breakdown on the what each attribution type includes and what each modelling targets.

| Term | Definition | Experiments compared | Framework | Application |
|------|------------|----------------------|-----------|-------------|
| **Event attribution for fire weather and burned area** | | | | |
| **Anthropogenic climate forcing** | Change in fire weather driven by anthropogenic emissions from greenhouse gases, land-use change and aerosols. As per (Ciavarella et al., 2018; Li et al., 2021) | **Factual:** HadGEM3-A_ALL: with natural forcing plus human emissions **Counterfactual:** HadGEM3-A_NAT **with** natural-only forcing from solar variability and volcanoes | HadGEM3-A attribution ensemble. 0.83 x 0.56 degree resolution | Fire weather (FWI) |
| | | | ConFLAME (Kelley et al. 2021; Barbosa et al. 2025b) with merged ERA5/HadGEM3-A product | Burned Area with ConFLAME |
| **Impacts attribution for fire weather and burned area** | | | | |
| **Total climate forcing** | Changes in FWI since pre-industrial | **Factual (2016-2025):** present-day climate from CanESM5 SSP585 **Counterfactual 1850-1859:** Pre-industrial climate simulation | CanESM5 CMIP6 ensemble | FWI |
| | Changes in BA due to climate change, irrespective of the cause of warming. As per ISIMIP (Intersectoral Impacts Model Intercomparison Project) (Mengel et al., 2021 and Frieler et al., 2024) | **Factual (2003-2019):** present-day climate (driven by GSWP3-W5E5 reanalysis), CO2, land-use and population **Counterfactual (2003-2019):** Historical climate detrended using seasonally-varying regression on global mean temperature (ATTRICI method, CO2 fixed at 1901 value, present-day land-use and population | ISIMIP3a impact attribution. 0.5 degree resolution | FireMIP ensemble and ConFLAME |
| **Socio-economic factors** | Changes in BA due to land-use and population change. As per (Burton, Lampe et al., 2024) | **Counterfactual (1901-1917):** Warming trend removed using ATTRICI method, fixed 1901 CO2, limited land use and population change **Counterfactual (2003-2019):** Warming trend removed using ATTRICI method, fixed 1901 CO2, present-day land use and population | | |
| **All forcings** | Changes in BA due to climate, land-use and population change. As per (Burton, Lampe et al., 2024) | **Counterfactual (1901-1917):** Warming trend removed using ATTRICI method, fixed 1901 CO2, limited land use and population change **Factual (2003-2019):** Historical climate driven by reanalysis | ISIMIP3a impact attribution | FireMIP ensemble |


### 5.1.2. Attributing Extremes in Fire Weather during 2024-25
We use two complementary approaches to attribute changes in the probability of high fire
weather, measured using the Canadian Fire Weather Index (FWI), to anthropogenic climate
change. The first method uses a targeted large-ensemble weather model simulation to
assess the influence of climate change on the 2024/25 fire seasons directly. The second




method applies a longer-term, probabilistic framework using simulations from a fully coupled Earth system model.

The first approach follows the same methodology used in the previous State of Wildfires report (Jones et al. 2024b). This is an established approach to attribute changes in the probability of high fire weather, measured using FWI, to anthropogenic climate forcing. This method has been previously used by the World Weather Attribution (Barnes et al., 2023; Barnes et al., 2024; Barnes et al., 2025), using outputs from the HadGEM3-A large ensemble (Ciavarella et al., 2018). Our approach builds on the methodology introduced by Stott et al. (2004) for attributing extreme weather events, and it has been applied in other attribution studies targeting fire weather, such as Li et al. (2021).

As outlined in **Section 4.1.1**, the FWI is used operationally and in research contexts to rate fire danger based on meteorological conditions. Due to the availability of model output variables we use maximum daily temperature at 1.5 m as a proxy for noon values, total daily precipitation, mean daily relative humidity at 1.5 m, and mean daily wind speed at 10 m, following Perry et al. (2022). We calculate the daily FWI for the months of 2024-25 peak BA anomaly for each focus region, using the same month and region for validation over the historical time series (1960-2013). Note that at time of writing, data for HadGEM3-A was only available till the end of 2024, so we do not report on Southern California fires using this method.

We validate and bias-adjust the model estimates of high FWI for the period 1960-2013 by comparing a 15-member HadGEM3-A ensemble with ERA5 reanalysis data (C3S, 2024) representing "observed" FWI. The 0.25 degree resolution observed FWI from ERA5 was coarsened by linear interpolation (calculated by extending the gradient of the closest two points) to match the 0.5 degree model grid. We compare the time series of individual components of the FWI **(Figure S49-S55)**, and the distribution of the modelled and observed FWI **(Figure S56-S58)**, and apply a simple linear regression to find the bias correction required for the 2023 model output. Before bias-adjustment, the modelled FWI is generally higher than the observed FWI for Amazonia and Congo, which modelled FWI compares more favourably to ERA5 in the Pantanal. The correction adjusts the trend and absolute value while maintaining variability, and the model successfully reproduces the observed distribution after applying the correction in each region (see **Supplementary Text S9**).

For the events occurring in the 2024 fire season, we calculate the FWI from the HadGEM3-A model simulations comprising 2 experiments of 525 members each, one driven by all forcings including historical greenhouse gas emissions, aerosols, zonal-mean ozone concentrations, land-use change and natural forcing (ALL), and a second counterfactual simulation with natural-only forcing from solar variability and volcanic emissions, and 1850 land-use (NAT) (**see Table 4**). By applying the bias-adjustment from the previous step, and comparing the fire weather in the two simulations to the 2024-25 observed FWI from ERA5, we calculate the change in probability of high fire weather due to anthropogenic climate forcing. The standard definition of "high fire weather" that we use is the 95th percentile of daily Fire Weather Index (FWI) values across all grid cells and days during the season. However, as in last year's report and in Burton et al. (2025), when the region is small or when climate conditions significantly influence the higher FWI in our counterfactual, leading to few ensemble members reaching higher FWI values, we need to adjust our definition of extreme. In this year's assessment, we apply the 90th percentile threshold for the Northeastern Amazonia and Congo regions, as the differences between the factual and counterfactual ensembles are so large that very few counterfactual members reach the 95th percentile of the factual distribution, making the calculation of risk ratios unreliable.



### 5.1.3.    Attributing Region-wide Extreme BA during 2024-25

We use the ConFLAME framework for direct BA attribution. For this report, we apply two configurations of the ConFLAME attribution framework to attribute anomalies in BA fraction during the peak burning months of the 2024-25 fire season:

- A **near real-time (NRT) setup** for targeting anthropogenic climate forcing, which largely mirrors the configuration used in the drivers attribution section (see **Section 4**), assesses how human influences affected the likelihood of BA via meteorological driver of fire conditions observed during the specific 2024 events. This setup targets the actual environmental conditions leading up to and during the events, providing the most up-to-date picture of climate and socioeconomic influences. By focusing on the precise timing and location of the event, the NRT configuration provides an up-to-date and high-resolution picture of how anthropogenic climate forcings have influenced the likelihood of extreme fire activity.
- The **Inter-Sectoral Impact Model Intercomparison Project (ISIMIP) 3a setup**, previously used with ConFire in last year's report. This setup enables the analysis of how often fire events such as those in 2024 might occur under environmental conditions from 2002 to 2019. While 2024 itself is excluded, we look for similar events in this earlier period to understand how likely they would be without the recent changes in climate and land use. This broader, long-term setup helps us assess how the background risk is shifting over time and complements the more event-specific analysis shown earlier. This setup also directly links to the future projections presented in **Section 6**, which also use ISIMIP. As an addition to last year's report's set up, our ISIMIP set up also includes changes in land use and cover (measured as the difference between tree cover and agricultural fraction since the previous year) in the direct socioeconomic forcing attribution (see **Table S3**).

As each configuration uses data that is somewhat similar to our Fire Weather (in the case of NRT) or FireMIP (when using ISIMIP) set ups, neither setup is fully independent of our other two modelling approaches. However, the fire modelling in ConFLAME captures different components of fire than FWI or FireMIP by attributing BA during the events themselves. The advantage of ConFLAME is that it bridges the gap between event-focused real-time attribution and global process-based fire models. That said, future iterations would benefit from incorporating more independent, preferably observation-driven input datasets to improve robustness and reduce potential structural alignment across methods.

Each attribution experiment involved training ConFLAME using "observed" or reanalysis driving data against MODIS BA (as described in Section 4). We then ran the framework with factual driving data followed by a separate run counterfactual with the effect we aim to attribute (e.g., all forcings, climate, or socioeconomic drivers) removed. We conducted paired ConFLAME factual and counterfactual predictive model simulations at monthly resolution, using a structure similar to that in **Section 4.1.2**, with specific drivers grouped into controls in **Table S1** and evaluated the model following Barbosa (2025; **Section 4.1.2**). We separately train ConFLAME on 50% of the data between 2003-2011 and perform evaluation on years 2012-2019. Further details of the model fitting and validation can be found in **Supplementary Text S5.1.3** and **Supplementary Text S9.1**, respectively.

To determine the impact of total climate forcing, socioeconomic factors and total forcing on increased BA during our focal events using the ISIMIP configuration, we conducted paired sampling of monthly BA in the target months (see Table 4). Total climate forcing's factual driving data uses the same 2003-2019 GSWP3-W5W5 reanalysis data used for training for factual, while we use detrended data for the counterfactual, whereas socioeconomic used detrended data 2003-2019 for factual and 1901-1917 for counterfactual. Total forcing used 2003-2019 from GWSP3-W5W5 for the factual and 1901-1917 from detrended



GWSP3-W5W5 for the counterfactual. We used paired sampling to account for uncertainty in the relationships between drivers and BA, ensuring co-variation between experiments (as in Kelley et al., 2021). In total, we drew 1,000 samples across the 17 years of each simulation, resulting in 17,000 paired samples.

We use two key metrics to assess how our target factors have influenced BA during extreme fire events. We report attribution metrics both for the entire region (reported in the main text, **Section 5.2.2**) and for "sub-regional extremes" - the grid cells with the top 5% of BA, to also assess how anthropogenic factors may have influenced the most severely affected areas (in **Supplementary Material S5.2.2**) The Amplification Factor (AF) tells us how much bigger (or smaller) the BA was because of a specific factor. It works by comparing factual BA for BAs as large or larger than what was observed during the target months versus counterfactual. Observed BA is calculated in a manner consistent with the model outputs, by averaging BA across either the entire region or the top 5% of BA within the target region and month. Observations are derived from monthly MCD64A1 data. In near-real-time (NRT) mode, we do this for the specific year of interest. In the ISIMIP setup, we compare across many years (2003-2019). An AF greater than 1 means climate change increased BA. For example, an AF of 2 means twice as much area burned. We calculate this across our model simulations and report both the central estimate (median) value and the range of uncertainties based on 10th to 90th percentiles. Because the Early Industrial factual simulation in our ISIMIP setup includes no human influence on the climate, we first adjusted the target event's BA to the level expected without climate change. This adjustment involved identifying the percentile of the observed BA in the factual simulation, and then finding the BA at that same percentile in the counterfactual simulation

For the NRT set up, we can also use the Risk Ratio (RR), which shows how much more (or less) likely the target factor made a fire event of this size. Similarly to **Section 5.1.2**, it compares the chance of seeing the observed BA under today's climate to the chance under a climate without human influence. A RR above 1 means climate change made the event more likely; a RR below 1 means it made it less likely.

### 5.1.4.    Attributing Background Changes in Burned Area this Century

We assess how BA has changed over recent decades due to climate and socio-economic drivers using the FireMIP ("Fire Model Intercomparison Project") attribution framework developed by Burton, Lampe et al. (2024). This method uses state-of-the-art global FireMIP models, employing each model's native fire scheme, to estimate the contribution of different drivers to BA by comparing simulated fire activity under different ISMIP3a experiments setup. We quantify the effect of *climate forcings* on BA by comparing the present-day factual burned area to the present-day counterfactual BA. The effect of *socio-economic forcings* is assessed by comparing the present-day of the counterfactual simulations to the early-industrial of the counterfactual simulations since long-term climate is stationary in these simulations. Lastly, we find the effect of *all forcings* by comparing the present-day factual BA to the early-industrial counterfactual BA.

The attribution focuses on changes in median monthly BA during 2003-2019 and uses a weighted multi-model ensemble, where weights reflect each model's ability to reproduce observed regional fire anomalies in GFED5 and FireCCI datasets. All results are reported as relative anomalies, and uncertainty is assessed via a random resampling of the weighted ensemble, including a stochastic parameter which accounts for uncertainty on the performance of the entire ensemble. This approach provides a robust and conservative estimate of trends, particularly suited to assessing regional-scale fire responses.



In contrast to last year's report, where results were reported for IPCC AR6 regions containing the focal fire zones, this year we refined the analysis by tailoring the attribution directly to the specific areas featured in the report. This regional adjustment enhances the relevance and interpretability of the attribution results for each case study.

For full details on the method, model evaluation, and baseline results across all IPCC regions, see Burton, Lampe et al. (2024).

## 5.2.    Results

### 5.2.1.    Extremes in Fire weather during the 2024-25 Focal Events

#### 5.2.1.1.    *Northeast Amazonia*

We find that the fire weather conditions in Northeast Amazonia during January-March 2024 were significantly more likely due to anthropogenic climate forcing, with the probability of experiencing fire weather at or above the levels observed during the event being 32 to 73 times higher in the factual simulations compared to the counterfactual simulations (**Figure 13**). A substantially larger proportion of the factual ensemble exceeds the observed 90th percentile of FWI from the ERA5 reanalysis than in the counterfactual ensemble (**Figure 13**), indicating that high fire weather conditions during early 2024 were much more likely in a climate influenced by anthropogenic emissions.

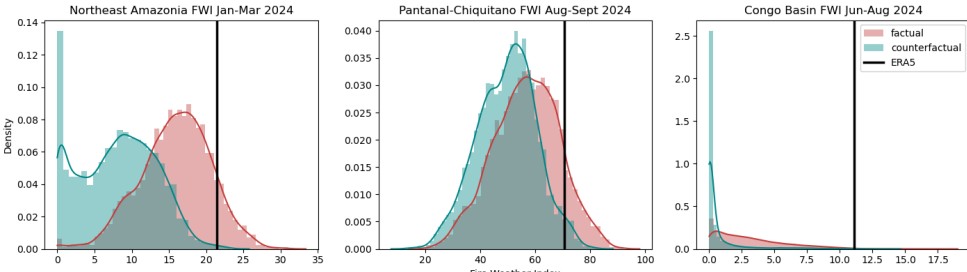

**Figure 13:** High fire weather conditions in 2024/25: Probability distributions of FWI in the HadGEM3 ensemble for the focal fire season in each region, comparing simulations with anthropogenic and natural forcings (red; factual) to natural-only forcings (teal; counterfactual). Black line shows ERA5 reanalysis. The x-axis shows the regional average of high-percentile FWI days: 89th percentile for Jan-Mar in Northeast Amazonia (left), 95th percentile for Aug-Sep in the Pantanal and Chiquitano (middle), and 90th percentile for Jun-Aug in the Congo Basin (right).

#### 5.2.1.2.    *Pantanal and Chiquitano*

The high fire weather conditions experienced during the peak anomaly in fire activity in August-September 2024 were 4.2-5.5 times more likely due to anthropogenic climate forcing (**Figure 13**). While this increase is smaller than that estimated for Northeast Amazonia, the narrower range suggests we have greater confidence that human influence increased the probability of extreme fire weather conditions in this region.

Our results largely agree with the rapid attribution analysis from the World Weather Attribution (WWA) initiative (Barnes et al., 2024), though with smaller uncertainty ranges, WWA found that the accumulated fire weather conditions, represented by the June Daily





Severity Rating (DSR), were 4.6 (1.1 to 20) times more likely due to human-induced climate change. The DSR, a fire-suppression oriented rescaling of the FWI, is commonly used to assess the cumulative fire weather danger over monthly timescales (Van Wagner, 1987). WWA focused on June conditions because of their role in setting up the severe fire season that followed, and their direct relevance to the large BA that severely impacted wildlife and livelihoods in the Pantanal. Observations also indicated a decrease in annual rainfall of -23.5% (−46% to +5%) in the region, though this trend was not reproduced by climate models (Barnes et al., 2024).

### 5.2.1.3.  Southern California

Due to the lack of data availability from HadGEM3-A for 2025, we were unable to perform bespoke FWI attribution analysis for Southern California. However, in previously published analysis, the rapid attribution study by WWA (Barnes et al., 2025) found that the extreme fire weather conditions (peak FWI) in the coastal southern California ecoregion surrounding Los Angeles during January 2025 were 1.37 (0.48 to 3.6) times more likely in comparison to the pre-industrial climate, suggesting that climate change may have lead to a moderate increase in fire weather, though causing a reduction in fire weather is also plausible and within the confidence range. As the impacts of Los Angeles fires related to extreme single days of wildfire spread, the monthly maximum FWI value averaged over the study region was used here. This result is complemented by the increasing likelihood of an extended dry season in the region. Decreased October-December precipitation allowed for protracted fuel drying, resulting in a more likely overlap between dry conditions and the winter Santa Ana winds. Observed trends (ERA5) in the October-December standardised precipitation index found that the dry conditions leading up to the LA fires were 2.4 (0.33 to 20.9) times more likely than in the pre-industrial climate. Using analogue-based attribution (Vautard et al., 2016), the cut-off-low circulation pattern associated with the strong Santa Ana winds around Los Angeles was found to have increased in likelihood by 2.5 (0.4 to 17) times.

### 5.2.1.4.  Congo Basin

The high fire weather conditions observed across the Congo Basin during June-August 2024 were unusual in both the factual and counterfactual simulations. Our analysis indicates that these conditions were 3.0-8.0 times more likely due to anthropogenic climate forcing (**Figure 13**). The entire FWI distribution in the factual ensemble is shifted toward higher values compared to the counterfactual ensemble. This means that across the full range of fire weather conditions, the probability of conditions conducive to burning is substantially greater in a climate influenced by human emissions.

## 5.2.2.  Region-wide extreme BA during 2024-25
### 5.2.2.1.  Northeast Amazonia

We find strong evidence that anthropogenic climate forcing contributed to increased regional BA during the January-March 2024 fire season in Northeast Amazonia. Our analysis shows a 96% likelihood (very likely under IPCC definitions of confidence) that BA was higher than it would have been without anthropogenic climate forcing (**Figure 14**). We estimate that regional BA was approximately 4.3 times larger (our *Amplification Factor*) than it would have been in a counterfactual world without anthropogenic climate forcing (**Figure 14; Table 5**), with a 90% confidence range of 1.02 to 25.32. While the central estimate suggests a quadrupling of BA, the wide uncertainty range reflects the natural variability of fire processes. Nonetheless, even the lower bound supports a small but clear increase.

We assess the risk ratio, the likelihood of an event like January-March 2024 occurring under current climate conditions versus a pre-industrial baseline (**Table 5**). Based on historical data



provided as evidence for the model, we estimate that a similar event is now 2.1 times more likely due to anthropogenic climate forcing. This figure captures the longer-term climate signal that would shape the overall frequency of such events. When we control for meteorological variability by comparing simulations with and without anthropogenic forcing but using identical weather patterns from 2024, we see slightly stronger effects (**Table 5**). The risk ratio rises to 2.7, and the upper bound of our Amplification Factor increases dramatically (over 100-fold in some ensemble members). This suggests that climate forcing alone could account for much, or possibly all, of the burning under certain conditions, although the central estimate remains close to our previous assessment.

Climate influence was widespread across Northeast Amazonia, most of the entire region showing a greater likelihood of increased BA due to anthropogenic forcing (**Figure 15**). The strongest attribution signal occurred in the Southern Guiana Shield Fringe Forests, where climate change was very likely (≥90% confidence) to have increased BA. These forests are particularly important due to their extensive areas of primary rainforest and high ecological sensitivity. In contrast, attribution confidence tapered to around 70-80% in the Guiana Coastal Plain, and only a few localized areas, particularly in savanna mosaics, showed weak or no signal.

The region's ecological heterogeneity, encompassing floodplain forests, natural grasslands, and savanna formations, means fire impacts vary considerably. Some savanna systems are naturally adapted to low-intensity surface fires (Alvarado et al., 2020; Pivello et al., 2021), but increased frequency and intensity of burning can overwhelm their resilience. Fire-sensitive ecosystems, such as humid forests and wetlands, are even more vulnerable, with increased fire pressure posing a long-term threat to ecosystem stability and biodiversity (Alvarado et al., 2020), and it is these ecosystems where anthropogenic climate forcing is most likely causing increase in burning.

For regional BA totals, the likelihood that socioeconomic drivers increased BA was 47% (**Figure 14**), indicating no clear signal that human landscape modification influences the extent of burning in seasons like early 2024. The estimated Amplification Factor was 1.08, but with a wide 90% confidence interval of 0.44 to 7.21 (**Table 5**). The wide confidence range,from potential halving of BA to a seven-fold increase, indicates that our model finds socioeconomic drivers to have a highly uncertain influence on regional fire activity during this period. This uncertainty likely reflects both the limited resolution of the socioeconomic variables used (e.g. population density, broad land cover classes) and the challenge of capturing the complex ways that human activities interact with fire. It is also possible that opposing effects such as suppression in one area versus ignition pressure in another, could be offsetting each other in regional statistics, though the modelling framework does not resolve these interactions explicitly.



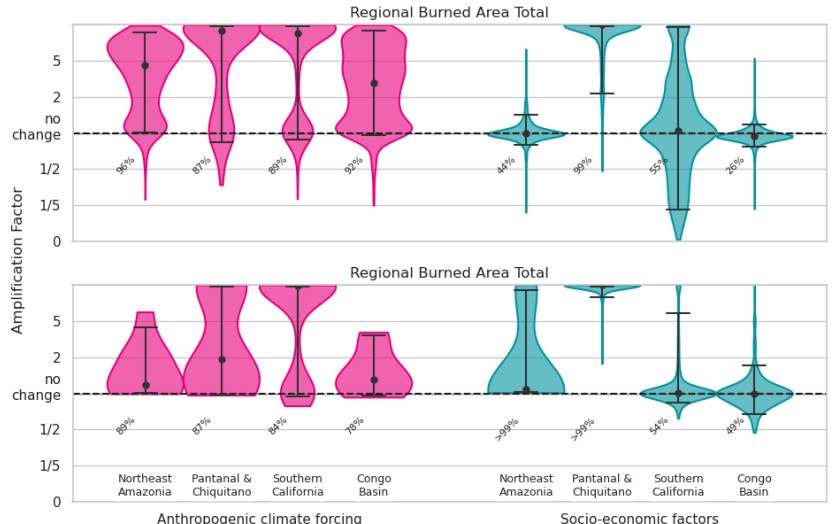

**Figure 14:** Probability density of the Amplification Factor (AF) for each region, showing how different factors influenced the extent of burning for each focal region. The top panel displays results for the entire region, while the bottom panel focuses on sub-regional extremes (defined as the grid cells in the top 5% of BA fraction). Anthropogenic climate forcing targets the 2024/25 focal moths using NRT set up with counterfactuals using all HadGEM ensemble members; socioeconomic factors uses the ISIMIP set up, looking at increased likelihood of 2024/255 like events in 2003-2019 with climate trends removed vs 1900-1917. An AF greater than 1 indicates that the factor contributed to an increase in burned area extent; a AF less than 1 indicates a reducing influence; a value near 1 suggests no change. Vases show probability distribution of AF, dots within each vase show central estimate and bars show 90th percentile confidence range The percentages lower left of each vase shows the likelihood of each factor increasing burned area.

### 5.2.2.2.      *Pantanal and Chiquitano*

The Pantanal and Chiquitano regions showed one of the strongest anthropogenic climate change signals of all focal regions studied here or in previous reports (Jones et al., 2024b). The likelihood that anthropogenic climate forcing increased the observed regional BA is estimated at 88% (**Figure 14**), indicating anthropogenic climate forcing likely drove an increase in BA (**Table 5**). The total BA was 34.5 times higher (our *amplification factor*) in the factual ensemble than in the counterfactual, although the wide uncertainty range of 0.84 to 100 suggests the effect of anthropogenic climate change could range from minimal to extremely large (**Table 5**). When internal meteorological variability is removed (using ensemble-mean), the estimated amplification factor remains largely unchanged. The model-based risk ratio for the event is 3.3, meaning the observed extent was roughly three times more likely due to anthropogenic climate change.

Climate influence was relatively consistent  across the region (**Figure 15**). Uniformity in attribution results may reflect the broad scale influence of anthropogenic climate change. It also suggests that climate change is amplifying fire risk even in areas with relatively intact ecosystems or seasonal wetlands, underscoring the vulnerability of these landscapes to the ongoing warming. However, the wide range in uncertainty highlights the need for improved observational data and better representation of fuel-moisture dynamics in fire-prone wetland mosaics such as Pantanal.

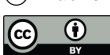



Socioeconomic factors show a very strong role for direct human influence in shaping BA anomalies during 2024-like events in the Pantanal and Chiquitano region. At the regional scale, the likelihood that socioeconomic factors increased BA is 99%, with an estimated amplification factor (AF) exceeding 100 (90% confidence interval: 2.12 to 100). This means that even under conservative estimates, human activity at least doubled BA during comparable fire years. In sub-regional extremes, the Amplification Factor range is even more extreme with a central estimate of more than 100 (lower 90% confidence bound of 16.24), with a similarly high likelihood (>99%) that human activity contributed. This implies that the vast majority of burning in these most severely affected areas was directly linked to socioeconomic drivers and would have been extremely unlikely in their absence.

These results confirm that direct human influences, such as land use effects and human ignition sources, can be as significant, or more so, than climate change in raising the likelihood of extreme wildfire events in the Pantanal-Chiquitano region. This is particularly important and promising because these factors can be directly addressed through local policies, incentives and enforcement actions, offering clear and locally isolated pathways for intervention and risk reduction alongside global action on climate change. The consistency between regional and sub-regional attribution indicates that these influences are not just diffuse but are concentrated in areas of greatest impact. Even the lower bounds of the confidence intervals provide compelling evidence that anthropogenic pressure substantially elevated fire outcomes.

These results agree with a growing body of evidence pointing to compounding non-linear effects of human and climatic drivers in the Pantanal (Marques et al., 2021, Barbosa et al., 2022, Santos et al., 2024). While this attribution includes some of the human drivers identified in the region, such as land use change, other key drivers, like wetland degradation and water extraction (which can intensify fire risk by drying out the landscape; Barbosa et al., 2022, 2025b), are not captured here.

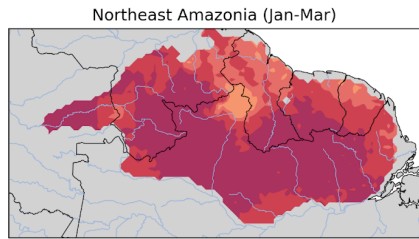

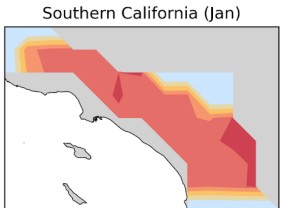

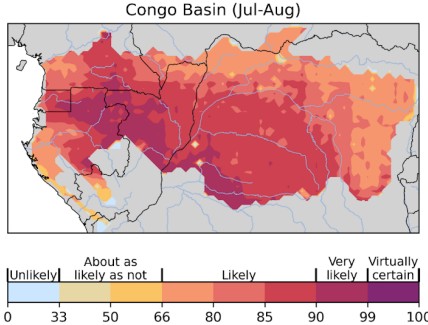

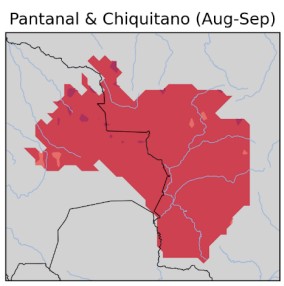





**Figure 15:** Regions where anthropogenic climate forcing most likely influenced fire activity during the 2024-25 fire season, based on the ConFLAME Near Real-Time setup. Maps show the probability that burned area (BA) was higher in the factual (climate change-influenced) scenario compared to the counterfactual (no climate change) scenario-based on the proportion of ensemble members where BA was greater in the factual than in the counterfactual scenario.. Results are shown for focal fire periods in each region: January-March 2024 for Northeast Amazonia; August-September 2024 for the Pantanal and Chiquitano; June-August 2024 for the Congo Basin; and January 2025 for Southern California. Colourbar descriptive labels are based on IPCC uncertainty definitions (Mastrandrea et al. 2010).

### 5.2.2.3. *Southern California*

Anthropogenic climate forcing likely contributed to the high levels of BA observed in Southern California in January 2025, with a likelihood of increased burning of 89%. The amplification factor (AF) was estimated at 24.8, though with a wide uncertainty range (90% confidence interval 0.89 to 100), indicating that the influence could have ranged from negligible to extremely large. Despite this spread, the ensemble-mean counterfactual results largely agree, reinforcing confidence that anthropogenic climate forcing increased the likelihood of the event. The risk ratio of 2.3 suggests that similar fire conditions are more than twice as likely in the present-day climate compared to a scenario without climate change. This elevated risk was in January, outside the region's typical peak fire season, suggesting that anthropogenic forcing may be expanding the seasonal window during which large fire events can occur.

There is no clear evidence that socioeconomic factors occurring on the landscape increased the likelihood of January 2025-like regional BA in Southern California during 2002-2019. The estimated likelihood of an increase is 55%, with a highly uncertain amplification factor (AF = 1.04 [0.17-85.58]). As with the climate attribution, this likely reflects the small size of the region and limited signal in long-term data.

### 5.2.2.4. *Congo Basin*

Anthropogenic climate forcing likely increased the total area burned across the Congo Basin during June to August 2024. The likelihood of an increase is estimated at 92%, with an amplification factor (AF) of 2.69, meaning the event-scale BA was nearly three times higher than it would have been without forcing. However, there remains some uncertainty: while the best estimate points to a substantial increase, the range spans from a very small influence to a more than 30-fold increase (90% confidence range of 0.96 to 33.96).

When we account for internal climate variability by averaging across all ensemble simulations (rather than using only the observed event conditions), the signal strengthens substantially. In this case, anthropogenic climate change appears to have increased BA by a factor of 15 (90% confidence range: 0.97 to over 100), with a risk ratio of 2.6, which shows a more consistent pattern of increased fire risk due to long-term warming and drying trends.Unlike other regions, where most of the uncertainty stems from how fire responds to environmental conditions, in the Congo Basin uncertainty in the meteorological response to climate change itself plays a larger role.

The influence of climate change also varied significantly within the region. The strongest signal appears in the southern parts of the Congo Basin, particularly the Southern Moist Forests, where our modeling frameworks suggest climate change very likely (90-95% likelihood), using IPCC terms definition **Figure 15**)  increased BA. Further north, in the DRC's northern moist forests, the likelihood was lower (50-80%), and in the Southern Gabon



transition forests, there was little to no signal. These spatial differences may reflect varying sensitivities to rainfall patterns, fuel conditions, or other landscape features, and highlight the importance of region-specific analysis.

There is no clear signal that socioeconomic factors increased BA during the June-August 2024 fires in the Congo. Across the region as a whole, the likelihood of increased burning due to population density and land-use change was 26%, with an amplification factor (AF) of 0.94 (90% confidence interval: 0.70 to 1.17), suggesting a small or even slightly dampening influence. At the sub-regional level, attribution remains uncertain. The likelihood of increased BA in the most affected grid cells was estimated at 62%, with an AF of 1.00 [0.68-1.69].





**Table 5:** Summary of attribution results for burned area (BA) and fire weather indices during key fire events across Northeast Amazonia (Jan-Mar 2024), Pantanal-Chiquitano (Aug-Sep 2024), Congo Basin (Jul 2024), Southern California (Jan 2025). Values are reported for both burned area (BA) across the full region and sub-regional extremes –the areas that saw the most burning (see Figure 5). Metrics include the amplification factor (AF; the ratio of BA under the influence of the assessed factor relative to the counterfactual), risk ratio (RR) of fire weather index during the events, and % change in annual mean (background) BA. Results are shown for different configurations: anthropogenic meteorological forcing (using near real-time and ensemble-mean setups), total climate forcing, and socio-economic factors. Values are reported as median [5th-95th percentile] ranges, with likelihoods indicating the confidence that the factor contributed to increased burning or extreme fire weather. Colours indicate IPCC-defined confidence or likelihood categories (Mastrandrea et al., 2010). Where likelihoods are not explicitly provided, colours reflect the lowest plausible category based on the reported confidence range.

| Variable | Metrics | Sources | Northeast Amazonia | Pantanal and Chiquitano | Southern California | Congo Basin |
|---|---|---|---|---|---|---|
| **Anthropogenic climate forcing** | | | | | | |
| Fire Weather Index | **Risk Ratio (RR)** | **HadGEM** | 31.96-72.64 | 4.16-5.45 | | 3.04-8.00 |
| | RR | CanESM5 | 1.9 [1.5, 53.3] | 12.3 [3.4, 76.9] | 1.7 [1.6, 1.8] | 1.3 [0.7, 1.7] |
| | RR | WWA | | 4.6 [1.1, 20]* | 1.37 [0.48, 3.6]+ | |
| | Intensity Delta | WWA | | +39% [13%, 71%]* | +5.7% [-10, 27]+ | |
| Burned Area (BA) | **Amplification factor (AF)** | **ConFLAME/ HadGEM ensemble** | 4.33 [1.02, 25.32] | 34.47 [0.84, >100] | 24.79 [0.89, >100] | 2.69 [0.96, 33.96] |
| | **RR** | | 2.1 | 3.3 | 2.3 | 1.6 |
| | AF | ConFLAME/ HadGEM mean | 13.25 [1.02, >100] | >100 [0.82,- > 100] | >100 [0.82, >100] | 14.76 [0.97, > 100] |
| | RR | | 2.7 | 3.5 | 2.9 | 2.6 |
| Areas of highest BA | AF | ConFLAME/ HadGEM ensemble | 1.17 [1.01, 5.13] | 1.91 [0.98, >100] | >100 [0.95, >100] | 1.29 [0.96,-3.32] |
| | R | | 2.2 | 2.4 | 2.9 | 1.8 |
| | AF | ConFLAME/ HadGEM mean | 1.11 [0.95,1.94] | >100 [0.96, >100] | >100 [0.98, >100] | 1.24 [0.84,-1.55] |
| | RR | | 1.6 | 3.7 | 3.8 | 1.3 |





| Variable | Metrics | Sources | Northeast Amazonia | Pantanal and Chiquitano | Southern California | Congo Basin |
|---|---|---|---|---|---|---|
| **Total climate change** | | | | | | |
| **BA** | **AF** | ConFLAME/ISIMIP | 1.01 [0.88,1.15] | >100 [2.73, >100] | 1.07 [0.68, 2.83] | 1.08 [0.95, 1.43] |
| Areas of highest BA | AF | ConFLAME/ISIMIP | 1.02 [0.94,1.13] | >100 [4.92, >100] | 1.00[0.91, 1.86] | 1.14 [0.87, 3,02] |
| Background BA | | FireMIP | -6% [-11%, - 2%] | 10% [6, 15%] | 7% [2%, 12%] | 54% [45%, 63%] |
| **Socio-economic factors** | | | | | | |
| **Burned Area** | **AF** | ConFLAME/ISIMIP | 0.99 [0.8, 1.41] | >100 [2.12, >100] | 1.04 [0.17, 85.59] | 0.94 [0.7, 1.17] |
| Max. Burned Area | AF | ConFLAME/ISIMIP | 1.02 [1.07, 1.13] | >100 [16.24, >100] | 1.00 [0.85, 6.,65] | 1.00[ 0.68, 1.69] |
| Background Burned Area | | FireMIP | 10% [3%, 17%] | -7% [-12%, -2%] | -3% [-7%, -1%] | -16% [-21%, -11%] |
| **All forcings** | | | | | | |
| Burned Area | AF | ConFLAME/ISIMIP | 0.99 [0.81, 1.47] | 1.08 [0.44, 7.21] | 1.05 [0.26, 64.26] | 1.01 [0.86, 1.42] |
| Max. Burned Area | AF | ConFLAME/ISIMIP | 1.01 [0.96, 1.10] | 1.04 [0.98, 8.26] | 1.00 [0.86, 12.16] | 1.06 [0.73-4.44] |
| Background BA | | FireMIP | 1% [-6%, 9%] | 3% [-2%, 9%] | 2% [-2%, 7%] | 25% [18%, 33%] |

| Virtually certain | Very likely | Likely | About as likely as not | Unlikely, | Very unlikely | Exceptionally unlikely |
|---|---|---|---|---|---|---|
| > 99% | >90% | >66% | 33-66% | <33% | <10% | <1% |

*WWA results for June DSR and

+ January max FWI





### 5.2.3. Background Changes in Burned Area this Century

We assess how climate and socio-economic drivers have influenced changes in background levels of BA for each focus region using the global fire model attribution framework introduced by Burton, Lampe et al. (2024), adapted this year to match the specific geographic areas analysed in this report (see methods in **Section 5.1.4**). Results represent the change in median monthly BA during 2003-2019 compared to a counterfactual scenario in which anthropogenic climate change or changes in socio-economic factors were removed. This is distinct from our analyses focussing on the attribution of individual focal events in **Sections 5.2.1** and **5.2.2**.

#### 5.2.3.1. Northeast Amazonia

Total climate forcing led to a modest but consistent decrease in background BA between 2003-2019, with a median change of -6% [-11%, -2%] compared to a counterfactual without climate change. Unlike the earlier attribution method (**Section 5.2.2**), which focused on extreme 2024-like events, this model captures long-term, background fire activity, including broader fuel-climate interactions.

The reduction in BA may reflect increased moisture or changes in vegetation structure that reduce flammability, though the exact mechanism is unclear. Recent observational analyses suggest a rise in wet-season (December to May) rainfall and a reduction in dry days in northern Amazonia over the past two decades (Barichivich et al., 2018; Almeida et al., 2017), which could contribute to these trends if captured in the climate inputs. The underlying models used in this attribution framework used here also features tighter coupling between vegetation, climate, and fire than the event-based approach, which may explain some of the differences, though it remains difficult to determine whether these are due to improved fuel representation or simply reflect a contrast between background and extreme conditions.

Socioeconomic changes are estimated to have increased the background BA in Northeast Amazonia by +10% [3%, 17%] in 2003-2019 compared to 1901-1917. This signal aligns well with the earlier analysis of 2024-like events (**Section 5.2.2.1**) but is more narrowly constrained, reinforcing the role of human-driven changes as a key influence on regional fire activity, as identified in many previous studies. For instance, recent studies on land use and fire dynamics in the Amazonia region points to rising fire activity associated with expanding agricultural areas, secondary vegetation, and newly deforested areas (Silveira et al., 2022). Human activities remain the primary source of ignition, mainly through practices such as deforestation, pasture maintenance, and crop field burning, often intensified under dry conditions (Lapola et al., 2023).

#### 5.2.3.2. Pantanal and Chiquitano

We find a modest but robust signal of climate-driven change in background fire activity. Between 2003 and 2019, total climate forcing is estimated to have increased the average BA by 10% [6%, 15%]. The relatively narrow confidence range suggests strong model agreement and indicates that the region's area burned has already been measurably affected by long-term climatic shifts. This aligns with broader lines of evidence that highlight the Pantanal's vulnerability to changes in rainfall patterns and dry season intensity, which influence both fuel availability and flammability (**Section 4.2**). These findings are also consistent with attribution results for extreme events in 2024 (**Section 5.2.2.2**), which also showed a high likelihood of increased burning, albeit with greater uncertainty.





We estimate that socioeconomic drivers contributed a reduction in background BA of 7% [-12%, -2%] compared to pre-industrial conditions. This suggests that long-term changes in land use and management, including shifts in agricultural practices, may have contributed to a modest but consistent suppression of average fire activity over the past two decades. The attribution of socioeconomic influence on BA in the Pantanal presents an interesting contrast with the attribution of focal event BA in the previous section, which suggests that socioeconomic factors very likely increased BA (**Section 5.2.2.2**). This contrast may point to important temporal and functional differences:

- Long-term socioeconomic changes, such as improved fire control in settled areas or changes in land use, could suppress background fire activity.
- Yet, during extreme conditions, these same systems may fail to contain fires, or different areas (e.g.the interface between private properties and protected areas , Barbosa et al., 2022) may dominate the fire signal.

Still, the disagreement raises a cautionary flag. While the two methods target different timescales and use different models, their confidence intervals do not fully overlap, suggesting that at least one framework may be underestimating uncertainty or missing key processes. It also reinforces the importance of using multiple, independent lines of evidence in attribution work and, specifically for the Pantanal, shows that more work is needed to assess the balance between human impact on background vs extreme BA along with the modelling techniques used to assess this.

### 5.2.3.3. Southern California

In Southern California, the models attribute a +7% [2%, 12%] increase in median background BA to total climate forcing. This is consistent with the attribution results for 2025-like events (**Section 5.2.2.3**), though with higher confidence. The agreement across these distinct approaches, despite targeting different fire outcomes (seasonal extremes vs general background activity), provides additional confidence that long-term climate change is influencing baseline fire conditions in the region.

Socio-economic influences contributed a -3% change in background BA, with an uncertainty range of [-7%, 1%]. While not statistically significant, this result is more tightly constrained than those from the earlier analysis of 2025-like events. The modest downward influence may reflect intensifying suppression capacity, declines in human-caused fires due to fire-prevention policies including those targeted to electrical utilities (Jorge et al., 2025; Abatzoglou et al., 2020), or other urban interface factors, though uncertainty remains high.

### 5.2.3.4. Congo Basin

In the Congo Basin, we estimate that total climate change has driven an increase in mean annual BA of 54%, with a tight confidence range of [45%, 63%]. This makes it one of the most robust signals of climate influence across the background fire analyses. These results are consistent with, though slightly stronger and more confident than, the attribution using 2024-like extreme events. The agreement between methods strengthens confidence that climate change is already amplifying baseline fire activity in the region.

This signal likely reflects a clear climate influence on fire-conducive weather, particularly in the southern part of the basin (**Section 4.2.2.2.4**). While fuel limitations played a role in moderating fire spread (**Figure 12**), the background increase in BA appears strongly tied to meteorological shifts linked to climate change.

Socioeconomic influences appear to have played a moderating role in background fire activity across the Congo Basin. In our process-based model analysis, socioeconomic drivers, including changes in land use, land cover, and population, led to a 16% reduction in





background BA between 2003-2019, with a 90% confidence range of -21% to -11%. This suggests a consistent and substantial dampening effect on fire, possibly reflecting a combination of land fragmentation, land use conversion, or reduced fire use. These results are broadly in line with, though more confidently constrained than, the amplification factor estimated for 2024-like events in the previous attribution method, which indicated limited influence from socioeconomic factors.

# 6. Seasonal and Multi-Decadal Outlook

## 6.1. Methods

### 6.1.1. Seasonal Forecasts

#### 6.1.1.1. *Fire Weather Index*

In **Section 4**, we introduced the use of seasonal forecasts of FWI and examined how they performed during the focal events of the 2024-25 fire season. In this section, we present global FWI forecasts from the ECMWF's SEAS5 seasonal prediction system for the months June-August 2025, extending the same approach employed in **Section 4** throughout the boreal summer months of 2025 (see **Section 4.1.1.2.1** for methods).

#### 6.1.1.2. *Burned Area*

In **Section 4**, we introduced the use of seasonal forecasts of burned areas using a combination of weather driver and ML and examined how they performed during the focal events of the 2024-25 fire season. In this section, we present global BA forecasts from the same system for the months July-September 2025, extending the same approach employed in **Section 4** throughout the boreal summer months of 2025 (see **Section 4.1.1.2.2** for methods).

### 6.1.2. Multi-Decadal Projections

#### 6.1.2.1. *Fire Weather Index at Future Global Warming Levels*

To calculate how the risk of fire weather extremes might evolve with future warming, we apply the same framework described in **Supplementary Material S5.1.1** but instead of comparing recent climate to the past, we compare it to a set of global warming levels: 1.5 °C, 2.0 °C, 3.0 °C, and 4.0 °C above recent past climate (2016-2025).

For each level of warming, we identify years in the CanESM5 ensemble where the smoothed 11-year running global mean temperature aligns with the target level, and then assess the frequency of extreme 7-day FWI events in those years, as per Liu et al. (2023b) and similar to Otto et al. (2018). Comparing this to the 2016-2025 climate baseline gives us a forward-looking set of Risk Ratios (RR) — RR1.5, RR2.0, etc. These indicate how much more likely such extremes become as the planet warms.

As with the attribution to past climate (**Section S5.1.1**), uncertainties are captured through bootstrapped confidence intervals, enabling meaningful comparison of future risks even when rare extremes are involved.

#### 6.1.2.2. *Burned Area in Future Emissions Scenarios*

In order to project future changes in BA, we extended the ConFLAME ISIMIP3a modelling approach used in **Section 5.1.3** to future decades under Shared Socioeconomic Pathway (SSP) scenarios SSP126, SSP370, and SSP585, following a similar protocol to UNEP



(2022a). We use the same optimised model as in **Section 5.1.3**, but here we employ
bias-corrected global climate model (GCM) outputs from ISIMIP3b (Frieler et al. 2025) for
prediction. While ISIMIP3a uses reanalysis data for historical analysis, ISIMIP3b employs
GCM data to project future climates and is designed for usage cases requiring a seamless
continuation of the historical period into future scenarios.
ISIMIP3b utilizes five bias-corrected GCMs, including historical model output up to 2014 and
future scenarios from 2015-2100 under the three SSPs. ISIMIP3b uses surface-based
meteorological outputs from ScenarioMIP simulations, which include future forcings from
greenhouse gases, aerosols, land-use change, and short-lived climate forcers. The five
GCMs used are: GFDL-ESM4 (Held et al., 2019), IPSL-CM6A-LR (Boucher et al., 2020),
MPI-ESM1-2-HR (Mauritsen et al., 2019), MRI-ESM2-0 (Yukimoto et al., 2019), and
UKESM1-0-LL (Tang et al., 2019; Sellar et al., 2019). As part of ISIMIP3b, each GCM is
bias-corrected as described in Lange (2019).
Future ISIMIP3b projections for socioeconomic drivers such as population density or land
use change were not available at time of analysis. As such, our simulations exclude future
changes in ignition sources or direct land-use modification on both fire and vegetation. To
simulate vegetation structure and fuel availability, the JULES-ES dynamic vegetation model
was run offline, driven by surface climate variables from each of the five bias-corrected
GCMs under each SSP scenario, and scenario-specific CO2 concentrations to represent
CO2 fertilization, along with prescribed nitrogen deposition but excluding changes in fertiliser
application, along with prescribed nitrogen deposition but excluding changes in fertiliser
application. The land cover output from JULES-ES was then bias-corrected (using the same
mapping procedure as **Section 5.1.3**, based on biases between JULES-ES driven by
reanalysis and VCF observations) to maintain consistency with the GCM bias-correction
procedures. Our approach provides a probability distribution of future BA representing the
uncertainty range from cross-model (GCM) spread in the response of climate and vegetation
to emissions for each scenario and year in the period 2010-2100. Years 2010-2014 were
adopted from the historical experiment for each GCM, and post-2014 from branched SSP
and model specific projections. We describe future changes as significant if the range across
GCM projections for a future period does not overlap with the range given by the GCMs for
2010s.
Using this driving data, we generate 1,000-member ensembles for each region and each
GCM/SSP combination, using the trained ConFLAME-ISIMIP model described in **Section
5.1.3**. For each 10-year period, we calculate the likelihood of extreme fires by determining
the fraction of years within each ensemble member where burned area during the event
months exceeds that of the observed focal event. We then average this exceedance fraction
across all 1,000 ensemble members to estimate the likelihood for that decade. This process
is repeated for each GCM and SSP.
For decades beyond 2010s, we then calculate the increase in the likelihood of 2024/25-like
events by taking the ratio of the exceedance frequency in each future decade relative to the
2010s baseline. This is analogous to the risk ratio used in **Section 4**, where the future period
acts as the "factual" and 2010s as the "counterfactual" baseline. Following methods outlined
in **Section 4**, we perform this analysis for the entire region and for "sub-regional extremes" -
the grid cells with the top 5% of BA.
Lastly, we calculated the integrated probability of experiencing a fire event of similar
magnitude to our target region within the expected lifespan of a citizen born in 2023 (the
year of the latest estimate). According to UN population statistics (United Nations Population
Division, 2023), life expectancy at birth is 75.8 years for Brazil, 79.3 years for the USA, and
61.9 years for the Democratic Republic of the Congo (DRC). While the Northeast Amazonia
and Congo Basin regions span multiple countries, most fire anomalies in these regions





occurred in Brazil and the DRC, respectively (**Figure 5**). To account for years beyond 2100 in the life expectancy of Brazil and the USA, we extrapolated the annual trend in event probabilities. The integrated probability is calculated as one minus the product of the annual probabilities of not experiencing a fire event like the focal event, across each year from 2025.

## 6.2.  Results

### 6.2.1.  Seasonal Forecasts of Fire Weather Index and Burned Area Anomalies

As of mid-2025, neither La Niña nor El Niño conditions are present in the tropical Pacific. Instead, the climate system has entered an ENSO-neutral phase, according to the latest report from the National Oceanic and Atmospheric Administration (NOAA, 2025c). This neutral phase is expected to persist through the remainder of summer, and into at least early autumn. While neutral ENSO conditions typically indicate a reduced influence of Pacific sea surface temperature anomalies on global weather patterns, the persistence of anomalously warm ocean conditions and other climate drivers may continue to exert significant influence on regional and global climate variability in the months ahead (Frölicher and Laufkötter, 2018).

May 2025 was the second-warmest May on record globally, with an average temperature of 15.79 °C, 0.53 °C above the 1991-2020 climate and 1.4 °C above pre-industrial levels (Copernicus Climate Changes Service, 2025). While this marked a brief drop below recent consecutive months exceeding 1.5 °C from pre-industrial record, it still reflects the persistent trend of global climate warming (Horton, 2025). Unusually low rainfall and soil moisture across northwestern Europe, including the UK, reached their lowest levels since 1871. This raises serious concerns about crop failures, potential water shortages and wildfire risk (European Commission Joint Research Centre, 2025; UK Environment Agency, 2025). Similar conditions were reported in the US, particularly across Arizona and Texas, where exceptional drought levels led to reservoir depletion, strict water restrictions, and increased wildfire activity (National Centers for Environmental Information, 2025; National Interagency Fire Center, 2025).

Starting from May, and according to the outlook for the Northern Hemisphere boreal summer of 2025 (June-July-August), anomalous fire weather conditions are anticipated across several key regions with high levels of confidence (in places reaching 80 %). Anomalous fire danger season is expected in Canada, US western states (also see National Interagency Fire Center, 2025), northeast Europe (notably the UK), and parts of Siberia (**Figure 16**). In the equatorial zone, persistent dryness and hydroclimatic anomalies are expected to increase fire danger (confidence level of 60% and higher) in Northeast Amazonia, the Congo Basin, and the Himalayan foothills (affecting areas of India and Nepal). In contrast, a relatively quiet fire season is projected for the Southern Hemisphere, with only Chile and southern Australia showing fire-prone conditions at a moderate level of confidence (>50%).

The BA anomaly forecast (bottom panel of **Figure 16**) displays a distinct pattern from that of FWI, as it models the expected fire response conditioned on both coincident and antecedent climate variables, based on region-specific statistical relationships. For instance, elevated probabilities of above-median BA are projected in the western part of South America, southern California, localized areas of Central America, and central North America. In central Asia, medium-to-high probabilities emerge, particularly in the eastern regions. In Africa, significant signals are observed over the central continent, while in Australia, elevated probabilities are mainly found in the northern regions. Over central Europe, despite a high FWI forecast, limited historical fire activity prevents reliable calibration of the climate-fire model, and therefore no BA forecast is issued for this region.





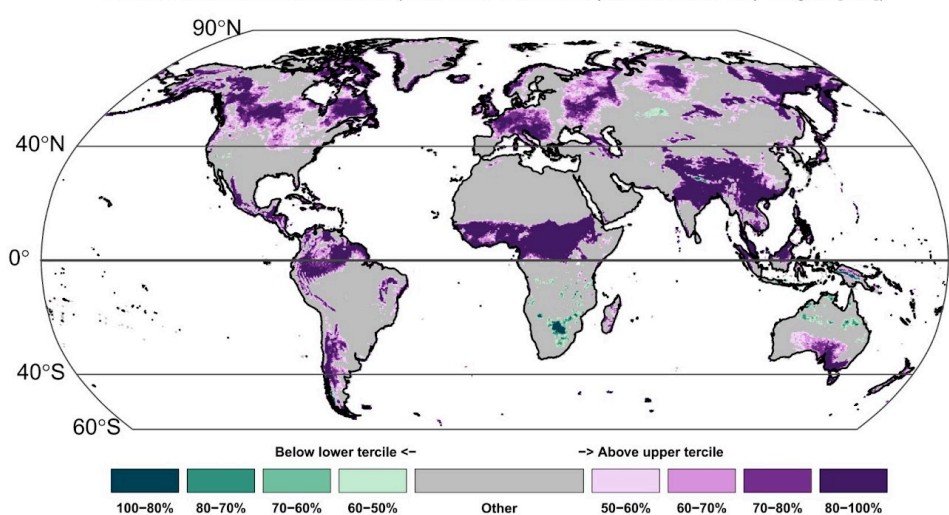

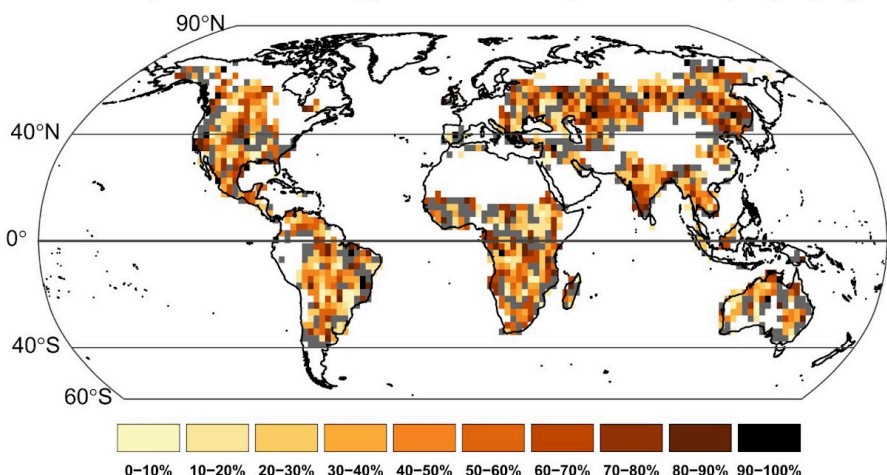

**Figure 16:** Seasonal prediction of Fire Weather Index (FWI) and burned area (BA) anomalies for the boreal summer of 2025 (June-July-August). Both forecasts are issued in June 2025 and are presented in probabilistic terms: FWI prediction shows the likelihood for increased (above the upper tercile) or decreased (below the lower tercile) fire-weather conditions; whereas BA prediction shows the probability of BA anomalies being above the climatological median. Grey areas are masked where insufficient BA statistics are available to perform the predicted mean.



### 6.2.2. Future Changes in Likelihood of Extreme Fire Weather Events

In three of the focal regions where climate change significantly increased the likelihood of a 2024-25-level fire weather event (**Section 5.2.1**), even greater increases are projected under future warming levels of 1.5 °C, 2 °C, 3 °C, and 4 °C (**Figure 17**).

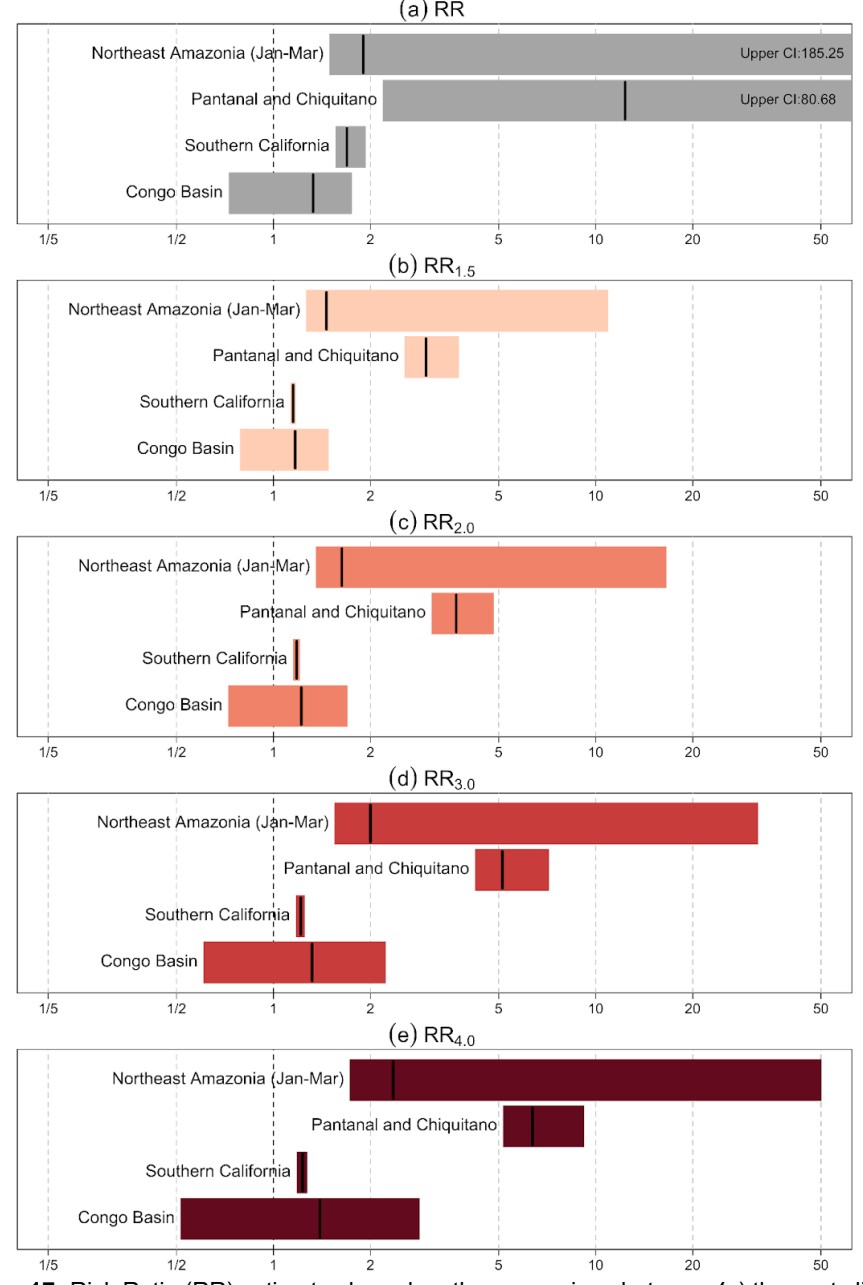

**Figure 17**: Risk Ratio (RR) estimates based on the comparison between **(a)** the past climate of 1850-1859 and the recent climate of 2016-2025, **(b)** the recent climate of 2016-2025 and



the period that global mean surface temperature (GMST) reached **(b)** 1.5 °C, (c) 2 °C, **(d)** 3 °C and **(e)** 4 °C for the four extreme wildfire events between 2024 and early 2025 using CanESM5. Bars show 95% confidence intervals (CIs) and central values are shown in bold.

### 6.2.2.1. Northeast Amazonia

In Northeast Amazonia the increased fire weather risk found in **Section 5.2.1.1** during January-March is projected to continue rising under future warming, with increases in probability of 1.5 (95% CI: 1.3-10.8), 1.6 (1.4-16.3), 2.0 (1.6-31.4) and 2.4 (1.7-49.5) at 1.5 °C, 2 °C, 3 °C, and 4 °C of warming, respectively . Compared to southern Amazonia, fires in Northeast Amazonia have gathered less attention from the scientific community and little is known about how future changes in fire weather conditions may impact this region.

Amazonia spans multiple countries, making coordinated fire governance particularly challenging. These countries often have differing political priorities and economic interests, which shape land use policies, enforcement capacity, and investment in fire monitoring and response systems. Such disparities can hinder the implementation of integrated fire management strategies, especially in border regions where transboundary fires may occur but fall under fragmented jurisdictional and institutional frameworks. These institutional and policy asymmetries introduce further uncertainty about how fire risk will evolve in a warming climate. As fire weather intensifies, the region's unique fire season and cross-border governance dynamics should be explicitly considered in fire risk assessments and regional adaptation strategies.

### 6.2.2.2. Pantanal and Chiquitano

The Pantanal and Chiquitano region, which showed the largest historical increase with 4.75 (95% CI. 4.2-5.5, **Section 5.2.1.2**), is set to continue to increase with global warming, with projected increases in probability of 3.0 (95% CI: 2.6-3.6), 3.7 (3.2-4.6), 5.1 (4.4-6.5), and 6.4 (5.4-8.3) at 1.5 °C, 2 °C, 3 °C, and 4 °C of warming, respectively (**Figure 17, b-e**). This is especially concerning for the Pantanal and Chiquitano, where fires are strongly driven by climate, particularly through extreme (Silva et al., 2022; Barbosa et al., 2022) and compound events (Ribeiro et al., 2022; Libonati et al., 2022). The ongoing reduction of wetlands in the Pantanal, often replaced by flammable grasslands (Damasceno-Junior et al., 2021), combined with the projected increase of fire weather conditions (Feron et al., 2024), may indicate a permanent shift in the landscape and its fire regime. This increases the vulnerability of fire-sensitive vegetation and wildlife habitats, while also threatening economic activities that rely on seasonal flooding.

### 6.2.2.3. Southern California

Southern California shows a similar pattern, with the likelihood of 2024-25 extreme fire weather being about 1.7 times higher (95% CI: 1.6-1.8) than in the past, and projected increases in likelihood ranging from 1.1 to 1.3 with rising global temperatures.

### 6.2.2.4. Congo Basin

In contrast, the Congo Basin shows a more modest and statistically non-significant change, with the likelihood of a similar extreme fire weather event to that of the 2024-25 season increasing by a factor of 1.3 from the past to the present. Future projections suggest a wide but uncertain range of change, between 0.5 and 2.7 depending on the warming level.





### 6.2.3. Future Changes in Likelihood of Extreme Fire Events

#### 6.2.3.1. Northeast Amazonia

By the 2040s, under SSP585, the likelihood of an event similar to those of the 2024-25 season increases modestly but significantly to 0.12-0.14%, a ~17% increase in frequency compared to the 2010s (**Figure 18; Table 6**). Other scenarios show smaller or even negligible changes over this period. By the end of the century, however, all scenarios project notable increases in event frequency. SSP585 shows the largest rise, with the probability of such an event nearly doubling (up to 1.92 times more frequent). SSP370, reflecting current emissions trajectories, projects a 1.19-1.57 times increase. In contrast, SSP126 illustrates the mitigation potential of low-emission pathways, limiting increases to just 1.09 times (under 10% increase) by 2100, significantly lower than under higher-emission scenarios. SSP370 only clearly diverges from SSP126 by late century (2090s), though the potential for larger increases appears earlier (**Figure 18**). This divergence between the two scenarios is especially pronounced when focusing on areas with the highest BA (top 5% of grid cells, **Figure S29**). These regions of extreme burning could see a doubling in fire extent by mid-century and at least doubling (potentially tripling) by 2100 under SSP370, with substantial overlap with SSP585 projections (where extreme BA could almost quadruple).

By 2100, SSP126 still shows marginal increases in the likelihood of BA events such as those in 2024 (**Figure 18**), though sub-regional extreme BA see much less significant change (**Table 6; Figure S29**), with frequency ranging from slight decreases (by a factor of 0.91) to modest increases (1.34).

These increases are mainly driven by projected declines in moisture availability (**Figure 18 Figure S29**). Although fuel availability is expected to decline somewhat, this only marginally offsets the rise in extreme BA likelihood across the region and has virtually no mitigating effect on the highest BA areas. No changes in fuel are statistically significant in our projections.

Most regions of Northeast Amazonia see increases in January-March (JFM) average BA by 2100 (**Figure S32**). However, under SSP126, increases in the north, French Guiana, Suriname, and Guyana, are less certain and, if they occur, are smaller. This is reflected in a decreased frequency of extremes across these areas (**Figure S32**). Under SSP370, climate change drives widespread increases in BA, with corresponding rises in extremes nearly everywhere except Roraima (Brazil). Most of Brazil and Venezuela are very likely to see increases in BA even under SSP126, with some moist regions showing rises in extremes under SSP126 and widespread increases under SSP370. Results for SSP585 are similar to those of SSP370, with widespread increases in BA and extremes throughout the region. Importantly, increases in extremes begin in some areas in the near future (**Figures S30-31**). By the 2030-2040s, Amapá (Brazil), northern Pará (Brazil), and southern Suriname are projected to experience more frequent extreme BA events and increased BA under the SSP585 scenario (**Figure S30**). Increases in BA are less certain but still likely under SSP370, with mitigation under SSP126 helping to limit these trends.

Finally, we explored what this means for people's lived experience (**Figure 19**). A person born 75.8 years ago (Brazil's current life expectancy) would have had a 33-36% likelihood of witnessing a fire event like January-March 2024 during their lifetime. This suggests that, although anthropogenic changes have increased the likelihood of such fires (see **Section 5**), these events remain far from certain. Even the modest increases in frequency projected under SSP126 would raise that lifetime likelihood to 41-55% for someone born today (i.e, 2025-2021). Under SSP370 (our current path), the chance rises substantially to 52-69%, and under SSP585, to 55-76%. There is also a substantial rise in the probability of experiencing

multiple such events within a lifetime, for example, under SSP370, there is a 17-32% chance
of seeing two such events, compared to just 6-8% for those born in the 1940s.

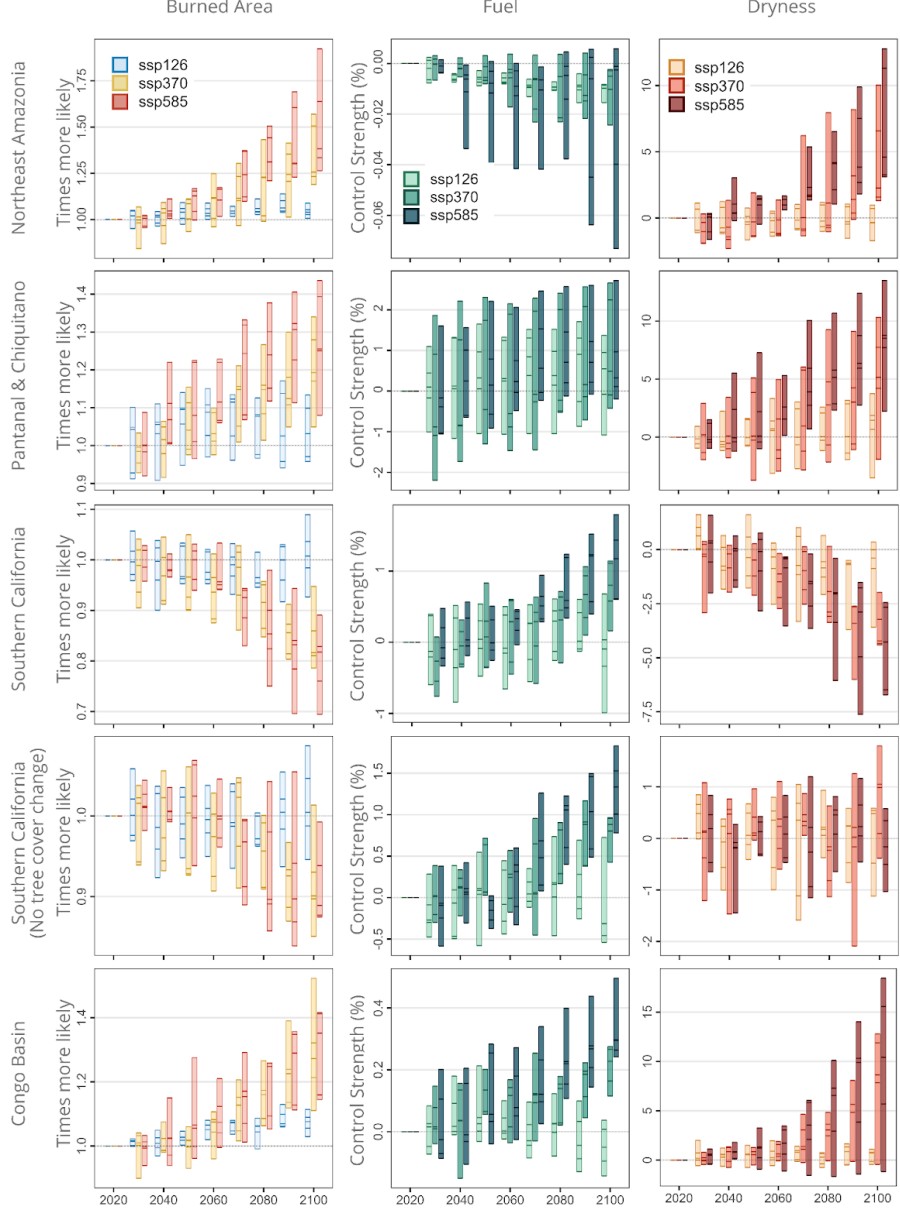

**Figure 18:** Future projections from ConFLAME of the change in likelihood of BA extent of
the magnitude seen in the 2024-25 season, along with the contribution of fuel and moisture
conditions in years in which BA exceeds the 2024-25 thresholds. Each set of bars shows
changes for each decade relative to the 2010-2020 baseline, with each bar representing a
different SSP scenario and the spread of bars indicating the variation across GCMs, with
individual bars representing different GCMs.

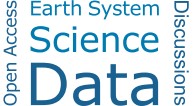

**Table 6:** Summary of the likelihood of extreme events today using reanalysis 'factual' and into the future using bias-corrected GCMs for our focal events identified in **Section 4.4.3**. Min and max report range across GCMs. We also determine how much more frequent the events will be at two different time horizons based on each model's likelihood in the future projections over likelihood during 2010-2020. Asterisks (*) indicate non-significant changes from 2010-2020 values. Colours show linear increase of likelihood (orange) and frequency (blue for less frequent, orange for more), where darker shade indicates higher values. The top half of the table displays projections using BA over the entire region, while the bottom shows projections for sub-regional extremes (grid cells with the top 5% BAs).

**All Region**

| Region | SSP | Represents | Likelihood(%/year during focal months) | | | | | | How much more frequent (multiplier) | | | |
|---|---|---|---|---|---|---|---|---|---|---|---|---|
| | | | 2010-2020 | | 2040-2050 | | 2090-2100 | | 2040-2050 | | 2090-2100 | |
| | | | min | max | min | max | min | max | min | max | min | max |
| Northeast Amazonia | Factual | observed | 0.073 | | | | | | | | | |
| | SSP126 | strong mitigation | | 0.12 | 0.12* | 0.14* | 0.12 | 0.13 | 0.97* | 1.09* | 1.01 | 1.09 |
| | SSP370 | middle-of-the-road | | 0.12 | 0.12* | 0.13* | 0.15 | 0.19 | 0.93* | 1.11* | 1.19 | 1.57 |
| | SSP585 | no mitigation | | 0.12 | 0.12 | 0.14 | 0.15 | 0.23 | 1 | 1.17 | 1.26 | 1.92 |
| Pantanal & Chiquitano | Factual | observed | 0.19 | | | | | | | | | |
| | SSP126 | strong mitigation | | 0.09 | 0.09* | 0.11* | 0.09* | 0.11* | 0.95* | 1.14* | 0.96* | 1.13* |
| | SSP370 | middle-of-the-road | | 0.08 | 0.09* | 0.11* | 0.1 | 0.12 | 0.98* | 1.15* | 1.05 | 1.34 |
| | SSP585 | no mitigation | | 0.08 | 0.09* | 0.11* | 0.1 | 0.13 | 0.97* | 1.22* | 1.08 | 1.44 |
| Southern California | Factual | observed | 0.38 | | | | | | | | | |
| | SSP126 | strong mitigation | | 0.34 | 0.33* | 0.36* | 0.31* | 0.38* | 0.95* | 1.03* | 0.93* | 1.09* |
| | SSP370 | middle-of-the-road | | 0.34 | 0.3* | 0.37* | 0.27 | 0.33 | 0.9* | 1.05* | 0.79 | 0.95 |
| | SSP585 | no mitigation | | 0.34 | 0.32* | 0.36* | 0.23 | 0.31 | 0.94* | 1.03* | 0.69 | 0.89 |
| Southern California - no tree cover change | Factual | observed | 0.42 | | | | | | | | | |
| | SSP126 | strong mitigation | | 0.38 | 0.38* | 0.4* | 0.37* | 0.42* | 0.95* | 1.04* | 0.95* | 1.09* |
| | SSP370 | middle-of-the-road | | 0.38 | 0.36* | 0.41* | 0.35* | 0.39* | 0.93* | 1.06* | 0.85* | 1.01* |
| | SSP585 | no mitigation | | 0.38 | 0.38* | 0.41* | 0.34* | 0.38* | 0.94* | 1.07* | 0.87* | 1.00* |
| Congo Basin | Factual | observed | 0.17 | | | | | | | | | |
| | SSP126 | strong mitigation | | 0.16 | 0.17* | 0.18* | 0.17 | 0.19 | 1* | 1.05* | 1.03 | 1.11 |
| | SSP370 | middle-of-the-road | | 0.17 | 0.17* | 0.19* | 0.21 | 0.26 | 0.93* | 1.06* | 1.11 | 1.52 |
| | SSP585 | no mitigation | | 0.17 | 0.17* | 0.21* | 0.2 | 0.26 | 0.96* | 1.28* | 1.15 | 1.42 |



**Sub-regional extremes**

| Region | SSP | Represents | Likelihood(%/year) | | | | | | How much more frequent (multiplier) | | | |
| --- | --- | --- | --- | --- | --- | --- | --- | --- | --- | --- | --- | --- |
| | | | 2010-2020 | | 2040-2050 | | 2090-2100 | | 2040-2050 | | 2090-2100 | |
| | | | min | max | min | max | min | max | min | max | min | max |
| Northeast Amazonia | Factual | observed | <0.01 | | | | | | | | | |
| | SSP126 | strong mitigation | 0.01 | 0.02 | 0.01* | 0.02* | 0.01* | 0.02* | 0.94* | 1.38* | 0.91* | 1.34* |
| | SSP370 | middle-of-the-road | 0.01 | 0.02 | 0.01* | 0.02* | 0.03 | 0.04 | 0.92* | 1.58* | 1.98 | 3.23 |
| | SSP585 | no mitigation | 0.01 | 0.02 | 0.02 | 0.02 | 0.03 | 0.05 | 1.15 | 1.64 | 2 | 3.6 |
| Pantanal & Chiquitano | Factual | observed | 0.01 | | | | | | | | | |
| | SSP126 | strong mitigation | 0.02 | 0.03 | 0.03 | 0.03 | 0.03 | 0.03 | 1.05 | 1.21 | 1.03 | 1.24 |
| | SSP370 | middle-of-the-road | 0.02 | 0.03 | 0.03* | 0.03* | 0.03 | 0.04 | 0.94* | 1.23* | 1.21 | 1.45 |
| | SS585 | no mitigation | 0.02 | 0.03 | 0.03* | 0.03* | 0.03 | 0.04 | 0.96* | 1.45* | 1.26 | 1.75 |
| Southern California | Factual | observed | 0.27 | | | | | | | | | |
| | SSP126 | strong mitigation | 0.24 | 0.26 | 0.24* | 0.25* | 0.23* | 0.27* | 0.96* | 1.03* | 0.94* | 1.12* |
| | SSP370 | middle-of-the-road | 0.24 | 0.26 | 0.22* | 0.27* | 0.2 | 0.24 | 0.91* | 1.08* | 0.82 | 0.97 |
| | SSP585 | no mitigation | 0.24 | 0.26 | 0.23* | 0.25* | 0.18 | 0.23 | 0.9* | 1.04* | 0.76 | 0.94 |
| Congo Basin | Factual | observed | 0.01 | | | | | | | | | |
| | SSP126 | strong mitigation | 0.01 | 0.01 | 0.01* | 0.01* | 0.01 | 0.01 | 0.92* | 1.94* | 1.02 | 1.42 |
| | SSP370 | middle-of-the-road | 0.01 | 0.01 | 0.01* | 0.01* | 0.02 | 0.05 | 0.91* | 1.37* | 1.59 | 5.07 |
| | SSP585 | no mitigation | 0.01 | 0.01 | 0.01* | 0.04* | 0.02 | 0.05 | 0.69* | 3.85* | 2.57 | 3.97 |



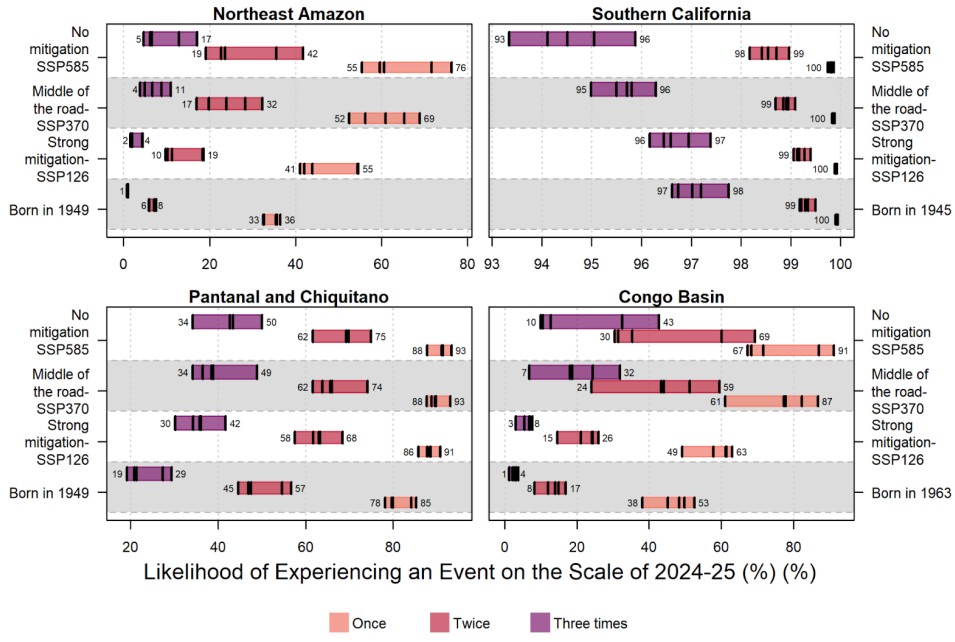

**Figure 19:** Likelihood of experiencing extreme fire events similar to those of 2024-2025 during the average lifetime of a citizen, based on current life expectancy (2023): Brazil (75.8 years, Northeast Amazonia, the Pantanal-Chiquitano), USA (79.3 years, Southern California), and Democratic Republic of Congo (61.9 years, Congo Basin). Bars show the probability of experiencing at least one, two, or three such events if born today under different scenarios: historical climate (bottom bar in each group), SSP126, SSP370, and SSP585 (subsequent bars, bottom to top). Black vertical lines indicate individual GCM estimates; bar heights show the range across models.

### 6.2.3.2. *Pantanal and Chiquitano*

By mid-century (2050), no scenario shows significant increases in the frequency of BA levels such as 2024 at the regional scale (**Table 6**). All scenarios project modest increases by this point: about 1.14-1.15 times more frequent in SSP126 and SSP370, with slightly higher increases in SSP585 (up to 1.22 times). However, substantial changes emerge later in the century (**Figure 18**). Under SSP370, the likelihood of these fires becomes significantly higher by the 2070s, with a 1.2-fold (20%) increase relative to historical conditions. By 2100, SSP585 shows the greatest increases, up to 1.44 times more frequent, while SSP370 projects 1.34 times (**Table 6**). SSP126 demonstrates clear mitigation potential, limiting increases to about 1.13 times, with no significant change throughout the century.

For areas with the highest BA (top 5% grid cells), future changes in the frequency of 2024-like events are significantly different from 2010-2020 for both mid-century (2050) and by 2100 (**Table 6**). Increases at the sub-regional level are larger than regional averages, though not as dramatic as in Northeast Amazonia: by the end of the century events such as those from the 2024-25 season are expected to increase 1.26 to 1.75 times under SSP585, while SSP126 keeps increases much smaller (1.03-1.24 times). SSP370 projections fall between these (1.21-1.45 times), demonstrating that mitigation could still meaningfully limit the occurrence of extreme fire. Increases in the likelihood of extreme BA in high burning





cells could begin as early as the 2030s under SSP126, driven in part by potential increases in fuel availability, though this effect could level off or reverse by mid-century (**Figure S29**). Under SSP370 and SSP585, increases in frequency of extreme BA start to take hold by the 2040s, though large changes may not emerge until after 2060.

These future extremes will mainly be driven by declining moisture availability over the entire region (**Figure 18**). For the most extreme BA areas, this moisture signal is less certain, and changes in fuel, though uncertain, could be large enough to modulate moisture effects (**Figure S29**).

Increases in BA will likely occur across the region by 2090 under all scenarios except in the wetland core of the Pantanal (**Figure S35**), where responses are much more uncertain. Areas of increased extreme fire behaviour exist even under SSP126, but most of the region is projected to see reductions or little change in extremes. In contrast, SSP370 drives widespread increases in extreme BA across almost the entire region, except the wetlands. However, as **Section 5** and other studies (e.g. Barbosa et al., 2022, 2025b) highlight, recent increases in extreme fire have been driven by the combined effects of climate change and wetland degradation, factor not considered in the future projections. This means increases in wetland fire extremes could arise sooner, even by the 2030s or 2040s under SSP126. Under SSP585, widespread increases in extreme BA may arise as soon as 2030 (**Figure S33**), and by 2040 even SSP126 shows large areas of the Pantanal and Chiquitano with much higher chances of a 1-in-100 event (**Figure S34**). Under the SSP126 scenario, the lower chances of extreme events by 2100 compared to mid-century (2040-2050) reinforce how strong mitigation strategies may alter wildfire trajectory throughout the 21st century in this region.

Finally, in terms of lived experience, someone born in the 1940s would already have had a high chance (78-85%) of witnessing a fire event like 2024 during their lifetime (**Figure 19**), with **Section 5** showing climate and human factors likely contributed substantially. Even under SSP126, this rises to 86-91% for someone born today. The difference is most striking for multiple-event likelihoods. Historically, someone born in the 1940s would have had a 19-29% chance of seeing three such events. Under SSP370, this rises sharply to 34-49%, similar to SSP585 (34-50%). Even under SSP126, the likelihood of seeing two such events exceeds 50% (58-68%), compared to 45-57% historically.

### 6.2.3.3. Southern California

While January 2025 fire activity was likely influenced by anthropogenic climate change (**Section 5.2.2.3**), future projections suggest that similar-scale BA extremes may become less frequent (**Table 6; Figure 18**). However, this depends strongly on how local vegetation responds to rising $CO_2$ and climate change.

Looking ahead, models do not project a significant increase in the frequency of these regional-scale extremes (**Figure 18**). In fact, under SSP370 - a scenario closely aligned with current emissions trajectories, the likelihood of 2025-like events in terms of January BA slightly declines by a factor of 0.79 to 0.95 by the 2090s versus 2010s. Similar trends are seen under SSP585, though with the potential for stronger decreases. SSP126, however, showed no robust change by the end of the century.

The projected decline in extreme fire activity in Southern California appears to be driven primarily by modelled increases in tree cover, which occurs even with GCMs with declining precipitation, suggesting that it is largely driven by $CO_2$ fertilisation and enhanced water-use efficiency (**Figure S28**). This effect is more pronounced in drier climates like Southern California, where rising $CO_2$ concentrations reduce water stress on plants and promote vegetation growth. While this leads to greater fuel loads, our framework also represents tree covers influences on fuel moisture, which can suppress fire risk tipping the balance toward





fewer extreme fire events in many model simulations. $CO_2$ concentrations are higher in SSP585 and SSP370 compared to SSP126, which explains why this effect is more pronounced in these scenarios. However, when tree cover is held constant at present-day levels, this signal weakens considerably. Under these "fixed tree" simulations, future projections of extreme fire activity become much more uncertain, with wide variation across scenarios all the way to the 2090s (**Figure 18**). Climate projections themselves for the region are mixed. Some models show increases in January precipitation and fewer dry days, while others suggest drier conditions (**Figure S42**). These divergent signals further contribute to uncertainty in fuel moisture and fire behaviour over the coming decade.

Our projections, therefore, rely on modelled tree and shrub cover from a global land surface model, which, while bias-corrected using historical observation (**Figure S28**), is primarily designed to capture broad-scale vegetation patterns. The model includes global plant functional types (PFTs) such as evergreen and deciduous shrubs, which encompass Mediterranean shrublands like those found in Southern California, but also represent structurally similar ecosystems in very different climatic and ecological settings (e.g., tropical savannas, tundra scrub). As a result, while the model tends to perform reasonably well in estimating total woody cover, it may not fully capture the fine-scale ecological gradients or the dominant shrubland dynamics that drive fire activity in this region. In particular, it may miss key features of chaparral systems and their interannual variability. Future work using regionally calibrated vegetation models or integrating remote sensing estimates of fuel structure may help increase confidence in projections for fire-prone shrub-dominated systems like Southern California.

Therefore, while our models suggest a potential future decrease in large-scale fire extremes in Southern California, this outcome depends on how burned area responds to increasing tree cover, and how vegetation itself responds to rising $CO_2$ and changing climate. Both relationships remain uncertain and will require further investigation. Understanding the evolving links between fuel load, fuel moisture, and ignition risk in the region is essential to refining future fire risk projections in this region.

### *6.2.3.4. Congo Basin*

By the 2050s, none of the emission scenarios project a significant increase in the frequency of regional-scale 2024-like fire events (**Table 6**). Both SSP126 and SSP370 project modest changes, ranging from slight decreases to increases of up to 1.28 times more frequent, though wide uncertainty means small decreases remain possible. Substantial increases emerge by 2100, especially under higher-emissions scenarios. Under SSP370, the likelihood of large fire events rises by 1.11-1.52 times, with SSP585 showing similar values. In contrast, SSP126 holds the increase to just 1.03-1.11 times, indicating a meaningful mitigation potential.

For the most extreme fire events (top 5% of grid cells), projected increases in frequency are more substantial (**Table 6**). No scenario shows significant differences by 2050. However, significant and potentially large changes emerge by 2100. Under SSP370, the frequency of these high-BA extremes could rise by up to 5 times relative to historical conditions (range: 1.59-5.07), slightly higher than the 4-fold increase under SSP585 (2.57-3.97). SSP126 limits this increase substantially to just 1.02-1.42 times. These results show that even under a mitigation pathway, some increase in extreme BA is likely, but the scale of that increase is drastically reduced.

The primary driver of increased fire risk in the region is declining moisture availability, with drier conditions projected across much of the basin (**Figure 18**). In the higher-emissions scenarios (SSP370 and SSP585), increased fuel availability may amplify this effect. For the



most extreme fire-prone areas, however, fuel controls show little change, suggesting that moisture stress will be the dominant factor shaping future fire behavior (**Figure S29**).

Spatially, increases in BA are relatively uniform across the region, though some local differences emerge (**Figures S39-41**). The eastern DRC may experience small decreases in July average BA, though increases remain more likely. In contrast, Gabon, Equatorial Guinea, and central DRC (particularly south of the Congo River) are projected to see the largest increases, with BA doubling or even quadrupling in some areas. Some of these increases, particularly along the Gabonese and Equatoguinean coasts, may begin as early as the 2030s.

In terms of lived experience, someone born in the DRC in 1963 with a life expectancy of 61.9 years, would have had a 38-53% chance of experiencing at least one event like that of July 2024 (**Figure 19**). For those born today, this rises to 49-63% under SSP126, 61-87% under SSP370, and as high as 67-91% under SSP585. The likelihood of experiencing multiple such events also increases markedly. Under SSP585, someone born today would have a 30-69% chance of seeing two events, and a 10-43% chance of seeing three. In contrast, SSP126 limits this to 15-26% for two events and just 3-8% for three, highlighting the powerful influence of mitigation. Indeed, the chance of seeing just one event under SSP126 is comparable to seeing two under SSP585.



# 7.    Conclusions: Summary of the State of Wildfires in 2024-25

## 7.1.    Extreme Wildfire Events of 2024-25

- **Global:** A total of 3.7 million km² burned globally during the 2024-25 fire season, 9% below the average of previous seasons (4.0 million km²), ranking 16th of all fire seasons since 2002. Despite the relatively low area burned, global fire carbon emissions were 2.2 Pg C, 9% above average and the 6th highest on record, driven by intense and high-emission fires in South America and Canada. This pattern reinforces a trend towards growing fire impacts in carbon-rich forest ecosystems, even during years with below-average fire extent globally.

- **South America:** South America experienced an unprecedented fire season setting a new record for carbon emissions. Emissions reached 263 Tg C (84% above average), with BA also 120,000 km² (35%) above average. Bolivia, Brazil, and Venezuela each saw high or record-breaking anomalies, with Bolivia setting national records for both BA and C emissions. Record fire activity occurred across multiple biomes including the Chiquitano dry forests, Pantanal wetlands, and southern and Northeast Amazonia. These fires were characterised by extremely large, fast-spreading, and intense events despite fire counts often being average or below average, highlighting a pattern of fewer but larger and more intense fires on the continent. Highlights:
    - **Northeast Amazonia (Focal Event):** Record-breaking fire activity affected the moist tropical forests north of the Amazon River and Rio Negro, including large portions of Venezuela, Guyana, Suriname, and northern Brazil. Several ecoregions experienced all-time highs in burned area or carbon emissions, with fire activity peaking March-April and again in late 2024. Air quality impacts and environmental degradation were reported across the region.
    - **Pantanal-Chiquitano (Focal Event):** Extreme fire season across Bolivia and adjacent Brazil, with the Chiquitano dry forest and Pantanal wetlands (the world's largest wetlands) seeing some of the largest fires on record. Bolivia experienced the highest national carbon emissions total ever recorded (187 Tg C), with the Santa Cruz department (Bolivia) alone responsible for 157 Tg C. Fires destroyed critical habitat, caused severe air pollution, and threatened biodiversity hotspots. The pantanal recorded PM2.5 concentrations of 903.2 μg/m³ in September 2024, 60 times the WHO daily standard.
    - **Amazonas State, Brazil:** A record-breaking year for fire activity in this moist tropical forest region. Fire counts were up +154% versus the long-term average, and BA and fire size reached record levels. The 95th percentile fire size anomaly was +60%. This was one of the few regions in South America where high fire counts *and* severe individual fire behaviour co-occurred.
    - **Mato Grosso and Mato Grosso do Sul States, Brazil:** Both states saw record-breaking fire intensity and rate of spread. In Mato Grosso, 95th percentile fire size was 266% above average, despite fire counts being 54% below average. Mato Grosso do Sul experienced record emissions (+323%) and fire growth rates, pointing to fast, intense fires likely driven by land-use change and drought.
    - **Pará State, Brazil:** This state recorded its highest-ever BA (36,000 km²) and major emissions anomalies (+61%). Fire activity expanded deep into forested areas, likely linked to land clearing. It was among the most significant subnational contributors to Brazil's fire totals in 2024-25.
    - **São Paulo State, Brazil:** Unusually high-intensity fires occurred despite a relatively small area burned. 95th percentile fire size and intensity both set new records. Carbon emissions were nearly double the historical average

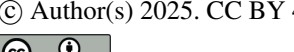



(+190%), driven by a combination of unseasonal drought and land-use pressures.

- **Bolívar and Delta Amacuro, Venezuela:** Two states in northeast Venezuela experienced record emissions and BA, with Bolívar seeing a +133% BA anomaly and Delta Amacuro impacted by early-season fire peaks. These fires affected swamp forests and grassland regions.
- **Coastal and Andean Ecuador, Peru, and Colombia:** Subnational analysis reveals record or high-ranking anomalies in 8 provinces of Ecuador, 7 regions of Peru, and multiple Colombian ecoregions. These include areas in southwestern Amazonia and the eastern Andean slopes, where record fire sizes and intensities occurred despite average fire counts.
- **Guyana and Suriname:** Six ecoregions in Guyana and two districts in Suriname experienced record fire counts and BA, contributing to the focal Northeast Amazonia event but deserving standalone mention given the extent and duration of the anomalies.

- **North America:** The 2024-25 fire season was the second most severe on record for North America, with total C emissions of 194 Tg C (112% above average) and BA of 31,000 km² (35% above average). Canada again saw extreme fire activity for the second year running, with 282 Tg C emitted and over 46,000 km² burned, second only to the record-breaking 2023-24 season. In the US, the catastrophic Palisades and Eaton Fires in California in January 2025, which killed at least 30 people, destroyed over 11,500 homes, and caused over $140 billion in damages. Highlights:
  - **Southern California, USA (Focal Event):** The most disastrous wildfire event in modern US history occurred in Los Angeles County in January 2025 during a severe Santa Ana wind event. The Palisades and Eaton Fires destroyed over 11,500 homes, killed at least 30 people, displaced over 150,000, and caused economic losses exceeding US$140 billion (including insured losses of US$20-75 billion). Fires also disrupted water supplies, worsened the housing crisis, and led to mass evacuations and air quality emergencies.
  - **Western Canada:** Northwest Territories, British Columbia, Alberta and Saskatchewan experienced their second-highest emissions year on record with a combined emissions anomaly of +191 Tg C and provincial anomalies in the range of +184-441%
  - **Mexico:** According to national statistics, Mexico experienced its worst wildfire season on record with over 8,000 wildfires and more than 16,500 km² burned. Particularly severe activity occurred in March-May, reportedly driven by drought and elevated temperatures. This record is not captured in our analyses based on global satellite products, warranting further investigation of the differences.
  - **Alberta, Canada:** Extreme wildfires in summer 2024 destroyed 358 structures and led to $1.23 billion in damages, second only to the Fort McMurray fire of 2016. The town of Jasper was evacuated. Two firefighter fatalities occurred.
  - **New York, USA:** In an unusual late-season outbreak, every borough experienced multiple wildfires during a two-week span in October-November 2024, an unprecedented fire signal in a densely populated urban environment.

- **Africa:** For the second consecutive year, fire extent in Africa was well below average, with BA in the African savannah biome 12% below average, the third lowest on record. However, several regions experienced notable fire anomalies, particularly the Congo Basin, northern Angola, and South Africa. Record-setting BA and C emissions were recorded in some regions of the Republic of Congo and the Democratic Republic of Congo. Despite the extent of these events, many went



under-reported in the media, reinforcing the importance of Earth observation-based monitoring. Highlights:
- ○ **Congo Basin (Focal Event):** Record fire activity and C emissions in the Republic of Congo and Democratic Republic of the Congo. Fires contributed to the region's highest primary forest loss since 2015 and caused hazardous air pollution, with DRC reporting $PM_{2.5}$ levels 11 times the WHO standards. Fires in western ecoregions such as the Atlantic Equatorial and Central Congolian lowland forests were particularly intense.
- ○ **South Africa:** Fires killed 34 people, including 6 firefighters, and destroyed thousands of livestock and homes. KwaZulu-Natal Province was particularly affected. High fuel loads from previous wet years reportedly contributed to the intensity.
- ○ **Côte d'Ivoire:** Fires in Séguéla (Worodougou region) burned 50,000 ha, destroyed homes and plantations, and killed 23 people. Other fatal incidents occurred in Bouna, Bongouanou, and Taabo.

- ● **Asia:** Overall, Asia experienced a below-average fire season, with BA 26% below average and C emissions 28% below average. However, significant regional extremes were observed. Highlights:
  - ○ **Nepal:** Nepal endured its second-worst fire season since 2002, with over 1,000 wildfires. Wildfires killed more than 100 people, with significant destruction of forests and homes. In the Lumbini Province, wildfires devastated 11,448 ha of forests and destroyed more than 230 houses and livestock shelters.
  - ○ **Northern India:** Uttar Pradesh experienced its most severe wildfire season on record, reportedly driven by crop burning, heatwaves, and dry fuel accumulation. Regional fires contributed to severe haze episodes in New Delhi in November 2024, with $PM_{2.5}$ concentrations exceeding 200 µg/m³ across large parts of Northern India (13 times the WHO daily standard).
  - ○ **Iran:** Worst fire season since 2002. Fires burned key national parks and forest areas. Carbon emissions, fire counts, and BA all reached record highs, reportedly driven by a combination of climate stress and human pressures.
  - ○ **South Korea and Japan:** Japan's largest wildfire in over 50 years took place in Iwate Prefecture in February 2025, destroying 221 buildings. South Korea's deadliest wildfires occurred in March 2025 (just outside of the 2024-25 fire season), killing 31 and damaging 4,000 homes.
  - ○ **Sichuan and Guizhou, China:** A fire in Sichuan lasted 14 days, displaced 3,000 people, and impacted multiple villages. Strong winds and dry spring conditions reportedly drove unusually large wildfires.
  - ○ **Heilongjiang and Jilin, China:** Record BA occurred in both provinces. Though not widely reported, these events underscore rising fire activity in northeast Asia, which has been linked to agricultural burning and shifting policy enforcement.
  - ○ **Republic of Sakha** and **Zabaikalsky krai, Russia**: Fires in these regions accounted for 65% of total forest area burned across Russia and forced 58 redeployments of firefighting resources involving 1,861 firefighters.

- ● **Europe:** Europe recorded its fourth lowest BA since 2002, with 30,000 km² burned (49% below average) and C emissions 22% below average. However, there were stark regional contrasts. Highlights:
  - ○ **Portugal:** Most destructive fire season since 2017. Over 137,000 ha burned, with 16 fatalities and €180 million in damages. Fires in September affected wildland-urban interface areas in the northwest. A 5,000 ha fire in Madeira entered the laurel forest, a rare cloud forest and UNESCO World Heritage



site. This incident highlighted the vulnerability of non-fire-adapted ecosystems under increasing fire pressure.
- **Serbia, North Macedonia, and Bulgaria:** Worst wildfire seasons in two decades. Large-scale fires led to EUCPM activations and widespread evacuations, including four fires >10,000 ha in North Macedonia alone.
- **Ukraine:** Nearly 1 million ha burned during 2024-25, mostly in conflict-affected eastern areas. Fires were likely exacerbated by warfare, with higher-than-usual forest losses reported.
- **Romanian Danube Delta:** An unusually dry winter led to 45,000 ha of wetlands burning in February 2025. Though a recurring phenomenon, this was one of the most extensive burn events yet, and emblematic of changing fire regimes in sensitive wetland ecosystems.
- **Turkey (Mardin Province):** A rapidly spreading fire in June 2024 burned farmland and villages, killing 15 people and injuring at least 70. It was one of the deadliest fire events in the Eastern Mediterranean this season.
- **Austria and Germany:** While Central Europe had a quiet fire year overall, Austria recorded its highest number of fires and largest BA since 2012, and Germany had a slightly above-average season, consistent with a slow but steady upward trend.

- **Oceania:** Oceania experienced a moderate fire season overall, but numerous high-impact events were recorded. Highlights:
  - **Western Australia:** Over 1,000 large fires burned ~470,000 ha amid record heat and severe dryness between Perth and Carnarvon. The Skeleton Rocks fire (44,000 ha) impacted long fire-interval ecosystems and a lithium mine, while the largest fire near Cervantes burned 80,000 ha and disrupted regional honey production. Manjimup fires affected over 42,000 ha of native forest and required interstate response.
  - **Central Australia:** Over 5.7 million ha burned by October 2024, including a 450,000 ha fire near Devil's Marbles that forced closures of major infrastructure. In January, 80,000 ha burned in the West MacDonnell Ranges, including national parks and Aboriginal land trusts.
  - **Victoria and Tasmania:** Severe dry lightning outbreaks triggered major fires in culturally sensitive landscapes. Victoria's Grampians National Park saw two-thirds of its area burned, and the Little Desert fire burned 90,000 ha in under 8 hours. Tasmania's northwest fires burned 100,000 ha, affecting the Tarkine and Cradle Mountain.
  - **Queensland:** Firefighters responded to 40 incidents at Mount Isa, with one fire burning over 100,000 ha for nearly two months. Smoke exposure caused hospital admissions and endangered species such as the Carpentarian Grasswren were threatened.
  - **New Zealand:** Peat fires at Whangamarino Wetland and Tiwai Peninsula each burned ~1,000 ha, likely generating significant $CO_2$ emissions after similar events in 2022 emitted 0.6 million tonnes $CO_2$.

## 7.2.    Focal Regions

In this year's report, our detailed analyses target three tropical regions and Southern California. The extreme nature of events in these focal regions are given in Section 2..
- **Northeast Amazonia** saw record forest fire activity, with burned area +332% above average, the highest since records began. Fires severely impacted Indigenous communities, displacing thousands and degrading air and water access.
- The **Pantanal-Chiquitano** experienced its worst fire season on record, with burned areas nearly triple the average and carbon emissions six times above average. Fires





affected both the Pantanal wetlands, the world's largest tropical wetland, and the Chiquitano dry forests of Bolivia. $PM_{2.5}$ pollution reached hazardous levels of 900 µg/m³, carrying strong potential for detrimental health and economic impacts.

- **Southern California** recorded catastrophic wildfire losses, with 30 deaths, 11,500 homes destroyed, and US$140 billion in total damages. $PM_{2.5}$ levels peaked at 483 µg/m³, triggering a regional housing and insurance crisis.
- The **Congo Basin** had its highest recorded fire activity at 28% above the annual mean, contributing to a +150% increase in primary forest loss in 2024 versus 2023. Fires were the main driver of deforestation but received minimal media or institutional attention, highlighting a broader lack of media coverage of fires affecting equatorial Africa.

## 7.3.    Impact Assessments

In this year's report, we incorporate new assessments of the impact of fires on society, specifically via the exposure of populations, physical assets, and carbon projects to fire and via smoke degrading air quality. Key findings from our analyses were as follows.

**Population exposure:**

- We estimate that ~100 million people were exposed to wildfire activity globally during the 2024-25 fire season, with the highest exposures in India and the Democratic Republic of the Congo (15 million each).
  - Uttar Pradesh (India) recorded the highest subnational exposure at 4.6 million people, a 146% increase over average, followed by Heilongjiang (China, 3.7 million) and Punjab (India, 3.6 million).
  - Despite severe fire seasons, Canada, Brazil, and Bolivia contributed modestly to global population exposure due to the remoteness of areas burned.
  - Other countries experiencing large *relative* anomalies in population exposure included: Jordan, Peru and Ecuador (Andes); Venezuela, Guyana, and Suriname (northern South America), Nepal and Niger.
- 20 thousand people were officially displaced according to IDMC displacement records, or 0.02% of those exposed according to our analysis. This reflects a gap between exposure and formal displacement, though true disruption is likely higher than in the IDMC records due to known issues with underreporting.
- Exposed communities may still suffer serious health, economic, and psychological consequences (e.g., missed income, increased debt, long-term health declines), even if they are not formally displaced.

**Physical asset exposure:**

- According to our analysis, an estimated US$215 billion in physical assets were exposed to wildfires in 2024-25. Top countries by asset exposure were India (US$44 billion), United States (US$26 billion), China (US$17 billion), South Africa (US$14 billion).
  - Other countries with high absolute asset exposure were: Mexico, Turkey, and Russia (~US$8 billion each).
  - Other countries experiencing large *relative* anomalies in physical asset exposure were: Pakistan, Sudan, Chad, Albania, Greece, Iraq, Syria, and Eritrea.
- US$57 billion in direct losses were recorded in the international disaster database EM-DAT, including $53 billion caused by fires affecting LA and southern California.
  - Direct financial losses are generally smaller than our estimates of physical asset exposure (the detection of fire in proximity to the built environment) because exposure is a measure of potential for loss, and not of loss itself.



○ In the case of southern California, recorded direct financial losses from fires were three times larger than our estimates of exposed physical assets due to the underestimation of asset density in our analysis. A lesson from this work is that analyses of exposure must account for the significant variation in the density of real estate value across states of the USA, and likely in other countries as well.

**Carbon project exposure:**

- The 2024 fire season saw record BA across forestry projects in the Voluntary Carbon Market (VCM): 169 of 927 projects (18%) experienced fire, the highest on record since 2001, with burned area in 2024 affecting 1.6% of project areas on average.
- 72% of projects experienced above-average drought contributing to elevated risk of fire, with 13% exceeding extreme drought thresholds (SPEI < -2).
- The 2024 fire season had an above average impact on carbon projects in Latin and northern America, average BA was recorded in Eurasia and below average in Africa. In addition to climate, land use and land cover changes and project activities also contributed to regional differences in observed extremes.
- Despite elevated BA in the latest fire season, 46% of all carbon projects experienced no fire in the entire period since 2001, while 67% experienced little fire (defined as <0.5% burned annually in the surrounding 50-km buffer).
- The 2024 season underscores that while high-integrity forest carbon projects remain a key climate change mitigation tool, the permanence of carbon stored or avoided is increasingly threatened by extreme fire years, especially under worsening climate extremes.

**Air quality:**

- Our analysis of air quality impact in this report focuses exclusively on the Pantanal-Chiquitano focal region, where population-weighted $PM_{2.5}$ exceeded the WHO daily standard (15 µg/m³) on 43 days between July to October (over a third of all days in the period) from July to October and peaked at a regional population-weighted average of 61 µg/m³ in September, with fires accounting for ~58% of the pollution. Smoke emissions from fires were the sole cause of exceedances of the WHO daily standard on 50 days in the period July-October.
- Wildfire smoke emissions exposed communities to extremely harmful air quality in various world regions, according to direct measurements (**Appendix A**). For example, communities in the Brazilian Pantanal, Southern California, Bolivia, and northern India were exposed to $PM_{2.5}$ concentrations of over 60, 30, 30, and 13 times the WHO daily standard of 15 µg/m³, respectively.

## 7.4. Diagnosing Causes and Assessing Predictability

- **Weather was the dominant driver of fire activity during all of the 2024-25 focal events** targeted in this report, contributing 40% to 70% of the explainable cause.
  - **Fuel** availability and dryness increased in importance during the most severe fires (up to 40% of explainability) and determined the final extent of BA.
  - **Ignitions** were consistently dominated by human influence, but they did not emerge as a primary cause of fire activity during the 2024-25 focal events (only around 10% of explainability).
- In **Northeast Amazonia** fire activity was predominantly driven by persistent, large-scale drought conditions that depleted deep soil moisture reserves. These droughts suppressed fuel moisture recovery for extended periods, even during rain periods. Soil moisture anomalies reached up to 3 standard deviations below the climatological mean, with values dropping to as little as 2% of average. The





prolonged drought significantly increased fuel dryness and, during the period of most intense burning, fuel importance rose up to 20% above its annual baseline. Fuel also determined the final burned area extent, contributing significantly to the observed anomalies in BA, accounting for up to 50% in the sub-regions where BA anomalies were most extreme. Human-caused ignitions were present but did not emerge as a leading cause of fire (10-20%). Their contribution remained limited and at times negative compared to what is considered usual (thus reducing the total extent), likely reflecting limited ignition opportunities or active suppression efforts to limit BA.

- In the **Pantanal-Chiquitano**, extreme fire activity was primarily driven by antecedent drought persisting since 2023. Deep soil moisture remained in the driest 15% of records, 1-2 standard deviations below average, despite wetter conditions in early 2024. Although February-April rains moistened surface fuels, they failed to recharge deeper layers. Weather dominated fire activity (71% average contribution), with fuel importance rising to 40% during the peak burning week in early August and explaining over 50% of final BA anomalies. Lightning played a minimal ignition role, often occurring in association with convective downpour. Human-caused ignitions, though still dominant, were lower than in previous years and at times limited burned area extent.

- In the **Congo Basin** extreme fire activity was driven by prolonged and severe drought persisting over recent years. The usual spring wet season (March-May) failed to occur, and the second wet season later in the year provided limited relief, leaving deep soil moisture up to 2 standard deviations below climatological norms. Weather was the dominant driver of fire activity, with rainfall 67% below and temperatures 90% above climatological averages, placing vegetation and soil dryness among the driest 1-2% of records (2003-2023). Human activity accounted for most fire ignitions but as for the other 2 tropical regions they were not the main causes of the fire severity and actually acted to reduce the final BA

- In **Southern California,** the 2024-25 fire season was marked by atypical seasonality, with extreme fire activity occurring in January well outside the usual summer peak. The Palisades and Eaton fires were driven by a rare convergence of weather, fuel, and ignition factors, each contributing significantly (weather: 40%, fuel: 30%, ignition: 20%). Despite preceding years of exceptional wetness, a short-lived but extreme drying of surface fuels (3 standard deviation below normal) and intense winds (3 standard deviation above normal) created highly flammable conditions. These fires ignited and spread rapidly at the wildland-urban interface, highlighting how brief windows of extreme weather can override generally moist background conditions and trigger major off-season events in these parts of the world.

- There were **distinct challenges to the forecasting of all focal events**:
  - In **Northeast Amazonia**, our models correctly identified two high-risk fire seasons, but most of the burning occurred during the first (February-April), not the second (August-November), despite similar fire danger forecasts. This disconnect highlights a key limitation: high fire danger does not always lead to high fire activity. Human factors, such as suppression, fire bans, or shifts in land use, likely played a role and are currently underrepresented in fire prediction systems.
  - In the **Pantanal and Chiquitano**, fires were closely linked to long-term drought conditions that dried out fuels months before the fire season peaked. Fire activity rose only after this slow build-up, meaning accurate forecasts required capturing both drought and fuel dynamics. While the general heightening of fire danger was picked up by the FWI, the machine-learning-based PoF model, which includes fuel conditions, better predicted when and where fires would actually occur.
  - In **Southern California**, fire prediction remains difficult without accounting for the 'whiplash effect' that arises from extreme fire weather following on from



wet periods with high vegetation productivity. A wet period led to vegetation
growth, followed by rapid drying and strong winds that enabled the January
fires. As in the Pantanal-Chiquitano, including fuel information helped the PoF
model identify higher-risk areas more accurately than the FWI.
○ In the **Congo Basin**, both FWI and PoF tended to overpredict fire danger.
While drought increased flammability, ignition remained limited, possibly due
to cultural practices, suppression efforts, or fewer ignition sources (though
reporting on such activities in this region is extremely limited). Here, human
activity and fuel moisture, more than fire weather, shaped outcomes. The FWI
system, which unlike PoF does not include these factors, was less effective in
predicting fire activity in the Congo Basin.

## 3704 7.5. Attribution to Global Change

● **Climate change has increased the likelihood of extreme fire events across all**
**focal regions studied.** The high fire weather and extreme levels of burning seen in
2024-25 were significantly more likely in a world with human-induced climate change.
● In **Northeast Amazonia,** we find that the extreme fire weather during January-March
2024 was 30-70 times more likely due to anthropogenic climate forcing, while the risk
of regional BA totals being as observed in the period was 2.1 times greater due to
anthropogenic climate forcing and the area burned by fires was four times greater.
○ Our attribution analysis shows high confidence that climate change played a
major role in Northeast Amazonia's record fire season. We are virtually
certain (>99%) that anthropogenic climate forcing increased the risk of
extreme fire weather, very likely (96%) that it amplified the area affected, and
likely (89%) that it increased the chance of the extreme burned area
observed.
○ While climate change has clearly enhanced the probability of extreme events
in the region, such as that seen in 2024, there was conversely no robust
evidence that climate change increased average annual BA totals in
Northeast Amazonia during 2003-2019.
○ An increase in annual average BA during 2003–2019 of up to 17% was
attributed to socioeconomic changes since 1900-1917, indicating that
long-term human activities have elevated typical fire levels in the region.
○ Overall, our attribution analyses suggest that climate change has enhanced
the likelihood of extreme fire events in the region, against a backdrop of
increased annual BA levels driven by socioeconomic change such as land
use/land cover change and human ignitions.
● In the **Pantanal and Chiquitano**, we find that the extreme fire weather
August-September 2024 was 4-5 times more likely due to anthropogenic climate
forcing, while the risk of regional BA totals being as observed in the period was 3.3
times greater due to anthropogenic climate forcing and the area burned by fires was
around 34 times greater.
○ Our attribution of extreme fire weather to climate change was made virtually
certain (>99%, IPCC definition), while the amplification of both extreme
burned area and region-wide burned area extent was attributed with likely
confidence (87%). Taken together, these findings provide strong evidence that
anthropogenic climate change raised the odds of the largest fire season on
record in the Pantanal-Chiquitano region.
○ In addition to the enhanced odds of extreme BA events, a 10% increase in
annual average BA during 2003-2019 was attributed to climate change.
○ At least a two-fold increase in BA during years with 2024-like fire conditions





was attributed to socioeconomic change, indicating that human activities have substantially increased the risk of widespread fire under extreme conditions. However, other analyses focusing on long-term annual average burned area suggest that some human-driven changes may have reduced typical annual fire activity. While these findings are not strictly contradictory since they examine different aspects of the fire regime, the contrast between them reduces confidence in attributing overall fire trends to socioeconomic drivers alone and points to the need for further investigation.

- Overall, extreme BA events in the Pantanal-Chiquitano, such as those seen in August-September 2024, are made more likely by climate change and are superimposed on broader background increases in fire extent related to climate change and possibly socioeconomic changes in the region.

- In **Southern California**, we find that the risk of regional BA totals being as observed during January 2025 was 2.3 times greater due to anthropogenic climate change and the area burned by fires was 25 times greater.
  - Our attributions of amplified BA extent during the event to climate change were all made with at least 89% confidence. It is therefore *likely* (per IPCC definitions) that anthropogenic climate change raised the odds of the costly wildfires in Southern California during January 2025.
  - The meteorological conditions during the event were previously studied by the World Weather Attribution (WWA) group, who reported that extreme fire weather conditions were also made more likely, by around 40%, with other indicators such as prolonged drought and delayed seasonal drying also showing climate influence (Barnes et al., 2025). We did not perform an independent attribution of fire weather here due to a lack of data required for construction of a counterfactual scenario in our attribution protocol.
  - In addition to the enhanced odds of extreme BA events, a 7% increase in annual average BA during 2003-2019 was attributed to climate change.
  - Our BA attribution approaches did not provide robust evidence that socioeconomic change affected average annual BA, though this is possibly due to the difference between the coarse model resolution and the fine scale over which effects would be expected at the wildland-urban interface in this region.
  - Overall, extreme BA events in Southern California, such as those seen in January 2025, are made more likely by climate change and are superimposed on broader background increases in fire extent related to climate change.

- In the **Congo Basin**, we find that the extreme fire weather July-August 2024 was 3-8 times more likely due to anthropogenic climate change, while the risk of regional BA totals being as observed in the period was 60% greater due to anthropogenic climate change and the area burned by fires was three times greater.
  - It is virtually certain that anthropogenic climate change contributed to the extreme fire weather observed during the 2024 season in the Congo Basin. The widespread extent of burned area was very likely influenced by climate change (92% likelihood), while the most extreme sub-regional burned area events were likely influenced (78% likelihood). Together, these findings indicate that climate change increased the odds of the largest fire season on record in the region.
  - In addition to the enhanced odds of extreme BA events, a more than 45% increase in annual average BA during 2003-2019 was attributed to climate change.
  - Our BA attribution approaches did not provide robust evidence that socioeconomic change affected average annual BA during 2003-2019 versus a pre-industrial counterfactual.



- Overall, extreme burned area events in the Congo Basin, such as those seen in July–August 2024, are made more likely by climate change and are superimposed on broader background increases in fire extent attributable to climate change, with no robust evidence that socioeconomic changes significantly altered recent fire activity.

## 7.6. Seasonal and Multi-Decadal Outlook

- **Fire weather and BA forecasts for boreal summer 2025 highlight several areas with elevated probability of anomalous fire danger.** Probabilities for anomalous fire prone seasons are high across Canada, northeast Europe (including the UK), and parts of Siberia. These conditions following the second-warmest May on record globally (1.4 °C above pre-industrial levels), with exceptional dryness and the lowest northwestern European rainfall since 1871.
  - In equatorial regions, forecasts show a more than 60% chance of anomalous fire weather conditions in Northeast Amazonia, the Congo Basin, and the Himalayan foothills.
  - In the US, severe drought conditions in Arizona and Texas are already leading to elevated fire activity in line with predicted anomalies in fire weather.
  - Seasonal outlooks of burned area anomalies coincide with fire weather anomalies in western South America, southern California, central North America, and eastern Central Asia.
  - Chile and northern Australia stand out with >50% confidence for anomalous fire activity during the boreal summer of 2025.
  - Despite high FWI in central Europe, we could not confidently predict a BA anomaly due to insufficient historical fire-climate data for reliable modelling.

- **In Northeast Amazonia, our climate model projections consistently indicate a rise in extreme wildfire risk by the end of the century.** Under a *middle-of-the-road* emissions pathway (SSP370), the frequency of regional BA totals on the scale of 2024 are projected to increase by up to 57% by 2100.
  - Also under SSP370, the greatest rate of increase (factor of 2-3 rise) is projected in the sub-regions that burned most extensively in the extreme event of 2024 (5% of model cells with greatest BA).
  - Under a *no mitigation* scenario (SSP585), an even sharper rise is projected, with a near-doubling of the frequency of extreme (2024-like) events at the regional scale. Greater rates of increase (up to a four-fold rise) are projected in the sub-regions that burned most extensively in 2024.
  - In contrast, limiting warming under a *strong mitigation* scenario (SSP126) effectively contains future fire risk. By 2100, the increased frequency of an extreme (2024-like) event is limited to 9%, with the sub-regions that burned most extensively in 2024 showing no significant change. This demonstrates the strong potential of climate action to mitigate the risk of future extreme fires in Northeast Amazonia.
  - Projections of future increased risks are not spatially uniform in any scenario. In some areas, such as Amapá and northern Pará in Brazil, and southern Suriname, increased extreme fire activity is projected as early as the 2030s under higher-emissions scenarios (SSP370 and SSP585). Even under SSP126, rises in extreme BA are projected for parts of the moist forest zone.
  - The frequency of extreme (2024-like) events is projected to rise only modestly in all scenarios through 2050, however by 2100 the increased risk under higher emissions scenarios (SSP370 and SSP585) clearly emerges from that of SSP126.



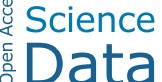

- In the **Pantanal and Chiquitano, our climate model projections indicate further increases in extreme wildfire risk by the end of the century.** Under a *middle-of-the-road* emissions pathway (SSP370), the frequency of regional BA totals on the scale of 2024 are projected to increase by up to 34% by 2100.
  - Also under SSP370, the greatest rate of increase (21-45% rise) is projected in the sub-regions that burned most extensively in the extreme event of 2024 (5% of model cells with greatest BA).
  - Under a *no mitigation* scenario (SSP585), an even sharper rise is projected, with a 44% rise in the frequency of extreme (2024-like) events at the regional scale. Greater rates of increase (up to a 75% rise) are projected in the sub-regions that burned most extensively in 2024.
  - In contrast, limiting warming under a *strong mitigation* scenario (SSP126) effectively contains future fire risk. By 2100, the increased frequency of an extreme (2024-like) event is limited to 13% and is not significant, while the sub-regions that burned most extensively in 2024 experience minimal increases in frequency (up to 24% rise). This demonstrates the strong potential of climate action to mitigate the risk of future extreme fires in the Panatanal-Chiquitano.
  - At the regional scale, only modest increases in the frequency of extreme (2024-like) fire seasons are projected by mid-century across all scenarios. However, by 2100, the increased risk becomes more pronounced under higher emissions pathways, with clear divergence between scenarios.
  - At the sub-regional level in the areas that burned most extensively, earlier increases in extreme fire risk could begin as soon as 2030.
  - Projections of future increased risks are not spatially uniform in any scenario. Geographically, widespread increases in BA are projected across most of the Panatanal-Chiquitano by 2100, though the response is considerably more uncertain in the Pantanal than in the Chiquitano. Some areas of increased extreme (2024-like) fire frequency may still emerge in the Pantanal even under SSP126.
  - It is important to note that these projections do not fully incorporate local *in situ* drivers, such as wetland degradation, which have already contributed to more frequent fires in recent years. Increases in fire activity might be expected to occur earlier than the models indicate, especially along the wetlands and adjacent drainage areas.

- **In Southern California, our climate model projections of future change in extreme (2024-like) fire events are highly uncertain.**
  - While high-emissions simulations under SSP585 and SSP370 suggest that extreme fire events could become less frequent over time, this strongly depends on how vegetation responds to rising $CO_2$ and a changing climate.
  - In particular, simulations suggest that increased tree cover driven by $CO_2$ fertilisation under higher-emissions scenarios (SSP585 and SSP370) may raise fuel loads while simultaneously increasing fuel moisture, with the overall effect being to reduce the likelihood of extreme fire events in our models.
  - However, when removing changes in tree cover, the projected future frequencies of extreme (2024-like) events become highly uncertain with no consistent direction of change under future scenarios.
  - There is a critical need for improved observation and modelling of how vegetation structure, fuel moisture, and local ecological processes shape fire behaviour in Southern California. Nonetheless, Southern California remains highly exposed to fire risk. Even under scenarios that suggest a decline in fire extremes, most residents alive today are still likely to experience multiple extreme fire seasons like 2025 in their lifetime. Stronger local adaptation and more regionally tailored research on climate-vegetation-fire interactions will



be essential to manage risk in the coming decades.

●   In the **Congo Basin**, **our climate model projections indicate that further**
**increases in extreme wildfire risk are likely by the end of the century.** Under a
*middle-of-the-road* emissions pathway (SSP370), the frequency of regional BA totals
on the scale of 2024 are projected to increase by up to 50% by 2100.

○   Also under SSP370, far greater rates of increase (up to a 5-fold rise) are
projected in the sub-regions that burned most extensively in the extreme
event of 2024 (5% of model cells with greatest BA).

○   Projections under SSP370 and SSP585 show similar levels of elevated risk,
indicating that mitigation efforts stronger than those implied by SSP370 are
likely needed to meaningfully reduce future fire risk.

○   In contrast, limiting warming under a *strong mitigation* scenario (SSP126)
effectively contains future fire risk. By 2100, the increased frequency of an
extreme (2024-like) event is limited to at most 11%, while the sub-regions that
burned most extensively in 2024 experience a far smaller increase in
frequency (up to 42% rise) than under higher emissions scenarios. This
demonstrates the strong potential of climate action to mitigate the risk of
future extreme fires in the Congo Basin.

○   Projections of future increased risks are not spatially uniform in any scenario.
Some of the largest projected increases, seen in Gabon, Equatorial Guinea,
and central DRC, may begin as early as the 2030s, with the frequency of
extreme (2024-like) events is projected to increase 2 to 4-fold by 2100. This
increase is driven largely by declining fuel moisture as climate change
reduces rainfall and increases dry spells across much of the region.

●   **Anthropogenic climate change has the potential to significantly increase future**
**fire risk for living generations, turning previously exceptional events into**
**events that are experienced several times in a generation.**

○   **Northeast Amazonia:** Our projections show that a person born in this region
today has a 41-55% likelihood of experiencing at least one extreme fire
episode on the scale of January-March 2024 in their lifetime under *strong*
*mitigation* scenario (SSP126). This likelihood rises to 52-69% under a
*middle-of-the-road* scenario (SSP370), and 55-76% under a *no mitigation*
scenario (SSP585). The odds of experiencing two or more such events are
considerably higher under *no mitigation* (19-42%) than under *strong*
*mitigation* (10-19%).

○   **Pantanal-Chiquitano:** Our projections indicate that a person born in this
region during the 1940s already had a ~78-85% likelihood of experiencing at
least one fire season like 2024. For someone born today, this likelihood rises
to 86-91% even under SSP126. Under SSP370, the likelihood of experiencing
at least two 2024-scale fire seasons rises to 62-74%, compared to 45-57% for
someone born in the 1940s, but even under low emissions, the chance of two
such events exceeds 58%. These findings highlight that while climate change
mitigation can reduce future fire risk, it is not sufficient on its own. Early
adaptation, ecosystem management, and stronger fire governance will be
essential to reduce future impacts.

○   **Congo Basin:** Our projections show that a person born in this region today
has a 49-63% likelihood of experiencing at least one fire season like July
2024 under SSP126. This likelihood rises to 61-87% under SSP370 and
67-91% under SSP585. The likelihood of experiencing multiple events differs
starkly across SSPs, with up to a likelihood of 43% for three events under
SSP585, compared to just 3-8% under SSP126.





## 7.7. Progress in the State of Wildfires Report

This report incorporates a number of major advances in our annual reporting on the State of Wildfires in the prior fire season. In **Section 2**, we added a new analysis of fire intensity to our extreme event identification variables, and we evaluated the dependence of our extreme event identification on the source of BA observation by incorporating data for 2019-2025 from two additional BA datasets (FireCCIS311 and VIRS VNP64A1), supplementing our consistent multi-decade analysis based on the MODIS BA dataset (MCD64A1). The contribution of regional expert knowledge was also formally recognised through the establishment of regional expert panels for each continent, with these panels consulted for their interpretation of results across all aspects of the report. We added **Section 3**, which presents an entirely new set of impact assessments relating to population exposure, asset exposure, carbon project exposure and air quality degradation. In **Section 4**, we expanded the analysis of the predictability of the focal event to include seasonal predictions of burned area, complementing the fire danger seasonal forecasts already provided. In **Section 5**, our main advancement was a new approach to attributing both extreme regional BA totals *and* sub-regional BA extremes directly to the 2024-25 focal events, made possible by developing near real-time counterfactuals and employing methodologies for aggregating probabilities across space. This represents a step-change versus our first report, which focussed on attributing sub-regional BA extremes only and substituted near real-time counterfactuals with less targeted counterfactuals for the 2003-2019 period. By creating more robust counterfactuals with observed events, and accounting for the stochastic nature of fire anomalies locations, we were able to more directly and confidently assess whether human influence made these specific fires more likely on regional scales. In **Section 6**, we extended our forward-looking capabilities by providing seasonal forecasts of BA, complementing the fire danger forecasts already presented in previous reports. We also added future projections of FWI at future global warming levels of 1.5-4.0°C, providing a clearer picture of how extreme wildfire risk may evolve in the coming decades.

This new report documents the progress made in the observation, diagnosis, modelling and attribution of extreme wildfire events and their impacts. As a community, our work is both driving innovation in the methods under use and prompting the development of new capabilities for the routine analysis of extreme wildfire events and their impacts. This new report builds on our inaugural report (Jones et al., 2024b) and documents the progress being made by the fire science community.

By combining cutting edge techniques in fire forecasting, prediction and modelling across the sections of our report, we compile multiple lines of evidence for a clear climate signal in recent fire extremes. Our complementary analyses consistently point to a strong role for climate change in driving extreme fire conditions, showing that human influence, both through climate change through socioeconomic change factors, are increasing fire risk and producing extreme wildfires. While individual methods sometimes diverge, particularly in regions like the Pantanal, where local socioeconomic factors emerge more clearly as drivers in some analyses, the overall convergence across independent lines of evidence builds confidence in the conclusion that climate changes exerts significant upwards pressure on the likelihood of extreme fire events.

These multiple lines of evidence show that human influence, often through climate change though sometimes through socioeconomic factors, are increasing fire risk and driving higher burned areas. Across every region analysed, we find clear signals that recent extreme wildfires are not purely natural events, but increasingly shaped by human-driven changes to climate and ecosystems.

A key strength of this report lies in its systematic evaluation of model performance across diverse regions of the globe. In this edition, for instance, we identify limitations in the



capacity of coarse-resolution air quality models to assess smoke exposure in small regions (**Section 3**), and show how projections of future fire activity can be strongly influenced by how models represent sensitive vegetation responses to uncertain climate changes (**Section 6**). In regions such as California, long-term projections are particularly sensitive to changes in tree cover, which can be affected by uncertainties in both climate inputs and modelled vegetation responses.

A rich body of observations, such as land surface and meteorological data are available to observe and model the effects of climatic change and variability on extreme fire likelihood, in particular following important advances in the modelling of fuel load and moisture dynamics during recent decades. However, a major outstanding barrier that consistently limits the effectiveness of our analyses, and those of the broader fire science community, is a severe lack of information regarding *in situ* human activities. Funding of projects that overcome this barrier is paramount and carries the greatest potential to drive a step-change in performance of fire models and predictive systems. Often, prediction and modelling analyses rely on basic indicators of human effects such as population density, which cannot sufficiently represent the diversity of relationships between people, their land uses, and the outcomes for fire ignitions and spread dynamics. Our work, and that of many others, highlights the need to develop global datasets that effectively represent the range of human-fire interactions that occur on Earth but with sufficient scalability to support regional and global analyses. Inevitably, there will be a trade-off between the geographical scalability and nuance of these datasets as they are developed.

Overall, our international collaboration routine catalogues fire extremes and annually evaluates the most extreme fire events of international relevance using state-of-the-art fire science tools. We provide a consistent stream of actional information to policymakers, disaster management services, firefighting agencies, and land managers, informing action on enhancing society's resilience to wildfires through investment in preparedness, mitigation, and adaptation.





## Appendix A: Year in Review by Continent

This appendix includes the review completed by regional expert panels to supplement our quantitative analyses of extremes in the 2024-25 fire season (**Section 2**). Details of the assembled panel are provided in **Table A1**, below.

**Table A1:** Experts contributing to the identification of extreme events and characterisation of the global fire season during March 2024-February 2025.

| Region | Co-authoring Experts | Country of Organisation / Nationality | Professional Background(s) | Supporting Expert Panellists | Others Consulted |
|---|---|---|---|---|---|
| Africa | Kebonye Dintwe | Botswana | | Lucy Amissah (Ghana), Sally Archibald (South Africa), Natasha Ribeiro (Mozambique), Tercia Strydom (South Africa) | |
| | Aya Brigitte N'Dri | Ivory Coast | | | |
| Asia | Cong Gao | China | | Bambang Saharjo (Indonesia), Sundar Sharma (Nepal), Raman Sukumar (India), Veerachai Tanpipat (Thailand), Bo Zheng (China) | |
| | Elena Kukavskaya | Russia | Research | | |
| Europe | Paulo Fernandes | Portugal | Research | Davide Ascoli (Italy), Stefan Doerr (UK), Julien Ruffault (France), Gavriil Xanthopoulos (Greece) | |
| | Cristina Santín | Spain | Research | | |
| | Johan Sjöström | Sweden | Research | | |
| North America | Crystal Kolden | USA | Research, Firefighting | Jacqueline Shuman (USA), Matt Jolly (USA), Piyush Jain (Canada), Chelene Hanes (Canada) | |
| | Mathieu Boubonnais | Canada | | | |
| | Victoria Donovan | USA | | | |
| Oceania | Hamish Clarke | Australia | Research, Environmental Management | | Simeon Telfer, South Australian Country Fire Service; Rui Feix, Western Australian Department of Fire and Emergency Services; Chris Collins, Tasmania Fire Service; Grant Pearce, Fire and Emergency New Zealand; David Field, New South Wales Rural Fire Service; Russell Stephens Peacock, QueenslandFire and Emergency Services; Maggie Towers, Northern Territory Police, Fire and Emergency Services |
| | Sarah Harris | Australia | Research, Emergency Management | | |
| South America | Liana Anderson | Brazil | Research | Dolors Armenteras (Colombia), Francisco de la Barrera (Chile), Mauro Gonzalez (Chile), Celso H.L. Silva-Junior (Brasil) | |
| | Carlos M. Di Bella | Argentina | Agronomist/Research | | |
| | Bibiana Bilbao | Colombia | | | |
| | Galia Selaya | Bolivia | Tropical Ecology/Research and action | | |





## A1. Africa

National and regional fire monitoring statistics are rarely recorded or made publicly available by fire agencies in Africa, meaning that our assessment of the latest global fire season largely focuses on the insights provided by global data analyses. According to these data, the total BA in Africa was approximately 2.4 million km$^2$ during the 2024-2025 fire season, 11.6% below the mean annual BA since 2002 (2.7 million km$^2$). Most of the BA occurred in non-forest (2 million km$^2$), with the remaining portion in the forest. Non-forest and forest BAs were 12% and 7% lower than the mean annual BA, respectively. The BA anomaly was notably larger in NHA (-14.6%) than in SHA (-9.1%). The relatively low BA in many parts of the continent could be a result of a combination of factors, though it aligns with a trend that has been attributed to the continued suppression of fire from expanding croplands (Andela et al., 2017) and to changing rainfall patterns across the continent (Zubkova et al., 2019).

Africa's most pronounced positive anomalies in BA and fire C emissions of the 2024-25 fire season were seen in the Congo basin and northern parts of Angola (**Figure 1, Figure 2; Table 2, Table 3**). BA in the Republic of Congo was 25% above average, the highest on record, and similarly fire C emissions were 25% above average (**Table 2, Table 3, Figure S43**). In the Democratic Republic of the Congo, the Mai-Ndombe and Sankuru provinces each experienced record levels of BA or fire C emissions with anomalies in the range of 36-58% (**Table 2, Table 3**). These anomalies were centred on several western ecoregions of the Congo Basin, including the Atlantic Equatorial coastal forests where BA was more than triple the annual mean, Western Congolian swamp forests where BA was twice the annual average and the Central Congolian lowland forests where BA was 77% above average, and the Northwestern Congolian lowland forests where BA was 55% above average. These results align with the recent report of the Global Forest Watch (World Resources Institute, 2025) which found that the Democratic Republic of the Congo (DRC) and the Republic of the Congo experienced their highest rates of primary forest loss since 2015. While loss to wildfire is a minor component of overall forest loss in the region (below 15%), for instance compared to the expansion of shifting cultivation, wildfires were the major explanation for the more than doubling (+150%) increase in forest loss in 2024 versus 2023.

The uptick in fires in the Congo basin can be linked in part to the enabling effect of record-breaking fire weather caused by drought in the region (**Section 2.2.2.1**), however a range of socioeconomic changes have also been underway and likely influenced the events of last year. Use and degradation of the forests for resources, often linked to an increase in related wildfire ignitions, is growing due to the extraction of timber to produce charcoal, clearing of land for the expansion of cash crops, and shortening or cessation of fallow periods in smallholder shifting cultivation systems (World Resources Institute, 2025). The potential effects of fires in this region on forest carbon stocks are globally significant (though they are yet to be quantified), with the region's swamp forests harbouring 30 billion metric tonnes of C in peat (Garcin et al., 2023). The 2024 IQAir World Air Quality Report highlighted that the Democratic Republic of the Congo had an annual average $PM_{2.5}$ concentration of 58.2 μg/m³, over 11 times higher than the World Health Organization's annual standard (IQAir, 2025). This indicates hazardous air quality levels, in part to the effects of elevated wildfire smoke emissions (IQAir, 2025). Despite the potentially large impacts on society and the environment, there was very limited news coverage on the impacts of these fires by national news outlets across the region. This underscores the importance of projects such as ours and the Global Forest Watch (World Resources Institute, 2025) using Earth Observations to routinely trace environmental extremes and highlighting events that would otherwise have gone under-reported.

The high BA in the Congo Basin during 2024-25 has implications for various initiatives supported by non-governmental organisations in the region, which aim to promote protection and sustainable management of tropical forests. For example, UNEP's Congo Basin



Sustainable Landscapes Programme (Green Policy Platform, 2025) supports action in
Cameroon, Central African Republic (CAR), the Democratic Republic of the Congo (DRC),
Equatorial Guinea, Gabon, and Republic of the Congo. In programmes such as this, wildfire
can sometimes be considered a secondary disturbance factor compared to other factors
such as clear-cut deforestation, but years such as 2024 demonstrate that large-scale
intermittent fires in the region can have lasting consequences for forest loss.
In Angola, BA and fire C emissions were 15-49% above average in the provinces of Moxico,
Huíla, and Bié and were either record-setting or high-ranking (**Figure S44**; **Figure 2**, **Figure
3**; **Table 2**). As discussed in **Section 2.2.2** and investigated formally for the Congo Basin in
**Section 4**, a particularly hot and dry fire season elevated the risk of fire in these regions and
coincided with broader social and economic factors promoting fire ignitions. The poor
economic situation in Angola over the past three years has prompted deregulation of the
charcoal industry and the harvesting of trees for charcoal production has risen, driving up fire
activity (Valor Económico, 2024; VisiteHuila, 2024). In addition, the government has been
promoting agriculture through financial programs, leading to the clearing of land through
shifting agriculture in Miombo woodlands (Fundo de Garantia de Crédito, 2024; World Bank,
2024b). In certain provinces, particularly Moxico in Angola, burning for hunting purposes is
also widespread and declining populations of prey have been linked to increased burning of
areas that were previously hunted less regularly (Papelo, 2024). These are just some of the
socioeconomic factors that may have contributed to the elevated availability of ignition
sources during 2024-25 fire season, when fire weather was particularly conducive to fire.
In Algeria, fires have killed and injured dozens and caused significant loss of life and
damage in recent years. At least 34 people were killed and several hundred were injured in
Bejaia province during the 2023-24 fire season (Jones et al., 2024b). However, in 2024-25, a
low number of fires were recorded and there were no casualties in Algeria, which could be
attributed to various factors such as the availability of better firefighting equipment, new fire
management policies, and a new law that was passed that imposes life imprisonment for
those caught deliberately starting forest fires (Serrah, 2024; The Arab Weekly, 2024).
Algerian authorities launched a wildfire prevention system that included 13 water-bombing
aircraft and 100 drones for monitoring and tracking firefighting operations. For instance, 26
fires were extinguished within 24 hours in the central and eastern regions of Algeria, with no
injuries or casualties reported (Gabriel, 2024).
In South Africa, the total BA was over 46,000km², which was 17% higher than the mean
annual BA. According to a report by the organisation Working on Fire (2024), the increased
intensity and frequency of these fires continue to challenge firefighting resources. The
2024-25 fire season broke records, with 2,750 firefighting teams dispatched, with a record
number of 34 people losing their lives, including firefighters. In KwaZulu-Natal Province, the
wildfires claimed the lives of 14 people, of whom 6 were firefighters who were trapped in a
blaze. In addition to the lost lives, thousands of people were displaced, over 2,050 livestock
destroyed, and critical infrastructure damaged. The high intensity fires in South Africa could
be due to a string of particularly high rainfall years that resulted in large accumulated grassy
fuel loads.
In Côte d'Ivoire, the overall BA in 2024-2025 was lower than the historical average, contrary
to what some national experts had expected following the long dry season which began
earlier than usual in the savanna areas of the country where fire is most widespread (N'Dri et
al., 2018, 2024; Soro et al., 2021). Nonetheless, the fire season was marked by an
above-average fire size distribution and there were several deadly events in the country's
main fire hotspots, with fires burning over 150,000 ha of forest, 2,800 ha of plantations, 109
ha of reforestation projects, and 107 properties in 2025 (CNDFB, 2025). In the department of
Séguéla (Worodougou region), wildfires in February 2024 destroyed 50,000 ha of natural
vegetation, 261 ha of cropland, 236 ha of cashew plantations, 19 homes, and claimed the





lives of 23 individuals across the villages of Sélakoro, Djénigbé, Touna, Djoman and Kondogo. In Bouna (Bounkani region), fires affected around 12,387 ha, of which 7,528 ha were forested, leading to additional humanitarian impacts. Three further fatalities were recorded between February and March 2024 in Bongouanou (Moronou region) and Taabo (Agnéby-Tiassa region). These impacts occurred despite the continued efforts of the *Comité National de Défense de la Forêt et de lutte contre les feux de Brousse* (CNDFB), such as the construction of firebreaks during the dry season and awareness campaigns. This reflects the challenges posed by expanding agricultural land and ignition sources, fire suppression policies that focus on fire exclusion to protect valuable crops (e.g. cashew nuts) but promote fuel build-up, and a lack of prescribed burning in Côte d'Ivoire's savanna ecosystems (Ruf et al., 2010; Soro et al., 2020; Kouassi et al., 2022). Generally, fire activity and BA have been declining across all ecoregions of Côte d'Ivoire, which has been attributed to conversion of savannas to agricultural lands and also bush encroachment in savanna areas (N'Dri et al., 2022; Douffi et al., 2021; Kouassi et al., 2022).

**A2. Asia**

The 2024-25 fire season in Asia was generally not an extreme one, with much of Asia experiencing typical or low BA. Nonetheless, there were regional extreme fire events in the fire season.

Iran emerged as a notable case, experiencing its most severe wildfire season since 2002, marked by record-breaking BA, number of fires, and carbon emissions at the national level (**Figure 2, Figure 3**). Ecologically sensitive regions were disproportionately affected, including Karkheh National Park in Khuzestan Province and the forests and rangelands of Ab Kenar and Khan Ahmad Basht in Kohgiluyeh and Boyer-Ahmad Province (Global Fire Monitoring Center, 2024). As one of the driest countries in the world, Iran experiences approximately 1,500 wildfire outbreaks annually, resulting in the burning of 15,000 ha of forest (Kheshti, 2020; Tavakoli Hafshejani et al., 2022). Human activities are the primary driver of wildfires nationwide, with deforestation, illegal logging, and accidental ignition contributing to the high incidence of fires (Masoudian et al., 2025). These anthropogenic pressures are compounded by systemic shortcomings against wildfires, including inadequate resource allocation and insufficient prevention measures, which challenge the protection of natural ecosystems (Iran International, 2024).

Nepal also endured its second-worst fire season since 2002 (**Figure S45**), with over 5,000 fires according to some sources (Bolakhe, 2024; and >1,000 individual fires in our analysis, **Figure S45**) causing more than 100 fatalities (Bolakhe, 2024). In Lumbini province, located in western Nepal, wildfires devastated 11,448 ha of forests and destroyed more than 230 houses and livestock shelters (Sanju Paudel, The Kathmandu Post, 2024). These catastrophic events were driven by extreme drought, prolonged heatwaves, and frequent lightning strikes (Karuna shechen, 2024). Concurrently, anthropogenic factors including agricultural residue burning, poachers' use of fires, and unintentional human negligence, have exacerbated wildfire occurrence (Shradha Khadka, Governance Monitoring Centre Nepal, 2024). Notably, Nepal's forest cover has doubled over the last three decades, increasing from 26% to 45% between 1992 and 2016 (Karan Deep Singh, The New York Times, 2022). While Nepal's afforestation initiatives represent a significant environmental achievement, addressing the escalating human-nature conflict and strengthening resilience to climate-induced disasters remain critical challenges for ensuring the sustainability of this fragile progress.

Northern India, bordering Nepal, also experienced extreme heatwave and drought in 2024, triggering unprecedented wildfire activity across several states (Reuters, 2024). Uttar Pradesh, for example, experienced its most severe wildfire season, marked by



record-breaking BA, carbon emissions, rate of growth, and fire size (**Figure A1**). Human activities, mainly land clearing and negligence, serve as the primary ignition source in Northern India, leading to uncontrolled wildfires. These fires are further exacerbated by the accumulation of dry pine needles in forests, which act as a ready fuel, and the steep Himalayan slopes, which accelerate the rate of fire growth (Vivek Saini, Climate Fact Checks, 2024). Agricultural practices in Northern India, a critical crop-producing region, have also contributed to the extreme wildfire season. Despite regulatory bans, post-harvest burning of crop residue has continued unabated in recent years (Arshad R. Zargar, CBS News, 2024). At the same time, temperature inversions coupled with Himalayan topographical blockages have trapped pollutants over Northern India. This phenomenon culminated in severe air haze episodes in New Delhi in November 2024, with $PM_{2.5}$ concentrations exceeding 200 μg/m³ across large parts of Northern India (CAMS, 2024).

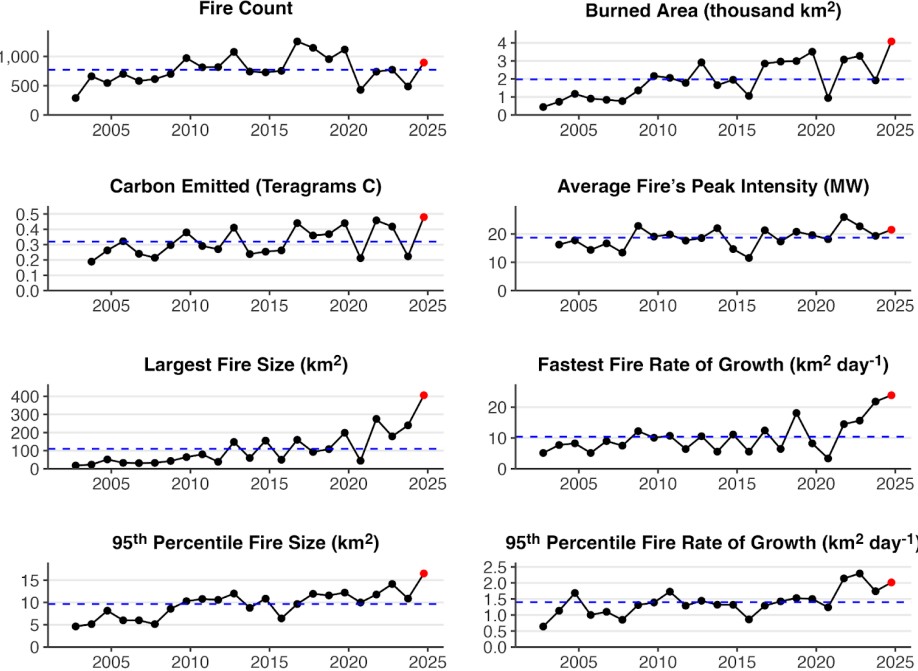

**Figure A1:** Summary of the 2024-2025 fire season in the Indian State of Uttar Pradesh. Time series show annual fire count, BA, C emissions totals within the region, as well as the average fire's peak fire intensity (95th percentile value of fire radiative power within fire perimeters), the 95th percentile fire size, fastest daily rate of growth, and 95th percentile fire daily rate of growth. Black dots show annual values prior to the latest fire season, red dots the values during the latest fire season, and blue dashed lines the average values across all fire seasons.

Although Russia generally experienced a typical fire season, several regions in Siberia recorded extreme fire activity. Two regions (Republic of Sakha and Zabaikalsky krai) accounted for 65% of the total forest area burned in Russia (Avialesookhrana, 2024) with 97% of the fires recorded in hard-to-reach areas according to official data from the Federal Forestry Agency (Rosleshoz, 2024). The high fire activity was associated with intense heat, decreasing precipitation and dry thunderstorms (Rosleshoz, 2024), which have become more frequent phenomena in Siberia in recent decades (Huang et al., 2024). Firefighting was complicated by strong winds and mountainous terrain (Rosleshoz, 2024). To attract





additional fire-fighting forces, federal emergency regimes were introduced from May 31 to November 8 in the Zabaikalsky krai and from June 28 to September 13 in the more northern Republic of Sakha, including in the Arctic Circle. In total, 139 redeployments involving 3,500 firefighters were carried out in 2024. The main causes of forest fires were lightning (48%), local population (39%) and fire transitions from other land categories (10%) (Rosleshoz, 2024). While the area burned in 2024 in the Republic of Sakha was not the highest compared to fire activity in the previous years, there is an increasing trend of fire activity and severity in the region over the last decade (ISDM, 2024), associated with weather anomalies (Tomshin and Solovyev, 2022) resulting in an increase in the duration of the fire season and the average area burned (Kirillina et al., 2020; Narita et al., 2021). The estimated total emissions for June 2024 were the third highest in the past two decades, following those of 2019 and 2020 (AMAP, 2024). In the Zabaikalsky krai, the total area burned in 2024 was about 7% of the area of the region, which is the highest value since 2010 (ISDM, 2025). Overall, both regions are considered hotspot areas of fire-induced change, where anthropogenic patterns and climate change are increasing ecosystem damage from wildfires and inhibiting recovery of natural ecosystems (Kukavskaya et al., 2016; Burrell et al., 2022).

Persistent dry and warm spring conditions in southwest China, particularly in Sichuan and Guizhou provinces, resulted in high-ranking BA anomalies (**Figure 2**). Strong winds exacerbated fire risks by increasing regional fire size and rate of spread, leading to large and fast-moving wildfires (Global Times, 2024). One of the most severe wildfires in Sichuan lasted 14 days, displacing more than 3,000 civilians across 11 villages and one community (Dou et al., 2024). Northeast China, including Heilongjiang and Jilin provinces, also experienced widespread wildfire anomalies during the spring season (**Figure 2; Table 2**). Contrary to the climate-driven wildfires in southwest China, these wildfires were predominantly anthropogenic originating from crop residue burning. The Chinese government implemented policies in 2013 and 2018 to control straw burning, a major contributor to air pollution, which initially achieved measurable success (Huang et al., 2021; Song et al., 2024). However, due to financial strain on rural communities and administrative pressures on local officials, recent policy adjustments have shifted from a zero-tolerance approach to a more flexible framework. This revised strategy permits controlled crop residue burning in designated areas during periods of low air pollution risk (Ding, Sixth Tone, 2025).

Earth observations data showed high-ranking BA anomalies, frequent fires, fires with large sizes, and rates of growth during 2024-25 in several regions of Lebanon, Palestine, Jordan, Iraq, Syria, United Arab Emirates, Philippines, and Laos (**Figure 2, Figure 4**), consistent with reports of persistent heatwave in these regions (Zachariah et al., 2024).

A drought that persisted from the 2024-25 fire season to the 2025-26 fire season has resulted in several highly impactful events in Asia (Faranda et al., 2025). Thus, from March 21st 2025, South Korea experienced its deadliest wildfires on record with very strong wind, burning across 11 regions and resulting in 31 deaths, 44 injuries, more than 3.3 thousand displaced people, and at least 4 thousand homes damaged (Yonhap, 2025).The wildfire in Iwate Prefecture, Japan, which started on February 26th 2025, was the country's largest wildfire in over 50 years, killing one person, destroying 221 buildings and forcing evacuation of over 4,5 thousand people (NHK, 2025). These events are not reviewed at length here, however they will be featured in future editions of the State of Wildfires report.

## A3. Europe

In 2024, wildfire activity in the European Union was close to the long-term average in terms of total BA, but characterized by strong regional contrasts; approximately 420,000 ha were burned, slightly above the 18-year average (San-Miguel-Ayanz et al., 2025), with some countries experiencing record-breaking seasons and others seeing minimal fire activity.





Across the continent, including in Turkey and Ukraine, a total of 1.82 million ha burned from March 2024 to February 2025 as recorded by the European Forest Fire Information System (2025), of which 48% pertain to large (>500 ha) fires. The EU Civil Protection Mechanism (EUCPM) was activated 16 times in response to wildfires, providing international assistance to Greece, Portugal, Cyprus, Bulgaria, Albania and North Macedonia (European Commission Emergency Response Coordination Centre, 2024).

The 2024 wildfire season in the Nordic and Baltic countries was the calmest in recent decades. While spring was drier and warmer than average in some areas, abundant summer precipitation limited fire spread. No major wildfire events were reported, and most incidents were confined to small wildfires caused by land-management activities (Swedish Firefighters, 2024). Likewise, wildfire activity in Western Europe during 2024 and early 2025 was subdued, as precipitation during spring and summer limited fire occurrence and spread across the region. France experienced one of its quietest seasons in recent decades, and similar conditions were observed in Belgium, the Netherlands, Ireland, and the UK (Global Wildfire Information System, 2025). The fire season was insignificant in Central Europe, because of wetter-than-average conditions during spring, especially in the Czech Republic and in Slovakia. However, Austria saw the highest number of fires and the largest BA since 2012 (Global Wildfire Information System, 2025) and Germany experienced a slightly above-average fire season, consistent with the trend of the previous five years. The most notable incident was a wildfire in Harz National Park in July, which led to the evacuation of approximately 500 people and involved 150 firefighters (Deutsche Welle, 2024).

In Southern Europe fire activity varied widely depending on seasonal precipitation and fire weather, with notable peaks in July-August (Balkans) and September (Portugal). In Portugal (**Figure S46**), 2024 was the most impactful year since 2017: 137,111 ha burned on the mainland, around 20% above the past decade's average, with 25 fires exceeding 1,000 ha, eight of which surpassed 5,000 ha (Instituto de Conservação da Natureza e Florestas, 2024). Most of these fires occurred as a sudden burst in mid-September in the northwest and under exceptional fire weather conditions (Instituto Português do Mar e Atmosfera, 2024). The Sever do Vouga complex and other major fires affected wildland-urban interface areas, resulting in 16 fatalities (Agência para a Gestão Integrada de Fogos Rurais, 2025), and €180 million in estimated losses across housing, infrastructure, forestry, and agriculture (Centro de Coordenação Regional Centro, 2024; Centro Pinus, 2024). Additionally, 48,272 ha of protected areas and Natura 2000 habitats burned (Gonçalves and Marcos, 2024). In Madeira island, a fire burned over 5,000 ha, entering the non fire-adapted laurel forest, a UNESCO World Heritage Site (Público, 2024).

BA in Spain, Italy and Greece was respectively 41, 51 and 73% of the 2012-2023 average (Global Wildfire Information System, 2025). In Greece, the drought lasted until mid November, lengthening the fire season and enabling unusual high-elevation fires in the north. Nonetheless, strengthened preparedness and fire suppression hindered the spread of many potentially large fires. The most destructive fire occurred near Varnavas in August, entering the NE suburbs of Athens and killing one person (Giannaros et al., 2024).

The 2024 fire season in the Balkans and Southeastern Europe was among the most severe in recent decades for several countries. Wildfire activity was substantial in North Macedonia, Serbia, Albania, Kosovo and Bulgaria, including multiple large-scale events requiring international firefighting assistance. In Albania, the largest wildfire surpassed 4,000 ha in the Dropull i Poshtëm region and the EU Civil Protection Mechanism was activated in response, with aerial support from Greece and Italy (Directorate-General for European Civil Protection and Humanitarian Aid Operations, 2024). Evacuations were carried out near the coastal town of Shengjin. Bulgaria experienced its worst fire season since 2007, with two wildfires in July destroying houses, the Sakar Mountain fire (Radio Bulgaria, 2024) and the Gorska Polyana fire (Novinite, 2024). North Macedonia (**Figure A2**) and Serbia faced their worst fire





seasons in over two decades, and a state of emergency was declared in the former
(Euronews, 2024), where four fires larger than 10,000 ha were recorded (European Forest
Fire Information System, 2025). On 16 August, Serbian authorities reported 135 active
wildfires within 24 hours (N1info, 2024). Other countries in the region, such as Croatia and
Montenegro, had seasons closer to the norm. In the Romanian Danube delta, and during an
unusually dry winter, 45,000 ha of wetlands burned in February 2025, a recurring
phenomenon with increasing extent (Volodymyr and Andiy, 2025).
BA in Turkey reached 270,000 ha, about 65% of the previous 12-years average (Global
Wildfire Information System, 2025) but with noticeable societal consequences. Most large
fires (up to 7000 ha) occurred in the province of Mardin (European Forest Fire Information
System, 2025), including a rapidly spreading fire that burned farmland and impacted villages
on 20 June, killing 15 and additionally injuring at least 70 people (The Nation, 2024). A fire
that started near the coastal city of İzmir on 15 August brought havoc to the wildland-urban
interface and ended up burning houses and injuring 78 people (Ozerkan, 2024).
Long periods of high fire danger combined with intensified aggression and scarcity of
firefighting resources set the scene in Ukraine. The fire season was severe in extent and
nearly 1 million ha burned between March 2024 and February 2025 (European Forest Fire
Information System, 2025). This is larger than the combined BA in all of Europe, Middle East
and North Africa (San-Miguel-Ayanz et al., 2025). As the majority of these fires are located
near the front line in the eastern part of the country, warfare was presumably a major driver
of their ignition, with forests seemingly accounting for a larger share of BA than in the recent
past (The Guardian, 2025). Nonetheless, higher BA had been recorded in the past, namely
>2 million ha in both 2014 and 2015 (Global Wildfire Information System, 2025).

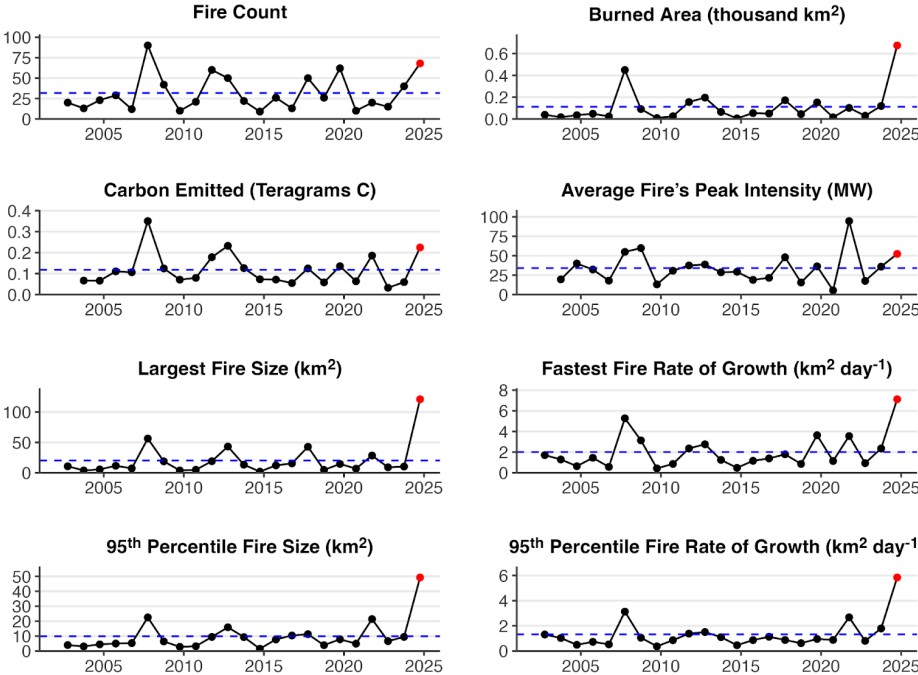

**Figure A2:** Summary of the 2024-2025 fire season in North Macedonia, as in **Figure A1**.





## A4. North America

Wildfires across North America were characterized by above average activity in Canada and the United States and a record-breaking season in Mexico in 2024-2025. This included multiple wildfires that resulted in substantial impacts to human communities, including the Eaton and Palisades wildfires, which are among the most destructive in California, USA history. Following a record breaking 2023 wildfire year, during which almost 150,000 km$^2$ burned, Canada once again experienced an above average wildfire season in 2024. A total of 5,686 wildfires burned approximately 46,000 km$^2$, marking the six-highest area burned since 1972 based on national records (Skakun et al. 2024). In the United States, over 64,000 wildfires burned over 36,000 km$^2$ in 2024, exceeding both the previous 5 and 10-year averages (NICC, 2024). The USA also recorded the second-highest number of Level 4 and 5 National Fire Preparedness days since 1990, reflecting elevated national fire suppression resource commitment associated with high potential for continued emerging wildfires (NICC, 2024). National fire records for Mexico suggest that the country experienced more that 8,000 wildfires in 2024, setting a record for area burned - over 16,500 km$^2$ - since record keeping began in 1998 (Comisión Nacional Forestal, 2025), though this record is not reflected in the global datasets compiled as part of this report.

Much of Canada experienced earlier-than-normal snowmelt in 2024, resulting in an early onset of the wildfire season. For example, parts of Alberta experienced snowmelt up to 30 days earlier than average. Drought conditions, which were prevalent across the country in 2023, persisted into 2024 in much of western Canada. Holdover fires from 2023, which smouldered through the winter in northern British Columbia, Alberta (**Figure A3**), and parts of the Northwest Territories, reignited in early spring due to warm and dry conditions (Kolden et al. 2025). This contributed to above average area burned and wildfire emissions in May (Copernicus Atmosphere Monitoring Service, 2024). Wildfires in May led to evacuations of Fort Nelson, British Columbia and Fort McMurray, Alberta - an area previously affected by Canada's costliest wildfire in 2016 (Canadian Forest Service, 2025).

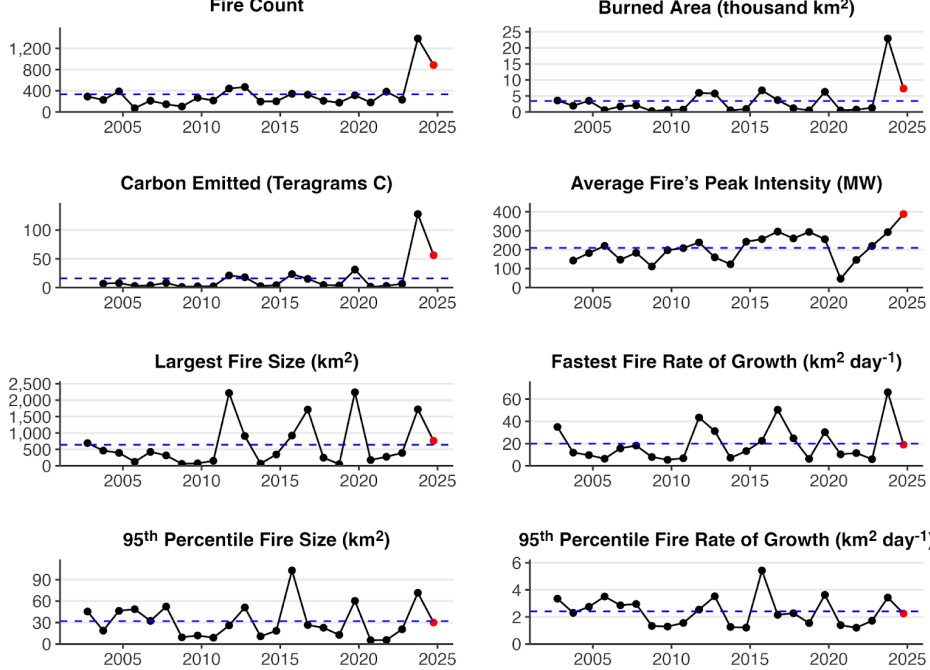


**Figure A3:** Summary of the 2024-2025 fire season in Alberta, Canada, as in **Figure A1**.

Most of the USA was characterized by normal to high precipitation at the start of 2024, with
minimal wildfire activity (NICC, 2024). A heat wave at the end of February 2024 in the
southern plains combined with strong winds and high fine fuel loads led to multiple large
wildfires, including the record-breaking Smokehouse Creek Wildfire in the Texas Panhandle
and western Oklahoma that burned over 4000 km² and resulted in two fatalities before
reaching 100% containment in March (Texas House of Representatives Investigative
Committee on the Panhandle Wildfires, 2024). Wildfire risk in the southern plains remained
elevated for several weeks. Warm and dry conditions in March led to an increase in activity
in the central Appalachians region of the eastern US, with the Virginia Department of
Forestry reporting over 100 wildfires in 48 hours. By early April, fire activity peaked for the
spring fire season in the southern and eastern US. Dry and windy conditions prompted
significant growth of large wildfires burning in New Mexico; however, fire activity remained
below average in the USA in May (NICC, 2024).
Wildfires in Mexico started increasing in March during Mexico's typical wildfire season.
Warm and dry conditions helped to fuel hundreds of wildfires (Comisión Nacional Forestal,
2025; NASA Earth Observatory, 2024b) contributing to Mexico's record breaking wildfire
season with anomalously high carbon emissions. Wildfire numbers peaked by mid-March
through early May (Comisión Nacional Forestal, 2025).
Wildfire activity remained high across Canada during the summer of 2024, with many
regions experiencing above-average area burned. Areas including New Brunswick in the
east, and the Northwest Territories, recorded area burned totals among the top five highest
since 1972 (Skakun et al., 2024). Hot and dry conditions in July resulted in wildfires forcing
the evacuation of Labrador City in Newfoundland and Labrador and John D'Or Prairie and



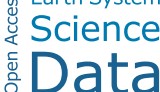

Fox Lake in Alberta (Canadian Forest Service, 2025). In late July during a period of extreme 99[th] percentile fire weather, a fast-moving wildfire resulted in the evacuation of the town of Jasper, Alberta and destroyed 358 structures resulting in an estimated $1.23 billion in damages - the second costliest wildfire in Canadian history (Kolden et al. 2025; Insurance Bureau of Canada, 2025). There were two fatalities in July related to fire suppression operations in Alberta and the Yukon.

Large fires continued to burn in northern regions of British Columbia, Alberta, and throughout the Northwest Territories throughout August and into the fall, resulting in the fourth, sixth, fifth highest area burned for these areas, respectively, since 1972 (Skakun et al., 2024). Significant fire activity also developed in Saskatchewan, Manitoba, and Ontario in August, and New Brunswick experienced the second highest area burned since 1972 (Skakun et al. 2024). In total, 91 wildfire-related evacuations took place across Canada during the 2024 season, affecting 56,000 people (Canadian Forest Service, 2025). According to estimates from the Copernicus Atmosphere Monitoring System, the 2024 wildfire season in Canada produced the second-highest total emissions recorded since 2003 - surpassed only by the record-breaking 2023 season (Parrington & Di Tomaso, 2024).

Wildfire activity began to pick up in the USA during the later part of June, with multiple fires in New Mexico resulting in several hundred structures burned (NICC, 2024), two fatalities, and over $1 billion in damages (NCEI, 2025). By July, an extreme and long-lasting heatwave across the western US spurred numerous large wildfires, including the Park Fire in Northern California that drove thousands to evacuate and destroyed over 700 structures (CALFIRE, 2025). Record breaking dry conditions in Oregon and Washington led to over 100 human-caused wildfires by early July (US Forest Service, 2024), contributing to a record setting year in BA and anomalously high carbon emissions in Oregon (**Figure S47**). Through July and August, hot and dry weather drove numerous large wildfires in the northwestern front range, including the Stone Canyon wildfire in Colorado that resulted in one fatality and multiple burned homes and the Remington wildfire in Wyoming that killed hundreds of cattle. During September, numerous dry lightning strike wildfires occurred in the northwestern US along with multiple wildfires in southern California associated with extreme heat, including the Airport Fire that resulted in 22 injuries and 194 damaged structures (CALFIRE, 2025).

Fall was anomalously warm and dry across much of the continental US, with 87% classified as abnormally dry or in drought by early November (NICC, 2024).The northeast US experienced hundreds of wildfires that interacted with densely populated regions in October and November coincident with record-dry conditions and warm temperatures across multiple states. For instance, New York City experienced its highest number of recorded wildfires during a two-week period, with every borough experiencing multiple wildfires. The conditions were unseasonable, with Connecticut, Massachusetts, and Rhode Island setting record red flag days, despite typical peaks for red flag days occurring in spring (NOAA, 2024). Massachusetts experienced its most active fall fire season in over 40 years (NICC, 2024). Two fatalities and hundreds of structures were destroyed before rainfall associated with an extratropical cyclone halted the fall fire season in the northeast in late November.

Wildfire activity remained low at the end of 2024 and beginning of 2025, except in southern California. Southern California is climatically prone to experiencing a downslope (katabatic) wind during the late autumn and winter months known locally as Santa Ana winds. Historically, the most devastating wildfires in California have occurred when a delayed onset of autumn precipitation coincides with a Santa Ana wind event (Kolden and Abatzoglou, 2018), but such concurrences are increasing in frequency with climate change (Goss et al., 2020). In November and December, Santa Ana wind events produced wildfires that burned nearly 10,000 ha and destroyed over 250 structures, however, this was just a precursor. In January 2025, the most disastrous wildfire event in modern US history occurred in Los Angeles County, California. Prolonged drought conditions, unseasonably warm winter



temperatures, and exceptionally powerful Santa Ana winds exceeding 140 km/h created extreme fire weather conditions (Barnes et al., 2025; CNN, 2025). Fire potential was also exacerbated by anomalously wet winters for two years prior, which increased the fine fuel load in the region. The potential for extreme wildfires to develop under dry downslope winds was predicted several days in advance, including by the National Interagency Fire Center (NIFC), the National Weather Service (NWS), and the Storm Prediction Center (SPC; see summary by Wikipedia, 2025) as well as by specialist commentators (e.g. Swain, 2025).

The two most destructive fires-Palisades and Eaton-that burned during the event occurred in the same general locations as destructive fires in 1961 and 1993 during other Santa Ana wind events. These two fires resulted in numerous outcomes with widespread and severe consequences. Among the most devastating were the high fatalities and extensive structure loss. Over 11,500 homes were destroyed across Los Angeles County, and at least 30 lives were lost, according to the Los Angeles County Coroner (2025). Specifically, the Palisades Fire damaged or destroyed nearly 8,000 structures, while the Eaton Fire impacted over 10,000 (CALFIRE, 2025; Wikipedia, 2025). The fires also triggered mass evacuations. At the peak of the crisis, at least 153,000 people were forced to evacuate, with up to 200,000 under evacuation warnings or orders (USGS, 2025b; NPR, 2025; Wikipedia, 2025).

In addition to human displacement and infrastructure damage, the fires severely affected both air and water quality. Air pollution reached hazardous levels, contributing to negative health outcomes for thousands. During the fires, peak $PM_{2.5}$ levels reached 483 µg/m³, an order of magnitude greater than the 35 µg/m³ daily standard set by the US Environmental Protection Agency, resulting in a prolonged period of hazardous air quality (California Air Resources Board, 2025). Municipal water supplies were similarly impacted, with water considered unsafe for tens of thousands of residents in the burned areas for several weeks following the fires (City of Pasadena, 2025). Beyond Los Angeles, the political fallout from the crisis led to federal orders to release over 8.3 million cubic meters of water from federal reservoirs further north in California. However, this water did not flow to southern California and was instead vital for irrigating crops in the state's heavily agricultural Central Valley (Levin et al., 2025).

The economic consequences were equally severe. Total economic losses were estimated at US$140 billion, factoring in property destruction, health costs, business disruption, and infrastructure damage, making this one of the most costly wildfire events in US history (LAEDC, 2025; UCLA Anderson School of Management, 2025). Wider economic disruption is also projected, with estimated losses of US$4.6-8.9 billion in economic output over five years, 25,000-50,000 job-years lost, and reductions in labour income of US$1.9-3.7 billion (LAEDC, 2025). The Palisades and Eaton fires directly affected nearly 2,000 businesses (LAEDC, 2025). Moreover, as Los Angeles hosts the largest port on the US Pacific coast, these fires disrupted broader supply chains connected to the Port of LA (ASU, 2025).

Insured losses added another layer of financial strain, with industry estimates ranging from $20 billion to $75 billion (PreventionWeb, 2025; Morningstar DBRS, 2025; Insurance Insider, 2025; UCLA Anderson, 2025). This placed substantial pressure on the already volatile home insurance market in California, as well as on most global re-insurers.

The fires also deepened Southern California's ongoing housing and affordability crisis. Thousands of affordable housing units were lost, worsening the existing housing shortage, displacing large numbers of low-income residents, and exacerbating the region's homelessness problem (Urban Land Institute, 2025; UCLA Anderson, 2025; Vox, 2025). This led to ripple effects, with mass displacement into surrounding communities and beyond in the months that followed (NYT, 2025).





Finally, the aftermath of the fires brought additional physical hazards in the form of debris flows. Given southern California's geology, the region is highly susceptible to erosion and debris flows following wildfires. Several such events occurred after high-intensity rainfall in the weeks following the fires, causing further damage and prompting hundreds of additional evacuations in and near the recently burned areas (USGSa, 2025).

## A5. Oceania

Oceania experienced relatively moderate levels of fire during the 2024-25 fire season, although there were still a series of high profile and high impact events across the region. Overall, however, the season did not reach the magnitude of the previous year, which ranked among the top 5 years for BA in Australia since 2002. Where fires occurred and had impacts, lightning and sustained dryness were prominent drivers (Bureau of Meteorology, 2024; Dowdy and Brown, 2025).

The 2024-25 fire season in Western Australia was characterised by record-high temperatures, variable rainfall, and significant soil moisture deficits in coastal areas of the South, Southwest, and West. Over 1,000 large fires burned about 470,000 ha, many in coastal shrubland and woodland over the ~800 km stretch from Gingin, north of Perth, to Carnarvon. The largest fire by area burned occurred near Cervantes in November, where fire ignited by a car crash went on to burn more than 80,000 ha and severely impact local honey production. In the inland Goldfields region at Skeleton Rocks, more than 44,000 ha of Mallee-heath vegetation of the Great Western Woodlands were burned (according to the Department of Fire Emergency Services (DFES), Rui Feix, pers. comm.). This fire reached extreme intensity, impacting fire-sensitive species and post-fire regeneration cycles in an ecosystem that requires long intervals to recover. A lithium mine in the area was also directly impacted by the fire. Four other large incidents were recorded in the shrublands of the Great Western Woodlands, further affecting these vulnerable ecosystems. Between December and March, numerous fires occurred in the grasslands of the Wheatbelt and Esperance, as well as in the forests of the Perth Hills. These included fires that collectively destroyed seven residential properties in areas east of Mundaring, Arthur River, Wooroloo, and Waroona. In February and March, lightning ignited several large bushfires in native forests and coastal shrubland around Manjimup. Some of these fires burned for up to five weeks and affected more than 42,000 ha, including areas of Shannon and D'Entrecasteaux National Parks (DFES, Rui Feix, pers. comm.). These incidents required significant aerial support and personnel deployments, including interstate assistance.

Above average rainfall was recorded in Central Australia, leading to expectations of strong grass fuel growth and another period of increased fire activity, after last year's above average season (Verhoeven et al. 2020; Ruscalleda-Alvarez et al. 2023). By the end of October 2024, over 5.7 million ha had burned, much of it stemming from an intense band of dry lightning stretching from the Northern Territory into Queensland in October (according to Northern Territory Fire and Emergency Services, Maggie Towers, pers. comm.). Many of these fires combined with a particularly large fire complex near Devil's Marbles Conservation Reserve (450,000 ha) (NTFES, Maggie Towers, pers. comm.). The fire threatened hotels and other infrastructure and caused temporary closure of a major highway. In late January 2025, a bushfire swept through the West MacDonnell Ranges, affecting approximately 80,000 ha across the Tjoritja/West MacDonnell National Park, Standley Chasm, the Tyurretye and Iwupataka Aboriginal Land Trusts, as well as nearby pastoral properties (NTFES, Maggie Towers, pers. comm.). Standley Chasm and sections of the Larapinta Trail were closed for several days while a 10-day multi-agency effort worked to contain the fire.

Queensland's north west saw heightened fire activity during Spring, with fire fighters responding to 40 incidents in Mount Isa alone. One of these fires burned for nearly two months, reaching over 100,000 ha according to Queensland Fire Department, (Russell

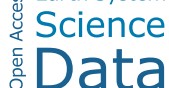

Stephens-Peacock, pers. comm.). The fires caused an increase in hospital admissions due
to respiratory illnesses and impacted mining operations, pastoral property and Lawn Hill
National Park. The fires affected the habitat and food sources of endangered species such
as the Carpentarian Grass wren, found only in north-western Queensland.
In 2024-25, eastern Australia, comprising southern Queensland, New South Wales (NSW),
and the Australian Capital Territory (ACT), experienced a notably warm period, with
significant rainfall variation across regions and seasons. Although temperatures were above
average in the austral spring, many areas received above average rainfall, thereby reducing
fire occurrence and impacts. Repeated dry lightning started a number of complexes of fires
in remote and difficult to access terrain across NSW, including areas like Lithgow, the
Hawkesbury, Bulga and around Tamworth. Despite the number of fires, NSW saw more
moderate fire weather than other parts of the country and Emergency Warnings were only
issued for three fires (according to New South Wales Rural Fire Service, David Field, pers.
comm.).
The south to south east of Australia (including the states of South Australia, Victoria and
Tasmania) experienced record dryness in the leadup to the fire season. Fires in Chappelvale
and Casterton-Edenhope in late spring signaled an early start to the fire season in Victoria.
In December a band of dry lightning ignited a number of fires including several in the
Grampians National Park. About 75,000 ha burned in the Grampians, affecting culturally and
ecologically sensitive areas. The coincidence of the fire with Christmas and the peak holiday
season led to major tourism losses and extensive community evacuations. This fire was
contained by January 6 but later in January another band of dry lightning passed through the
west of the state, this time affecting the western side of the Grampians burning another
almost 60,000 ha (according to Country Fire Authority, Sarah Harris, pers. comm.). By
season's end over two thirds of this important National Park was impacted by fire. Another
significant fire occurred on December 26, a public holiday, in Little Desert National Park in
the state's west. This fire was an extremely fast-moving fire with approximately 65,000 ha
burning in less than eight hours and a final area burned of 90,000 ha (according to Country
Fire Authority, Sarah Harris, pers. comm.). These fires required interstate deployments to
assist in the fire fight. The fire season concluded with challenging fires that burned through
rugged terrain in the Gippsland area, impacting the World Heritage-listed Budjim National
Park with its significant cultural heritage. Several planned burns escaped during the season,
highlighting the significant dryness of the area.
In South Australia dry lightning storms in early February combined with severe drought
conditions to cause the Wilmington fire, which burned approximately half of Mount
Remarkable National Park. Firefighting efforts reduced the impact to human, ecological and
cultural assets. Lightning storms in February and March also caused an above average
number of fires in eastern parts of South Australia. While impacts were limited, firefighting
resources were strained. (South Australia, Country Fire Service, Simeon Telfer pers. comm.)
Tasmania faced a significant bushfire season, with up to 100,000 ha burned in the state's
northwest, including sensitive ecosystems such as the Tarkine rainforest and the alpine
vegetation around Cradle Mountain (according to Tasmania Fire Service, Chris Collins, pers.
comm.). Sparked by intense lightning storms in remote and rugged terrain, the fires required
interstate support to assist with firefighting efforts. The blazes led to evacuations, threatened
heritage sites and caused major disruptions to local businesses and the tourism industry.
In New Zealand the 2024-2025 fire season was moderate, with a couple of minor fires at the
end of the 2023/24 fire season (Mar-Jun 2024) and a few more significant fires during the
2024/25 fire season (Jul 2024-Feb 2025). A key feature was the occurrence of a couple of
significant wetland fires that burned large areas of peatland (2,271 ha) and damaged flora
and fauna habitat. These fires occurred at Whangamarino Wetland, Waikato (central North



Island) in October 2024 and Tiwai Peninsula, Southland (southern South Island) in late January 2025, with both fires just over 1,000 ha. The fires followed two major peatland fires in 2022 at Kaimaumau in the far north (2,434 ha), and Awarua in the south (890 ha and close to this season's Tiwai fire) (according to Fire and Emergency New Zealand, Grant Pearce, pers. comm.). Carbon emissions are likely to be high, given the 2022 fires were estimated to release more than 620,000 t $CO_2$ (Pronger et al., 2024). There were a number of other noted fires in a mixture of vegetation types including in Canterbury, Northland and North Otago. However, unlike recent years, there were no major house loss incidents, with just a few homes and outbuildings lost across the multiple fires.

## A6. South America

The 2024-25 fire season was a remarkable year for fire in South America, with seven of its 13 countries reaching new records in BA since 2002 and widespread records in the fire size, growth rate and intensity distributions (**Figure 3**; **Figure 4**). Anomalies in BA commenced early in 2024 and persisted through November in some regions (**Figure S4**). As discussed in Section 2.2.2 and Sections 4-6, intense drought and fire weather affected much of South America during the 2024-25 fire season and this drought occurred at a time when socioeconomic factors are increasingly cited as drivers of shifting fire regimes and timing. The event is part of a trend towards an earlier onset of the fire season since 2020, with new record fire counts set for the months of March to May in 2020 and for January in 2022, based on monitoring by Brazil's National Institute for Space Research (INPE) since 1998 (INPE, 2025). During 2024, January, February and June presented the second highest value on record (previous record during 2003 for January and 2007 for the other months, respectively). Fires have expanded into new territories, driven by a combination of climate variability, shifting land-use practices, and governance challenges, as discussed in the study cases, below.

Across South America, the number of fire hotspots recorded by the Queimadas/INPE system (511 thousand hotspots in 2024) rivalled the previous record set in 2010 (523 thousand hotspots) (INPE, 2025). Compounding climate and human drivers likely led to a widespread extreme fire year across the continent in 2024-2025. The land use fire dependent practices, associated with new deforestation frontiers during an extreme drought year amplified the fire crisis. Amidst rising socioeconomic and environmental impacts of fires in the region, researchers have been calling on governments across the globe to rethink strategies for combating the root causes of extreme wildfires, from climate change to fire-free agricultural practices (UNEP, 2022). Increases in extreme droughts with already vulnerable forest due to extreme climatological events are expected and therefore controlling ignition sources are the only immediate measure for preventing 2024-like scenarios. In this context, major fire events in terms of largest fire size emerged in many parts of Brazil, in Peru, Ecuador and Bolivia during the 2024-25 fire season, with unprecedented levels of BA and exceptional fire weather (**Figure S2**).

In Brazil, one of the most intense droughts in decades, combined with the expansion of the agricultural frontier in Amazonas and Pará states (Santos et al., 2023), caused fires to persist nearly year-round (**Figure S2.4**). In northern Brazil, including much of the Amazon biome, several states such as Amazonas and Pará experienced their largest BA on record. Other states, including Mato Grosso, São Paulo, and Paraná, recorded their highest fire extent in a single year. In the Pantanal biome, Mato Grosso do Sul experienced the second-largest BA extent on record, the fourth in rank in fire size and fifth regarding the fastest growth. This resulted in estimated losses to agribusiness caused by the fires amounting to R$ 1.2 billion (~ $222 million) (Câmara, 2024). In addition, Pantanal recorded particulate matter concentrations of 903.2 μg/m³ in September 2024 (Viana et al., 2024), which is 60 times higher than the World Health Organization (WHO) standards. Efforts to contain the flames lasted 78 days and involved the National Force, local communities,



environmental organizations, and state fire brigades (Nunes, 2025). The response faced significant challenges, particularly in remote border areas with difficult access and complex logistics. Providing support to isolated populations was especially difficult, with reported cases of respiratory illnesses worsened by smoke exposure, as well as emotional distress, including stress and anxiety (Nunes, 2025).

São Paulo and Mato Grosso state, both centres of large-scale crop production, experienced the fourth-larged BA extent on record. Regarding fire intensity, 2024 was the record for São Paulo, Paraná, Mato Grosso do Sul, Rio de Janeiro and Roraima, and the second in the rank for Amazonas and Goias. All together, from the southeast to the north of the country, records in one or more fire metrics were observed during this period, placing Brazil in a state of fire emergency.

In general, early fire season onset and long duration occurred across most Brazilian regions, with the first month of anomalous fire ranging from March in most of the Amazonian states and extending through to December. In fact, more fire hotspots were detected in February and March 2024 than in any year since 1998 (INPE, 2025). Record fire counts were observed across states covering more than half of Brazil's territory, and represented a threat almost during the entire year, posing challenges for managing the wildfires response and combat. By August 2024, the National Centre for Early Warning of Natural Disasters (CEMADEN, 2024) pointed out that the drought, covering Amazonia to the southeast, initiated in the second half of 2023, was already one of the strongest in decades. Data from the National Secretariat for Civil Protection and Defence (S2ID, 2024), in December 2024 pointed out that there were 21 of the 27 states with a recognized decree either in state of emergency or calamity due to the drought, affecting more than 520 municipalities in the country. These conditions brought widespread devastation across Brazil in 2024, impacting urban and rural communities and affecting an estimated 18.9 million people nationwide (CNM, 2024). Fire disaster forced 10,700 people from their homes, resulting in housing instability and severe disruptions to livelihoods. Thousands more were affected by the breakdown of essential services, such as school closures (CNM, 2024). Although Brazil does not have an official database on wildfire-related fatalities, existing records point to a rising death toll (Carvalho et al., 2025). Estimates have identified 186 deaths between 2020 and 2024, with 38 in 2024 alone. However, the actual number is likely higher due to underreporting.

Notably, the state of São Paulo in Brazil recorded 8,712 hotspot fires, the highest number since 1998 (INPE, 2025). August and September together accounted for approximately 70 % of these detections (6,134), roughly four times the 1998-2023 August average (914 hotspot fires) and three times the corresponding September average (848 hotspot fires). According to a study by the Amazon Environmental Research Institute (IPAM, 2024), of the 2,600 hotspots fires recorded in the state of São Paulo between August 22 and 24, 81% were in areas of agricultural use - drawing attention to the fact that the state recorded, on the 23rd August alone, more hotspots than the entire Amazon biome. In an even more alarming interval, analysed images from the geostationary satellite indicate that the smoke columns in western São Paulo appeared in just 90 minutes, between 10:30 AM and 12:00 PM on the 23rd August, and, on that same day, the number of fires jumped from 25 to 1,886 hotspot fires, reinforcing the hypothesis of orchestrated action and the unprecedented intensity of these fires.

Amazonas state, the largest of Brazil's Amazonian states, can be pointed out as an epicentre of wildfires and its impacts. During 2024, it was ranked as first in number of fires (**Figure 4**) and presented a historical record of fire occurrence in June, July and August, consecutively, since the monitoring began in 1998 (INPE, 2025). Moreover, it was the 3rd year with the fastest fire growth rates, with a fire season lasting for 8 months. It has been estimated that fires affected over 790,000 ha of forests, approximately 39% of the affected area, especially in the southern region of the state (Alencar et al., 2022). The Amazonas





state has been facing an increase in the deforestation rate since 2021, mainly in the southern region, following the pressure and political speech of Brazil's BR-319 Highway paving. The lack of governance, associated with illegal logging, land grabbing and public lands invasion, are some of the drivers of the fire peaks observed in the region (Fearnside, 2022). Moreover, it has been estimated that the population from this region has been exposed to aerosols emitted from the wildfires causing a pollution of up to 113 µg/m³, 653% above the 15 µg/m³ standard set by the WHO, according to the data from the Atmosphere Monitoring Service (CAMS) and Copernicus Climate Change Service (C3S).

A recent report from the Global Forest Watch (World Resources Institute, 2025) also showed widespread high levels of forest loss (stand-replacing fire extent) to wildfire in 2024 in the Amazon biome (including both Brazil and neighbouring Amazonian countries). The highest rates of forest loss since 2016 was observed, with total forest loss more than doubling in 2024 versus 2023 and 60% of those losses were attributed to wildfires. Note that Global Forest Watch data define "forest loss" as the complete removal of tree canopy, including areas affected by stand-replacing fires, but do not capture more subtle or partial fire-related degradation. As a result, the data may overestimate deforestation while underestimating degradation, limiting understanding of the broader ecological impacts of wildfires on forests. Moreover, Indigenous communities were disproportionately affected by wildfires in 2024, a year that recorded the highest number of fires in territories inhabited by isolated Indigenous peoples (COIAB, 2024). In 2024, fires in Indigenous lands in Brazil increased by 81% compared to 2023, accounting for the largest share of Amazonia fires at 24% (Alencar et al., 2024). In Roraima, uncontrolled fires in indigenous lands have degraded air quality, ravaged crops, homes, and native vegetation leading to food and water insecurity (ISA, 2024). The fires have further worsened the ongoing humanitarian crisis in the Yanomami Territory, Brazil's largest Indigenous land. Local organizations estimate that at least 70,000 people across urban and rural communities were affected by the lack of access to clean water, a result of the compounded impacts of drought and fire (WWF-Brasil, 2024).

The implications of extreme fire activity in Amazonia extend beyond immediate ecological damage. As a globally significant carbon sink and a key part of the terrestrial hydrological cycle, the Amazon stores an estimated 100-120 Pg of carbon (Malhi et al., 2006). Intensified fire regimes risk accelerating forest degradation, potentially triggering a biome-scale shift from net carbon sink to a significant carbon source, releasing several petagrams of carbon and exacerbating global warming through positive feedbacks (Gatti et al., 2021). Fire-driven environmental degradation also poses public health risks and economic instability. Biomass burning increases respiratory illness, especially among populations exposed to prolonged smoke (Campanharo et al., 2019, 2021). Economically, fire reduces agricultural productivity, damages infrastructure, and undermines regional development, compounding poverty and inequality. Costs extend to firefighting programmes and personnel (Morello et al., 2020), as well as hospitalisations from respiratory or other fire-related conditions (Machado-Silva et al., 2020). Rising fire activity may also weaken the effectiveness of forest conservation and restoration policies, including those tied to international climate agreements, threatening long-term mitigation and adaptation efforts.
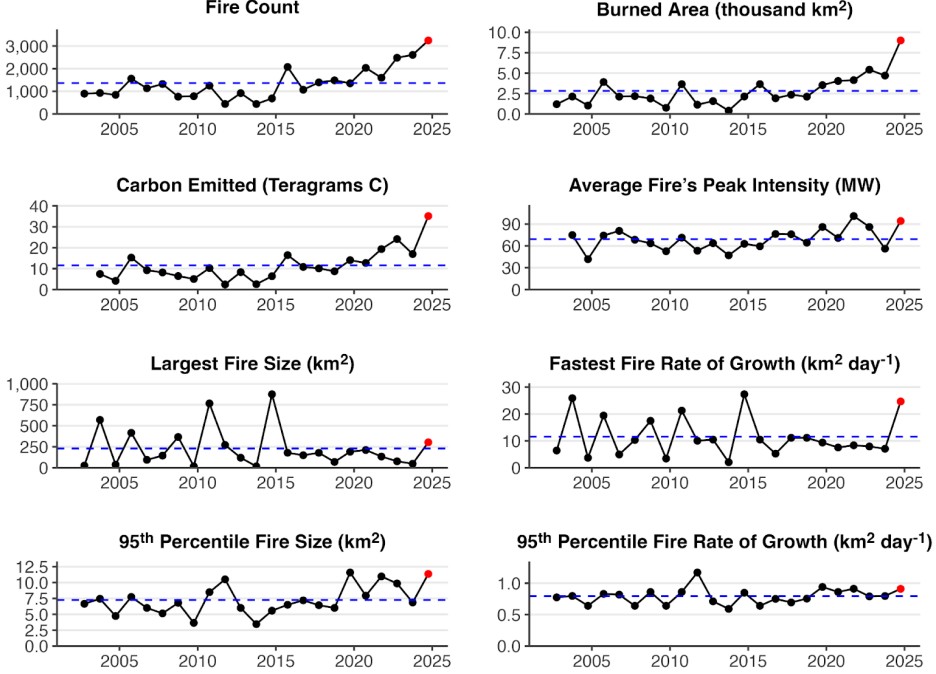

**Figure A4:** Summary of the 2024-2025 fire season in Brazil's Amazonas State, as in **Figure A1**.

Bolivia endured one of its worst fire seasons on record by many measures, intensified by the El Niño phenomenon, record temperatures, and accelerating deforestation and contributing significant carbon emissions to the atmosphere (**Figure A5**). These conditions intensified fire outbreaks, especially in ecologically vulnerable regions such as the Chiquitania and Amazonian lowlands (Ruiz, 2025). The cumulative number of fire hotspots in 2024 was 923,464, with 77% occurring in Santa Cruz (Chiquitano dry forest), 19% in Beni (Amazonian lowlands), 1.6% in La Paz, and the rest in other departments including Pando (north Amazonia) (CEJIS, 2024). A recent report from Global Forest Watch (World Resources Institute, 2025), found that forest loss in Bolivia tripled in 2024 versus 2023 and was many times over the annual mean since 2002, with 60% of those losses related to wildfires. The forest fires have been attributed to a combination of one of the most severe droughts on record as well as a number of socioeconomic factors and government policies that encourage agricultural expansion, such as the lifting of soy and beef export quotas, removal of import taxes on agrochemicals and machinery (World Resources Institute, 2025).

The number of ha burned was a contentious issue in the country. According to the NGO Fundación Tierra (2024), in September 2024 about 10 million ha were burned. The wildfires extended up to November. In early January 2025 an independent group of experts of the national journal (El Deber, 2025) reported that 14 million ha burned based on the MODIS Terra sensor. In April 2025, the ministry of environment officially reported 12.6 million ha burned in 2024, 12% of the country's territory, with 57% corresponding to primary forest and 43% pastures and agricultural lands (Ministerio de Medio Ambiente y Agua, 2025). Although wildfires have been occurring regularly in Bolivia over the past decade, the events of 2024 have been the most catastrophic to date. The event is considered the second megafire after



the one that occurred in 2019. Indigenous lands, protected areas, and fiscal lands were among the most impacted categories. The Global Forest Watch cited a lack of early warning systems and adequate firefighting resources as a factor contributing to high rural exposure to fire and urban exposure to wildfire smoke (World Resources Institute, 2025). An investigation by Fundación Tierra (2025) reports that wildfires in Bolivia are mostly intentional, with 66% being maliciously set and 34% resulting from out-of-control slash-and-burn agricultural practices.

The Bolivian Air Contamination Index reached 537 in the city of Cobija, northern Bolivia (Silva Trigo, 2024), corresponding to a $PM_{2.5}$ concentration of over 500 µg/m³ (24-hour average), a level considered extremely harmful and impactful to the health of millions of people in the region and beyond. As a result, the government declared a sanitary emergency. In addition to the extensive environmental destruction and incomparable biodiversity loss, these fires have significantly increased atmospheric carbon emissions, exacerbating regional and global climate challenges. It is important to note that laws and regulations in Bolivia encourage agricultural and livestock expansion and are lenient towards the use of fire (Yifan He et al. 2025). Encroachment and illegal land occupation are also pointed to as causes of provoked wildfires in Bolivia. Efforts in the legislative branch to prohibit or amend these regulations have not been successful thus far. Therefore, there is a looming risk that similar events may occur again in the near future.

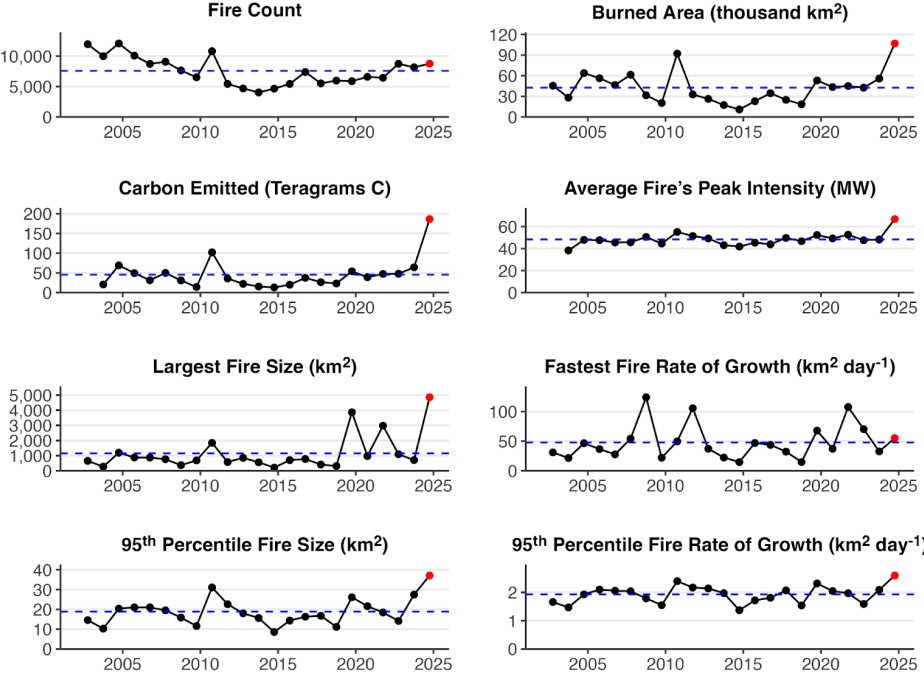

**Figure A5:** Summary of the 2024-2025 fire season in Bolivia, as in **Figure A1**.

In early 2024, Venezuela experienced its most intense wildfire season on record, with over 30,000 active fires between January and March (NASA FIRMS, 2025). Unlike Brazil, Venezuela's peak fire season runs from December to April, driven by the northward shift of the Intertropical Convergence Zone (ITCZ; Katz & Giannini, 2010; Ramírez & Gómez, 2021),





which were intensified by the 2023-2024 El Niño , one of the strongest in decades, creating an extreme fire weather window (NOAA CPC, 2024). Fires have historically been concentrated in the Orinoco Llanos, a savanna-dominated region covering approximately one-third of Venezuela, where fire is used for agricultural purposes and grazing (Bilbao et al., 2020). More recently, deforestation in tropical forests south of the Orinoco has fueled large fires, like those seen in 2019 (Lizundia-Loiola et al., 2020). In 2024, wildfires impacted nearly all ecosystems, from Amazonian humid forests in Bolívar and Amazonas (5,600+ fires, including Canaima National Park), to flooded savannas in Apure, cloud forests in Henri Pittier, and an estimated 36,000 ha of Caribbean pine lost in Uverito, Latin America's largest plantation (Ciudad CCS, 2024; Lozada, 2024). Since 2019, Venezuela's National Parks Institute (INPARQUES) has promoted an intercultural Integrated Fire Management (IFM) strategy, coordinated by an intersectoral team involving government agencies, local communities, and researchers (Bilbao et al., 2022). With support from FAO and RAMIF (under ACTO), this national IFM system aims to strengthen fire management efforts in response to Venezuela's vast ecological and territorial complexity, as well as to the increasing extreme fire weather conditions projected for the region , including higher temperatures, prolonged dryness, and lower humidity (Feron et al., 2024).

A fundamental challenge in the wildfire crisis affecting Bolivia and Venezuela is the complexity of managing fires in border regions. Many of the most affected areas are located along international boundaries, where coordination between neighboring countries is often inadequate or inefficient. The lack of standardized protocols, difficulties in sharing real-time information, and disparities in firefighting capacities create significant logistical and operational challenges. Fires in these areas are particularly difficult to control due to overlapping jurisdictions and administrative barriers that delay response efforts. This is also the case in other regions in south america, such as the trinational frontier with Acre, Peru and Bolívia (Pismel et al., 2023) and at the Pantanal region. Without improved cross-border collaboration, enhanced communication channels, and harmonized fire management strategies, these transboundary wildfire zones will remain highly vulnerable, exacerbating the broader crisis in South America.

Ecuador presented the peak in BA during 2024, with an anomaly of 166%, the highest on record. Official governmental reports from the National secretariat of risk management, there were almost 6,000 wildfires, 83 thousand ha of burned vegetation, 1,663 affected people, 47 people hurt, 6 deaths, 45,000 animals killed and over 5,000 animals affected, according to the National Secretariat for Risk Management (SitRep., 2024). These events were attributed to the extreme drought and land use and land cover conversion fire dependent practices.

In 2024, there were 13,400 hotspots in Peru, which was 1,000 more than in 2023 (Caceres et al. 2024). Of the total, 49% were in natural areas such as forests or other natural covers, 35% in non-forest vegetation types, and 12% in anthropogenic areas. The maximum number of hotspots occurred in September. According to Caceres et al. (2024), in August and September, the regions most impacted by fire were Ucayali, Madre de Dios, Huánuco, San Martín, and Loreto, all belonging to the Amazonia region. In November 2024, the number of wildfires totaled 1,798, with over 80,000 ha burned, 35 people and a countless number of animals died in the events (Castillo, 2024, Informe Defensorial n.° 225). The severity of the wildfires was reflected by Castillo (2024) using information from INDECI (Institute of Civil Defense).

In the extreme south of South America, fires in Patagonia started in early 2025, continuing a recent trend that aligns with an 80% increase in BA since 2002. In Argentina, the 2024-25 fire season was the most destructive in decades for northern Patagonia. By late February 2025, more than 30,000 ha had burned across Río Negro and Neuquén provinces, primarily affecting Lanín and Nahuel Huapi National Parks (Greenpeace, 2025). Extreme fire behavior was driven by prolonged drought, anomalously high temperatures, and intense westerly



winds. Nearly all ignition sources were anthropogenic, amid conditions of critical fuel dryness (Greenpeace, 2025). The Patagonia 2024-25 fire campaign represents the most extensive and intense in decades, underscoring the combined influence of climate extremes and human pressures.

In Chile, fire occurrence and BA were lower during the 2024-25 fire season than in recent years. The 2024 season reached a BA of 73,834 ha compared to the 429,103 ha burned in 2023. However, in February 2024, the Valparaíso Region experienced a record-setting catastrophic fire associated with extreme weather conditions (high temperatures and strong winds), affecting wildland-urban interface areas with significant material losses and more than 30 deaths (González et al. 2024). Central and south-central Chile have experienced an intense and uninterrupted megadrought since 2010, which has increased the size and severity of wildfires (Garreaud et al 2017; González et al. 2018; Bowman et al. 2019). Priority steps to advance solving this problem are restoring and managing forest vegetation and removing highly flammable forest plantations to move towards less fire-prone landscapes.

## 8.    Competing Interests Statement

SV is a member of the editorial board of Earth System Science Data. The authors declare no further conflict of interest.

## 9.    Acknowledgements

The authors thank all contributing panellists of the regional expert panels (Table A1): Lucy Amissah, Dolors Armenteras, Davide Ascoli, Sally Archibald, Francisco de la Barrera, Stefan Doerr, Mauro Gonzalez, Chelene Hanes, Piyush Jain, Matt Jolly, Natasha Ribeiro, Julien Ruffault, Bambang Saharjo, Sundar Sharma, Celso Silva Junior, Jacqueline Shuman, Tercia Strydom, Raman Sukumar, Veerachai Tanpipat, Gavriil Xanthopoulos, Bo Zheng. The regional panel of Oceania thank the following for their contributions to the identification and description of key events in the 2024-25 fire season: Telmo António; Chris Collins; David Field; Rui Feix; Russell Stephens Peacock; Grant Pearce; Simeon Telfer; Maggie Towers. We thank the working groups "FLARE: Fire science Learning AcRoss the Earth System" and "TerraFIRMA: Dummies Guide to using Fire Models" for contributing to defining the report scope and establishing contributor links. We thank Luca Minello for his assistance with the analyses of C project exposure to fire. We thank the CNDFB (Comité National de Défense de la Forêt et de lutte contre les Feux de Brousse) for their contribution to the characterisation of the fire season in West Africa. We acknowledge the use of GrammarlyGO (www.grammarly.com) and ChatGPT-4 (https://chatgpt.com/) to identify improvements in language use and writing style only.

## 10.    Author Contributions

**Conceptualisation and Project Administration:** DIK, CAB, FdG, MWJ.

**Data Curation, Resources, and Software:** DIK, CAB, FDG, MWJ, MLFB, EBr, JRM, ZL, ASIB, EBu, AC, EDT, JE, AJH, SL, GM, YQ, FRS, CBS, MAT-V, DvW, JBW, EW, NA, EC, LG, MP, MT, GRvdW, SV.

**Formal Analysis:** DIK, CAB, FDG, MWJ, MLFB, EBr, JRM, ZL, KB, IJMF, LF, AJH, SL, GM, YQ, FRS, CBS, MAT-V, RV, DvW, JBW.

**Visualisation:** DIK, CAB, FDG, MWJ, MLFB, JRM, ZL, KB, IJMF, LF, SL, AL, FRS,  MAT-V, RV, JBW, NA.



**Writing - Original Draft Preparation:** DIK, CAB, FDG, MWJ, MLFB, EBr, JRM, ZL, ASIB, AC, EdT, IJMF, LF, AJH, TK, TRK, SL, YQ, PSS, FRS, CBS, MAT-V, RV, JBW, EW, BB, MB, GC, CMdB, KB, VMD, SHar, EAK, BN'D, CS, GS, JS, JA, NA, DSH, SHan, SM, MP, MT, LOA, HC, PMF, CAK.

**Writing - Review & Editing:** All authors.

# 11. Data Availability

**Section 2:** BA data from NASA's MODIS BA product (MCD64A1) are extended from Giglio et al. (2018) and are available at Giglio et al. (2021, https://lpdaac.usgs.gov/products/mcd64a1v061/, last access: 6 August 2025). GFED4.1s fire C emissions data are extended from van der Werf and are available at https://globalfiredata.org/ (last access: 6 August 2025). GFAS fire C emissions data are extended from Kaiser et al. (2012) and are available at https://confluence.ecmwf.int/display/CKB/CAMS+global+biomass+burning+emissions+based+on+fire+radiative+power+%28GFAS%29%3A+data+documentation (last access: 6 August 2025). Global Fire Atlas are extended from Andela et al. (2019) and are available at Andela and Jones (2025, https://doi.org/10.5281/zenodo.11400061, last access: 6 August 2025). Regional summaries of the MODIS BA, GFED4.1s, GFAS, and the Global Fire Atlas are presented here and are available at Jones et al. (2025, https://doi.org/10.5281/zenodo.15525674, last access: 6 August 2025). Regional summaries of FWI anomalies are available from Turco et al. (2025, https://doi.org/10.5281/zenodo.15538595, last access: 6 August 2025). Studies utilising our regional summaries should cite both the current article and the primary reference for the variable(s) of interest: Giglio et al. (2018) for the MODIS MCD64A1 BA product; Andela et al. (2019) for the Global Fire Atlas; Giglio et al. (2016) for active fire observations of FRP from MOD14A1 and MYD14A1; Lizundia-Loiola et al. (2022) for the FireCCIS311 BA product; van der Werf et al. (2017) for GFED4.1s fire C emissions; Kaiser et al. (2012) for GFAS fire C emissions; Vitolo et al. (2020) for FWI from the ECMWF ERA5 reanalysis. **Section 3:** Regional summaries of population and physical asset exposure are available from Steinmann et al. (2025a, https://doi.org/10.5281/zenodo.15755007, last access: 6 August 2025). **Section 4 (and subsequent sections):** The input meteorological data used for training the PoF model, listed in Table S1, are taken from the ERA5-Land dataset, openly available through the Copernicus Climate Change Service (C3S, 2019; https://doi.org/10.24381/cds.e2161bac; accessed 6 August 2025). The fuel characteristic dataset, updated from McNorton and Di Giuseppe (2024), is available from the ECMWF (2025; https://doi.org/10.24381/378d1497, last access: 6 August 2025). Model driving data and re-gridded BA target data for ConFLAME, for section 4, 5 and 6, are available from Barbosa et al. (2025a; https://doi.org/10.5281/zenodo.15721434, last access: 6 August 2025), with ConFLAME driver assessment data for Northeastern Amazonia and Pantanal & Chiquitano available from Barbosa et al. (2025c; https://doi.org/10.5281/zenodo.16786041, last access: 6 August 2025) and Southern California and Congo Basin from Kelley et al. (2025a; https://doi.org/10.5281/zenodo.16789657, last access: 6 August 2025). Data for the FWI seasonal forecast used in Section 4 and 6 are available from the Copernicus Emergency Management Service (CEMS, 2025; https://doi.org/10.24381/cds.b9c753f1, last access: 6 August 2025). **Section 5 (and subsequent sections):** Historical (1960-2013) HadGEM3-A are available from the Met Office (2025; http://catalogue.ceda.ac.uk/uuid/99b29b4bfeae470599fb96243e90cde3, last access: 6 August 2025). ConFLAME NRT attribution outputs are available from (Kelley et al. 2025c; https://doi.org/10.5281/zenodo.15641876, last access: 6 August 2025). FireMIP / ISIMIP driving and output data is available from the Inter-Sectoral Impact Model Intercomparison Project (ISIMIP; https://data.ISIMIP.org/, last access: 6 August 2025). **Section 6 (and subsequent sections):** ConFLAME future burned area projections are available from





(Kelley et al., 2025b; https://doi.org/10.5281/zenodo.15807587, last access: 10 August 2025). Data and scripts used to produce Fire Weather Index (FWI) projections at different global warming levels are available from Liu & Eden (2025; https://doi.org/10.5281/zenodo.15790287, last access: 6 August 2025).

## 12.    Code Availability

**Section 3:** Code for regional summaries of population and physical asset exposure have been made available by Steinmann et al. (2025b; https://doi.org/10.5281/zenodo.15831766, last access: 6 August 2025). **Section 4 (and subsequent sections):** ConFLAME attribution and future projections framework (Kelley et al., 2021; Barbosa et al., 2025b) is available from Barbosa et al. (2025a; https://doi.org/10.5281/zenodo.16790787, last access: 6 August 2025). The PoF model used in Section 4 is from ECMWF implementation. A simplified version with the main scripts for data processing, model training, and analysis are archived in a publicly accessible repository https://doi.org/10.24433/CO.8570224.v1 with documentation to facilitate replication of the results. **Section 5 (and subsequent sections):** The code used to produce the FWI attribution results is available from Kelley et al., 2024 (https://doi.org/10.5281/zenodo.11460379, last access: 6 August 2025). The FWI code used to generate the figures in section 4 can be accessed via the ECMWF GitHub (https://github.com/ecmwf-projects/geff; last access: 6 August 2025). Code used for the FireMIP attribution results, along with processed ISIMIP data, can be found at https://doi.org/10.5281/zenodo.16779167 (Lampe & Burton 2025), with methods documented in Burton, Lampe et al. (2024). The current version of ibicus, used for JULES-ES bias correction, is available from PyPI (https://pypi.org/project/ibicus/, last access: 6 August 2025) and is described in detail in https://ibicus.readthedocs.io/en/latest/ (last access: 6 August 2025). Model code and evaluation for bias-correction of JULES-ES model output can be found at Spuler and Wessel (2025, https://doi.org/10.5281/zenodo.15792440, last access: 6 August 2025).

## 13.    Financial Support

DIK and MLFB were funded by the UK Natural Environment Research Council (NERC) as part of the LTSM2 TerraFIRMA project and NC-International programme (NE/X006247/1) delivering National Capability. CAB and EW were funded by the UK Department for Science (DSIT) Met Office Hadley Centre Climate Programme and the DSIT Innovation & Technology International Science Partnerships Fund (ISPF; UK Met Office Climate Science for Service Partnership [CSSP] Brazil). FDG and JRM were funded by a service contract (n 942604) issued by the Joint Research Center on behalf of the European Commission. MWJ was funded by the UK NERC (NE/V01417X/1). ASIB was funded by the DSIT Innovation & Technology International Science Partnerships Fund (ISPF; UK Met Office Climate Science for Service Partnership [CSSP] Brazil). CBS was funded by the Swiss Innovation Agency Innosuisse (53733.1 IP-SBM). EAK was funded by the State Assignment Project (FWES-2024-0040). EBr was funded by the UK NERC ARIES Doctoral Training Partnership (NE/S007334/1). GM was funded by the FAPESP (2019/25701-8 and 2023/03206-0). HC was funded by the Westpac Scholars Trust via a Westpac Research Fellowship and by the Australian Research Council via an Industry Fellowship with the Victorian Department of Energy, Environment, and Climate Action, the Victorian Country Fire Authority and Natural Hazards Research Australia (IM240100046). JBW was funded by the UK Engineering and Physical Sciences Research Council (EPSRC; 2696930). LOA was funded by the São Paulo Research Foundation (FAPESP; 2021/07660-2 and 2020/16457-3), and the Brazilian National Council for Scientific and Technological Development (CNPq; 409531/2021-9 and 314473/2020-3). MAT-V and MT were funded by the Spanish Ministry of Science and Innovation (MCIN; MCIN/AEI/10.13039/501100011033; ONFIRE PID2021-123193OB-I00) and by the European Regional Development Fund (ERDF; project "A way of making Europe"). MP and EDT are part of the Copernicus Atmosphere Monitoring Service, which is

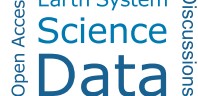

operated by the European Centre for Medium-Range Weather Forecasts (ECMWF) on behalf of the European Commission as part of the Copernicus Programme. MT was funded by the Spanish Ministry of Science, Innovation and Universities through the Ramón y Cajal (RYC2019-027115-I). PF was supported by the Portuguese Foundation for Science and Technology (FCT; UID/04033/2023 [Centre for the Research and Technology of Agro-Environmental and Biological Sciences] and LA/P/0126/2020 [https://doi.org/10.54499/LA/P/0126/2020]). RV was funded by the Brazilian National Council for Scientific and Technological Development (CNPq; #443285/2023-3) and the Brazilian Institute of Environment and Renewable Natural Resources (IBAMA) / Federal University of Rio de Janeiro (UFRJ; #968711). SM was funded by the Dragon Capital Chair on Biodiversity Economics. SV and YQ were funded by a European Research Council Consolidator Grant (101000987). TRK was funded by Schmidt Sciences, LLC.

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
