# Peer review of "State of Wildfires 2024-25"

_Earth System Science Data, 2025_

## Referee Comment (RC1)

**Key findings & recommendations**

1) This report represents a superb scientific achievement by the authors. It is an extremely impressive compendium of the rapidly changing nature and impacts of fire on Earth. Congratulations.

2) I do not think the report currently derives maximum impact from this work as a tool of communication beyond the research community. I am not sure the report fully delivers on its stated aim "to deliver actionable information to *policy and practice stakeholders and wider society*" (my emphasis). I recommend that:

   a. The authors begin the report with a 2-page (max) executive summary that communicates absolute key findings as concisely as possible, with internal hyperlinks to relevant report sections. This content could be drawn from the excellent section 7.
   b. The authors substantially revise the background section to make it clearer, cleaner and simpler to non-specialist audiences.
   c. The authors produce 1 or more high impact visual summarise of the report: e.g., an infographic executive summary or infographics summarising their 4 case studies.
   d. The authors consider reducing the complexity of figures in the main report text, with secondary findings moved to appendices. Some figures are too small to be easily read, in any case.

3) There are significant limitations in the handling of direct human impacts on fire regimes. These should be addressed before publication. I recommend that:

   a. The authors conduct a literature review of human fire use & management for each of their focal regions and use findings from this to inform discussions of how human factors may drive model-observation divergence or act as confounding factors in attribution. An existing meta-analysis such as DAFI (see end of this document) could provide a quick index that allows rapid identification of relevant literature.
   b. And/or the authors ask for input from their expert panels on this, particularly in section 4, where human factors are postulated as potential confounding issues explaining model-observation error.
   c. The authors redo model runs for section 5, ignoring population density changes [hold them constant] and instead focus on land cover change as a directly attributable global change process within the limits of current models and data.

Examples of the consequences of this in the substance of the report include:

- No mention of the civil conflict in the DRC as a potential contributor to the fire anomaly.
- No mention of the legacy of over-suppression in the California fires. Where suppression is discussed, it is discussed as having an immediate negative forcing on burned area, neglecting this longer-term fuel feedback. Lack of account for this kind of longer-term interaction is a substantial limitation to the discussion of results and attribution.
- Tendency towards weakly substantiated comment on policy interventions around direct human impacts on fire, and / or, general discussion of human factors as confounders without reference to specific knowledge and data on the focal regions [timing of different forms of agricultural fires in the Amazon basin, shifting cultivation in the Congo, e.g,].

4) There is a potential conflict of interest regarding discussion of carbon markets that should be disclosed. This is primarily a matter of due process & transparency rather than substance, with one exception that is noted below.

**Major comments**

**1) Science**

- The science of this report is generally excellent. Highlights include: assessments of which aspects of fire regimes (spread rate, numbers of fires, fire size) drove BA anomalies, the use of expert panels to identify focal events, the novel methods used to identify damages and exposure, the use of PoF to hindcast focal events. Etc.

- One important point is I think it is vital that you give due weight to regions that do not experience anomalous fire years. E.g., Eurasia, section 2.2.1.5. Just as in the sections explaining the fire regime context of anomalous burned area, can you show that there were factors constraining the impact of climate change? Were there fewer fires? Or, were there anomalous wet years? It's great that you report these "no results" for the year, but a little focus on their constraining factors would be good as well to ensure a rounded presentation of all the data.

- In general, I think it is very reasonable to focus on large and extreme fires (wildfires). However, there are times when a lack of explicit differentiation between such events and agricultural fires, e.g., is problematic. This is particularly true, in my view for your (otherwise excellent) population & asset exposure calculations. You note the region with most people exposed is Uttar Pradesh (India), however it is extremely well documented (and indeed reflected in the GLOCAB/GFED5 crop fires product) that such fires are predominantly generated through rice-stubble burning. These are not climate driven events, nor wildfires as such. This lack of distinction may partly explain the large difference between exposed and displaced persons. Similarly, in the DRC (where you note recurrent exposure), this is likely related to widespread shifting cultivation fires.

- The hindcasting section (4) is generally excellent. It is reasonable, in my view, to acknowledge that difficulties in understanding the direct human drivers of constrain the certainty of results (e.g. lines 1788-1797, 1911-1918). However, I think you need to do better than speculate on potential human causes of model-observation divergence, and rather point to explicit documented evidence about land use practices in the focal regions. Evidence around human fire use & management is increasingly systemised (see Kasoar et al., 2023 Table 1 for an overview; https://doi.org/10.1016/j.crsus.2024.100128). Such systematic data could be used to identify studies of possible direct human drivers in each focal region. At a simple level, the omission of mention of the armed conflict in the DRC as a potential driver of fire, e.g., is notable. This may explain the unattributed anomalies in northeastern DRC (line 2177). Similarly, there are data

available on the seasonality of shifting cultivation fire use in the region (see refs at end), which seems to be a partial explanation of the difficulty of forecasting there.

- The attribution section (5) is generally very good. However, human population density has an ambivalent relationship to human fire use and management (Perkins et al., 2024; doi https://doi.org/10.5194/gmd-17-3993-2024). These are much more driven by land user objectives and socioeconomic context; factors which are not accounted for in your attribution. By contrast, land cover change does have some more predictable impacts (croplands being suppressive of large fires, e.g.). Therefore, I think it would be much stronger simply to attribute burned area to land use changes, and ignore population density. Whether you include pop density or not, you cannot currently claim to be attributing changes in burned area to "socio-economic factors", because population density is a fundamentally unsuitable measure of these. But you can do so for land use change. You note this issue in lines 2619-2630, but as a structural limitation in the modelling it would be better to account for this systematically by narrowing the focus of your attribution, in my view. In the minor comments I have noted specific pieces of comment on results that I think are not sound for these reasons – in my view you should limit your comment to those aspects of the system you are robustly modelling.

**2) Communications**

- Overall, I am not sure the report rises to its stated aim "to deliver actionable information to *policy and practice stakeholders and wider society*" (my emphasis). Overall, the key messages & headline findings of the report are not clearly enough communicated for non-specialist audiences, in my view. This is not a problem unique to the SOW – IPCC reports have been critiqued by the policy community along similar lines, after all.

- The background section is too dense for non-academics, and I don't think it would provide such readers with a clear sense of how fire is changing globally and how this year fits in. Page 5 is stronger in this regard than page 4, which suggests to me the issue is in being crystal clear on the absolute headline message of the report and how you wish to communicate it.

- I wonder if you have mapped your key audiences & key messages for the report? The Abstract reads to me a bit like you are framing this in the context of the Carbon budget. But is this the most important message for your stakeholders (the feedback of fires on further climate change)? The key findings in section 3

may be more attention grabbing. I have had a go at providing an overall report message in the minor comments. Feel free to disregard.

- In any case, I would replace the abstract with a 2-page high level executive summary that tries to nail down your key findings and messages, accompanied by 1 or 2 high-impact graphics. This could be, in effect, a crunched down version of (the excellent) section 7.

- A contents page would be extremely useful as well!

- Similarly, your figures are scientifically logical and appropriate. However, some of them they could be more concise to make the information in the report more impactful. E.g., Figure 2/3 – how much information is added by the Global Administrative regions panels? The figure is too small to see clearly and without looking it up, an average reader is not going to know what these are. I might argue the same for ecoregions, which could also go in a supplementary figure. Once you get into the details of attribution I understand that more detailed figures are required.

**3) Transparency of competing interests**

- Finally, I would expect to see authors from BeZero Ltd note their role in the industry in competing interests, given explicit discussion of carbon projects (which is itself very useful). A phrase such as "High-integrity forest carbon projects can help to mitigate global climate change, and we find some evidence that these interventions are also reducing fire risk locally." (lines 1540-1541) may well be true. However, it is also in line with the commercial interests of BeZero to make such a claim -> i.e. promoting the value of *high integrity* carbon projects and hence the importance of ratings. Readers of the report should be aware of this potential conflict so they can transparently evaluate the claims made. In addition, if the specific claim regarding reduced local fire risk is to be made, it needs substantiation & should be placed in the context of documented cases of carbon projects having led to displacement of fire-dependent livelihoods in the global South (e.g. Croker et al., 2023; https://doi.org/10.1029/2023EF003552).

**Minor comments**

**Section 1**

**Abstract**

Given the likely very large underestimate of BA in the MODIS product (dealt with from line 477), I don't think it makes sense to quote the headline BA numbers in km2. A % change relative to the historical trend would make more sense to me. Alternatively, use the GFED5 number for the headline totals, and global fire atlas metrics for other aspects of the regime?

The sentence about Southern California (lines 144-146) is a bit confusing – are you describing some theoretical modelled and observed burned area here?

I don't think it's accurate to describe SSP370 as middle of the road. "Medium-high" is a better description. SSP245 is explicitly "middle of the road" in the narratives. This change should be made throughout the report.

**Introduction**

I don't think this framing (lines 174-177) is quite right. After all, global burned area is still seemingly declining, so at least on one measure, the potential for wildfires isn't growing (the phrase "potential of wildfires" is itself a little ambiguous to me). I think a more accurate/sharp key message could be: "There is a global transition from low-intensity, low-emission and low-harm grassland fires, to high-intensity, high-emissions and high-harm forest fires... This is a long-term transition, driven by a combination of climate change and land use change, which continued forcefully in 2024-25." You do make this point, but not until the first paragraph of your results!

More broadly, to me, describing lots of increases in burned area in paragraph 1, but then contrasting this with the overall declining global trend (paragraph 2) is confusing to the non-specialist reader.

The first two paragraphs are also too long for a non-specialist audience. Important messages, such as that globally aggregated BA is increasingly less important as a measure of contemporary fire regimes get hidden in the middle of paragraphs and likely skipped over.

The opening of the second paragraph mixes discussion of the trend towards destructive fires in fuel-rich environments with the overall declining trend. This all needs reframing for clarity, to my mind.

Could we have some figures in the background section to capture these key trends? Simple clear charts to communicate the key overall dynamics/trends of contemporary fires (e.g., grasslands vs forests, tropics vs extra-tropics?).

On page 5, paragraphs 2-4 (lines 228 – 267) are much stronger than the opening, to my mind. Each with a simple clear message succinctly communicated. These could all be a symbol & key message in an infographic.

Finally, I think some really key, fundamental points are missing. I know that we (researchers) all know that fire is both a natural ecosystem process & man-made, but you should make this somewhere in the material about declining burned area in grasslands.

**Objectives of this Report**

These objectives are very useful. Bear in mind that "suppression" is a jargon term to non-academics, who may know this as "fire-fighting".

The key methodologies section belongs lower down the report, in my view. I would instead replace this with a clear indication of the logic of the *structure* of the report, and how this delivers against your objectives. You could then include a table summarising the objectives, the relevant report section (s) and the methods used.

The paragraph 367-372 is a bit repetitive of para 319-322. We should know what the report is trying to achieve by line 367.

**Section 2**

**Methods**

This all seems very good and appropriate.

The use of expert panels to complement EO data is impressive.

As noted above, it of course makes sense to use global fire atlas data as the primary source in a report focused on extreme fires, which enables the excellent discussion of fire spread rates and sizes etc. However, I am not clear why you do not report total burned area from GFED5?

To my mind, given that you used summary values for FRP (if I have understood correctly) it belongs in section 2.1.2.1, not 2.1.2.2.

**Results**

*Overall*

See major comment on Eurasia & providing context for negative results.

Why are selected ecoregions chosen for Figure 1, and then long tables for administrative regions? Figure 1 is much more impactful communication.

Similarly, Figure 4 contains lots of interesting information, but is too messy for me to understand in current format. Fewer panels, bigger panels, more aggregated regions, or some combination, please.

Line 1205, I think "intriguing" is the wrong word here – "important".

*South America*

The point about fast fire spread driving increased BA across Brazil vs increased fires in the Amazon is fascinating. To my mind that speaks to fire for explicit deforestation vs uncontrolled wildfires spreading purely on climatic factors.

*America*

The section on drivers of extreme fire anomalies is extremely interesting, as per South America.

*Africa*

The increase in numbers of fires also seems likely to me to reflect shifting cultivation fires (the dominant mode of human fire use in the region; see end refs) growing out of control due to anomalous climate conditions. They were likely undetectable by MODIS before, hence the apparent increase in fire number – which may actually be due to increased fire size & hence increased detection.

**Section 3**

This section is generally excellent. Figure 6 is very good as a high-level summary.

See major comments on wildfires vs agricultural fires distinction & carbon markets section.

**Section 4**

See major comment on human dimensions of fire in your hindcasts.

It is fine not to have detailed human factors in your models (because nobody has yet built this). However, given it seems to play an important role in PoF & ConFLAME model error [in part because the models do an excellent job with biophysical factors] I think you should consider being more explicit about this limitation in the methods section, similar to the excellent section on EO product uncertainty. At present there are several scattered references to missing human processes in the results (e.g., suppression in California [but only as an immediate not long-term forcing], lines 2157-2159, Panantal lines 2128-2131). It would be cleaner and clearer to address this upfront in the methods.

Furthermore, I think where you note possible human factors as explanations of model-observation divergence, you should do so in specific reference to published literature. E.g., in the Amazon (lines 1788-1797), there is well-documented evidence of escaped pasture fires as drivers of forest-edge degradation (see refs list). This matches your pattern of ignitions driving fires at the eastern edge of the Amazon (Figure 10). To dig deeper, the seasonality of pasture vs agricultural residue fires in the region could be identified from field data to see whether these match the error in model seasonality predictions (see refs).

Incidentally, there is an interesting & important discussion to be had about whether current fire activity in the Amazon case study is indicative of deliberate deforestation, or perhaps more worryingly, of the transition towards forest degradation and inherent instability under a changing climate (see Lapola et al., 2023; https://www.science.org/doi/10.1126/science.abp8622)

**Section 5**

See major comment on how what exactly you are able to attribute on the human dimensions. I have noted sections where this limitation leads to, in my view, unjustified comment in your results section, below.

As a general comment on the Pantanal case study, in meta-analyses of human fire use, there tends to have been very little in this region historically (too wet). As such, I am not at all surprised to see that it is the event most clearly attributable to climate change (lines 2649-2651).

However, because of the limitations of your socio-economic analysis, I think the comment on possible policy intervention focused on constraining direct human

influence may be true (lines 2678-2687), but cannot be justified on the basis of your analysis, which is limited to coarse patterns of land cover change.

Similarly, in the Congo, there is not much relationship between increasing population density and changes in fire use in shifting cultivation. Increased population density can shorten fallow periods (leading to increased fire use), but can also lead to sedenterisation, and hence less use of fire for field preparation. As such, your comment on socioeconomic factors broadly is not warranted (lines 2755-2760).

Again, I'm afraid I don't think the analysis in lines 2828-2840 is robust. You aren't representing socioeconomic drivers beyond land cover change. (Sorry to repeat myself).

**Section 6**

This section is excellent. Because you held land cover and population density constant – you are spared my comment on the limitations of this! I think this is actually better in any case as it allows focus on those dimensions of fire JULES currently captures in a robust way.

**Section 7**

This section is excellent & could form the basis of a revised executive summary.

However, after revising section 5, I think the sections here on socio-economic attribution need to be cut or substantially revised.

**Key relevant papers for each study region from DAFI/recent lit:**

**Northeast Amazonia**

Cano-Crespo et al., suggests escaped pasture fires as the largest source of wildfires in the Amazon - https://doi.org/10.1002/2015JG002914

Jakimow et al., suggests that pasture burning in the Southern Amazon is focused around the main dry season (Aug-Oct) https://doi.org/10.1016/j.rse.2017.10.009

Brunel et al., suggests that pasture burning in the coastal savanna regions is clustered around December-Jan. https://doi.org/10.1016/j.rama.2021.08.003

Cammelli et al., important study to understand behavioural and economic drivers of smallholder fire use in the region - https://doi.org/10.1016/j.gloenvcha.2020.102096

Bilbao et al., 2019, details current management and policy challenges around fire, and makes strong cross-community suggestions for ways forward - https://www.mdpi.com/2571-6255/2/3/39

**Congo Basin**

Molinario et al., shifting cultivation and forest loss in the Congo – importantly this study highlights how subsistence farming is often synergistic with increased logging & mining (i.e. you have to feed the workers). https://www.mdpi.com/2073-445X/9/1/23; see also https://iopscience.iop.org/article/10.1088/1748-9326/10/9/094009/meta

Tyukavina et al., on shifting cultivation in the Congo basin - https://doi.org/10.1126/sciadv.aat2993

Examples of fire increasing under conflict: https://doi.org/10.1080/1747423X.2011.565372, https://www.scirp.org/journal/paperinformation?paperid=27362,

**California**

I recommend including citation of this modelling study on the long-term impact of over suppression on fuels. https://www.nature.com/articles/s41467-024-46702-0

**Panantal**

There is generally less literature in this region, owing to the wetness & lack of use of susceptibility to fire. Here are the two papers from DAFI, but perhaps indirect impacts including drainage are those to focus on (as already mentioned in the text).

Devisscher et al, - https://doi.org/10.1111%2Fgeoj.12261

McDaneil et al., https://doi.org/10.1080/08941920500248921

---

## Referee Comment (RC2)

**Review of essd-2025-483**

**State of Wildfires 2024-25**

*General comments*

The authors are to be commended for their efforts to so comprehensively summarise the complex and complicated state of wildfires on our planet, to wrangle the datasets and models needed to make sense of a phenomenon that is at once a natural part of our world and a terrifying threat to our existence. However, this document is a beast to read and having read through pretty much all of the 100 pages of the main part of the paper (I admit to skipping some parts just to maintain my sanity), I wonder if any of the authors have in fact read the thing in its entirety. There is a very large amount of repetition in the document. Much of this comes from the nature of the methods employed, being similar in many of the different components being presented, but these could be streamlined to a great extent—this might not reduce the overall length by much but would improve the readability considerably; we don't need to be told more than once that FWI is the Canadian Fire Weather Index and we don't need yet another explanation of its origin.

In the same vein to improve the reading experience, I would strongly suggest the authors consider moving their Conclusions section to the very front of the paper. I found this section to be the most readable and to convey all the important information any reader of the article would want without the need to wade through the pages and pages of detail on method before getting to the juicy stuff—I think it works better than the Abstract in this because it hasn't been so condensed; maybe call it an extended abstract or similar. This may break with the standard article structure of this journal and the previous State of Wildfires (I've not read it), but would make future editions of this report so much easier to ingest. The detail of the methods could then follow.

Some other high-level comments:

1. I don't quite see the point of the climate projections for the four selected extreme event localities. Unpacking what happened to them this year makes perfect sense but why would one care more about the seasonal and multi-decadal outlooks for these areas than anywhere else?
2. In section 5 on Attribution to Global Change, be consistent in the language used to present level and extent of attribution. Section 5.2.1 deals largely with 'probability of occurrence being x times higher' or 'times more likely' than the counterfactual, which may be the same thing but is confusing. 5.2.2. suddenly presents attribution as 'x% likelihood of whereas The introduction of Amplification Factor without an explanation of what it was threw me while I was trying to unpack the probabilities that were presented.
3. In many of the analyses, there is much discussion of the comparison of predicted 'hotspots' versus actual 'hotspots' (being satellite derived). But's not clear how these 'hotspots' compare with actual fire events. Most of the discussion assumes that hotspots and fire events are synonymous, but they are not. Can the discussion be made more specific to fire events, since that is what the metric being assessed is?
4. Similarly, there is much discussion throughout in which analysis of fire weather is conflated with ignition potential, particularly in regard to the use of the Canadian Fire Weather Index. The FWI is not designed to provide any insight into ignition potential or

fire occurrence and so it is no surprise it does not perform well in this role. Since it is recognised as such (e.g. L1634), why do so many of the analyses persist with it? Has there been any published validation of the FWI for such uses? This is particularly important since it is stated on L1633 that FWI does not consider vegetation state in biomes other than boreal forest. And yet not 40 lines later ( L1674) it states the FWI successfully identifies regions with elevated fire danger aligning with BA anomalies outside the boreal forests. There needs to be some consistent application of tools such as FWI and their treatment in the ms.

5. Reliance upon FWI and PoF is largely going to highlight fire weather as the cause of fire events because they are largely (and specifically in the case of the FWI) fire weather products. Extrapolating these (particularly FWI) to fire occurrence a long bow to draw.

6. Despite the effort to explain what is meant by 'extreme' fire there is still no robust definition that I could find in the ms. A full and complete definition of the following words is required at the beginning of the ms:
   a. Extreme fire event
   b. Extreme fire area/extent
   c. Extreme fire emission
   d. Extreme fire activity
   e. Extreme wildfire
   f. Extreme wildfire season

7. Similarly, fire intensity is not defined (e.g. L383). Fire intensity can be interpreted any number of ways which are not compatible (e.g. fireline intensity (kW/m) and fire radiative intensity (kW/m^2). Please define and use consistently words like 'fire intensity' and similar (e.g. intensity of fire).

8. It is understood that MODIS is being phased out, to be replaced by VIIRS and Sentinel 3, etc. Since it is intended for this series of articles to continue can the authors provide a plan for enabling the future results to be comparable with current and past results using MODIS, even if it is a period of parallel presentation if necessary.

9. In the results section, be consistent in your comparative language. In some places, quantitative comparisons are number of times greater while in others it is given as a percentage, sometimes in the same sentence (see L912). This is related to Point 2 above in the attribution section.

10. Be consistent in the presentation of large numbers. In some cases these are given as numerals, in others they are spelled out in full. e.g. 290 thousand ha versus 290000 ha.

11. Be consistent in presentation of units. In some cases there is a space between the value and the unit, and in others there is not.

12. There is also a mix of tense throughout the ms, shifting sometimes in the same sentence between present and past tense.

13. FWI seems to be defined every time it is introduced (e.g. L356 & L 719 & L1602 to name a few). It only needs to be defined once.

*Specific comments:*

L124: Superscript '2'. Define what you mean by 'extreme fire seasons'.

L160: Delete apostrophe.

L167: Should this not be 'climate change and land use attribution'? Currently it reads like the attribution has already been made.

L179: 'extremely large, fast-moving or intense fires'. Is this your definition of 'extreme fire'? Seems circular.

L190: Define 'extreme fire extent'. I don't think 'extent' in this context is correct as it refers to the outer bounds of something, not the total area, which is what I think is intended.

L243: What about marine sinks? How do they compare?

L249: 'India, the EU27 or the USA'. Do you mean together or separately? Not clear.

L251: 'greater vegetation mortality'. How true is this? How was it determined? In many fires across the globe, vegetation is not necessarily killed by fire—often it is the dead vegetation that is burning.

L264-267: Not clear. Are you referring to use of fire as a land management tool (be it traditional or otherwise)? Be explicit here.

L277-281: Other mechanisms such as ignition risk reduction, pre-suppression, rapid early/initial attack are also employed.

L325: Insert 'the' before 'context'

L329: Change 'four extremes' to 'four extreme events'

L334: Predict what for each focal event? By the looks of what is presented, you are attempting to predict fire occurrence. Please add that here.

L345: When you say 'size and rate of growth' are you talking average, median or peak values?

L356: How suited is FWI to this purpose? Has it been validated for this use? Provide references.

L383: 'fire intensity' needs to be defined as there are many different interpretations of it.

L384: What is the functional difference between 'BA' and 'fire size'?

L395-409: This needs to come earlier, with a concise definition in the abstract.

L397: 'comprehensive'. Do you mean 'rigorous'? In regard to impacts are you referring to direct impacts or long-term impacts?

L413: Define what you mean by 'extreme fire weather days'.

L442: For those of us who are not EO experts, what does 0.1 degree translate to metres at, say, the equator?

L451: What does 'scaled to emissions' mean?

L455: As with L383, 'intensity of the fires' needs to defined. How is it determined from BA and daily growth rate? It certainly would not be the 95th percentile of actual fire intensity.

L458: Can you explain exactly how FRP is directly related to 'the fire's intensity and fuel consumption' and how these are measured remotely?

L464: How can four daily measures of FRP provide information on fire intensity enough for a meaningful estimate of 95th percentile?

L469-472: As mentioned in the general comments, what is the plan to eventually transition over to VIIRS or Sentinel and what will it mean for consistency of reporting in this series? Point to section 2.1.1.4 for comparison of the various products.

L479: Not just small fires, but short-lived fires.

L481: An increase of global BA by 93% is huge.

L482: Given such a large increase in BA by including small fires, it is difficult to believe variability and trends in regional BA do not significantly differ when small fires are included. How can this be?

L483-485: This is not clear. What is the generation of biases got to do with anything and why is it in line with the sensitivity of different sensors? Clarify.

L488: 'deemed highly suitable'. But is it the preferred option given the access to other sources of data? If you could use other products, would they produce better results? Seems like an attempt to sweep the problem under the carpet.

L490-497: How can uncertainties in a product be determined by comparing the uncertainties in other similar products? Uncertainty at some point needs to be related to actual burnt area that may or may not be observed remotely. What are the implications for the continued use of MODIS?

L502-505: It is not clear what you mean here. Clarify.

L505-50513: This essentially repeats the first half of this paragraph.

L515-534: So what is the uncertainty in FRP data?

L528: Define 'MIR'

L547: Repeats L380.

L551: Do you mean boreal spring?

L558: How are these determined and validated?

L590-594: This is scary, if the new products are largely the same as the old products with similar uncertainties and no improvements. Is this correct?

L595: What does 'generally replicated' mean?

L597-599: Repeats previous statement. Where are improvements to come from?

L620-626: This is a reasonable approach, but for outyears beyond the current ms, will you maintain the top 5 or the upper quartile? They are close enough now, but in a few years where you need 6 fires to be a quartile, what will you do? Best to decide now if you will now and forever take the top 5 or the upper quartile.

L629: 'event's peak'. To be clear, to which event are you referring?

L644-648: Unnecessary detail. At the scale we are talking, most cells will contain components of each combustion mode.

L675-680: Repeats L400.

L726: Explain why you chose 95%. Previously it was the upper quartile. Be consistent in what you consider 'extreme'.

L760: What is the basis for this conclusion?

L763: You start off by saying '3.7 million km^2 burned globally, but now at the end of the paragraph you say '*at least* 3.7 million km^2). Say this first.

L781: here and throughout: What do you mean by '(sub)tropical'? Is it tropical or subtropical or both? Be clear.

L781: The use of the same numerical value (290,000 km^2) for two things in this paragraph is a tad confusing. Is this correct?

L788/Figure 1: Is there any reason why the secondary y-axis cannot be scaled such that the heights of the boxes cannot also be read as relative anomaly (%) instead of using an additional symbol that is almost impossible to see against the blue?

L882-892: How do you know that these BA weren't burned by multiple individual fires that had coalesced?

L892: 'intensity value'. What is this? Explain.  See L458.

L1044/Figure 5: Put the panels of this figure in the same order in which they are presented in the text (i.e. Amazonia, Pantanal, SoCal, Congo). (Or reorder the text to match the figure).

L1061-1063: Do we need this sentence, since it's already been explained?

L1068: What is 'climatological mean'?

L1079-1080: How was this determined?

L1092: January- March 2024 is largely outside your year of study.

L1128-1132: What is the difference here between 'homes' and 'structures'? Can you present consistently as one or the other, rather than both?

L1143-1147: Not a sentence. What about the water from the federal reservoirs?

L1176: 'prior year to two'. What does this mean?

L1266: Should this sub-head be 'Carbon Offset Projects Exposure' to be fully correct? See L1268.

L1290: 'a toxic mix of gases'. The mix is not toxic but some of the gases in it are toxic. Include volatile organic compounds (VOCs) in the list.

L1294: Are cancers in this list?

L1307: What do you mean by 'prescribed'? I would avoid this word in connection with fires as many planned burns are called 'prescribed' fires.

L1462:  Something missing from 'the northern tropical of South America'.

L1487: Delete comma after 'Note'.

L1497: As per L1266.

L1499: Change to '…cost-effective climate change mitigation…'

L1525: Everywhere else you refer only to BA, not burned area. Consistency.

L1601: FWI is not designed for predicting fire occurrence, for many of the reason stated in the analysis L1631-1641. Why was it used in the first place? Seems criminal to abuse such a venerable system in this way and then complain about it.

L1619: Do you mean predictions of wildfire occurrence?

L1628-1630: Performance of PoF should be compared against actual, not FWI.

L1634-1635: What do you mean 'too far in advance of actual fire emergence'?

L1674: This sentence conflicts with what is presented in L1631-1641.

L1681-1694: Is this detail necessary? Suggest cutting.

L1713-1724: Repeats L343-L365.

L1744: How is this error determined?

L1751-1753: Fuel moisture does not lead to increased vegetation growth.

L1755: '…lots of trees…' Huh?

L1761: In many of the subsection in these results, way too much background is provided. This could be pruned considerably.

L1768: How is number of 'hotspots' converted to number of fires? These are not necessarily the same. What are you suddenly referring to hotspots instead of fires?

L1776-1777: Are these not parts of two separate fire seasons? 2023/4 and 2024/5?

L1781: 'fire hotspots' Are these separate fires?

L1788-1797: This paragraph is essentially discussing the difference between fire weather and ignition potential. Fires do not occur without both but each does not influence the other.

L1803: How useful is this? Are these separate fire ignitions?

L1814: 'worst water crisis'. What is this? Too little or too much water? Poor quality? Be specific.

L1828-1829: How does this translation work? How well do they compare?

L1877: This is true only if the total number of fires remains constant.

L1938: Change 'controls' to 'factors'. Topography is critical for influencing fire intensity.

L2345: Insert 'Initiative' before '(Barnes'.

L2354: Why mean daily relative humidity? Why not minimum to match the maximum air temperature?

L2537: Is there a reference for DSR? Van Wagner 1987 is a reference for FWI, not DSR.

L2553: Change 'lead' to 'led'.

L2583: What is 'Amplification Factor' and why is it suddenly being introduced here? It should at least be introduced in the methods section of this chapter. It needs a full explanation since it becomes an important metric.

L2610-2617: Why is this paragraph here? It is superfluous background.

L2888: It's not clear why this section is here. Did you do it for the previous State of Wildfires? If so, can you analyze the quality of the predictions you did for this fire season? I'm not sure how detailed outlooks for regions that burned this season will be of much relevance next year.

L3326: This chapter is an excellent summary of the breadth and depth of the document and deserves to be placed at the very beginning of the document, perhaps with some streamlining.

L3670: Add 's' to 'deviation.

L3671: Ditto.

L3692: 'vegetation growth'. There is *always* vegetation growth when it rains. What about it? Is there a missing adjective?

L3710: Change 'find' to 'found'.

L3737: Delete 'made'.

L4024: Insert comma after 'data'.

L4029: Insert 'research' before 'projects'.

L4040: Should this read '...international collaborators routinely...'?

---

## Referee Comment (RC3)

**General comments**

*The State of Wildfires* represents an important community effort to provide a scientifically rigorous account of key wildfire events during the 2024–25 fire season. The ambition and scope of the report are commendable, and it is encouraging to see such a broad range of expertise contributing to this work. I agree with the other reviewers that the report is both timely and significant. However, before recommending publication, I believe several issues should be addressed to ensure the report fully achieves its goal of delivering actionable insights for stakeholders and society.

**Non-scientific comments:**

1) **Readability and clarity**

I share Reviewer 2's view that the report is dense and difficult to follow. Beyond being heavy, the presentation sometimes obscures both intention and interpretation, which is very important given the target audience of this work. This does not stem from excessive technicality, but rather from how the material is structured and presented. I recommend targeted editing:

   a. **Abstract/Executive Summary:** The abstract is very dense does not facilitate clear and quick understanding. I think the suggestion of using the current conclusion for this as suggested by Reviewer 2 is a very good one, as I agree it very well written. I would also suggest presenting results in order of impact: (i) human fatalities and losses, (ii) emissions, and (iii) attribution results. Additionally, I recommend reconsidering the phrase "global fire activity" in the opening paragraph:

*L121: "The State of Wildfire Project systematically tracks and analyses global fire activity and this, its second annual report, covers the March 2024 to February 2025 fire season."*

   Whilst the report does acknowledge regions with lower-than-average fire activity, it does not focus on average global fire activity, and negative extremes are also not the focus of the report. Unless these themes are developed further, for example by highlighting cases where high fire weather coincided with unusually low burnt area, or highlighting regions of constant trends, it may be better to only mention the focus on (positive) extreme events here. Since most of the report emphasizes positive extreme fires, without transparently acknowledging this here, the framing could unintentionally bias readers toward viewing fire occurrence only in negative (and extreme) terms, which could have major policy implications.

   b. **The role of fire and defining "extremes":** This relates to the previous point about unintentional bias. From my reading, the report fails to explicitly state that globally fire is *(i)* unavoidable and *(ii)* a necessary part of Earth systems and biodiversity. This absence risks reinforcing a simplistic view that "more fire is bad" and "less fire is good," which may mislead stakeholders. I think some discussion of this context is needed. Related to this, the introduction directly

begins discussing "extreme fire events" without first defining "extreme." These terms should be defined clearly at first mention.

*Line 609*: *"(i) as relative anomalies (expressed in %) from the annual mean during all previous March-February periods since 2002 (2003 for fire C emissions); (ii) as standardised anomalies (standard deviations) from the annual mean during all previous March-February periods since 2002 (2003 for C emissions); (iii) as a rank amongst all March-February periods since 2002 (2003 for fire C emissions), March 2024-February 2025 inclusive."*

Given that many regions have very long fire return intervals, the short record used here should be acknowledged explicitly and transparently. I would like to see discussions of the implications and limitations of this. I suggest framing these as extremes "within the lifespan of an individual today" which both clarifies the scope and makes the results more relatable while transparently acknowledging the limitation. I think this type of framing would also make the future attribution figure where the likelihood of experiencing an event if some was born *today* vs born at other dates link quite nicely. However this is just a suggestion, I am not sure exactly what the best way to get around this is!

I also appreciate the use of regional expert panels. This approach mitigates the limitations of the satellite products and incorporates diverse forms of knowledge and expertise, which is desperately needed. However, it should be noted that "extremes" identified in this way often reflect vulnerability and resilience shaped by factors beyond climate such as governance, funding, and policy decisions, which are not strictly biophysical conditions.

*Line 102*: *"Examples of extremes that can be captured by expert assessment (but not by Earth observations) include: suppression difficulty; fatalities and structure loss; impacts on human health and wellbeing; impacts on agricultural and other economic sectors; impacts on biodiversity, and; impacts on diverse ecosystem services such as recreation, tourism, or other cultural values."*

*Line 680*: *"This includes (but is not limited to) wildfires that impacted society by causing fatalities, evacuations, displacement (e.g. homelessness), direct structure or infrastructure loss or damage, degradation of air or water quality, loss of livelihood, cultural practice or other ways of life, and loss of economic productivity. This definition also includes (but is not limited to) wildfires that impact the environment via disturbance to vulnerable ecosystems, biodiverse areas, or ecosystem services such as C storage."*

For example, many of the things mentioned here to not relate directly to the climate drivers. Some discussion of this, and how it takes more of a risks definition of extreme rather than a biophysical one would be nice.

c. **Focal point events:** The analyses from the focal events were difficult to follow because of the fragmented presentation of the results, which felt at time quite repetitive. I recommend restructuring the Results section into 1) Global results and 2) Focal event case studies. Each case study could then follow a consistent template with:
   - The regional context
   - The description of fire event

- Causes (from previous literature)
- Prediction analyses and causal inferences (PoF and ConFLAME)
- Attribution results

*Line 1961: "Northeast Amazonia experienced an exceptionally severe fire season between January and April ( Figure 10 ), driven by extreme drought which started in 2023, intensified by the combined effects of El Niño and the Atlantic Meridional Mode, which brought unusually high temperatures and suppressed rainfall."*

*Line 1996: "According to our Sparky-PoF analysis, the extreme fire activity during the 2024-25 fire season in the Pantanal-Chiquitano (described in Section 2.2.2.2 ), was mainly the result of extremely dry weather which had started since 2023…"*

Grouping all the information for each focal event would clarify what comes from background literature versus what is novel (PoF, ConFLAME), which is sometimes difficult to decipher. This would also reduce repetition and open space for deeper analysis, including integration of more of the excellent figures in S4.2. The recent California wildfire studies (McNorton et al., 2025) demonstrate how impactful such in-depth case analyses can be, and adopting a similar approach here would greatly strengthen the report.

**Scientific comments (major):**

**1) Attribution of human influence**

I agree with Reviewer 1 that the current approach to human attribution is problematic. Their suggestions for improvement are excellent, and I simply want to reiterate their concerns.

**2) Implication of single year analysis**

By its very design, the report focuses on wildfire events in an individual year. From my understanding, the weather/fuel/land use conditions relating to *that* specific year are then used to make causal inferences. This is fine; however, we know that many of the drivers of individual wildfire events can result from multi-year to decadal process such as fuel accumulation, shifting vegetation composition, and land use change as well as policy decisions such as suppression, and these will not occur on yearly timesteps. This especially relates to fuel conditions, such as fuel load but also fuel continuity. This point is especially important attribution work, both for the causal attribution of the focal events and the future the attribution results. I believe in nearly all the focal events, a multi-year process is described in the introduction to that event, and climate modes such as El Nino are also mentioned. It would be nice to clarify if PoF or ConFLAME take these multi-year processes into account? If they do not, it seems to me that the results are going to be bias towards attributing weather as the driving cause of fire events, regardless of how much longer-term processes may play a role, given that this is the only factor which operates on a yearly timestep. More discussion is needed to clarify if the current methods capture these long-term drivers, and if they do not, this limitation needs to be addressed. In the future, it would be nice to see future reports can account for the multi-year process (i.e. in both the ranking, forecasting and attribution).

**3) Discussion of the model limitations:**

It appears that both PoF and ConFLAME cannot reproduce the southern Californian focal event or the Northern Amazonia one (Figure S17). More evaluation and transparent discussion of this (and what it means) is needed.

*Line 1788: "Interestingly, both the PoF and FWI systems failed to capture a lull in fire activity during the second emergence in August-November of fire-conducive conditions showing the limitations of forecasting fire activity rather than fire danger."*

**4) Attribution analysis:**

Given that ConFLAME does not capture some of the focal events (above point), some more evaluation is needed on the model's ability to reproduce the contemporary fire record in specific case study locations. Are interannual variability, as well as previous extremes in the observed record captured, and are there benchmarking metrics available for this? I may have missed this, if so, ignore this comment.

**Scientific comments (minor but would like to see addressed in future reports):**

**1) Buffering in ranking and additional metrics**
**a. Fire regimes and variability**

Fire regimes diverge dramatically, and this includes their year-to-year variability in fire activity. It would be nice to see an analysis of the relative spread in the observational record for the different fire metrics, as I think this would contextualize some of the results. For example, in regions with long fire return intervals and very low burnt area averages, a single year (early in the record) can disproportionately alter the mean, adding uncertainty to the signal and potentially obscuring more recent extremes. In contrast, regions with consistent fire properties make extreme years more robustly interpretable.

**b. Trended data**

The authors discuss the overall trends in fire both inside and outside tropical regions, which are a result of climate, vegetation and land use change. Given that we know there is a trend in recent fire data, is the baseline mean approach appropriate? Complementing this with rolling windows (e.g., 5-year averages) or highlighting long-term trends in focal regions could add context. This would strengthen the policy relevance by distinguishing between consistent year-to-year patterns (with trends) and one-off anomalies that a mean value alone cannot capture.

**2) Discussion of climate modes**

Given that the satellite record is so short, it would be useful to also contextualize the rankings in the context of climate modes. Do certain areas consistently show higher rankings in El Nino years for example? If so, what are the implications of this? I don't think this is something that necessarily needs to be addressed in this report, however, it should be considered for future reports, as may explain/influence the ranking results.

**3) Fire weather vs actualized fire**

I would be very interested to see a discussion about the differences in the FWI rankings and the BA rankings. Comparing Figure 4 and Figure S2, it becomes very clear that the high ranking FWI regions do not necessarily correlate to the high ranking FWI (which is to be expected). Whilst South America does emerge very clearly in high rankings for both, Southern Africa, the coast regions of Western and Northern Africa, Central Europe, Scandanavia, and Russia all show very high rankings in FWI but low rankings in BA. This seems to be anomalous in some of these regions, where the reported percentage reductions are huge:

*Line 993: "BA was around 50 thousand km 2 (71%) below average in the Asian temperate grassland, savannah and shrubland biome, 42 thousands km 2 (62%) below average in the Asian xeric shrublands"*

This appears to me like another point where a focus on longer-term processes, as well as a discussion of fire regimes and fire ecology more generally would help explain these mismatches. Relating to the previous points, it would be interesting to relate these regions of mismatch to previous fire years, or, for example, highlight them as regions in which, given that *this* year we saw unusually low fire activity, excess fuel may have built-up, which may, in combination with the right weather conditions, increase the likelihood of extremes in the next few years. Such an approach would also yield policy-relevant insights, identifying areas where fire management attention may be needed not just because of recent fires, but also because of reduced fire activity in recent years that could elevate future risk.

---

## Author Comment (AC1)

**Reviewer 1**

**Key findings & recommendations**

 This report represents a superb scientific achievement by the authors. It is an extremely impressive compendium of the rapidly changing nature and impacts of fire on Earth. Congratulations.

Thanks! It takes huge effort and commitment from an international team, and it is wonderful to see this recognised.

We also appreciate the time and effort involved in reviewing such a large document, and we thank the reviewer very much for their service to the peer review system.

- 2) I do not think the report currently derives maximum impact from this work as a tool of communication beyond the research community. I am not sure the report fully delivers on its stated aim "to deliver actionable information to *policy and practice stakeholders and wider society*" (my emphasis). I recommend that:
  - a. The authors begin the report with a 2-page (max) executive summary that communicates absolute key findings as concisely as possible, with internal hyperlinks to relevant report sections. This content could be drawn from the excellent section 7.
  - b. The authors substantially revise the background section to make it clearer, cleaner and simpler to non-specialist audiences.
  - c. The authors produce 1 or more high impact visual summarise of the report: e.g., an infographic executive summary or infographics summarising their 4 case studies.
  - d. The authors consider reducing the complexity of figures in the main report text, with secondary findings moved to appendices. Some figures are too small to be easily read, in any case.

We are grateful for this helpful set of connected comments, which indeed raise important points about how we communicate our findings to broader audiences.

We agree that the original stated aim, "to deliver actionable information to policy and practice stakeholders and wider society", was not phrased appropriately. This highlight a broader need to revise the first paragraph of the "Objectives of this Report" section, which now reads: "The State of Wildfires report brings together the latest science on extreme fire monitoring, prediction, and modelling to track how wildfire impacts on society and the environment are changing, and to explain the drivers behind these shifts. It forms the foundation of the wider State of Wildfires Project, which aims to deliver insights on climate, land use, and fire management policy to decision-makers and practitioners, ultimately supporting stronger societal and environmental resilience to wildfire. In this edition of the State of Wildfires Report we:" This revised phrasing more appropriately flags that this report presents the scientific evidence that can/should underpin the transfer of information from science to policy and practice, while also clarifying that we are not directly instructing policymakers here.

We also appreciate the reviewer's suggestions regarding the dissemination of our findings to broader (non-scientific) audiences, such as an executive summary and visual communication tools like interactive apps and infographics. In fact, we are very much encouraged by these comments because the planning for these activities is already underway as we prepare for the public launch of this report. This year, we have broadened and deepened the range of materials that we will be providing to non-scientific audiences, including and extending from the list of suggestions given by the reviewer. Our website, due for launch imminently, will serve as an access hub for the full suite of materials serving a range of audiences and allowing them to access the findings of our work in an appropriate form which aligns with their requirements. As part of this effort, we have consulted with the science communications teams at our institutions and also commissioned materials from experts in scientific outreach and knowledge exchange.

Nonetheless, we must emphasise that those activities are treated in separation from the scientific reporting activities of our State of Wildfires Project. The report presented here is intended for a scientific audience predominantly and hence the use of specialised scientific terminology and more complex plots is, in our view, entirely appropriate in this forum. While we do hope to draw interest in the scientific report from particularly dedicated and well-informed audiences beyond academia, such as the experts and officials operating within governmental departments or fire management agencies/bodies, we expect their initial contact with the report's findings to be in the form of our Summary for Policymakers, our website, or our other outreach and knowledge exchange materials.

In short; please be assured that we are making significant efforts to ensure that the findings from this report are accessible to non-scientific audiences. Given this description of our broader strategy, we hope the reviewer will not be too disappointed that we did not fully implement their suggestions a-d. Nonetheless, please note that we did make figures 2-4 clearer in light of the reviewer's concerns about readability (e.g. through font size adjustments). We also paid close attention to the reviewer's specific comments below and addressed many of them as advised.

- 3) There are significant limitations in the handling of direct human impacts on fire regimes. These should be addressed before publication. I recommend that:
  - a. The authors conduct a literature review of human fire use & management for each of their focal regions and use findings from this to inform discussions of how human factors may drive model-observation divergence or act as confounding factors in attribution. An existing metaanalysis such as DAFI (see end of this document) could provide a quick index that allows rapid identification of relevant literature.

Indeed it is important to describe both the bioclimatic and human contexts that shape fire regimes in each of our focal regions. As such, we added detail on bioclimatic and human factors of each region's fire regime, including human fire use and management practices, to section 2.2.2, drawing on literature review including from DAFI:

**Northeast Amazonia:**

Fire regimes in northeastern Amazonia reflect interactions among ecosystems, human activity, and climate. Ecological heterogeneity, spanning humid and floodplain forests, natural grasslands, and savanna formations, produces marked variation in fire impacts. Savannas and forest-savanna transition zones in Roraima, Venezuela, and the Guianas are relatively fire-adapted, historically experiencing low-intensity surface burns at multi-year intervals (Alvarado et al., 2020; Pivello et al., 2021). Yet their resilience declines under more frequent or intense burning. Fire-sensitive ecosystems such as humid forests and wetlands are even more vulnerable, with elevated fire pressure threatening long-term stability and biodiversity (Alvarado et al., 2020). In these ecosystems, anthropogenic climate forcing is most likely driving increases in fire activity. Most ignitions now arise from human use, particularly pasture management, with escaped fires the dominant source of wildfires (Cano-Crespo et al., 2015). Burning follows distinct seasonal patterns: August-October in the southern Amazon (Jakimow et al., 2018), December-January in coastal savannas (Brunel et al., 2021), and January-March in the northeast (Carvalho et al., 2021). These dynamics help explain the late peak of the 2024 event. Climatic variability remains the principal temporal control (Brando et al., 2020; Berenguer et al., 2021), while land-use practices and socio-economic drivers shape fire use (Cammelli et al., 2020). Addressing these coupled drivers requires management that integrates ecological context, community practices, and regional policy frameworks (Bilbao et al., 2019).

**Pantanal and Chiquitano:**

In the Pantanal, the annual cycle of flooding (October–March) and drying (April-September) plays a central role in shaping fire regimes. During the wet season, extensive inundation keeps most of the landscape too moist to burn. As waters recede, grasses and savanna vegetation dry out, creating windows of flammability, but under normal conditions fires remain patchy and largely restricted to grasslands, savannas and wetland margins (Damasceno-Junior et al., 2022). When this cycle is disrupted, often by multi-year droughts, large areas stay dry for longer, exposing grasslands, forests, and even peat-rich soils to extensive burning. In the Chiquitano, surface fires are the dominant type, frequently originating in deforested or agricultural areas before spreading into forest edges. Human activity is the primary ignition source in both regions (Romero-Muñoz et al., 2019; Menezes et al., 2022), with escaped agricultural and pasture-renewal fires driving many of the catastrophic events of recent years. These coupled processes illustrate that fire activity in the Pantanal and Chiquitano is no longer governed by climate alone but increasingly by altered hydrology, land-use frontiers, and the intensity of human fire use (Barbosa et al. 2022).

**Southern California:**

Southern California's fire regime reflects the interaction of Mediterranean-climate vegetation, frequent extreme weather, and dense human presence. The region is dominated at lower elevations by chaparral shrublands that historically experienced fire return intervals of several decades, and frequent-fire forests at higher elevations.

Southern California's chaparral shrubland ecosystem is distinctly different from the frequent-fire dry forests elsewhere in the western US, where over-suppression is often discussed as a driver of extreme burning (Keeley and Fotheringham 2001) Over-suppression is less likely to have influenced the January 2025 fires. Pre-colonial Indigenous burning shaped fire patterns and fuel distribution (Keeley 2002), but Euro-American settlement and ongoing urban expansion have altered ignition patterns and increased the frequency of fire (Keeley et al. 1999). Fire suppression resources are extensive but overwhelmed during extreme fire weather, particularly where land use and climate 'whiplash' between extremely wet and dry years produce heavy fuel loads (Swain et al. 2025). Consequently, the contemporary fire regime is characterized by highly variable burned area, frequent wildland-urban interface impacts, and substantial sensitivity to meteorological extremes such as katabatic 'Santa Ana' winds. This context highlights that while human activity strongly influences fire occurrence and exposure, the underlying bioclimatic and ecological conditions continue to govern fire behaviour in southern California (Jin et al. 2015, Parks et al., 2015).

**Congo Basin:**

The Congo Basin region here refers to the moist tropical forest ecoregions of equatorial Africa (Figure 5). Here, fires have historically been rare because short dry seasons and high moisture constrain fuel availability and limit the natural ignitions and spread of fires (Wimberly et al., 2024). Interannual fire variability is positively correlated with higher temperatures and atmospheric drying, and widespread outbreaks were recorded under the anomalously warm and dry conditions of some El Niño events, such as in 2015-2016 (Wimberly et al., 2024; Dwomoh et al., 2019; Verheaghen et al., 2016). Recent satellite observations demonstrate increasing fire occurrence across multiple Congolian ecoregions, particularly in the central lowland and swamp forests, where active fire detections approximately doubled between 2003 and 2021 (Wimberly et al., 2024). These trends are closely associated with deforestation and fragmentation in the central and western parts of the basin, which are largely driven by small-scale agriculture and logging (Shapiro et al., 2021, 2023). Land-use change alters canopy structure and understory microclimates, increasing the likelihood that anthropogenic ignitions will spread into forested areas (Zhao et al., 2021; Dwomoh & Wimberly, 2017). The contemporary fire regime is thus characterised by rising exposure of tropical forests to anthropogenic ignitions, heightened sensitivity to climate extremes, and growing implications for carbon storage, biodiversity, and local livelihoods (Wimberly et al., 2024).

b. And/or the authors ask for input from their expert panels on this, particularly in section 4, where human factors are postulated as potential confounding issues explaining model-observation error.

Thanks, yes, the expert panels have as ever been extremely helpful in this regard.

c. The authors redo model runs for section 5, ignoring population density changes [hold them constant] and instead focus on land cover change as a directly attributable global change process within the limits of current models and data.

We thank the reviewer for this suggestion and also note that it is an important and relevant extension of their recent work. (Perkins et al., 2024) previously criticised the use of population density as a predictor of human influence on fire in DGVMs and FireMIP models, on the basis that these models prognostically define a single global relationship between population density and fire that cannot be expected to realistically capture the diversity of human-fire relationships that exist on Earth (i.e. important aspects of local human agency, such as fire management practices or shifting cultivation dynamics). We fully understand and appreciate the reviewer's concerns surrounding the universal application of a single prognostic relationship between population density and fire.

However, we urge the reviewer to consider that the way population density is treated in our ConFLAME attribution framework is very different from the parameterisations in DGVMs/FireMIP models, and we argue that it remains appropriate to include local population density as a predictor **provided that** it is handled as in the ConFLAME framework when analysing local extreme fire events. ConFLAME treats population density in a more detailed and flexible way than in DGVMs/FireMIP models, specifically as follows:

- Regional optimisation and non-linear response: ConFLAME fits population effects separately for each region, and allows a hump-shaped, bi-directional, or negligible relationship depending on our regional optimisation. This year's setup further separates urban and rural population density, capturing distinct ignition and suppression pressures (Table S3).
- Proxy role for human influence: ConFLAME treats population density not as a direct causal mechanism, but as a proxy for factors such as ignition pressure, suppression capacity, or landscape fragmentation (Barbosa et al., 2025; Kelley et al., 2021). Where population density does not explain observed dynamics, the model assigns it negligible weight to the variable.
- Uncertainty framing: Because ConFLAME uses a Bayesian framework, weak predictors (as population density appears to be in the Congo Basin) do not contribute to model output, and noisy predictors widen the uncertainty distribution rather than forcing misleading conclusions. In this sense, the framework is designed to say when the evidence is insufficient, and our results should be interpreted in that light. In fact, almost all statements on socioeconomic attribution point to uncertainty and need for more work, with Pantanal and Chiquitano being the only region where make an socioeconomic policy assertion which is also backed up by prior research.

Nonetheless we recognise the need for clarity when discussing the modelled relationships between population density and fire, and so we take the following actions to address the reviewer's comments:

1. In Section 4.1.2 and in Supplementary Text 4, we describe at length how our framework deals with population density as a flexible, non-prognostic predictor of fire and how the unresolved effects stemming from important aspects of local human agency (e.g. fire

management practices or shifting cultivation dynamics) feed into the uncertainty term of model such that the weight given to population density as a predictor is not artificially inflated.

"While PoF is trained globally, ConFLAME is trained separately for each region to capture regional variation in the relationship between fire drivers and BA. This design is particularly important because global parameterisations, such as the use of population density as a proxy for human influence, are known to mask regional differences in ignition practices, land-use regimes, and fire management (Perkins et al. 2024). By focusing on regional training, ConFLAME can more directly capture how local ecological conditions (e.g. vegetation type, biomass structure). Population desnity inparticular is split between urban and rural population densities, and BAs response to them is represented with flexible, non-linear response curves, allowing them to act as both ignition-related drivers (alongside lightning, crop and pasture fractions) and suppression/fragmentation controls (alongside crop, pasture, and urban extent). This formulation captures region-specific, potentially humped relationships between population density and burned area, reflecting how ignition pressure and suppression capacity vary across different human—landscape context."

2. We performed a sensitivity test as suggested by the reviewer in which population density was excluded from the ConFLAME model runs. We find that removing population density had little influence on our overall attribution conclusions, except for in the Pantanal–Chiquitano region where results are more sensitive. Please see the results tables below, which will continue to be available through this public discussion.

In addition to ConFLAME, we draw on the FireMIP ensemble to provide an independent line of evidence. FireMIP models each represent human influences differently (e.g. through land use, population density, or suppression), and the ensemble is weighted by regional performance. This means that, all else being equal, models which fail to capture local fire—human interactions contribute less to the combined result, while those that reproduce observed dynamics are given greater weight. As with ConFLAME, missing processes do not create spurious signals but instead appear as wider uncertainty ranges. We acknowledge that FireMIP includes only a limited set of socioeconomic processes and thus cannot fully resolve human-fire relationships; however, it provides a useful cross-check against single-model approaches, ensuring that attribution conclusions are not overly dependent on one framework. By applying both ConFLAME and FireMIP, we follow a multi-model strategy that explicitly represents uncertainty, highlights where signals are consistent across approaches, and identifies regions where future model development on human—fire relationships is most needed.

We note that the ConFLAME approach, using similar predictors to those employed here, has been used in several regions previously, including in (Barbosa, 2024; Barbosa et al., 2025; Kelley et al., 2024, 2019, 2021; UNEP et al., 2022), and independent approuches have also shown the value of using population metrics in South American Biomes (Ferreira Barbosa et al., 2022). For the current application of ConFLAME, in the context of an assessment report which is not seeking to introduce fundamental changes to existing published methodologies, this approach (and the predictors used) is recognised as one of the most advanced methods employed to simulate fire (Huntingford et al., 2025). This year's configuration does advance somewhat in this area, by incorporating land cover

change and a more nuanced population representation (urban vs rural). In addition, we also note that we are strictly following the standard community-led definition of "socioeconomic factors" from the ISIMIP protocol (<a href="https://www.isimip.org/protocol/3/">https://www.isimip.org/protocol/3/</a>) (Burton et al., 2024; Frieler et al., 2024), and is consistent with last year's report. We agree that this remains a partial picture of human-fire interactions, but using the standard protocol allows comparability across studies while continuing to refine the representation of human drivers in fire attribution. This is not to say that we would oppose revisions to the protocol in future.

As a continuation of this response, please see also our full response to comment "Major comments > Science > Bullet #5", which includes examples of the text that we have added to clarify how our framework deals with simple and coarse predictors such as population density.

Attribution results from ConFLAME under the ISIMIP protocol, including regional optimization of urban and rural population density.

| Variable               | Northeast
Amazonia | Pantanal and
Chiquitano | Southern
California | Congo Basin       |  |  |  |
|------------------------|-----------------------|----------------------------|------------------------|-------------------|--|--|--|
| Total climate change   |                       |                            |                        |                   |  |  |  |
| ВА                     | 1.01 [0.88,1.15]      | >100 [2.73, >100]          | 1.07 [0.68, 2.83]      | 1.08 [0.95, 1.43] |  |  |  |
| Areas of highest BA    | 1.02 [0.94,1.13]      | >100 [4.92, >100]          | 1.00[0.91, 1.86]       | 1.14 [0.87, 3,02] |  |  |  |
| Socio-economic factors |                       |                            |                        |                   |  |  |  |
| Burned Area            | 0.99 [0.8, 1.41]      | >100 [2.12, >100]          | 1.04 [0.17, 85.59]     | 0.94 [0.7, 1.17]  |  |  |  |
| Max. Burned Area       | 1.02 [1.07, 1.13]     | >100 [16.24, >100]         | 1.00 [0.85, 6.,65]     | 1.00[ 0.68, 1.69] |  |  |  |
| All forcings           |                       |                            |                        |                   |  |  |  |
| Burned Area            | 0.99 [0.81, 1.47]     | 1.08 [0.44, 7.21]          | 1.05 [0.26, 64.26]     | 1.01 [0.86, 1.42] |  |  |  |
| Max. Burned Area       | 1.01 [0.96, 1.10]     | 1.04 [0.98, 8.26]          | 1.00 [0.86, 12.16]     | 1.06 [0.73-4.44]  |  |  |  |

Attribution results from ConFLAME under the ISIMIP protocol, without using population density. Bold and underlines results would change the IPCC defined attribution statement. See Table 5 for definition.

| Variable             | Northeast
Amazonia | Pantanal and
Chiquitano | Southern
California | Congo Basin       |  |  |  |
|----------------------|-----------------------|----------------------------|------------------------|-------------------|--|--|--|
| Total climate change |                       |                            |                        |                   |  |  |  |
| ВА                   | 1.02 [0.97,1.21]      | >100 [1.07, >100]          | 1.09 [0.70, 2.70]      | 1.08 [0.99, 1.31] |  |  |  |

| Areas of highest BA    | 1.02 [0.98,1.31]  | >100 [3.12, >100] | 1.00 [0.92, 1.70]  | 1.03 [0.94, 1.35] |  |  |  |
|------------------------|-------------------|-------------------|--------------------|-------------------|--|--|--|
| Socio-economic factors |                   |                   |                    |                   |  |  |  |
| Burned Area            | 0.98 [0.77, 1.21] | 2.39 [1.42, >100] | 0.88 [0.15, >100]  | 4.73 [0.12, >100] |  |  |  |
| Max. Burned Area       | 1.00 [0.97, 1.29] | 3.12 [0.99, >100] | 1.00 [0.81, 1.75]  | 4.81 [1.98, >100] |  |  |  |
| All forcings           |                   |                   |                    |                   |  |  |  |
| Burned Area            | 0.99 [0.81, 1.47] | 1.08 [0.44, 7.21] | 1.05 [0.26, 64.26] | 1.01 [0.86, 1.42] |  |  |  |
| Max. Burned Area       | 1.01 [0.96, 1.10] | 1.04 [0.98, 8.26] | 1.00 [0.86, 12.16] | 1.06 [0.73-4.44]  |  |  |  |

Examples of the consequences of this in the substance of the report include:

- No mention of the civil conflict in the DRC as a potential contributor to the fire anomaly.

**Please see our answer to your more specific comment below**

No mention of the legacy of over-suppression in the California fires. Where suppression
is discussed, it is discussed as having an immediate negative forcing on burned area,
neglecting this longer-term fuel feedback. Lack of account for this kind of longer-term
interaction is a substantial limitation to the discussion of results and attribution.

The legacy of over-suppression is a key issue in many frequent-fire forests of the western United States. However, the situation in southern California is somewhat different from these forested regions. The January 2025 Los Angeles event occurred primarily in or adjacent to shrubland rather than in forests where fire exclusion legacies are most relevant. In these chaparral shrubland systems, fire suppression has not created a deficit of fire as has occurred in dry, frequent fire forests. Rather, there is a substantial body of literature showing that fire frequency has increased in these shrublands over the past century due to a combination of substantially increased human ignitions and a conversion of native species to non-natives (Keeley, 2002a, b; Keeley et al., 1999; Keeley and Fotheringham, 2001). Further, the 2025 LA Fires were primarily an urban conflagration type of wildfire (Calkin et al., 2023; Wei et al., 2023), wherein a fire may start in nearby wildland vegetation, but are ultimately maintained by the built environment itself (i.e., houses). The urban conflagration was facilitated by the combination of unusually delayed winter precipitation into January (i.e., drought conditions) and the katabatic 'Santa Ana' winds that allowed ignitions in wildland vegetation immediately adjacent to suburban/urban housing to easily ignite housing and landscaping (Barnes et al., 2025). As such, there is little evidence that the history of over-suppression in California frequent-fire forests contributed to the 2025 LA Fires.

We agree that the lack of explicit representation of these longer-term fire—fuel—management interactions is a limitation of the current modelling framework. For ConFLAME, processes such as these that are not explicitly represented are

incorporated into the "error-based uncertainty" framework outlined in (Kelley et al., 2021), so that their potential influence is captured in the uncertainty ranges presented, though these uncertainties would not be attributed to human factors in the models current form (see responses to specific comments below). So in this case, if we were to include representation of long term human-fuel-fire dynamics in future model development, it may help constrain uncertainties in the future.

Both systems do not explicitly store past years' conditions, but are informed by prognostic variables derived through physical modelling. This means that multi-year processes are implicitly represented, as the fuel state reflects the legacy of antecedent weather, vegetation dynamics, and land—climate interactions. For example, the long-term accumulation of fuels following moist periods is captured in the fuel product, even though attribution is made at the time of the fire. In this framework, such effects are attributed to the fuel state rather than directly to the prior weather. This raises the conceptual point: should we consider the driver "fuel today," or "weather from previous seasons," given that one is a prerequisite of the other? Our approach takes the state variable on the day of the event, which inherently includes past influences.

**To clarify this we have added:**

"Sparky-PoF inherently reflects long-term conditions, for example antecedent weather and multi-year processes are expressed in the fuel state on the day of the event. In such cases, e.g. where prior weather manifests through its influence on fuel accumulation, it is therefore categorised as a fuel driver rather than as weather itself. We assign past conditions that build up fuel loads to the fuel category, while shorter-term processes such as drying are attributed to weather, though the boundary between these two timescales is not always clear."

 Tendency towards weakly substantiated comment on policy interventions around direct human impacts on fire, and / or, general discussion of human factors as confounders without reference to specific knowledge and data on the focal regions [timing of different forms of agricultural fires in the Amazon basin, shifting cultivation in the Congo, e.g,].

The reviewer has very kindly identified most of these instances in their detailed review below (thank you! Makes replying easier). See see responses to specific comments.

4) There is a potential conflict of interest regarding discussion of carbon markets that should be disclosed. This is primarily a matter of due process & transparency rather than substance, with one exception that is noted below.

We agree and have added the following text to the Competing Interests Statement:

"NA is an employee of BeZero Carbon Ltd., a carbon credit ratings agency for the voluntary carbon market."

**Major comments**

**Science**

• The science of this report is generally excellent. Highlights include: assessments of which aspects of fire regimes (spread rate, numbers of fires, fire size) drove BA anomalies, the use of expert panels to identify focal events, the novel methods used to identify damages and exposure, the use of PoF to hindcast focal events.

Etc.

Thanks! Indeed lots of innovations are being driven by this effort.

• One important point is I think it is vital that you give due weight to regions that do not experience anomalous fire years. E.g., Eurasia, section 2.2.1.5. Just as in the sections explaining the fire regime context of anomalous burned area, can you show that there were factors constraining the impact of climate change? Were there fewer fires? Or, were there anomalous wet years? It's great that you report these "no results" for the year, but a little focus on their constraining factors would be good as well to ensure a rounded presentation of all the data.

We agree that a stronger focus on the constraining factors explaining *non-anomalous* fire years is needed across the fire sciences, where long-term trends and extreme events have consistently received greater attention than explanations of non-events (as is particularly obvious in the case of attribution studies).

Our regional expert panels were constructed in part to assist us in providing the context behind observations such as this. In the case of Eurasia, for example, our panellists were presented with evidence from global Earth observations that Eurasia's broadly unremarkable fire season (save for some clear exceptions).

No clear consensus emerged from our regional panel for Eurasia, signalling that a dedicated analysis would be required to pick apart the drivers. Given the reports considerable length, the enormous effort required to deliver it, and the very clear focus on studying extremes, a combination of practical limitations and strategic priorities prevent us from performing a dedicated analysis for every "non-event" in detail (indeed, we only had capacity this year to study four extreme focal events).

To address the comment as directly as is feasible with the information to hand, we have now added details of the fire count and fire size anomalies for statements about key biomes in the section for Eurasia in section 2.2.1.5, thus providing further context as to the patterns underlying the report BA and C emissions anomalies.

We do encourage standalone work to conduct more detailed investigations of the causes of "non-events", and many of the global datasets which we create in Sections 2 and 3 and the model code we provide in later sections are openly available to support such research objectives.

Relatedly, we note that greater attention to extreme *negative* anomalies would be valuable in our research field. The likelihood of "extreme low fire" years is probably shifting in many regions due to global change factors, with important implications (for example, a shifting frequency of 'respite years' when fire managers can allocate resources to prescribed burning programmes or other pre-emptive measures).

Also in response to comments from reviewer #3, who identified the separation between anomalous FWI and anomalous BA in some locations from section S2.2.1, we added the following text to section S2.2.1:

While the present report focuses primarily on explaining focal events that did emerge as extremes, we recognise the underexplored value of examining the factors that constrain fire occurrence in regions where anomalously high fire weather might otherwise be expected to drive extremes in burned area and associated carbon emissions. Future iterations of the State of Wildfires assessment may therefore consider giving greater emphasis to understanding why such extremes did not materialise. That said, this type of analysis has not, to our knowledge, been a common approach in fire science to date. For example, we are not aware of any formal attribution studies focusing on non-extreme fire events, in contrast to the growing number of attribution studies of extreme events. It may therefore be more appropriate for such investigations to be pursued initially as a dedicated exercise, whether within our network or by others.

• In general, I think it is very reasonable to focus on large and extreme fires (wildfires). However, there are times when a lack of explicit differentiation between such events and agricultural fires, e.g., is problematic. This is particularly true, in my view for your (otherwise excellent) population & asset exposure calculations. You note the region with most people exposed is Uttar Pradesh (India), however it is extremely well documented (and indeed reflected in the GLOCAB/GFED5 crop fires product) that such fires are predominantly generated through rice-stubble burning. These are not climate driven events, nor wildfires as such. This lack of distinction may partly explain the large difference between exposed and displaced persons. Similarly, in the DRC (where you note recurrent exposure), this is likely related to widespread shifting cultivation fires.

This is an excellent point that we address by integrating the following explanation into section 3.2.1:

"We note that our analysis of population exposure to fire captures exposures to all fire types (see Section 3.1.1), including wildland fires but also fires observed in agricultural settings (e.g. crop stubble removal or for pasture maintenance) or in shifting cultivation systems, which generally pose low direct risks of fatality or injury. The use of fire for agricultural burning is widely documented in Uttar Pradesh (Shyamsundar et al., 2019) and shifting cultivation is a widespread practice in the Congo Basin (Molinario et al., 2015; Tyukavina et al., 2018), with these modes of fire use dominating over other uses in the respective regions (Millington et al., 2022). As a result, some regions ranked highly for population exposure in our analysis are particularly susceptible to inflated estimates due to the prevalence of exposure to relatively low-risk fire types. Recent work has begun to address this issue by

quantifying exposure specifically to higher-intensity fires (Teymoor Seydi et al., 2025), an approach we intend to adopt in future editions of this report."

The hindcasting section (4) is generally excellent. It is reasonable, in my view, to acknowledge that difficulties in understanding the direct human drivers of constrain the certainty of results (e.g. lines 1788-1797, 1911-1918). However, I think you need to do better than speculate on potential human causes of model-observation divergence, and rather point to explicit documented evidence about land use practices in the focal regions. Evidence around human fire use & management is increasingly systemised (see Kasoar et al., 2023 Table 1 for an overview; <a href="https://doi.org/10.1016/j.crsus.2024.100128">https://doi.org/10.1016/j.crsus.2024.100128</a>). Such systematic data could be used to identify studies of possible direct human drivers in each focal region. At a simple level, the omission of mention of the armed conflict in the DRC as a potential driver of fire, e.g., is notable. This may explain the unattributed anomalies in northeastern DRC (line 2177). Similarly, there are data available on the seasonality of shifting cultivation fire use in the region (see refs at end), which seems to be a partial explanation of the difficulty of forecasting there.

This is a very good point. The discussion now highlights the possibility of performing a more causal analysis that merges hindcast results with available data on human practices. Such a study would require extending the analysis further back in time, not just 2024 to assess how human practices have correlated with or changed fire patterns.

Given the scope and size of the report, we need to focus on summarising established knowledge rather than conducting deep analyses of every aspect so while these investigations could be possible they are outside the scope of the report. However, it is a fair point that we should point to possible avenues where our findings could inspire separate efforts. We fully agree that systematic datasets, such as those referenced e.g. (Kasoar et al., 2024), provide an excellent foundation for future work to better capture direct human drivers.

As for the comment about the ongoing conflict in DRC we notice that most of the burning took place in areas outside the zones directly affected by conflict. That said, the overall instability in the country may still influence fire activity through socio-economic factors. Making a direct causal link is difficult, though, given the very limited reporting of fire events in these regions. We therefore decided to refer to the conflict only as a possible factor that could modulate ignitions.

We have included all of these considerations in the text as follows:

In particular, the FWI fails to capture the complex interactions among fuel availability, ignition sources, and human activity. Although the PoF incorporates some of these elements, it also struggles in regions where human presence and behaviour are rapidly changing. In such cases, while most burning occurred outside conflict zones (Trigg et al., 2012), the broader instability in the region may still influence local fire activity, challenging the predictive capabilities of data-driven systems if not trained on the most recent data. This

limitation is especially evident in areas where natural ignitions are infrequent and fuel dynamics, rather than weather alone, drive fire occurrence and behaviour (Figure S13). In the future, these inherent limitations may be addressed by incorporating more fire-specific socio-economic factors. We are aware that datasets providing more detailed information on human practices are becoming available (Kasoar et al., 2024), and these may help constrain and improve forecast skill going forward. Such datasets could also provide a valuable basis for further exploring the links between fire predictability and human influences, building on the data presented in this report.

• The attribution section (5) is generally very good. However, human population density has an ambivalent relationship to human fire use and management (Perkins et al., 2024; doi https://doi.org/10.5194/gmd-17-3993-2024). These are much more driven by land user objectives and socioeconomic context; factors which are not accounted for in your attribution. By contrast, land cover change does have some more predictable impacts (croplands being suppressive of large fires, e.g.). Therefore, I think it would be much stronger simply to attribute burned area to land use changes, and ignore population density. Whether you include pop density or not, you cannot currently claim to be attributing changes in burned area to "socio-economic factors", because population density is a fundamentally unsuitable measure of these. But you can do so for land use change. You note this issue in lines 2619-2630, but as a structural limitation in the modelling it would be better to account for this systematically by narrowing the focus of your attribution, in my view. In the minor comments I have noted specific pieces of comment on results that I think are not sound for these reasons – in my view you should limit your comment to those aspects of the system you are robustly modelling.

As discussed above in relation to "Key findings & recommendations > Comment 3c", we agree that globally parameterised population density alone cannot be considered a direct representation of land-user objectives or broader socioeconomic context. To reiterate: In ConFLAME specifically, population density is not treated as a stand-alone causal variable but as a locally calibrated, associational predictor. Rural and urban population density are fitted with flexible, non-linear response functions, and influence the model only where there is empirical support. Where population density is uninformative, it receives negligible weight, and the Bayesian framework ensures this is reflected as additional uncertainty rather than a spurious signal. We have made this clearer by adding the following in section 5.1.3, where we described ConFLAME based attribution:

"As an addition to last year's report's set up, our ISIMIP set up also includes (i) changes in land use and cover (measured as the difference between tree cover and agricultural fraction since the previous year) and (ii) a separation of urban and rural population density in the socio-economic forcing attribution (see Table S3). The former allows us to capture direct effects of land-use change on fire, rather than only static land-use states, while the latter enables us to represent non-linear relationships between population density and fire, including both positive and negative influences. Together, these developments represent methodological advances introduced in this year's report. As noted in Section 4.1.2, ConFLAME is optimised regionally and, for population density in particular, provides a more

effective parameterisation than would be possible with a globally optimised model (Perkins et al., 2024)."

The new sensitivity tests (see responses and results tables under ""Key findings & recommendations > Comment 3c") show that excluding population density has little effect on most regions, but in the Pantanal and Chiquitano in particular the model's confidence decreases, consistent with independent studies that identify settlement patterns, road proximity, and agricultural expansion as strong local predictors of fire occurrence (Barbosa, 2024; Barbosa et al., 2025; Devisscher et al., 2016; Ferreira Barbosa et al., 2022; Tomas et al., 2021). Conversely, in much of the Amazon, fire is more strongly linked to deforestation and land-use change than to population density per se (Pessôa et al., 2023; Singh et al., 2022), though in sub-regional extremes, including our population measures helps.

For the FireMIP ensemble, rerunning experiments without population density is not feasible given the scale of coordination, but the framework mitigates this limitation by weighting models according to regional performance. Models that poorly capture local human—fire interactions contribute less to ensemble results, while uncertainty is expanded where processes are under-represented. This ensures that signals attributed to human drivers are conservative and uncertainty is made explicit, as now explained in section 5.1.4:

"Uncertainty is evaluated through random resampling of the weighted ensemble, including a stochastic parameter that captures uncertainty in overall ensemble performance. This weighting reduces the influence of models that fail to capture local fire—population or fire—bioclimate relationships, but does not fully resolve structural gaps in the ensemble. In particular, weak performance for socio-economic drivers may widen overall uncertainty without attributing it to the specific process. As a result, FireMIP provides a conservative estimate of regional-scale fire responses, complementing our more detailed but regionally focused approaches."

In response to the reviewer's concern about terminology, we have revised the text to clarify that in our study "socioeconomic attribution" refers primarily to land use and land cover change, which are directly represented, with population density included as a regionally optimised proxy for ignition, suppression, or fragmentation processes (Kelley et al., 2019, 2021). We explicitly state that this remains an evolving area, and that population density should not be read as a complete measure of socioeconomic influence. Future integration with more process-based approaches (e.g. WHAM!) will be essential to reduce this uncertainty and better capture land user objectives and governance, by adding the following to section 5.1.1 "Overview of Attribution Approaches"

"A key challenge that has rarely been addressed before is how to represent socio-economic influences on fire within a framework that can also attribute climate change. This is a relatively new area of research, and progress has been limited by data availability, the complexity of human—fire interactions and how to represent these in models. The absence of variables capturing human agency, such as deliberate ignitions, suppression, or governance, has made it difficult to capture the ways people influence fire (Perkins et al. 2024). For instance, previous attempts to incorporate socio-economic drivers have often relied on global parameterisation of population density as a simple predictor of fire activity. However, this relationship is shaped by local cultural, political, and economic contexts and can

therefore give misleading results. In this report we extend last year's framework by taking a more detailed and regionally grounded view of how people influence burning through using urban and rual population densities ( see section 5.1.3) and using none-linear response mechanisms (section 4.1.2). This represents an important step towards capturing socio-economic influences more realistically, but it is still only a partial representation. Considerable work remains to incorporate human agency and broader socio-economic context (Millington et al. 2022), and we see this as an active area of development for future reports."

The reviewer has also provided specific comments for individual regions, in cases where coarse population density of land use variables may fail to capture the unique human-fire relationships in those regions. We respond to those comments individually below.

**Communications**

- Overall, I am not sure the report rises to its stated aim "to deliver actionable information
  to policy and practice stakeholders and wider society" (my emphasis). Overall, the key
  messages & headline findings of the report are not clearly enough communicated for
  non-specialist audiences, in my view. This is not a problem unique to the SOW IPCC
  reports have been critiqued by the policy community along similar lines, after all.
- The background section is too dense for non-academics, and I don't think it would provide
  such readers with a clear sense of how fire is changing globally and how this year fits in.
  Page 5 is stronger in this regard than page 4, which suggests to me the issue is in being
  crystal clear on the absolute headline message of the report and how you wish to
  communicate it.
- I wonder if you have mapped your key audiences & key messages for the report? The
  Abstract reads to me a bit like you are framing this in the context of the Carbon budget.
  But is this the most important message for your stakeholders (the feedback of fires on
  further climate change)? The key findings in section 3 may be more attention grabbing. I
  have had a go at providing an overall report message in the minor comments. Feel free to
  disregard.
- In any case, I would replace the abstract with a 2-page high level executive summary that tries to nail down your key findings and messages, accompanied by 1 or 2 high-impact graphics. This could be, in effect, a crunched down version of (the excellent) section 7.

Please see our response to your important points in you "key findings and recommendations section" - in short: this publication in ESSD is definitely aimed at scientists, but yes we have absolutely mapped messages to non-scientific audiences with help from comms professionals, and we describe our broader comms and outreach strategy above. It is difficult to cover all angles of interest in a short space, but we do want to hit as many of them as possible by discussing a range of impact metrics that would be of interest to different research communities (i.e. magnitude/scale of BA/emissions, fatalities, costs, evacuations - it should vary in each reporting year depending on our findings and the nature of the fire season).

A contents page would be extremely useful as well!

We enquired with the publisher; this is not permitted.

Similarly, your figures are scientifically logical and appropriate. However, some of them they could be more concise to make the information in the report more impactful. E.g., Figure 2/3

 how much information is added by the Global Administrative regions panels? The figure is too small to see clearly and without looking it up, an average reader is not going to know what these are. I might argue the same for ecoregions, which could also go in a supplementary figure. Once you get into the details of attribution I understand that more detailed figures are required.

It is important to provide the sub-regional panels because this emphasises the variability that can occur within countries for many of the variables presented. Part of the premise of our report is to highlight events that go under-reported, and we find that presenting data on smaller geographical units is helpful in highlighting regions that experienced extremes that are smoothed out when looking at large geographical regions such as countries. If readers are inspired to look closer, and to possibly download the data layers that we provide to support investigations into specific regional cases, then we see this as a positive. Hence, on this particular point, we must disagree with the reviewer's suggestion to remove the plots for smaller geographical units.

Nonetheless we appreciate the need to improve the readability of these plots and so we have made efforts to do so as part of our revisions (e.g. increased text size).

**Transparency of competing interests**

• Finally, I would expect to see authors from BeZero Ltd note their role in the industry in competing interests, given explicit discussion of carbon projects (which is itself very useful). A phrase such as "High-integrity forest carbon projects can help to mitigate global climate change, and we find some evidence that these interventions are also reducing fire risk locally." (lines 1540-1541) may well be true. However, it is also in line with the commercial interests of BeZero to make such a claim -> i.e. promoting the value of high integrity carbon projects and hence the importance of ratings. Readers of the report should be aware of this potential conflict so they can transparently evaluate the claims made. In addition, if the specific claim regarding reduced local fire risk is to be made, it needs substantiation & should be placed in the context of documented cases of carbon projects having led to displacement of fire-dependent livelihoods in the global South (e.g. Croker et al., 2023; https://doi.org/10.1029/2023EF003552).

Important point - thanks! Please see our response above regarding the new declaration of interest.

We also replaced the sentence flagged here as follows: "High-integrity forest carbon projects can help to mitigate global climate change, and some prior work has shown that these interventions can reduce fire risk locally (Croker et al., 2023)."

**Minor comments**

**Section 1**

**Abstract**

Given the likely very large underestimate of BA in the MODIS product (dealt with from line 477), I don't think it makes sense to quote the headline BA numbers in km2. A % change relative to the historical trend would make more sense to me. Alternatively, use the GFED5 number for the headline totals, and global fire atlas metrics for other aspects of the regime?

At present, GFED5 data are only available through 2020 (<a href="https://zenodo.org/records/7668424">https://zenodo.org/records/7668424</a>). Our section 2 focuses on taking multiple BA and C emissions estimates from products that are continuously and publicly available through February 2025. Also, GFED5 is only available at 0.25 degree spatial resolution, which is not sufficient for a consistent analysis across all regional layers in our analyses. Moving forwards it is more likely that VIIRS VNP64A1 will be the most appropriate product for our analysis.

Of course, this raises a predicament in communications. Our solution in this instance is to remove the BA total from the abstract as suggested and allow readers to take the full context from the later methods and results sections.

The sentence about Southern California (lines 144-146) is a bit confusing – are you describing some theoretical modelled and observed burned area here?

Thanks, we have clarified this sentence so that it conforms with how this result is reported for the other regions.

I don't think it's accurate to describe SSP370 as middle of the road. "Medium-high" is a better description. SSP245 is explicitly "middle of the road" in the narratives. This change should be made throughout the report.

We have amended the name as suggested

**Introduction**

I don't think this framing (lines 174-177) is quite right. After all, global burned area is still seemingly declining, so at least on one measure, the potential for wildfires isn't growing (the phrase "potential of wildfires" is itself a little ambiguous to me). I think a more accurate/sharp key message could be: "There is a global transition from low-intensity, low-emission and low-harm grassland fires, to high-intensity, high-emissions and highharm forest fires... This is a long-term transition, driven by a combination of climate change and land use change, which continued forcefully in 2024-25." You do make this point, but not until the first paragraph of your results!

More broadly, to me, describing lots of increases in burned area in paragraph 1, but then contrasting this with the overall declining global trend (paragraph 2) is confusing to the non-specialist reader.

The first two paragraphs are also too long for a non-specialist audience. Important messages, such as that globally aggregated BA is increasingly less important as a measure of contemporary fire regimes get hidden in the middle of paragraphs and likely skipped over.

The opening of the second paragraph mixes discussion of the trend towards destructive fires in fuel-rich environments with the overall declining trend. This all needs reframing for clarity, to my mind.

Could we have some figures in the background section to capture these key trends? Simple clear charts to communicate the key overall dynamics/trends of contemporary fires (e.g., grasslands vs forests, tropics vs extra-tropics?).

On page 5, paragraphs 2-4 (lines 228 – 267) are much stronger than the opening, to my mind. Each with a simple clear message succinctly communicated. These could all be a symbol & key message in an infographic.

Finally, I think some really key, fundamental points are missing. I know that we (researchers) all know that fire is both a natural ecosystem process & man-made, but you should make this somewhere in the material about declining burned area in grasslands.

Thanks for this collection of comments, which were a very helpful steer. We have re-structured the entire introduction in a manner that, we feel, builds a clearer narrative with punchier messages. Your paraphrased suggestion above ("There is a global transition from low-intensity, low-emission and low-harm grassland fires, to high-intensity, high-emissions and highharm forest fires... This is a long-term transition, driven by a combination of climate change and land use change, which continued forcefully in 2024-25.") is certainly what we were going for as a set of messages built up across the introduction, so we thank you for the prompt to deliver these messages more clearly.

It is worth bearing in mind that this article is not itself intended for non-specialists (though we may hope to attract some more dedicated readers through our production of comms and outreach materials).

**Objectives of this Report**

These objectives are very useful. Bear in mind that "suppression" is a jargon term to non-academics, who may know this as "fire-fighting".

The key methodologies section belongs lower down the report, in my view. I would instead replace this with a clear indication of the logic of the *structure* of the report, and how this delivers against your objectives. You could then include a table summarising the objectives, the relevant report section (s) and the methods used.

In this section we had aimed to emphasise our project's ambition to stimulate advancement and growth of this research field, and so we have made our motivations for presenting this

clearer via a clarifying sentence at the beginning of the paragraph:

"The delivery of this report and its objectives relies on critical datasets and models developed by the research community over many decades. By applying these tools to the challenge of studying extreme wildfires, we not only gain insights into their strengths and limitations for studying extreme fires, but also drive scientific and technological innovation that advances our ability to monitor, explain, and predict such events."

The paragraph 367-372 is a bit repetitive of para 319-322. We should know what the report is trying to achieve by line 367.

Thanks, our revisions (see response to major comment above) lessen this repetition.

**Section 2**

**Methods**

This all seems very good and appropriate.

The use of expert panels to complement EO data is impressive.

Thanks - it's been a fantastic experience to work with the regional panels and a very unique element of this report.

As noted above, it of course makes sense to use global fire atlas data as the primary source in a report focused on extreme fires, which enables the excellent discussion of fire spread rates and sizes etc. However, I am not clear why you do not report total burned area from GFED5?

At present, GFED5 data are only available through 2020 (<a href="https://zenodo.org/records/7668424">https://zenodo.org/records/7668424</a>). Our section 2 focuses on taking multiple BA and C emissions estimates from products that are continuously and publicly available through February 2025. Also, GFED5 is only available at 0.25 degree spatial resolution, which is not sufficient for a consistent analysis across all regional layers in our analyses. Moving forwards it is more likely that VIIRS VNP64A1 will be the most appropriate product for our analysis.

To my mind, given that you used summary values for FRP (if I have understood correctly) it belongs in section 2.1.2.1, not 2.1.2.2.

The FRP information is presented in section 2.1.2.2 because FRP observations are initially summarised within the individual fire perimeters of the Global Fire Atlas, prior to averaging across all GFA fires regionally. This methodology would be challenging to introduce without already introducing the GFA dataset (as we do in section 2.1.2.2).

**Results**

**Overall**

See major comment on Eurasia & providing context for negative results.

Please see our response to your major comment above regarding the results for Eurasia.

Why are selected ecoregions chosen for Figure 1, and then long tables for administrative regions? Figure 1 is much more impactful communication.

We explored this option but the region names were too long to feature as axis labels of a column plot.

The tables do also enable us to showcase the more complete range of the information we are providing in the datasets generated by this report (beyond just the anomaly statistics visible as in Figure 1).

Similarly, Figure 4 contains lots of interesting information, but is too messy for me to understand in current format. Fewer panels, bigger panels, more aggregated regions, or some combination, please.

We have taken action to make the figure more readable (bigger panels, bigger font sizes).

We decided against presenting coarser regional aggregations (e.g. countries) because we prefer to emphasise that there are large sub-national patterns in the data.

The dataset underlying this plot includes data for all regional layers that we used in the report, and also we are providing a range of interactive visualisations in our outreach materials (e.g. updated from <a href="https://www.uea.ac.uk/climate/climate-data/state-of-wildfires">https://www.uea.ac.uk/climate/climate-data/state-of-wildfires</a>) to facilitate easy switching between layers for broader (e.g. non-scientific) audiences.

Line 1205, I think "intriguing" is the wrong word here – "important".

Corrected to "important".

**South America**

The point about fast fire spread driving increased BA across Brazil vs increased fires in the Amazon is fascinating. To my mind that speaks to fire for explicit deforestation vs uncontrolled wildfires spreading purely on climatic factors.

**America**

The section on drivers of extreme fire anomalies is extremely interesting, as per South America.

**Africa**

The increase in numbers of fires also seems likely to me to reflect shifting cultivation fires (the dominant mode of human fire use in the region; see end refs) growing out of control due to anomalous climate conditions. They were likely undetectable by MODIS before, hence the apparent increase in fire number – which may actually be due to increased fire size & hence increased detection.

Thanks, there are certainly some really interesting patterns emerging in these sections for the "beyond burned area" metrics and we generally agree with the reviewer's interpretation

of the results. By providing these statistics, we of course hope that we inspire the formation of hypotheses, so we're pleased to see this happening and hope this stimulates investigation of the more specific regional drivers at play. We think that more direct evidence must be built before making concrete statements/inferences here in section 2, and we prefer to stick here to a description of what was observed based on the methods described. However, some of the regions highlighted above do become focal regions of later report sections where we attempt to diagnose the drivers of extremes – this is where further comment on specific regional circumstances seems more appropriate.

**Section 3**

This section is generally excellent. Figure 6 is very good as a high-level summary.

See major comments on wildfires vs agricultural fires distinction & carbon markets section.

Thanks! Good points. Please see our responses to both major comments above.

**Section 4**

See major comment on human dimensions of fire in your hindcasts.

Thank you we have addressed this comment in the previous section

It is fine not to have detailed human factors in your models (because nobody has yet built this). However, given it seems to play an important role in PoF & ConFLAME model error [in part because the models do an excellent job with biophysical factors] I think you should consider being more explicit about this limitation in the methods section, similar to the excellent section on EO product uncertainty. At present there are several scattered references to missing human processes in the results (e.g., suppression in California [but only as an immediate not long-term forcing], lines 2157-2159, Panantal lines 2128-2131). It would be cleaner and clearer to address this upfront in the methods.

This is an important note and as requested by the reviewer deserved a more detailed explanation which has now been added:

It is important to stress that the representation of human influence on fire is very crude in both systems, with PoF relying only on static maps of population density, road networks, and land use and ConFLAME using time varying maps of urban and rural population density and land. Both methods are still missing a representation of human influence on fire that is far more complex and often shaped by cultural and political constraints. This causality is reflected only in observed fire occurrence rather than being explicitly represented as drivers leading to fire outcomes. These limitations are likely to introduce uncertainties that are much larger than those currently associated with the predictions, but they remain difficult to quantify due to the limited availability of detailed datasets on direct fire use, including suppression efforts. Neither system is free of limitations. Still, this dual-model setup allows for a more robust assessment of fire causes across different spatial and temporal scales, with prediction of occurred fires providing a fine-scale measure of fire activity and BA an integrated assessment of landscape impacts. ConFLAME's Bayesian framework additionally helps address some of these limitations by propagating uncertainty in the estimated contribution of drivers into the modelled probability distribution, which we use directly in our analysis. This means that part of the variability in how human influences shape burning is reflected in the likelihood ranges reported by ConFLAME. Therefore, uncertainties in human-driven effects are incorporated into the analysis. However these reflect uncertainty within the model's current driver set, rather than fully capturing structural uncertainties linked to missing human agency information (e.g. suppression activities, cultural fire use, or policy change). As such, while the posterior ranges provide a useful quantitative benchmark, they likely underestimate the true uncertainty associated with human influence on fire.

Furthermore, I think where you note possible human factors as explanations of mode/lobservation divergence, you should do so in specific reference to published literature. E.g., in the Amazon (lines 1788-1797), there is well-documented evidence of escaped pasture fires as drivers of forest-edge degradation (see refs list). This matches your pattern of ignitions driving fires at the eastern edge of the Amazon (Figure 10). To dig deeper, the seasonality of pasture vs agricultural residue fires in the region could be identified from field data to see whether these match the error in model seasonality predictions (see refs).

Thank you for pointing this out. We were just not aware of the suggested references and have included these into the possible explanation on the differences reported. The new text reads as follows

Interestingly, both the PoF and FWI systems failed to capture a lull in fire activity during the second emergence of fire-conducive conditions between August and November, highlighting the limitations of forecasting fire activity rather than fire danger. In this region, ignitions are believed to be largely driven by escaped pasture burning Cano-Crespo et al. (2015), which typically occurs between August and October Jakimow et al. (2018). The models may have learned and reproduced this seasonal behaviour, but such patterns can be disrupted by changes in human practices. One possible explanation is that these conditions fell outside the usual burning cycles—for example, in agricultural areas where fires are often timed around harvest, the prolonged drought may have reduced crop yields and therefore fire use. This suggests that the models missed the quiet September period because they incorporate only limited information on human ignition patterns, land ownership and land use types, and less-documented factors such as fire suppression, policy interventions, and cultural burning practices Lapola et al. (2023). These gaps underscore the need for improved datasets on human activity, which could significantly enhance fire prediction (Jones et al., 2022).

Incidentally, there is an interesting & important discussion to be had about whether current fire activity in the Amazon case study is indicative of deliberate deforestation, or perhaps more worryingly, of the transition towards forest degradation and inherent instability under a changing climate (see Lapola et al., 2023; https://www.science.org/doi/10.1126/science.abp8622)

Once again thank you for pointing out this study we were not aware of and which has now been referenced in this section

**Section 5**

See major comment on how what exactly you are able to attribute on the human dimensions. I have noted sections where this limitation leads to, in my view, unjustified comment in your results section, below.

As a general comment on the Pantanal case study, in meta-analyses of human fire use, there tends to have been very little in this region historically (too wet). As such, I am not at all surprised to see that it is the event most clearly attributable to climate change (lines 2649-2651). However, because of the limitations of your socio-economic analysis, I think the comment on possible policy intervention focused on constraining direct human influence may be true (lines 2678-2687), but

cannot be justified on the basis of your analysis, which is limited to coarse patterns of land cover change.

For a small piece of added context: the Pantanal-Chiquitano focal region covers both the Greater Pantanal region, which is only partially wetlands and has a significant agricultural economy, and Chiquitano dry forests where fire has long been present. These aspects of the region's fire regime are now discussed in section 2.2.2.1 (per our response to major comments above).

Please see also our response to major comments regarding our broader socio-economic setup above. We agree that our socio-economic indicators are coarser than would be ideal, but the uncertainty structure of the framework still allows us to test a broad range of plausible human influences on fire outcomes. In the Pantanal–Chiquitano, our findings are consistent with independent analyses of recent fire trends and extreme events (e.g. the 2020 fires, (Ferreira Barbosa et al., 2022)), which similarly conclude that direct human activities strongly amplify fire risk in this region along with a Bayesian GLM approach which does include more human and wetland dynamics (Barbosa 2024). This consistency gives us confidence that the policy implications we highlight—focusing on ignition management, land-use enforcement, and wetland conservation—are robust when considered alongside the wider literature(Ferreira Barbosa et al., 2022; Marques et al., 2021; Menezes et al., 2022). We have revised the text accordingly to make this supporting evidence explicit:

"Our socio-economic analysis is necessarily based on relatively coarse indicators (population density and land-use change), and so should be viewed as a restricted but internally consistent representation of direct human influence. Nonetheless, the framework captures a wide range of plausible anthropogenic effects and indicates a substantial role for people in shaping fire outcomes. Importantly, this finding is consistent with independent evidence from recent fire extremes in the Pantanal, including the 2020 event (Barbosa et al., 2022; Barbosa, 2024), which shows that land management, ignition patterns, and water extraction have amplified fire risk alongside climate pressures. Previous studies highlight that management of ignition sources, enforcement of land use regulation, and reduction of wetland degradation and water extraction can reduce fire vulnerability in this region (Barbosa et al. 2022; Menezes et al. 2022 ,Marques et al 2021). Thus, while global mitigation remains essential, locally targeted actions represent concrete and tractable pathways for reducing future fire risk"

Similarly, in the Congo, there is not much relationship between increasing population density and changes in fire use in shifting cultivation. Increased population density can shorten fallow periods (leading to increased fire use), but can also lead to sedenterisation, and hence less use of fire for field preparation. As such, your comment on socioeconomic factors broadly is not warranted (lines 2755-2760).

In our revised text, we have clarified that our socio-economic proxies should be interpreted cautiously in this region, and explicitly noted that population density alone does not resolve the different ways fire is used in agricultural systems. We also now highlight that while agricultural and land-cover change indicators capture broad-scale patterns, they cannot represent management practices such as ignition for field preparation or suppression under different governance contexts. The revised section therefore frames our analysis as capturing only coarse patterns, and points to the

need for future work that can incorporate these more detailed socio-economic processes:

"There is no clear signal in our indicators (population density and land-use change) that socioeconomic factors increased BA during the June-August 2024 fires in the Congo Basin. The likelihood of increased burning was 26% regionally (AF = 0.94, 90% CI: 0.70–1.17), suggesting a small or even slightly dampening influence. At the sub-regional level, attribution remains uncertain, with estimates of 62% likelihood of increased BA in the most affected grid cells (AF = 1.00 [0.68–1.69]). These results should be interpreted cautiously, as our indicators cannot capture the full dynamics of the region. For instance, population density alone does not represent how local conflicts (Meddour-Sahar, et al. 2013; Trigg et al. 2011), shifting cultivation practices vary with fallow length or sedenterisation (Molinario et al. 2020); agricultural land-use fractions capture broad changes in cover but not day-to-day fire use for field preparation; and land-cover change highlights large-scale transitions but not how governance or local management influence burning (Tyukavina et al. 2018; Perkins et al., 2024). In the Congo, these processes interact in complex and sometimes opposing ways, shorter fallow periods can increase fire use, while sedenterisation can reduce it (Perkins et al., 2024). Incorporating such dynamics into future frameworks will be important for improving and tightening socio-economic attribution in this region."

Again, I'm afraid I don't think the analysis in lines 2828-2840 is robust. You aren't representing socioeconomic drivers beyond land cover change. (Sorry to repeat myself).

In the revised text, we have clarified that our estimates for long-term background BA reflect only the processes captured by our indicators, and we explicitly note the contrast with focal-event BA results, where socio-economic factors are more likely to increase BA. This difference illustrates the temporal and functional complexities of human—fire interactions, and reinforces that uncertainty in process representation is a key consideration when interpreting attribution results. Our aim is not to overstate the influence of human drivers, but to provide a transparent and internally consistent assessment while highlighting areas where further methodological development is needed.

"We estimate that socioeconomic drivers, here represented by mianly land-use, contributed a modest reduction in background BA of 7% [-12%, -2%] compared to pre-industrial conditions. This suggests that long-term changes in land use may have contributed to a slight suppression of average fire activity over the past two decades. It is important to stress that our analysis is limited to land-cover change and population density proxies, and does not capture the full spectrum of socio-economic drivers such as ignition practices, fire suppression, or governance which may also influence fire activity. The contrast with focal-event BA attribution, which indicates that socioeconomic factors very likely increased BA (Section 5.2.2.2), may therefore reflect differences in timescale, the specific processes captured by our indicators, or areas where our representation of human influence is incomplete. Region wide BA above, for example, also incorporates land-use change and is locally optimised, more detailed population density-based representations. However, as noted in Section 5.2.2.2, even there we miss crucial aspects of local human agency, which can vary substantially in space and time across the region (Perkins et al. 2024).

**Section 6**

This section is excellent. Because you held land cover and population density constant – you are spared my comment on the limitations of this! I think this is actually better in any case as it allows focus on those dimensions of fire JULES currently captures in a robust way.

**Section 7**

This section is excellent & could form the basis of a revised executive summary.

Indeed, it absolutely does form the basis of our outreach materials.

However, after revising section 5, I think the sections here on socio-economic attribution need to be cut or substantially revised.

We thank the reviewer for this suggestion. We note, however, that the current text on socioeconomic attribution largely communicates the high degree of uncertainty in these analyses, essentially flagging "we don't know" where the available indicators are insufficient. To do this, we have explicitly used uncertainty-based frameworks that allows us to quantify the range of plausible human influences on burned area. These framework is applied consistently across regions and time periods, meaning that when evidence is weak, the attribution reflects that uncertainty rather than making unsupported claims.

Where possible, we have incorporated the reviewer's points, alongside the knowledge of regional experts, to indicate the key missing processes and data that would be required to increase confidence in socioeconomic attribution. This includes, for example, local human agency, agricultural practices, water extraction, and seasonal dynamics. In this way, the text does not overstate the conclusions, but rather transparently identifies the limitations and provides guidance on how future work could reduce uncertainty.

Given this, we believe only minor adjustments are needed to make the language less assertive, rather than removing the section entirely, as it plays an important role in highlighting both the current evidence and the path forward for improving regional socioeconomic fire attribution.

**\*Key relevant papers for each study region from DAFI/recent lit:**

**Northeast Amazonia**

Cano-Crespo et al., suggests escaped pasture fires as the largest source of wildfires in the Amazon - https://doi.org/10.1002/2015JG002914

Jakimow et al., suggests that pasture burning in the Southern Amazon is focused around the main dry season (Aug-Oct) https://doi.org/10.1016/j.rse.2017.10.009

Brunel et al., suggests that pasture burning in the coastal savanna regions is clustered around December-Jan. https://doi.org/10.1016/j.rama.2021.08.003

Cammelli et al., important study to understand behavioural and economic drivers of smallholder fire use in the region - https://doi.org/10.1016/j.gloenvcha.2020.102096

Bilbao et al., 2019, details current management and policy challenges around fire, and makes strong cross-community suggestions for ways forward https://www.mdpi.com/2571-6255/2/3/39

**Congo Basin**

Molinario et al., shifting cultivation and forest loss in the Congo – importantly this study highlights how subsistence farming is often synergistic with increased logging & mining (i.e. you have to feed the workers). <a href="https://www.mdpi.com/2073-445X/9/1/23">https://www.mdpi.com/2073-445X/9/1/23</a>; see also https://iopscience.iop.org/article/10.1088/1748-9326/10/9/094009/meta

Tyukavina et al., on shifting cultivation in the Congo basin - https://doi.org/10.1126/sciadv.aat2993

Examples of fire increasing under conflict:

https://doi.org/10.1080/1747423X.2011.565372,

https://www.scirp.org/journal/paperinformation?paperid=27362,

**California**

I recommend including citation of this modelling study on the long-term impact of over suppression on fuels. https://www.nature.com/articles/s41467-024-46702-0

**Panantal**

There is generally less literature in this region, owing to the wetness & lack of use of susceptibility to fire. Here are the two papers from DAFI, but perhaps indirect impacts including drainage are those to focus on (as already mentioned in the text).

Devisscher et al, - https://doi.org/10.1111%2Fgeoj.12261

McDaneil et al., https://doi.org/10.1080/08941920500248921

**Reviewer 2**

**General comments**

The authors are to be commended for their efforts to so comprehensively summarise the complex and complicated state of wildfires on our planet, to wrangle the datasets and models needed to make sense of a phenomenon that is at once a natural part of our world and a terrifying threat to our existence. However, this document is a beast to read and having read through pretty much all of the 100 pages of the main part of the paper (I admit to skipping some parts just to maintain my sanity), I wonder if any of the authors have in fact read the thing in its entirety. There is a very large amount of repetition in the document. Much of this comes from the nature of the methods employed, being similar in many of the different components being presented, but these could be streamlined to a great extent—this might not reduce the overall length by much but would improve the readability considerably; we don't need to be told more than once that FWI is the Canadian Fire Weather Index and we don't need yet another explanation of its origin.

Thank you very much for your attention to our comprehensive report and for your constructive feedback on readability.

We took the decision to split the report sequentially according to the branches of analysis employed and this does indeed lead to some challenges in avoiding overlap. We are covering a lot of ground in the report and so some repetition is inevitable. Sometimes, this can be beneficial to some readers who wish to 'dip' into one section of the report, e.g. because they are interested in attribution methods; in these cases, flicking back-and-forth to an earlier section of the report to understand a definition is likely to be cumbersome.

However we note that in some cases repetition is categorically not necessarily, most obviously in the case of describing the history and derivation of the FWI across several sections. We have now removed the explanation of FWI everywhere except where it is first introduced.

We also addressed any other specific cases of repetition flagged in minor comments below.

In the same vein to improve the reading experience, I would strongly suggest the authors consider moving their Conclusions section to the very front of the paper. I found this section to be the most readable and to convey all the important information any reader of the article would want without the need to wade through the pages and pages of detail on method before getting to the juicy stuff—I think it works better than the Abstract in this because it hasn't been so condensed; maybe call it an extended abstract or similar.

This may break with the standard article structure of this journal and the previous State of Wildfires (I've not read it), but would make future editions of this report so much easier to ingest. The detail of the methods could then follow.

We agree that the Conclusions section provides an accessible overview of the study's most important findings and can serve as an "extended abstract" of sorts. As the reviewer notes, reordering the paper in this way would depart from the standard structure used by ESSD and in previous State of Wildfires reports. For consistency and in line with guidance from the journal, we have retained the conventional structure (Abstract–Introduction–Methods–Results–Conclusions).

However, we very much take this comment on board for future iterations. As suggested, we will discuss with the ESSD editorial team whether a restructured format might be feasible in future editions of the report.

We also want to highlight that we are addressing this need for accessibility through complementary outputs released alongside the scientific report. The Conclusions (Section 7) will be used as the basis of our Summary for Policymakers and other outreach materials, which are co-released with the report and designed to engage audiences outside of the scientific community. This strategy aligns with our responses to review 1: while the ESSD paper is primarily aimed at a scientific readership, we ensure that key findings are communicated to non-specialist stakeholders through tailored summaries, visual products, and our project website.

**Some other high-level comments:**

1. I don't quite see the point of the climate projections for the four selected extreme event localities. Unpacking what happened to them this year makes perfect sense but why would one care more about the seasonal and multi-decadal outlooks for these areas than anywhere else?

We agree that the primary role of this report is to unpack what happened during this year's fire events. However, it is equally as important to provide a forward-looking perspective for the same regions. This serves several purposes:

**Policy and stakeholder demand** – Future projections of wildfire risk were highlighted as a major component of understanding global fire threats in the UNEP *Spreading Like Wildfires* report (where the reviewer was a lead editor). They are also one of the most common requests we receive after major wildfire events.

More specifically, such projections form a crucial piece of evidence for adaptation planning and for international mechanisms under the UNFCCC process, including those associated with adaptation funds and Loss & Damage. Stakeholders working in this area have explicitly requested that such material be included, which enhances the policy relevance of this report.

**Methodological progression** – The projections included here extend and enhance methods first trialled in the UNEP report, providing a worked example of how near-term and multi-decadal risk assessment can be linked directly to attribution of recent events. This strengthens the demonstration value of the report, showing how similar approaches could be applied elsewhere.

We note that teh need to extend projections to cover future impacts of fire, not just othe ccurrence of fire itself, was discussed extensively in last year's report. We did not cover that in this report to avoid repetition, but we now include areference in the introduction of the projections section to last year's report, to highlight where development priorities were discussed previously.

2. In section 5 on Attribution to Global Change, be consistent in the language used to present level and extent of attribution. Section 5.2.1 deals largely with 'probability of occurrence being x times higher' or 'times more likely' than the counterfactual, which may be the same thing but is confusing. 5.2.2. suddenly presents attribution as 'x% likelihood of whereas

We have revised Section 5.2.1 to clarify when we refer to events being "more likely" (i.e., changes in the probability of occurrence of specific FWI values) and when we refer to "higher" FWI (i.e., changes in the magnitude of the index itself).

Section 5.2.2 uses three distinct attribution metrics. We have improved the description of these metrics in Section 5.1.3, presenting them in bullet form for easier reference and to make it clearer how each metric is calculated and interpreted.

We report attribution results both for the entire region (reported in the main text, **Section 5.2.2**) and for "sub-regional extremes" - the grid cells with the top 5% of BA, to also assess how anthropogenic factors may have influenced the most severely affected areas (in **Supplementary Material S5.2.2**). We use three complementary metrics to assess how our target factors have influenced burned area (BA) during extreme fire events:

- 1. Amplification Factor (AF) quantifies how much larger or smaller BA was because of the considered driver. It compares factual BA for events as large or larger than the observed target months with counterfactual BA. An AF > 1 indicates an increase due to the driver (e.g. AF = 2 → twice as much area burned, or an AF = ½ → half as much burned area due to the target factor). We calculate this across our model simulations and report both the central estimate (median) value and the range of uncertainties based on 10th to 90th percentiles.
- 2. **Likelihood of Attribution** the probability that BA was higher under the factual (with-forcing) world than it would have been in the counterfactual (without-forcing) world, expressed as a percentage.
- 3. For the NRT set up, we can also use the **Risk Ratio (RR)** which expresses how many times more likely an event of comparable BA was under factual versus counterfactual conditions. Similarly to **Section 5.1.2**, it compares the chance of seeing the observed BA under today's climate to the chance under a climate without human influence. A RR above 1 means climate change made the event more likely; a RR below 1 means it made it less likely.

Observed BA is calculated in a manner consistent with model outputs by averaging BA either across the entire region or, for sub-regional extremes, across the top 5 % of grid cells. Observations are taken from the monthly MCD64A1 dataset. In near-real-time (NRT) applications, the comparison is made for the specific year of interest; in the ISIMIP setup, the comparison spans 2003–2019.

The introduction of Amplification Factor without an explanation of what it was threw me while I

was trying to unpack the probabilities that were presented.

Amplification factor is already described in the results, but we have added a reminder at the first mention in the results. See response to specific comments below

3. In many of the analyses, there is much discussion of the comparison of predicted 'hotspots' versus actual 'hotspots' (being satellite derived). But's not clear how these 'hotspots' compare with actual fire events. Most of the discussion assumes that hotspots and fire events are synonymous, but they are not. Can the discussion be made more specific to fire events, since that is what the metric being assessed is?

We agree there is a distinction between hotspots and fire events, the modelling approach is constrained to satellite retrievals which are hotspot detections. It should be noted however that the system isn't based on raw hotspots detections, for which many hotspots can equate to a single fire but rather to a gridded hotspot product whereby multiple or a single hotspot within a 9km grid cell is classified as a single fire event. This partially overcomes this problem to an extent, although in the cases of large fire events spanning more than 9km^2 it does mean multiple "hotspots" might be classified by the model as multiple fires. We have introduced the concept of a cell with observed fire activity as a proxy for fire and explained this in the ms.

4. Similarly, there is much discussion throughout in which analysis of fire weather is conflated with ignition potential, particularly in regard to the use of the Canadian Fire Weather Index. The FWI is not designed to provide any insight into ignition potential or fire occurrence and so it is no surprise it does not perform well in this role. Since it is recognised as such (e.g. L1634), why do so many of the analyses persist with it? Has there been any published validation of the FWI for such uses? This is particularly important since it is stated on L1633 that FWI does not consider vegetation state in biomes other than boreal forest. And yet not 40 lines later (L1674) it states the FWI successfully identifies regions with elevated fire danger aligning with BA anomalies outside the boreal forests. There needs to be some consistent application of tools such as FWI and their treatment in the ms.

You are right that there is a certain messiness in the way fire tools are used and interpreted but this is a reflection of the limitation of current systems which the ms aims to bring to light. The Canadian FWI has clear limitations, it was never designed to capture ignition potential or to be universally applicable across ecosystems, yet it remains the backbone of many operational systems worldwide. Because of this, we felt it was important to reflect both the traditional reliance on FWI and its limitations in the report, as this is the reality of current practice.

At the same time, we wanted to highlight how newer approaches, including ML-based methods, can address some of these gaps. In some cases, the FWI does appear to capture fire danger outside boreal regions, but this tends to occur when the main driver is large-scale drought. In contrast, when the key driver is a vegetation anomaly under otherwise average weather conditions, the FWI is much less effective. This reflects the very real challenges operational forecasters face every day and the need to complement traditional tools with more advanced approaches.

Our aim, therefore, was not to blame the FWI as a non perfect or universally valid tool, but rather to show how it continues to be used, where it can and cannot provide insight, and how its limitations motivate the advances we are working toward.

5. Reliance upon FWI and PoF is largely going to highlight fire weather as the cause of fire events because they are largely (and specifically in the case of the FWI) fire weather products. Extrapolating these (particularly FWI) to fire occurrence a long bow to draw.

The PoF is not a metric of fire danger but rather of fire activity, and this is a fundamental distinction. PoF output is expressed as the number of predicted hotspots or rather cells that have seen at least one hotspot (refer to previous comment), a metric that can be directly verified against observations.

In contrast, the FWI is a measure of landscape flammability and not of fire occurrence, although it has long been used in this way given that it is used in fire forecasting worldwide. So we wanted to acknowledge its broad usage in this context

The distinction between PoF and FWI was already described in the manuscript.

"PoF is an example of a new generation of indicators based on machine learning methods that have recently been created to produce more informative operational predictions of wildfire (Shmuel et al., 2025; Di Giuseppe, 2023). One of the practical advantages of PoF is that it can directly output a prediction of the number of fire hotspots when averaged over vast areas which is directly comparable to active fire observations."

To complicate matters further, the FWI, while developed for operational information systems, is indeed a very useful indicator for identifying large-scale fire-prone conditions, making it valuable for climate studies, trend analyses, and attribution work. However, since it does not capture small-scale fire activity, as it lacks the granularity of information provided by fuel characteristics, its much less skilful than the PoF for the purpose for which it was actually developed.

For the next edition of the report, given the fast-moving pace of AI adoption in fire forecasting, we might consider avoiding the use of the FWI for this specific purpose

- 6. Despite the effort to explain what is meant by 'extreme' fire there is still no robust definition that I could find in the ms. A full and complete definition of the following words is required at the beginning of the ms:
  - a. Extreme fire event
  - b. Extreme fire area/extent
  - c. Extreme fire emission
  - d. Extreme fire activity
  - e. Extreme wildfire
  - f. Extreme wildfire season

A formal definition of what is considered 'extreme' is indeed a challenge to our project, and one that was given a full section of dedicated discussion and reasoning in our first report ("Appendix B: Frontiers in observing and modelling extreme fire occurrence and impact" > "B1.1 Definition of extreme fire events". Please note this is also a point raised by a reviewer last year, and one which we responded to in the public discussion.

In this report, to avoid repetition and considering that there have been no substantial advances in an objective definition of what is 'extreme' in the past reporting cycle, we cite the section of review from our first report. The text now reads:

We reviewed the range of approaches that can be taken to identify extreme wildfire events in our inaugural report (see Appendix B1.1A of Jones et al., 2024b). A universally accepted objective definition of "extreme" remains elusive, reflecting a series of data- and knowledge-oriented challenges. Data-oriented challenges include the absence of consensus on quantitative criteria, with no universally applicable thresholds for size, severity, or other measurable properties; pronounced geographic variability, as regional fire regimes dictate the relevance of particular thresholds; evolving definitions that have progressively expanded to encompass a broader range of fire types and behaviours under changing climatic conditions; and context dependence, whereby interpretation is contingent on ecosystem characteristics, historical fire regimes, and benchmarks such as return intervals or ecosystem damage. Knowledge-oriented challenges centre on the lack of agreement over qualitative criteria, including fire behaviour and impacts; the proliferation of overlapping terminology (e.g. "megafire," "catastrophic fire"); the influence of linguistic and cultural context on interpretation and reporting; the shaping of scientific terminology by societal discourse, necessitating accessibility to diverse audiences; and the limited rigour, clarity, and standardisation evident in existing definitions. Recognising these complexities and the need for transdisciplinary processes to establish robust, standardised criteria in future work, this report maintains a deliberately broad and flexible definition of extreme.

Although this comment may not entirely satisfy the reviewer's for a formal definition, we stress that we are transparent about the limitations of these definitional issues. We hope the new text can highlight the need for urgency and progress in the coming years.

7. Similarly, fire intensity is not defined (e.g. L383). Fire intensity can be interpreted any number of ways which are not compatible (e.g. fireline intensity (kW/m) and fire radiative intensity (kW/m^2). Please define and use consistently words like 'fire intensity' and similar (e.g. intensity of fire).

Thanks - yes that is indeed an important point. We felt that the most appropriate place to address this definition was in Section 2.1.1 "Earth Observations of Fire". The observational data used in our report derive from MOD14A1/MYD14A1 FRP, and we clarify the meaning as follows:

The underlying data for this analysis were daily observations of fire radiative power (FRP) from the NASA active fire products MOD14A1 and MYD14A1 (Giglio et al., 2016). These products report FRP as the instantaneous rate of radiative energy release from actively burning fires, expressed in megawatts (MW) per pixel. Conceptually, FRP provides a satellite-derived measure of the combustion rate, and is therefore more closely aligned with fire radiative intensity (W  $m^{-2}$ ) than with measures of fireline intensity (W  $m^{-1}$ ) used more frequently in field studies, fire behaviour modelling, and active fire management.

In addition, we added "(with respect to fire radiative power; see section 2.1.1)" to the line identified by the reviewer above so that the reader knows where to ascertain the more complete information above.

8. It is understood that MODIS is being phased out, to be replaced by VIIRS and Sentinel 3, etc. Since it is intended for this series of articles to continue can the authors provide a plan for enabling the future results to be comparable with current and past results using MODIS, even if it is a period of parallel presentation if necessary.

Indeed we are making plans for this transition. With respect to BA products, the harmonisation of MODIS and VIIRS data at appropriate resolutions was recently assessed to be "highly feasible" given the extremely strong correlations between the two NASA products (<a href="https://www.sciencedirect.com/science/article/pii/S0034425725004109">https://www.sciencedirect.com/science/article/pii/S0034425725004109</a>). The FireCCI project is also working to deliver a harmonised BA product based on relationships seen between FireCCI51 (MODIS-based) and FireCCIS311 (Sentinel 3-based) datasets.

Regarding individual fire properties, currently derived from the Global Fire Atlas based on MODIS BA data, the outlook is less clear but we are beginning to plan a harmonisation exercise. Further investigation is required to understand whether statistically-summarised properties (e.g. 95th percentile fire size) within spatial units (e.g. ecoregions, countries) can be harmonised across MODIS and VIIRS eras. We are in the process of running Global Fire Atlas with the new VIIRS BA product, and we will thereafter check the feasibility of harmonising spatially-aggregated individual fire properties subject to validation over overlapping time periods.

We hope that the public response provided to this comment can serve as a reference point to the more dedicated readers interested in our future planning. We have not altered the main text because no results or conclusions presented in the current submission are contingent on harmonised datasets at this stage. Our future reports will of course report in full on any methodological advances that are employed going forwards.

9. In the results section, be consistent in your comparative language. In some places, quantitative comparisons are number of times greater while in others it is given as a percentage, sometimes in the same sentence (see L912). This is related to Point 2 above in the attribution section.

Sorry, but we didn't consider this minor editorial point to be a high priority, and so we have not acted on this.

10. Be consistent in the presentation of large numbers. In some cases these are given as numerals, in others they are spelled out in full. e.g. 290 thousand ha versus 290000 ha.

Thank you we have adopted "thousand km²" throughout and also converted all numbers from ha to km² for consistency throughout the report.

11. Be consistent in the presentation of units. In some cases there is a space between the value and the unit, and in others there is not.

Yes thank you for pointing this out. We have gone through the manuscript and try to homogenised units where it is practical - e.g. please see above.

12. There is also a mix of tense throughout the ms, shifting sometimes in the same sentence between present and past tense.

Thanks for raising this issue. We have fixed this as much as we can, but without specific notation of the lines where this matters most, it is difficult to act on this comment. Comments from professional copy editors may help to address issues like this during the proofing stage.

13. FWI seems to be defined every time it is introduced (e.g. L356 & L 719 & L1602 to name a few). It only needs to be defined once.

We apologise for having overlooked this. We have now cleaned up the MS and FWI is only introduced in section 2

Specific comments:

L124: Superscript '2'. Define what you mean by 'extreme fire seasons'.

The number was removed in response to reviewer 1

L160: Delete apostrophe.

**Deleted**

L167: Should this not be 'climate change and land use attribution'? Currently it reads like the attribution has already been made.

Changed to read "It analyses key regional events in Southern California, Northeast Amazonia, Pantanal-Chiquitano, and the Congo Basin, assessing their drivers, predictability, and the influence of climate change and land us"

L179: 'extremely large, fast-moving or intense fires'. Is this your definition of 'extreme fire'? Seems circular.

Please see our response to the general comment above. As it happens, this sentence was removed in our broader re-framing of the introduction section, in response to comments from Reviewer #1.

L190: Define 'extreme fire extent'. I don't think 'extent' in this context is correct as it refers to the outer bounds of something, not the total area, which is what I think is intended.

We don't see an issue with the use of 'extent' here, with extent being the area covered by something (in this case, fire). This is per Cambridge, Merriam-Webster, Collins, etc, definitions available online.

L243: What about marine sinks? How do they compare?

We have included Marine in the paragraph and made a comment that extreme fires may also affect ocean carbon uptake "Emerging evidence further suggests that extreme fire events can influence

marine carbon sinks, through deposition of pyrogenic aerosols and nutrients that alter ocean biogeochemistry, creating feedbacks between terrestrial fires and the global carbon cycle (Tang et al. 2021)"

L249: 'India, the EU27 or the USA'. Do you mean together or separately? Not clear.

The use of 'or' signifies that it can be only one of the listed places, but to clarify further we added an 'Oxford comma': "India, the EU27, or the USA"

L251: 'greater vegetation mortality'. How true is this? How was it determined? In many fires across the globe, vegetation is not necessarily killed by fire—often it is the dead vegetation that is burning.

**Modified the sentence:**

"However, shifts in fire regime towards more widespread and severe fires have contributed to a reversal of terrestrial carbon budgets from sinks to sources in some regions, driven by the enhanced disturbance of vegetation and soils carbon stores (Zheng et al., 2021; Gatti et al., 2021; Nolan et al., 2021a; Phillips et al., 2022; Harrison et al., 2018; Jones et al., 2024a)"

The methods range from bottom-up modelling to top-down Earth obs and atmospheric inversions, and so for brevity we prefer to point the reader to the range of papers cited rather than expand on the various lines of evidence that supports the statement here.

L264-267: Not clear. Are you referring to use of fire as a land management tool (be it traditional or otherwise)? Be explicit here.

This sentence was removed during revisions to the introduction requested by other reviewers, but clarifying discussion along these lines now appears in the following paragraph:

Not all landscape fires are 'bad fires'. Many ecosystems are fire-adapted, with flora that have developed competitive advantages to defend against damage from fire or to resprout or regenerate after fire, and fauna that exploit the habitats created by fire-adapted vegetation (Kelly et al., 2020; Pausas and Keeley, 2023). As Pausas et al. (2025) note, fire is a "defining feature of our biosphere, having appeared when the first plants colonized the land, and it continues to occur across the planet at different frequencies and intensities". In addition, fire has played a vital role in the success of the human species, from its early domestication for cooking, warmth and protection, through millennia of cultural burning to shape landscapes and resources (Bowman et al., 2011; Pyne, 2011). Small-scale intergenerational fire use continues to be used by Indigenous and traditional communities around the world, and to label all fire as 'bad fire' would risk erasing culturally embedded stewardship, stigmatising traditional practices and cultural values, undermining livelihoods and biodiversity, and increasing future wildfire risk by preventing the low-intensity cultural burns that maintain habitat mosaics and keep hazardous fuels in check (Carmenta et al., 2021; Barlow et al., 2020; Pascoe et al., 2024). The practice of low-intensity prescribed burning, which recognises the need for fire on fire-adapted landscapes, is applied in many world regions for the purpose of hazardous fuel reduction or for the rejuvenation of vegetation aligned with vegetation adaptations, often with inspiration from cultural burning practices (Hiers et al., 2020; Hsu et al., 2025). Nonetheless, trends towards larger, more intense or more severe fire properties have the potential to push fire-adapted ecosystems

towards the edge of their physiological range (Kelly et al., 2020; Pausas et al., 2025). At the same time, low-intensity controlled burning are, in some regions, facing shrinking windows of weather conditions in which low-intentional burns can be safely maintained (Fernandes et al., 2013; Swain et al., 2023; Di Virgilio et al., 2020).

L277-281: Other mechanisms such as ignition risk reduction, pre-suppression, rapid early/initial attack are also employed.

Thanks those have also been integrated now

L325: Insert 'the' before 'context'

Done, thanks

L329: Change 'four extremes' to 'four extreme events'

Done, thanks

L334: Predict what for each focal event? By the looks of what is presented, you are attempting to predict fire occurrence. Please add that here.

Done, thanks

L345: When you say 'size and rate of growth' are you talking average, median or peak values?

Clarified "regional statistics of individual fire properties such as ..."

L356: How suited is FWI to this purpose? Has it been validated for this use? Provide references.

Added references to (Abatzoglou et al., 2019)) and (Clarke et al., 2022).

L383: 'fire intensity' needs to be defined as there are many different interpretations of it.

Please see response to general comment above.

L384: What is the functional difference between 'BA' and 'fire size'?

BA is the aggregated extent of all fires in a region. Fire size is the extent of individual fires.

L395-409: This needs to come earlier, with a concise definition in the abstract.

Room is extremely sparse in the abstract but this is now mentioned in the introduction:

As well as collating records of extreme fire activity from Earth observations, we convened regional expert panels for each continent to help identify events considered extreme in terms of their social, economic, and ecological impacts, thereby capturing important dimensions of fire activity not always visible in satellite data.

L397: 'comprehensive'. Do you mean 'rigorous'? In regard to impacts are you referring to direct impacts or long-term impacts?

Comprehensive was our intention - in the sense that focussing exclusively on a few measurable quantities/qualities risks excluding fires that are extreme in other ways. Hence we stick with 'comprehensive'. We're referring to all forms of impact - e.g. a fire with potential to cause significant excess death over a more protracted period is also of interest.

L413: Define what you mean by 'extreme fire weather days'.

Added clarification:

(95th percentile)

L442: For those of us who are not EO experts, what does 0.1 degree translate to metres at, say, the equator?

Added clarification:

(approximately 11km at the equator)

L451: What does 'scaled to emissions' mean?

Added clarification:

For the post-2016 period in GFED4.1s, emissions estimates rely solely on MODIS active fire data, with pixel-level scaling factors trained on the relationship between active fire detections and burned-area-driven emissions during 2003-2016.

L455: As with L383, 'intensity of the fires' needs to defined. How is it determined from BA and daily growth rate? It certainly would not be the 95th percentile of actual fire intensity.

Please see our response to the general comment above - a clarification was added.

L458: Can you explain exactly how FRP is directly related to 'the fire's intensity and fuel consumption' and how these are measured remotely?

Please see our response to the general comment above - a clarification was added.

L464: How can four daily measures of FRP provide information on fire intensity enough for a meaningful estimate of 95 th percentile?

All FRP measurements from within a Global Fire Atlas footprint are considered - not just one value from one point in space.

L469-472: As mentioned in the general comments, what is the plan to eventually transition over to VIIRS or Sentinel and what will it mean for consistency of reporting in this series? Point to section 2.1.1.4 for comparison of the various products.

Please see our response above.

L479: Not just small fires, but short-lived fires.

**Added clarification: "or short-lived"**

L481: An increase of global BA by 93% is huge.

L482: Given such a large increase in BA by including small fires, it is difficult to believe variability and trends in regional BA do not significantly differ when small fires are included. How can this be? L483-485: This is not clear. What is the generation of biases got to do with anything and why is it in line with the sensitivity of different sensors? Clarify.

L488: 'deemed highly suitable'. But is it the preferred option given the access to other sources of data? If you could use other products, would they produce better results? Seems like an attempt to sweep the problem under the carpet.

L490-497: How can uncertainties in a product be determined by comparing the uncertainties in other similar products? Uncertainty at some point needs to be related to actual burnt area that may or may not be observed remotely. What are the implications for the continued use of MODIS?

In response to the cluster of important comments above, we made wholesale changes and clarification to the first two paragraphs of section 2.1.1.2, which now read as follows:

We note that the MODIS BA product data used in our analyses of anomalies in BA and individual fire properties (via the Global Fire Atlas) are known to be conservative due to the limitations to detecting small or short-lived fires (e.g. agricultural fires) based on surface spectral changes at 500m resolution. Recent work has shown that including detections of small active fires increases global BA estimates by 93% (Chen et al., 2023). However, variability and trends in regional BA totals using datasets that include small fires do not differ significantly from the variability and trends present in the MODIS MCD64A1 BA product because the corrections made for small fires are consistent over time (Chen et al., 2023). In this report, we require a BA product that has global coverage and ideally a spatial resolution that can be aggregated within geographical divisions (e.g. Table 1), and also a temporal consistency over a multi-decadal time series up to the present year. The MODIS MCD64A1 BA product meets these needs and allows us to address our research questions, though we caution that the absolute values of BA reported by the MODIS BA product are underestimated due to small fire omission.

Uncertainties in the BA estimation can be decomposed into observational uncertainty and parameter uncertainty. Observational uncertainty arises due to errors of omission (missed detections of true fires) and commission (incorrect identification of fires that did not occur). The NASA MODIS MCD64A1 was previously assessed to have a 40% commission error and 73% omission error, with an overall 50% under-detection of BA versus higher resolution Landsat imagery due to omitted small fires (based on analysis from 558 sites selected by probability sampling). Note that these uncertainties are highly consistent with the reported 93% increase in BA in the GFED5 BA product versus MCD64A1, which occurs due to small fire omissions in MCD64A1 (Chen et al., 2024). Other work comparing the ESA FireCCI51 BA product (based on 250 m MODIS retrievals) with the high resolution ESA FireCCISFD11 BA product (based on 20m Sentinel-2 retrievals) also suggests an approximately 50% under-detection of BA in sub-saharan Africa (Chuvieco et al., 2022). Parameter uncertainty arises from the range of methodological choices that can be made when producing a global BA product based on the same or similar observational inputs. These uncertainties can be quantified by comparing different global BA products using similar observations as input. For

instance, the estimates of global BA from the NASA MODIS MCD64A1 product (based on 500m MODIS observations) are 20% lower than the BA estimates provided by the Copernicus Land service (2025; based on 300 m Sentinel-3 observations with similar sensor properties and with no corrections for small fires as in ESA FireCCI products).

L502-505: It is not clear what you mean here. Clarify.

Following changes above this sentence is redundant and so was removed.

L505-50513: This essentially repeats the first half of this paragraph.

This is discussing GFAS, not GFED, and there are some differences related to how the two different products are derived (e.g. active fires vs. burned area).

L515-534: So what is the uncertainty in FRP data?

Freeborn et al. (2013) quantified overall MODIS FRP uncertainty by analysing >400,000 near-simultaneous duplicate fire detections from consecutive satellite scans, comparing overlapping measurements to assess variability mainly in relation to a fire's position within the sensor's point spread function. They found uncertainties of ~27% (at 1 standard deviation) for individual fire pixels, but showed through simulations that these decline rapidly with aggregation, falling to ~17% for two-pixel clusters and below 5% once ~50 or more pixels are included, making large-fire or regional FRP estimates more robust than single-pixel estimates.

L528: Define 'MIR'

"mid-infrared (MIR)" was already defined 8 lines prior.

L547: Repeats L380.

Thanks, reworded to flag that this paragraph explains our reasoning for this choice.

L551: Do you mean boreal spring?

Thanks, yes, now added.

L558: How are these determined and validated?

The values are simply a count of the number of ignition points identified in the Global Fire Atlas. Validation was performed by (Andela et al., 2019) as part of the Global Fire Atlas data description paper and so we have added a citation.

L590-594: This is scary, if the new products are largely the same as the old products with similar uncertainties and no improvements. Is this correct?

Thanks for the good catch, here - it was incorrectly stated that the regional BA totals were very similar, but in fact it is the *anomalies* in BA totals that were very similar. We have thus clarified:

With very few exceptions, we find a high level of consistency between the MCD64A1, FireCCIS311, and VIIRS VNP64A1 BA products with regards to both the relative magnitude of regional BA anomalies, the geographical distribution of those anomalies, and the rankings of BA in the 2024-25 fire season versus previous fire seasons since 2019 (Figure S1; Jones et al., 2025).

L595: What does 'generally replicated' mean?

Replaced with "consistent"

L597-599: Repeats previous statement. Where are improvements to come from?

Doesn't seem to be a precise replication as it describes a future outlook, but we have re-phrased to reduce this sense of replication.

L620-626: This is a reasonable approach, but for outyears beyond the current ms, will you maintain the top 5 or the upper quartile? They are close enough now, but in a few years where you need 6 fires to be a quartile, what will you do? Best to decide now if you will now and forever take the top 5 or the upper quartile.

Completely agree with this foresight, and it is something we are planning to address in our future workshops alongside topics such as bridging the MODIS-VIIRS/Sentinel eras.

L629: 'event's peak'. To be clear, to which event are you referring?

Thanks, re-phrased for clarity:

"we identified the 'peak' month as the maximum anomaly between monthly BA values in March 2024-February 2025 and the climatological mean monthly values from the prior March-February periods"

L644-648: Unnecessary detail. At the scale we are talking, most cells will contain components of each combustion mode.

Thanks, removed.

L675-680: Repeats L400.

We cut down the text in the paragraph of L400 to reduce repetition.

L726: Explain why you chose 95%. Previously it was the upper quartile. Be consistent in what you consider 'extreme'.

Just as a clarification: The upper quartile is (as written) only used for visualisation purposes - we do not state that it is our definition of extreme events. (paraphrased text: "For visualisation purposes, we identified regions in which the latest fire season ranked in the top 5 of all annual fire seasons on

record [...] a top-5 ranking translates approximately to a fire season in the upper quartile of those on record."

Regarding the 95th percentile threshold for FWI specifically - this is a very broadly metric in studies of extreme fire weather. We added a sentence "The 95th percentile value for extreme fire weather has been used in many prior publications (Abatzoglou et al., 2019; Jones et al., 2022)."

L760: What is the basis for this conclusion?

The referenced work shows a 40% rise in BA in forests and a 50% rise in C emissions per unit area from forest fires, aggregating to a 60% rise in forest fire C emissions. We made minor changes to the sentence:

These anomalies, signifying lesser fire extent but more severe fires than average, are consistent with a reported trend towards increased fire extent and intensity in forests globally (Jones et al., 2024a)

L763: You start off by saying '3.7 million km^2 burned globally, but now at the end of the paragraph you say 'at least 3.7 million km^2). Say this first.

'At least' now used in both instances.

L781: here and throughout: What do you mean by '(sub)tropical'? Is it tropical or subtropical or both? Be clear.

We added a sentence to clarify:

The notation "(sub)tropical" is an abbreviation of "tropical and subtropical" and is used consistently in this report.

L781: The use of the same numerical value (290,000 km^2) for two things in this paragraph is a tad confusing. Is this correct?

Yes, a bit awkward but those are the correct values.

L788/Figure 1: Is there any reason why the secondary y-axis cannot be scaled such that the heights of the boxes cannot also be read as relative anomaly (%) instead of using an additional symbol that is almost impossible to see against the blue?

Scaling the second axis further would suppress that the bar heights become smaller which is not desirable. See for example the lower panel where the triangle representing % anomaly is already at the top of the plotting space.

L882-892: How do you know that these BA weren't burned by multiple individual fires that had coalesced?

We have used the Global Fire Atlas (as explained in the methods section). Different approaches can result in different individual fire definitions but we have clarified our use of the Global Fire Atlas in the methods section and so by definition all fires are as mapped by the Global Fire Atlas algorithm - details of how fires are distinguished are available from the cited paper, (Andela et al., 2019)...

L892: 'intensity value'. What is this? Explain. See L458.

Simplified to "intensity".

L1044/Figure 5: Put the panels of this figure in the same order in which they are presented in the text (i.e. Amazonia, Pantanal, SoCal, Congo). (Or reorder the text to match the figure).

Sorry but in our view this was not a high priority versus other important comments raised here, and so we have not changed the figure or the text.

L1061-1063: Do we need this sentence, since it's already been explained?

We think this helps readers to navigate the text so would prefer to keep it.

L1068: What is 'climatological mean'?

Clarified "For seven months in a row, the observed burned area was greater than the historical average for those same months, based on the reference climatology since 2002."

L1079-1080: How was this determined?

This is the only logical explanation: if C emissions reduce, but by a smaller proportion than BA, then the emissions per unit BA must be smaller than usual.

L1092: January- March 2024 is largely outside your year of study.

Indeed this was awkward but we didn't want the region to be excluded from our reporting just because it lies in the relatively few world regions where the dominant fire season is split across two fire seasons - we took a flexible stance here.

L1128-1132: What is the difference here between 'homes' and 'structures'? Can you present consistently as one or the other, rather than both?

Different cited sources were reporting different things - One was reporting homes (i.e. only structures in which people live). The other was reporting structures (any type of building including homes, shops, warehouses, etc). So, the terminology used currently is true to the facts presented.

L1143-1147: Not a sentence. What about the water from the federal reservoirs?

Corrected to "were released from"

L1176: 'prior year to two'. What does this mean?

"Prior two years" clarified

L1266: Should this sub-head be 'Carbon Offset Projects Exposure' to be fully correct? See L1268.

"Carbon Projects" is fine in our opinion.

L1290: 'a toxic mix of gases'. The mix is not toxic but some of the gases in it are toxic. Include volatile organic compounds (VOCs) in the list.

Thanks added.

L1294: Are cancers in this list?

No this is a list of short term effects which have received greater attention in the literature.

L1307: What do you mean by 'prescribed'? I would avoid this word in connection with fires as many planned burns are called 'prescribed' fires.

Clarified "Fire emissions in CAMS derive from..."

L1462: Something missing from 'the northern tropical of South America'.

"Parts" added thanks.

L1487: Delete comma after 'Note'.

Done thanks.

L1497: As per L1266.

Sorry but it is unclear what the suggestion is here.

L1499: Change to '...cost-effective climate change mitigation...'

Done thanks.

L1525: Everywhere else you refer only to BA, not burned area. Consistency.

Thanks good spot.

L1601: FWI is not designed for predicting fire occurrence, for many of the reason stated in the analysis L1631-1641. Why was it used in the first place? Seems criminal to abuse such a venerable system in this way and then complain about it.

Please refer to the comment above. We are not misusing the FWI; if anything, we are emphasizing that it is a strong metric for large-scale identification of landscape flammability. Our intention is to highlight both its strengths and its limitations: while it is valuable for assessing broad fire-prone conditions, it is not a measure of fire occurrence and cannot capture small-scale fire activity.

L1619: Do you mean predictions of wildfire occurrence?

Yes, corrected

L1628-1630: Performance of PoF should be compared against actual, not FWI.

Both methods are validated against fire activity, they are not compared against each other. They are shown in the same validation exercise.

L1634-1635: What do you mean 'too far in advance of actual fire emergence'?

High fire weather was reported ahead of actual fire activity. We have explained this more clearly now.

L1674: This sentence conflicts with what is presented in L1631-1641.

Sentence reworded

L1681-1694: Is this detail necessary? Suggest cutting.

It was removed in the interest of maintaining focus

L1713-1724: Repeats L343-L365.

It was removed in the interest of maintaining focus

L1744: How is this error determined?

The error in Sparky-PoF is not actually derived through an ensemble forecast as described in the original text. This is an approach that is possible but would only represent NWP and fuel uncertainty and not structural uncertainty of the model. Instead we estimate model uncertainty by first computing the error with observations then we apply the relative error to each of the derived attribution values. The text has been updated to reflect this.

A representation of PoF attribution uncertainty is made by applying a relative error derived from the comparison of observations and forecast.

Meanwhile, ConFLAME quantifies uncertainty from two sources: (i) parameter uncertainty, arising from the Bayesian inference of the relationships between drivers and burned area, which generates posterior distributions for each fitted function; and (ii) stochastic uncertainty, which reflects the inherent variability in ignition and spread processes, and is critical for representing extreme fire events. By combining these two components, ConFLAME provides confidence intervals that account both for uncertainty in the estimated role of drivers and for irreducible randomness in fire occurrence. We have added:

In ConFLAME, uncertainty has two main components. First, the uncertainty in driver—response relationships: because ConFLAME uses a maximum entropy, Bayesian inference framework, the strength and form of associations between predictors (e.g. climate, land use, population density) and burned area are not fixed but estimated from the data. This generates posterior distributions for each fitted function, which translate into confidence intervals for their contribution to fire probability. Second, the model explicitly incorporates stochastic uncertainty, which is particularly important for extreme fire events. This reflects the inherent randomness in ignition and spread under similar conditions, and ensures that variability in event BA is not smoothed away. Together, these components produce confidence intervals that account both for parameter uncertainty and for the stochasticity of fire occurrence which is irreducible locally (i,e gridcell), but can reduce over scale (i.,e over focal regions)

L1751-1753: Fuel moisture does not lead to increased vegetation growth.

Reworded for clarity. We meant rainfall producing fuel accumulation not fuel moisture

L1755: '...lots of trees...' Huh?

Removed:D

L1761: In many of the subsection in these results, way too much background is provided. This could be pruned considerably.

We have streamlined this section, but we would like to point out that this is not a standard manuscript. The Methods section focuses solely on the methodologies and does not normally introduce the meteorological conditions of the regions. In this case, however, including such background information is necessary to contextualize the result.

L1768: How is the number of 'hotspots' converted to the number of fires? These are not necessarily the same. What are you suddenly referring to hotspots instead of fires?

Sorry for the confusion. We have introduced the concept of a cell with observed fire activity as a proxy for fire. We have explained this in the method. Also please refer to the comment before

L1776-1777: Are these not parts of two separate fire seasons? 2023/4 and 2024/5?

As this report is global, we have defined a SOW year. This period was chosen carefully, as February represents the global minimum in fire activity based on climatology. While this inevitably means that in a few locations the defined year may cut across local fire seasons, such overlaps are unavoidable in a report covering a full annual cycle. Moreover, in many tropical regions the concept of a defined fire season does not strictly apply, as fire-prone conditions can occur throughout the year.

L1781: 'fire hotspots' Are these separate fires?

These are cells on a 9x9 km box that have recorded fire activity. For very large fires we might have cells that split the fires into separate instances. This is now explained in the methods

L1788-1797: This paragraph is essentially discussing the difference between fire weather and ignition potential. Fires do not occur without both but each does not influence the other.

It is not entirely true that ignition and weather are uncorrelated, as favorable weather conditions can increase ignition potential. However, we agree the connection might be weak in the tropical regions. This discussion focuses on the different components that determine fire activity: fire weather, fuel availability, and ignition potential. Omitting any of these reduces predictive skill. While these factors are not fully independent and each influences the others, for the sake of building an explainability model we separate them and analyze them in isolation. This is precisely the aim of this section.

L1814: 'worst water crisis'. What is this? Too little or too much water? Poor quality? Be specific.

We have added 'availability' so now it reads worst water availability crisis

L1828-1829: How does this translation work? How well do they compare?

The translation performs well in terms of reliability. At large scales, the number of cells with fire is comparable to the total number of observed fires. This is fully validated in (Di Giuseppe et al., 2025), which is referenced here.

L1877: This is true only if the total number of fires remains constant.

L1938: Change 'controls' to 'factors'. Topography is critical for influencing fire intensity.

**Changed**

L2345: Insert 'Initiative' before '(Barnes'.

**Inserted**

L2354: Why mean daily relative humidity? Why not minimum to match the maximum air temperature?

Modelled CF data is not available, and we have included note in this sentence "Due to the availability of model output, which is typically only available on a daily temporal resolution, variables ...

L2537: Is there a reference for DSR? Van Wagner 1987 is a reference for FWI, not DSR.

We have checked, Van Wagner 1987 does also define DSR

L2553: Change 'lead' to 'led'.

**corrected**

L2583: What is 'Amplification Factor' and why is it suddenly being introduced here? It should at least be introduced in the methods section of this chapter. It needs a full explanation since it becomes an important metric.

The amplification factor is introduced on line 2450 in the methods section of the original manuscript, with reference to equations in the supplement: "The Amplification Factor (AF) tells us how much bigger (or smaller) the burned area was because of a specific factor. It works by comparing factual burned area for values as large or larger than those observed during the target months against the counterfactual."

We now provide a reminder of what it represents when first introduced in the results, clarifying that it expresses "how much larger or smaller burned area is because of climate change."

L2610-2617: Why is this paragraph here? It is superfluous background.

This has been removed though some details have been merged into section 2.2.2.1, which provides an overview of the focal regions selected

L2888: It's not clear why this section is here. Did you do it for the previous State of Wildfires? If so, can you analyze the quality of the predictions you did for this fire season? I'm not sure how detailed outlooks for regions that burned this season will be of much relevance next year.

Thank you for this comment. Last year we provided an overview of the upcoming fire season to showcase the availability of seasonal FWI forecasts. With the launch of the State of Wildfire website and considering the timeframe constraints of a published paper, we also realised that this section may be removed from future SOW reports, as this information will likely become available through a more dynamic web interface.

L3326: This chapter is an excellent summary of the breadth and depth of the document and deserves to be placed at the very beginning of the document, perhaps with some streamlining.

See response to main comment

L3670: Add 's' to 'deviation.

added

L3671: Ditto.

added

L3692: 'vegetation growth'. There is *always* vegetation growth when it rains. What about it? Is there a missing adjective?

The sentence now read "higher antecedent rainfall can lead to greater vegetation growth"

L3710: Change 'find' to 'found'.

corrected

L3737: Delete 'made'.

removed

L4024: Insert comma after 'data'.

added

L4029: Insert 'research' before 'projects'.

added

L4040: Should this read '...international collaborators routinely...'?

Yes thanks fixed

**Reviewer 3**

**General comments**

The State of Wildfires represents an important community effort to provide a scientifically rigorous account of key wildfire events during the 2024–25 fire season. The ambition and scope of the report are commendable, and it is encouraging to see such a broad range of expertise contributing to this work. I agree with the other reviewers that the report is both timely and significant. However, before recommending publication, I believe several issues should be addressed to ensure the report fully achieves its goal of delivering actionable insights for stakeholders and society.

Thank you very much for your comments.

**Non-scientific comments:**

**1) Readability and clarity**

I share Reviewer 2's view that the report is dense and difficult to follow. Beyond being heavy, the presentation sometimes obscures both intention and interpretation, which is very important given the target audience of this work. This does not stem from excessive technicality, but rather from how the material is structured and presented. I recommend targeted editing:

a. **Abstract/Executive Summary:** The abstract is very dense does not facilitate clear and quick understanding. I think the suggestion of using the current conclusion for this as suggested by Reviewer 2 is a very good one, as I agree it very well written.

Regarding the suggestion to move the current conclusions section to the abstract: Please see our full response to reviewer #2, above, who made the same suggestion. To summarise, we must use the conventional ESSD structure; moving the Conclusions to serve as an Abstract is not permitted under journal guidelines. However, we acknowledge this important point shared by the reviewers, and we will continue to explore this option with the journal for future report iterations. Please be assured that broader (non-academic) audiences will be able to access this information up-front in our Summary for Policymakers document (which includes a longer-format Executive Summary). It is encouraging to hear that the reviewers think the conclusions section is effective, because we base the summary for Summary for Policymakers document directly on the conclusions section.

[...cont...] I would also suggest presenting results in order of impact: (i) human fatalities and losses, (ii) emissions, and (iii) attribution results. Additionally, I recommend reconsidering the phrase "global fire activity" in the opening paragraph:

L121: "The State of Wildfire Project systematically tracks and analyses global fire activity and this, its second annual report, covers the March 2024 to February 2025 fire season."

Whilst the report does acknowledge regions with lower-than-average fire activity, it does not focus on average global fire activity, and negative extremes are also not the focus of the report. Unless these themes are developed further, for example by highlighting cases where high fire weather coincided with unusually low burnt area, or highlighting regions of constant trends, it may be better to only mention the focus on (positive) extreme events here. Since most of the report emphasizes positive extreme fires, without transparently acknowledging this here, the framing could unintentionally bias readers toward viewing fire occurrence only in negative (and extreme) terms, which could have major policy implications.

In order to flag our concentration on extreme events, the sentence flagged for clarification has been replaced with: "The State of Wildfire Project systematically tracks global and regional fire activity in each annual fire season, analyses the causes of prominent extreme wildfire events, and projects the likelihood of similar events occurring in future climate scenarios.".

We appreciate the suggestion to re-consider the use of the term "fire activity". We do acknowledge that it is sometimes used to refer to "active fire hotspots" in certain branches of fire science. Unfortunately, it is virtually impossible to avoid terminological overlaps given the general academic obsession with definitional obscurities. What we mean by 'fire activity' here is essentially a whole list of variables: burned area, carbon emissions, fire sizes, fire rates of spread, fire intensities .... Etc... It is not practical to list these in an abstract that is already described as 'dense'. "Fire activity" seems an imperfect though pragmatic summary of the multiple fire-related variables we are tracking and so, after consideration, we decided not to change the term.

b. The role of fire and defining "extremes": This relates to the previous point about unintentional bias. From my reading, the report fails to explicitly state that globally fire is (i) unavoidable and (ii) a necessary part of Earth systems and biodiversity. This absence risks reinforcing a simplistic view that "more fire is bad" and "less fire is good," which may mislead stakeholders. I think some discussion of this context is needed. Related to this, the introduction directly begins discussing "extreme fire events" without first defining "extreme." These terms should be defined clearly at first mention.

Thanks, good point. We added the following paragraph, explicitly stating that not all fire is bad, to the introduction:

Not all landscape fires are 'bad fires'. Many ecosystems are fire-adapted, with flora that have developed competitive advantages to defend against damage from fire or to resprout or regenerate after fire, and fauna that exploit the habitats created by fire-adapted vegetation (Kelly et al., 2020; Pausas and Keeley, 2023). As Pausas et al. (2025) note, fire is a "defining feature of our biosphere, having appeared when the first plants colonized the land, and it continues to occur across the planet at different frequencies and intensities". In addition, fire has played a vital role in the success of the human species, from its early domestication for cooking, warmth and protection, through millennia of cultural burning to shape landscapes and resources (Bowman et al., 2011; Pyne, 2011). Small-scale intergenerational fire use continues to be used by Indigenous and traditional communities around the

world, and to label all fire as 'bad fire' would risk erasing culturally embedded stewardship, stigmatising traditional practices and cultural values, undermining livelihoods and biodiversity, and increasing future wildfire risk by preventing the low-intensity cultural burns that maintain habitat mosaics and keep hazardous fuels in check (Carmenta et al., 2021; Barlow et al., 2020; Pascoe et al., 2024). The practice of low-intensity prescribed burning, which recognises the need for fire on fire-adapted landscapes, is applied in many world regions for the purpose of hazardous fuel reduction or for the rejuvenation of vegetation aligned with vegetation adaptations, often with inspiration from cultural burning practices (Hiers et al., 2020; Hsu et al., 2025). Nonetheless, trends towards larger, more intense or more severe fire properties have the potential to push fire-adapted ecosystems towards the edge of their physiological range (Kelly et al., 2020; Pausas et al., 2025). At the same time, low-intensity controlled burning are, in some regions, facing shrinking windows of weather conditions in which low-intentional burns can be safely maintained (Fernandes et al., 2013; Swain et al., 2023; Di Virgilio et al., 2020).

Line 609: "(i) as relative anomalies (expressed in %) from the annual mean during all previous March-February periods since 2002 (2003 for fire C emissions); (ii) as standardised anomalies (standard deviations) from the annual mean during all previous March-February periods since 2002 (2003 for C emissions); (iii) as a rank amongst all March-February periods since 2002 (2003 for fire C emissions), March 2024-February 2025 inclusive."

Given that many regions have very long fire return intervals, the short record used here should be acknowledged explicitly and transparently. I would like to see discussions of the implications and limitations of this. I suggest framing these as extremes "within the lifespan of an individual today" which both clarifies the scope and makes the results more relatable while transparently acknowledging the limitation. I think this type of framing would also make the future attribution figure where the likelihood of experiencing an event if some was born *today* vs born at other dates link quite nicely. However this is just a suggestion, I am not sure exactly what the best way to get around this is!

I also appreciate the use of regional expert panels. This approach mitigates the limitations of the satellite products and incorporates diverse forms of knowledge and expertise, which is desperately needed. However, it should be noted that "extremes" identified in this way often reflect vulnerability and resilience shaped by factors beyond climate such as governance, funding, and policy decisions, which are not strictly biophysical conditions.

Line 102: "Examples of extremes that can be captured by expert assessment (but not by Earth observations) include: suppression difficulty; fatalities and structure loss; impacts on human health and wellbeing; impacts on agricultural and other economic sectors; impacts on biodiversity, and; impacts on diverse ecosystem services such as recreation, tourism, or other cultural values."

Line 680: "This includes (but is not limited to) wildfires that impacted society by causing fatalities, evacuations, displacement (e.g. homelessness), direct structure or infrastructure loss or damage, degradation of air or water quality, loss of livelihood, cultural practice or other ways of life, and loss of economic productivity. This definition also includes (but is not limited to) wildfires that impact the

environment via disturbance to vulnerable ecosystems, biodiverse areas, or ecosystem services such as C storage."

For example, many of the things mentioned here to not relate directly to the climate drivers. Some discussion of this, and how it takes more of a risks definition of extreme rather than a biophysical one would be nice.

Thanks. We added the following to methods section 2.1.3.1: "We stress that extremes identified in this way often reflect vulnerability and resilience shaped by factors beyond climate, such as governance, funding, and policy decisions, which are not strictly biophysical conditions."

- c. Focal point events: The analyses from the focal events were difficult to follow because of the fragmented presentation of the results, which felt at time quite repetitive. I recommend restructuring the Results section into 1) Global results and 2) Focal event case studies. Each case study could then follow a consistent template with:
  - The regional context
  - The description of fire event
  - Causes (from previous literature)
  - Prediction analyses and causal inferences (PoF and ConFLAME) Attribution results

Line 1961: "Northeast Amazonia experienced an exceptionally severe fire season between January and April (Figure 10), driven by extreme drought which started in 2023, intensified by the combined effects of El Niño and the Atlantic Meridional Mode, which brought unusually high temperatures and suppressed rainfall."

Line 1996: "According to our Sparky-PoF analysis, the extreme fire activity during the 202425 fire season in the Pantanal-Chiquitano (described in Section 2.2.2.2), was mainly the result of extremely dry weather which had started since 2023..."

Grouping all the information for each focal event would clarify what comes from background literature versus what is novel (PoF, ConFLAME), which is sometimes difficult to decipher. This would also reduce repetition and open space for deeper analysis, including integration of more of the excellent figures in S4.2. The recent California wildfire studies (McNorton et al., 2025) demonstrate how impactful such indepth case analyses can be, and adopting a similar approach here would greatly strengthen the report.

Thanks. This comment has generated a lot of discussion within the core team, and it is absolutely something that will be on the agenda for our de-brief on this year's reporting cycle prior to the kick-off of the next reporting cycle in Spring 2026.

For this iteration of the report, to ensure its timeliness, we have decided to stick with the current layout but indeed we can see the advantage of your proposed separation of the global elements and the analyses applied to focal regions.

**Scientific comments (major):**

**1) Attribution of human influence**

I agree with Reviewer 1 that the current approach to human attribution is problematic. Their suggestions for improvement are excellent, and I simply want to reiterate their concerns.

See responses to reviewer 1

**2) Implication of single year analysis**

By its very design, the report focuses on wildfire events in an individual year. From my understanding, the weather/fuel/land use conditions relating to that specific year are then used to make causal inferences. This is fine; however, we know that many of the drivers of individual wildfire events can result from multi-year to decadal process such as fuel accumulation, shifting vegetation composition, and land use change as well as policy decisions such as suppression, and these will not occur on yearly timesteps. This especially relates to fuel conditions, such as fuel load but also fuel continuity. This point is especially important attribution work, both for the causal attribution of the focal events and the future the attribution results. I believe in nearly all the focal events, a multi-year process is described in the introduction to that event, and climate modes such as El Nino are also mentioned. It would be nice to clarify if PoF or ConFLAME take these multi-year processes into account? If they do not, it seems to me that the results are going to be bias towards attributing weather as the driving cause of fire events, regardless of how much longer-term processes may play a role, given that this is the only factor which operates on a yearly timestep. More discussion is needed to clarify if the current methods capture these long-term drivers, and if they do not, this limitation needs to be addressed. In the future, it would be nice to see future reports can account for the multiyear process (i.e. in both the ranking, forecasting and attribution).

We feel this is a very pertinent question. The PoF system does not explicitly include a variable with memory in the sense of storing past years' conditions, but it is informed by variables that are prognostically derived through physical modelling. This means that multi-year processes are implicitly represented, as the state of fuels at any given time reflects the legacy of antecedent weather, vegetation dynamics, and land—climate interactions.

For example, the long-term accumulation of fuels following moist periods is captured in the fuel product, even though the attribution is made at the time of the fire. In this framework, such effects are attributed to the fuel state rather than directly to the prior weather that enabled its build-up. This raises an important conceptual point: should we consider the driver "fuel today," or "weather from previous seasons," given that one is a prerequisite of the other? Our current approach takes the state variable as observed on the day of the event, but in doing so, it inherently includes the influence of past conditions that shaped that state.

This distinction is important for attribution. While the PoF framework focuses on the conditions directly linked with the fire event, the prognostic nature of the fuel variables ensures that multi-year processes such as fuel accumulation and continuity are not ignored, but are expressed in the attribution to fuel. Future work may benefit from making this linkage between short-term and long-term drivers more explicit.

To clarify this point we have added this to the ms

Sparky-PoF inherently reflects long-term conditions, for example antecedent weather and multi-year processes are expressed in the fuel state on the day of the event. In such cases, e.g. where prior weather manifests through its influence on fuel accumulation, it is therefore categorised as a fuel driver rather than as weather itself. We assign past conditions that build up fuel loads to the fuel category, while shorter-term processes such as drying are attributed to weather, though the boundary between these two timescales is not always clear

Like the PoF framework, ConFLAME relies on the same prognostic fuel inputs, which already incorporate the legacy of antecedent weather, vegetation dynamics, and land–climate interactions. This means that multi-year processes such as fuel accumulation or continuity are implicitly represented, even though the driver assessment itself is conducted at the time of the fire. In our framework, these effects are expressed through the state of the fuels at the time of the event, rather than being attributed directly to the weather or vegetation processes that enabled their build-up.

In addition, the Bayesian structure of ConFLAME ensures that these legacies are not treated deterministically. The posterior distribution explicitly quantifies both the uncertainty in how fuels relate to burned area and the stochastic variability that drives extreme outcomes. In practice, this means that antecedent processes embodied in the fuel predictors are expressed as a distribution of possible influences, rather than collapsed into a single fixed weight. This reduces the risk of biasing attribution toward weather-only explanations and helps maintain a balanced representation of both short-term and longer-term drivers.

Future work could extend this further by more explicitly linking fuel states to their antecedent processes, for example by incorporating direct memory terms or process-based socioeconomic drivers. For now, however, ConFLAME's Bayesian design ensures that multi-year processes embedded in fuel predictors are propagated through the posterior, allowing them to be reflected in the attribution results rather than ignored.

**3) Discussion of the model limitations:**

It appears that both PoF and ConFLAME cannot reproduce the southern Californian focal event or the Northern Amazonia one (Figure S17). More evaluation and transparent discussion of this (and what it means) is needed.

Line 1788: "Interestingly, both the PoF and FWI systems failed to capture a lull in fire activity during the second emergence in August-November of fire-conducive conditions showing the limitations of forecasting fire activity rather than fire danger."

The forecast for Southern California, despite its off-season occurrence, was very good with both systems (see validation plot available in supplementary material for ConFlame). However, the forecast for the Northern Amazon was accurate only for the first wave of fire activity. The second wave was predicted but did not occur. Our expert panel suggested that this discrepancy was due to changing patterns of human ignitions in the region at different times of the year. The first reviewer also contributed to interpreting this mystery of shifting human behaviours, and thanks to his suggestions, the text in this section has been reworded as follows.

Interestingly, both the PoF and FWI systems failed to capture a lull in fire activity during the second emergence of fire-conducive conditions between August and November, highlighting the limitations of forecasting fire activity rather than fire danger. In this region, ignitions are believed to be largely driven by escaped pasture burning Cano-Crespo et al. (2015), which typically occurs between August and October Jakimow et al. (2018). The models may have learned and reproduced this seasonal behaviour, but such patterns can be disrupted by changes in human practices. One possible explanation is that these conditions fell outside the usual burning cycles—for example, in agricultural areas where fires are often timed around harvest, the prolonged drought may have reduced crop yields and therefore fire use. This suggests that the models missed the quiet September period because they incorporate only limited information on human ignition patterns, land ownership and land use types, and less-documented factors such as fire suppression, policy interventions, and cultural burning practices Lapola et al. (2023). These gaps underscore the need for improved datasets on human activity, which could significantly enhance fire prediction (Jones et al., 2022).

**4) Attribution analysis:**

Given that ConFLAME does not capture some of the focal events (above point), some more evaluation is needed on the model's ability to reproduce the contemporary fire record in specific case study locations. Are interannual variability, as well as previous extremes in the observed record captured, and are there benchmarking metrics available for this? I may have missed this, if so, ignore this comment.

We now include the standard evaluation procedure for Bayesian-based attribution models introduced by (Barbosa et al., 2025) and used in last year's report. The key aim of this evaluation is to ensure that the model's posterior distribution adequately represents the observed fire record in each case-study region — a necessary condition for robust attribution analysis.

This evaluation has two main components:

- 1. **Coverage of observations:** The model's posterior uncertainty range should encompass the observed burned area. In other words, the historical observations should fall within the range of outcomes generated by the model.
- 2. **Lack of systematic bias:** The observed values should be distributed randomly within the posterior, rather than clustering consistently at one edge, which would indicate bias.

Meeting these criteria demonstrates that the model sufficiently captures interannual variability and historical extremes to support attribution statements. In some regions, such as Southern California, this approach results in relatively wide uncertainty ranges, reflecting limited predictability. However, this uncertainty is accounted for in the subsequent attribution step: what matters is whether there is a statistically significant difference between the factual and counterfactual probability distributions. If such a difference exists, attribution conclusions remain valid even when overall uncertainty is high, provided the two evaluation criteria are met.

The updated evaluation, together with the attribution results already presented, shows that these criteria are met in all study regions, including Southern California, allowing us to make robust attribution statements.

Below we show evolution just for Southern California from driver assessment, as the full evaluation is now an extra 12 pages of supplment. But full evaluation across regions can be found in the review supplement section 9.

Posterior evaluation diagnostics for ConFLAME in Southern California. Each panel shows three sets of diagnostics. Top row: Observed annual average BA for December–February (left), compared to the 5th (middle) and 95th (right) percentiles of simulated BA. Middle row: Scatterplot of observed BA (x-axis) versus likelihood of observations under the posterior (y-axis), where high observed BA should correspond to high likelihood values; spatial maps of the 5th (middle) and 95th (right) percentiles of observation likelihood across all months. Bottom row: Scatterplot of observed BA (x-axis) versus posterior-simulated BA (y-axis), with vertical ranges representing the 5th percentile, interquartile range, and 95th percentile; a map of the average posterior rank position of observations (ideal ≈0.5); and a map of the significance of deviations from 0.5, indicating where bias may be present.

The 2025 Los Angeles fires produced the highest burned area (BA) observation in the record, concentrated in the northern part of the region, where the 10-year mean BA reached ~0.3%. The model posterior captures this hotspot, but also indicates that, given the training data, elevated BA is plausible in the southern portion of the region. Posterior uncertainty ranges are wide in both sub-regions, spanning from negligible burning to values of 1–10% BA. Despite this spread, the probability of the observed values under the posterior, P(Obs|Model), is very high (>0.92) across the

region. Observations fall predominantly within the central mass of the posterior distributions, though with a slight tendency toward underestimation at the upper extreme (observations align, on average, with the 69nd percentile of the posterior for the most extreme events, i.e. 99–100% quantiles). This mild bias remains within acceptable limits according to the criteria of Barbosa et al. (2025), supporting the conclusion that the posterior adequately represents both the central tendency and extreme tail of the observed distribution in Southern California.

Scientific comments (minor but would like to see addressed in future reports):

**1) Buffering in ranking and additional metrics a. Fire regimes and variability**

Fire regimes diverge dramatically, and this includes their year-to-year variability in fire activity. It would be nice to see an analysis of the relative spread in the observational record for the different fire metrics, as I think this would contextualize some of the results. For example, in regions with long fire return intervals and very low burnt area averages, a single year (early in the record) can disproportionately alter the mean, adding uncertainty to the signal and potentially obscuring more recent extremes. In contrast, regions with consistent fire properties make extreme years more robustly interpretable.

We respond to part a and b in conjunction, below, as part b appears to continue on from part a (unless we are mis-interpreting the intention behind the main question "is the baseline mean approach appropriate?" below - apologies if so).

**b. Trended data**

The authors discuss the overall trends in fire both inside and outside tropical regions, which are a result of climate, vegetation and land use change. Given that we know there is a trend in recent fire data, is the baseline mean approach appropriate? Complementing this with rolling windows (e.g., 5-year averages) or highlighting long-term trends in focal regions could add context. This would strengthen the policy relevance by distinguishing between consistent yearto-year patterns (with trends) and one-off anomalies that a mean value alone cannot capture.

We agree that situating a given year within both the immediate past and the broader two-decade context can be informative in some regions/circumstances. We note several considerations in relation to this point.

First, the presentation of rankings already provides a partial means of addressing cases where an early-timeseries extreme could otherwise diminish the apparent significance of a more recent event. For example, even if the largest value occurred in 2002, an extreme in the final year would still appear as the second-highest in the record, ensuring that the year is clearly flagged as notable. This ranking approach therefore mitigates, to some extent, the risk that percentage anomalies understate the relevance of more recent extremes when judged against a 21st-century baseline.

Second, Figure S1 does provide recent context, and the accompanying data files include a broad set of statistics. These cover the mean and standard deviation of the observed metrics across the full

timeseries, as well as the raw annual values, which allow users to calculate anomalies or assess extremity using whichever baseline they deem most relevant.

While it would be possible to incorporate additional statistics directly in the main text, doing so would substantially increase complexity. Using different baselines for different regions introduces inconsistencies that can complicate interpretation, while shortening the baseline period exacerbates the challenge of limited sample sizes relative to longer fire return intervals. Given that the report has already been described as "a beast," there is a trade-off between comprehensiveness and clarity that we must carefully balance in future iterations. We will give this due consideration in advance of our next reporting cycle.

Finally, it is worth clarifying that the issue raised is most acute at fine spatial resolutions, where the probability of recording no fire in many years but a very large event in a single year is high. When aggregating to larger units (e.g. Canada as a whole rather than a  $5 \times 5$  km grid cell), this probability diminishes, as there will almost always be some fire activity across the broader region. Importantly, none of the focal regions in this or the previous report are characterised by persistently low fire activity at the spatial scales assessed, and we therefore do not expect the reviewer's concern to materially affect our dedicated analyses of focal events.

**2) Discussion of climate modes**

Given that the satellite record is so short, it would be useful to also contextualize the rankings in the context of climate modes. Do certain areas consistently show higher rankings in El Nino years for example? If so, what are the implications of this? I don't think this is something that necessarily needs to be addressed in this report, however, it should be considered for future reports, as may explain/influence the ranking results.

Thanks. This is a great idea, though it deserves some dedicated thought because there is actually a lot more variability within the same ENSO faces (El Niño, La Niña) than is often appreciated - e.g. this recent paper showing very weak precip changes in the 2023/24 El Niño (

<a href="https://www.nature.com/articles/s43247-025-02584-8">https://www.nature.com/articles/s43247-025-02584-8</a>). Also, though fire does respond to ENSO phase (e.g. <a href="https://www.nature.com/articles/s41558-017-0014-8">https://www.nature.com/articles/s43247-025-02584-8</a>) in a predictable pattern globally, the correlations can generally be characterised as 'apparent... but weak'. Differences in the delay in the response across the regions makes an analysis quite a challenging exercise!

One paper that does do what you are suggesting is this one focussed on Amazonia <a href="https://www.nature.com/articles/s41467-017-02771-y">https://www.nature.com/articles/s41467-017-02771-y</a>, concluding that droughts (tied to climate modes aligning) are increasingly governing fire extent in the Amazon via influence on wildfires. However these results have also been contradicted by reports showing that any effect of drought from climate mode phases are dwarfed by IAV in deforestation fires (<a href="https://www.nature.com/articles/s41598-021-82158-8">https://www.nature.com/articles/s41598-021-82158-8</a> ).

In summary; this is a difficult space to explore. It might be something to sort out in a separate study before we attempt to do this in our annual assessment report.

**3) Fire weather vs actualized fire**

I would be very interested to see a discussion about the differences in the FWI rankings and the BA rankings. Comparing Figure 4 and Figure S2, it becomes very clear that the high ranking FWI regions do not necessarily correlate to the high ranking FWI (which is to be expected). Whilst South America does emerge very clearly in high rankings for both, Southern Africa, the coast regions of Western and Northern Africa, Central Europe, Scandanavia, and Russia all show very high rankings in FWI but low rankings in BA. This seems to be anomalous in some of these regions, where the reported percentage reductions are huge:

Indeed this is important and the regions you highlight are all (or at least mostly?) discussed in Section 2.2.1. To supplement the existing discussion in Section 2.2.1, we now make a clear position statement that we think distinctions like these should be explored more enthusiastically in future.

While the present report focuses primarily on explaining focal events that did emerge as extremes, we recognise the underexplored value of examining the factors that constrain fire occurrence in regions where anomalously high fire weather might otherwise be expected to drive extremes in burned area and associated carbon emissions. Future iterations of the State of Wildfires assessment may therefore consider giving greater emphasis to understanding why such extremes did not materialise. That said, this type of analysis has not, to our knowledge, been a common approach in fire science to date. For example, we are not aware of any formal attribution studies focusing on non-extreme fire events, in contrast to the growing number of attribution studies of extreme events. It may therefore be more appropriate for such investigations to be pursued initially as a dedicated exercise, whether within our network or by others.

Line 993: "BA was around 50 thousand km 2 (71%) below average in the Asian temperate grassland, savannah and shrubland biome, 42 thousands km 2 (62%) below average in the Asian xeric shrublands"

This appears to me like another point where a focus on longer-term processes, as well as a discussion of fire regimes and fire ecology more generally would help explain these mismatches. Relating to the previous points, it would be interesting to relate these regions of mismatch to previous fire years, or, for example, highlight them as regions in which, given that *this* year we saw unusually low fire activity, excess fuel may have built-up, which may, in combination with the right weather conditions, increase the likelihood of extremes in the next few years. Such an approach would also yield policy-relevant insights, identifying areas where fire management attention may be needed not just because of recent fires, but also because of reduced fire activity in recent years that could elevate future risk.

Following on from our responses above: We acknowledge the view of multiple reviewers that greater focus should be placed on explaining negative or non-extremes. We support this; however, applications of existing methods in this area would be novel and some capabilities may need to be built (not only by us, but by the wider field). Hence, we are not implementing the study of non-extreme of negative extremes in the current report.

If, however, there is indeed an extreme +ve fire season in the coming years worthy of direct focus in our future reports, then we will clearly at that stage investigate the role of low fire activity in prior years as a contributor to fuel build-up.

While it would be possible to comment on the possible explanations for the low fire seasons in the highlighted biomes, the explanations have not really been explored properly through dedicated analyses, and so we do not feel we can say anything too concrete here; the list of plausible explanations for a -ve fire season could be extremely large (not only about previous fire seasons). With brevity in mind, we opt not to go down the route of writing a shopping list of explanations.

Finally, please consider that this section primarily serves as a "world tour" of fire statistics from the 2024-25 fire season. Later sections delve into specific events. Further comments about the factors driving fire extremes (e.g. heat, drought) are occasionally provided in the current section, but this is generally only if the regional panel raised these factors as explanatory factors in Appendix A.

**References**

Abatzoglou, J. T., Williams, A. P., and Barbero, R.: Global emergence of anthropogenic climate change in fire weather indices, Geophys. Res. Lett., 46, 326–336, https://doi.org/10.1029/2018gl080959, 2019.

Andela, N., Morton, D. C., Giglio, L., Paugam, R., Chen, Y., Hantson, S., van der Werf, G. R., and Randerson, J. T.: The Global Fire Atlas of individual fire size, duration, speed and direction, Earth Syst. Sci. Data, 11, 529–552, https://doi.org/10.5194/essd-11-529-2019, 2019.

Barbosa, M. L. F.: TRACING THE ASHES: UNCOVERING BURNED AREA PATTERNS AND DRIVERS OVER THE BRAZILIAN BIOMES, PhD, Instituto Nacional de Pesquisas Espaciais, 2024.

Barbosa, M. L. F., Kelley, D. I., Burton, C. A., Ferreira, I. J. M., da Veiga, R. M., Bradley, A., Molin, P. G., and Anderson, L. O.: FLAME 1.0: a novel approach for modelling burned area in the Brazilian biomes using the maximum entropy concept, Geosci. Model Dev., 18, 3533–3557, https://doi.org/10.5194/gmd-18-3533-2025, 2025.

Barnes, C., Jain, P., Keeping, T. R., Gillett, N., Boucher, J., Gachon, P., Heinrich, D., Kirchmeier-Young, M., and Boulanger, Y.: Disentangling the roles of natural variability and climate change in Canada's 2023 fire season, Environ. Res.: Climate, 4, 035013, https://doi.org/10.1088/2752-5295/adec0f, 2025.

Burton, C. A., Lampe, S., Kelley, D., Thiery, W., Hantson, S., Christidis, N., Gudmundsson, L., Forrest, M., Burke, E., Chang, J., Huang, H., Ito, A., Kou-Giesbrecht, S., Lasslop, G., Li, W., Nieradzik, L., Li, F., Chen, Y., Randerson, J., Reyer, C. P. O., and Mengel, M.: Global burned area increasingly explained by climate change, Nat. Clim. Chang., 1–7, https://doi.org/10.1038/s41558-024-02140-w, 2024.

Calkin, D. E., Barrett, K., Cohen, J. D., Finney, M. A., Pyne, S. J., and Quarles, S. L.: Wildland-urban fire disasters aren't actually a wildfire problem, Proc. Natl. Acad. Sci. U. S. A., 120, e2315797120, https://doi.org/10.1073/pnas.2315797120, 2023.

Clarke, B., Otto, F., Stuart-Smith, R., and Harrington, L.: Extreme weather impacts of climate change: an attribution perspective, Environ. Res.: Climate, 1, 012001, https://doi.org/10.1088/2752-5295/ac6e7d, 2022.

Croker, A. R., Woods, J., and Kountouris, Y.: Community-Based Fire Management in East and Southern

African Savanna-protected areas: A review of the published evidence, Earths Future, 11, https://doi.org/10.1029/2023ef003552, 2023.

Devisscher, T., Malhi, Y., Rojas Landívar, V. D., and Oliveras, I.: Understanding ecological transitions under recurrent wildfire: A case study in the seasonally dry tropical forests of the Chiquitania, Bolivia, For. Ecol. Manage., 360, 273–286, https://doi.org/10.1016/j.foreco.2015.10.033, 2016.

Di Giuseppe, F., McNorton, J., Lombardi, A., and Wetterhall, F.: Global data-driven prediction of fire activity, Nat. Commun., 16, 2918, https://doi.org/10.1038/s41467-025-58097-7, 2025.

Ferreira Barbosa, M. L., Haddad, I., da Silva Nascimento, A. L., Máximo da Silva, G., Moura da Veiga, R., Hoffmann, T. B., Rosane de Souza, A., Dalagnol, R., Susin Streher, A., Souza Pereira, F. R., Oliveira e Cruz de Aragão, L. E., Oighenstein Anderson, L., and Poulter, B.: Compound impact of land use and extreme climate on the 2020 fire record of the Brazilian Pantanal, Glob. Ecol. Biogeogr., 31, 1960–1975, https://doi.org/10.1111/geb.13563, 2022.

Frieler, K., Volkholz, J., Lange, S., Schewe, J., Mengel, M., del Rocío Rivas López, M., Otto, C., Reyer, C., Karger, D., Malle, J., Treu, S., Menz, C., Blanchard, J. L., Harrison, C. S., Petrik, C. M., Eddy, T., Ortega-Cisneros, K., Novaglio, C., Rousseau, Y., Watson, R. A., Stock, C., Liu, X., Heneghan, R., Tittensor, D. P., Maury, O., Büchner, M., Vogt, T., Wang, T., Sun, F., Sauer, I. J., Koch, J., Vanderkelen, I., Jägermeyr, J., Müller, C., Rabin, S. S., Klar, J., del Valle, I. V., Lasslop, G., Chadburn, S., Burke, E., Gallego-Sala, A., Smith, N., Chang, J., Hantson, S., Burton, C., Gädeke, A., Li, F., Gosling, S., Schmied, H. M., Hattermann, F., Wang, J., Yao, F., Hickler, T., Marcé, R., Pierson, D., Thiery, W., Mercado-Bettín, D., Ladwig, R., Ayala-Zamora, A. I., Forrest, M., and Bechtold, M.: Scenario setup and forcing data for impact model evaluation and impact attribution within the third round of the Inter-Sectoral Model Intercomparison Project (ISIMIP3a), Geosci. Model Dev., https://doi.org/10.5194/gmd-17-1-2024, 2024.

Huntingford, C., Kelley, D. I., and Barbosa, M. L. F.: A call to refine fire attribution: expanding the FAR statistic to capture the complexity of Los Angeles extreme fires, Environ. Res. Lett., 20, 091003, https://doi.org/10.1088/1748-9326/adf12b, 2025.

Jones, M. W., Abatzoglou, J. T., Veraverbeke, S., Andela, N., Lasslop, G., Forkel, M., Smith, A. J. P., Burton, C., Betts, R. A., van der Werf, G. R., Sitch, S., Canadell, J. G., Santín, C., Kolden, C., Doerr, S. H., and Le Quéré, C.: Global and regional trends and drivers of fire under climate change, Rev. Geophys., 60, https://doi.org/10.1029/2020rg000726, 2022.

Kasoar, M., Perkins, O., Millington, J. D. A., Mistry, J., and Smith, C.: Model fires, not ignitions: Capturing the human dimension of global fire regimes, Cell Rep. Sustain., 1, 100128, https://doi.org/10.1016/j.crsus.2024.100128, 2024.

Keeley, J. E.: Fire management of California shrubland landscapes, Environ. Manage., 29, 395–408, https://doi.org/10.1007/s00267-001-0034-y, 2002a.

Keeley, J. E.: Native American impacts on fire regimes of the California coastal ranges, J. Biogeogr., 29, 303–320, https://doi.org/10.1046/j.1365-2699.2002.00676.x, 2002b.

Keeley, J. E. and Fotheringham, C. J.: Historic fire regime in southern California shrublands, Conserv. Biol., 15, 1536–1548, https://doi.org/10.1046/j.1523-1739.2001.00097.x, 2001.

Keeley, J. E., Fotheringham, C. J., and Morais, M.: Reexamining fire suppression impacts on brushland fire regimes, Science, 284, 1829–1832, https://doi.org/10.1126/science.284.5421.1829, 1999.

Kelley, D., Gerard, F., Dong, N., Burton, C. A., Argles, A. P. K., Li, G., Whitley, R., Marthews, T., Roberston, E., Weedon, G., Lasslop, G., Ellis, R. J., Bistinas, I., and Veenendaal, E.: Fire, environmental and anthropogenic controls on pantropical tree cover, Commun. Earth Environ., 5, 714, https://doi.org/10.1038/s43247-024-01869-8, 2024.

Kelley, D. I., Bistinas, I., Whitley, R., and Burton, C.: How contemporary bioclimatic and human controls change global fire regimes, Nat. Clim. Chang., 2019.

Kelley, D. I., Burton, C., Huntingford, C., and Brown, M. A. J.: Low meteorological influence found in 2019 Amazonia fires, 2021.

Marques, J. F., Alves, M. B., Silveira, C. F., Amaral E Silva, A., Silva, T. A., Dos Santos, V. J., and Calijuri, M. L.: Fires dynamics in the Pantanal: Impacts of anthropogenic activities and climate change, J. Environ. Manage., 299, 113586, https://doi.org/10.1016/j.jenvman.2021.113586, 2021.

Menezes, L. S., de Oliveira, A. M., Santos, F. L. M., Russo, A., de Souza, R. A. F., Roque, F. O., and Libonati, R.: Lightning patterns in the Pantanal: Untangling natural and anthropogenic-induced wildfires, Sci. Total Environ., 820, 153021, https://doi.org/10.1016/j.scitotenv.2022.153021, 2022.

Perkins, O., Kasoar, M., Voulgarakis, A., Smith, C., Mistry, J., and Millington, J. D.: A global behavioural model of human fire use and management: WHAM! v1. 0, Geoscientific Model Development, 17, 3993–4016, 2024.

Pessôa, A. C. M., Morello R.S., T. F., Silva-Junior, C. H. L., Doblas, J., Carvalho, N. S., Aragão, L. E. O. C., and Anderson, L. O.: Protected areas are effective on curbing fires in the Amazon, Ecol. Econ., 214, 107983, https://doi.org/10.1016/j.ecolecon.2023.107983, 2023.

Singh, M., Sood, S., and Collins, C. M.: Fire dynamics of the Bolivian Amazon, Land (Basel), 11, 1436, https://doi.org/10.3390/land11091436, 2022.

Tomas, W. M., Berlinck, C. N., Chiaravalloti, R. M., Faggioni, G. P., Strüssmann, C., Libonati, R., Abrahão, C. R., do Valle Alvarenga, G., de Faria Bacellar, A. E., de Queiroz Batista, F. R., Bornato, T. S., Camilo, A. R., Castedo, J., Fernando, A. M. E., de Freitas, G. O., Garcia, C. M., Gonçalves, H. S., de Freitas Guilherme, M. B., Layme, V. M. G., Lustosa, A. P. G., De Oliveira, A. C., da Rosa Oliveira, M., de Matos Martins Pereira, A., Rodrigues, J. A., Semedo, T. B. F., de Souza, R. A. D., Tortato, F. R., Viana, D. F. P., Vicente-Silva, L., and Morato, R.: Distance sampling surveys reveal 17 million vertebrates directly killed by the 2020's wildfires in the Pantanal, Brazil, Sci. Rep., 11, 23547, https://doi.org/10.1038/s41598-021-02844-5, 2021.

UNEP, Popescu, A., Paulson, A. K., Christianson, A. C., Sullivan, A., Tulloch, A., Bilbao, B., Mathison, C., Robinson, C., Burton, C., Ganz, D., Nangoma, D., Saah, D., Armenteras, D., Driscoll, D., Hankins, D. L., Kelley, D. I., Langer, E. R. L., Baker, E., Berenguer, E., Reisen, F., Robinne, F.-N., Galudra, G., Humphrey, G., Safford, H., Baird, I. G., Oliveras, I., Littell, J., Kieft, J., Chew, J., Maclean, K., Wittenberg, L., Anderson, L. O., Gillson, L., Plucinski, M., Moritz, M., Brown, M., Soto, M. C., Flannigan, M., Costello, O., Silva, P. S., Fernandes, P., Moore, P., Jandt, R., Blanchi, R., Libonati, R., Archibald, S., Dunlop, S., McCaffrey, S., Page, S., Gonzãfâilez, T. D. T., Sokchea, T., and Charlton, V.: Spreading like Wildfire: The Rising Threat of Extraordinary Landscape Fires, edited by: Sullivan, A., Kurvits, T., and E., B., United Nations Environment Programme and GRID-Arendal., 2022.

Wei, Y., Gannon, B., Young, J., Belval, E., Thompson, M., O'Connor, C., and Calkin, D.: Estimating WUI exposure probability to a nearby wildfire, Fire Ecol., 19, https://doi.org/10.1186/s42408-023-00191-6, 2023.